# Unzipped chromosome-level genomes reveal allopolyploid nematode origin pattern as unreduced gamete hybridization

Dadong Dai [1,2,4], Chuanshuai Xie [1,2,4], Yayi Zhou [1,3], Dexin Bo [1,3], Shurong Zhang [1,2], Shengqiang Mao [1,3], Yucheng Liao [1,3], Simeng Cui [1,2], Zhaolu Zhu [1,3], Xueyu Wang [1,2], Fanling Li [1,2], Donghai Peng [1,2] ✉, Jinshui Zheng [1,3] ✉ & Ming Sun [1,2] ✉

The formation and consequences of polyploidization in animals with clonal reproduction remain largely unknown. Clade I root-knot nematodes (RKNs), characterized by parthenogenesis and allopolyploidy, show a widespread geographical distribution and extensive agricultural destruction. Here, we generated 4 unzipped polyploid RKN genomes and identified a putative novel alternative telomeric element. Then we reconstructed 4 chromosome-level assemblies and resolved their genome structures as AAB for triploid and AABB for tetraploid. The phylogeny of subgenomes revealed polyploid RKN origin patterns as hybridization between haploid and unreduced gametes. We also observed extensive chromosomal fusions and homologous gene expression decrease after polyploidization, which might offset the disadvantages of clonal reproduction and increase fitness in polyploid RKNs. Our results reveal a rare pathway of polyploidization in parthenogenic polyploid animals and provide a large number of high-precision genetic resources that could be used for RKN prevention and control.

Most animals, plants, and fungi create offspring through sexual reproduction, while asexual reproduction is a rare yet widely distributed trait in eukaryotes[1–3]. Although why most organisms maintain sexual reproduction remains an unanswered question, some hypotheses, such as the faster rate of adaptive evolution and elimination of deleterious mutations than those of asexual species are hypotheses with empirical support[3–5]. However, a small number of eukaryotes maintain obligate parthenogenetic reproduction, and hybridization and polyploidization generate unique combinations of genetic variation in parthenogenetic species[3]. Due to a lack of recombination, compared with their sexual relatives, parthenogenetic species usually exhibit weak competitiveness during environmental adaptation and are considered to accumulate harmful mutations by a mechanism

known as Muller's ratchet[6,7]. Therefore, asexual reproduction is considered an evolutionary dead end. However, there are also some exceptions.

Parthenogenesis is often associated with hybridization and polyploidization[3,8], which may help asexual species escape Muller's ratchet[7]. Therefore, some parthenogenetic species can gain an advantage by increasing their ploidy level and can even surpass their diploid sexual relatives in adaptability[9]. Polyploidization is an important driving force for speciation and adaptation, which has been well reported in many polyploid species[10–12]. An interesting example is *Otiorhynchus scaber*, a weevil that has both diploid sexual reproduction and triploid and tetraploid parthenogenesis; polyploid parthenogenic species have a stronger survival advantage over their diploid

[1]State Key Laboratory of Agricultural Microbiology, Hubei Hongshan Laboratory, Huazhong Agricultural University, Wuhan 430070, China. [2]College of Life Science and Technology, Huazhong Agricultural University, Wuhan 430070, China. [3]Hubei Key Laboratory of Agricultural Bioinformatics, College of Informatics, Huazhong Agricultural University, Wuhan 430070, China. [4]These authors contributed equally: Dadong Dai, Chuanshuai Xie. ✉e-mail: donghaipeng@mail.hzau.edu.cn; jszheng@mail.hzau.edu.cn; m98sun@mail.hzau.edu.cn

sexual relatives[9]. One of the hypotheses for the origin of parthenogenesis and polyploidy in weevils is that polyploidy results from nondisjunction of haploid chromosome sets (like unreduced gametes)[9]. In addition, species with polyploidy and parthenogenesis in nature are not unique cases.

Plant-parasitic nematodes (PPNs) are estimated to cause approximately $80–173 billion in economic losses to global crop production annually[13–15], and they pose a threat to global food security[16]. Root-knot nematodes (RKNs, *Meloidogyne* genus) are the most devastating PPNs, as they can infect the roots of almost all vascular plants[13]. RKNs exhibit a variety of reproduction strategies from amphimixis to obligate parthenogenesis and multiple ploidy levels from diploid to tetraploid[17]. The Clade I tropical RKNs[18], such as *M. incognita* (Mi), *M. arenaria* (Ma) and *M. javanica* (Mj), are usually characterized by obligate mitotic parthenogenesis and allopolyploidy. Polyploid RKNs show a stronger survival advantage, such as wider geographical distributions and host ranges, than their diploid sexual relatives[17,19]. Early studies hypothesized reticulate evolution and hybrid origins in polyploid RKNs[20], and many work verified this hypothesis[21,22]. Recently, several hypotheses about the origin of polyploid RKNs have been proposed: (1) facultatively parthenogenetic RKNs (meiotic) are the ancestor of obligate parthenogenetic RKNs (mitotic) through either intra- or interspecific hybridization following suppression of meiosis during oocyte maturation; (2) polyploids probably originated from the hybridization between an unreduced diploid maternal gamete and a haploid paternal gamete; (3) the reticulate hybrid origin of apomictic RKNs might result from combinations of closely related females with more diverse paternal lineages[22]; (4) some researchers propose that Ma and Mj are sister to each other while others propose that Mi and Ma as sisters; and (5) Mi, Ma and Mj all descend from hypotriplicated hybridization events which involved the addition of another copy of the genome to an existing diploid genome[19,22–24]. Previous studies found that the mitochondrial genomes of Mi, Ma, and Mj diverged very little, suggesting that their hybridization events share a recent common maternal ancestor[22]. Additionally, it is reported that interspecific hybridization between two facultative meiotic parthenogenetic diploid RKNs can occur under both laboratory and natural conditions and produce sterile offspring[25], suggesting that the hypothesis of a hybrid origin from parthenogenetic species of polyploid RKNs is highly plausible. However, due to the high fragmentation of draft genomes[14,22,26] and high variability of karyotypes[17], genetic information on multiple subgenomes is limited; thus, the specific origin and hybridization processes of polyploid RKNs and their genomic changes after hybridization remain largely unknown.

In this study, we found no canonical nematode telomeric repeats in these RKN genomes. Instead, we identified two putative novel types of repeats, respectively in the diploid and polyploid species enriched at the extremities of contigs or scaffolds. Our further cytogenetic analysis, along with a recent cytogenetic study[27], collectively confirmed a terminal position of the repeats on Mi chromosomes, suggesting that they are telomeric repeats. After two rounds of assembly, we obtained 4 unzipped chromosome-level genomes for polyploid RKNs, and 1 haploid telomere-to-telomere (T2T) genome for *M. graminicola* (Mg), a nematode with both sexual reproduction and meiotic parthenogenesis, which serves as an outgroup reference for polyploid nematodes. Unzipping the genome into several subgenomes is crucial for a comprehensive and realistic representation of polyploid RKN genomes. Our further analysis based on subgenomes revealed the origin history and karyotype evolution of polyploid RKNs during the hybridization process. We then investigated the chromosomal variation and reconstructed the ancestral chromosome of polyploid RKNs and determined the levels of homologous gene expression after polyploidization. Our findings shed light on the evolutionary mechanisms of obligate parthenogenetic animals and provide accurate and complete genomic resources for the study and control of PPNs.

## Results

### Exploration of the genome structure of Clade I RKNs

To reveal the origin and evolution of the most destructive RKNs, we collected 5 nematodes, including 4 mitotic parthenogenetic allopolyploids (Mi, Mj, and two Ma), and 1 facultative meiotic parthenogenetic diploid nematode (Mg) with a stable karyotype[28] as an outgroup (Supplementary Fig. 1). We sequenced these 5 samples using the Illumina and PacBio platforms, yielding 9.2–22.1 Gb of short-read data and 15.5–33.9 Gb of long-read data (Supplementary Data 1). To determine the phylogenetic position of our samples, we first assembled the mitochondrial genome for these species and constructed a phylogenetic tree alongside other published data (Supplementary Fig. 2a and Supplementary Data 2). The mitochondrial tree showed that most species was monophyletic, consistent with the species identity. However, we noticed that the currently defined Ma was not monophyletic but composed of two lineages, and our two Ma samples were located in these two lineages, respectively (Supplementary Fig. 2a, b). Moreover, a kmer-based genome survey suggested that one of our Ma samples was triploid, while the other one was tetraploid (Supplementary Fig. 3). The results even suggested that two Ma samples were different species that cannot be distinguished with SCAR markers (Supplementary Fig. 1), but were clearly different based on their mitochondrial genomes (Supplementary Fig. 2). Considering that the published Ma genome is tetraploid[22], we proposed naming the triploid Ma sample as Ma 3n and the tetraploid sample as Ma 4n in the rest of our study.

We performed de novo assembly of the sequences for the diploid Mg and polyploid species using PacBio long reads, and contigs were corrected with Illumina paired-end reads. The assembly size of Mg was 45 Mb, which was consistent with the haploid genome size estimated by k-mer (Supplementary Data 3, Illumina data) and a little bigger than previously published ones genome size (45 Mb vs. 36.86–41.5 Mb), with a longer N50 value (826 kb vs. 20.4–294 kb)[29–31]. The genome size of the polyploid species ranged from 213 Mb to 312 Mb, which was consistent with the whole genome size estimation based on k-mer[32] (Supplementary Data 3) and flow cytometry[22], suggesting the homoeologous subgenomes have been correctly separated during genome assembly. Based on Hi-C data, we successfully anchored the 87 contigs of Mg onto 18 chromosomes (Fig. 1a and Supplementary Data 4). Due to the high complexity of the chromosome structure, not all the subgenomes of polyploid species could be directly constructed at the chromosome level based on Hi-C data. Therefore, we attempted an exploratory assembly to clarify the genome structure of polyploid species. We combined the collinearity between the contigs of polyploid species and the diploid Mg genome as well as the Hi-C matrix to construct scaffolds of polyploid species (Supplementary Fig. 4). This drat preliminary assembly was used to elucidate the genome structure of polyploid species and the phylogenetic relationships between those homologous chromosomal sequences. Finally, a total of 48–67 scaffolds of polyploid species with N50 values of 4.1–4.4 Mb were obtained and named assembly version 1 (assembly v1, Supplementary Fig. 5 and Supplementary Data 5). Our assembly version 1 was verified to have high completeness by CEGMA[33] (95.1–96%), BUSCO[34] (50.6–51.4% for genome level, higher than other published *Meloidogyne* genomes, Supplementary Fig. 6 and Supplementary Data 6), and read mapping (97.9–98.2% mapping ratio for RNA-seq short reads). Read depth analysis showed that the depth of most scaffolds was consistent with the overall depth of the whole genome, except for 3 scaffolds in Mi, suggesting that assembly version 1 was mostly complete with a low degree of collapsed subgenome (Supplementary Fig. 7).

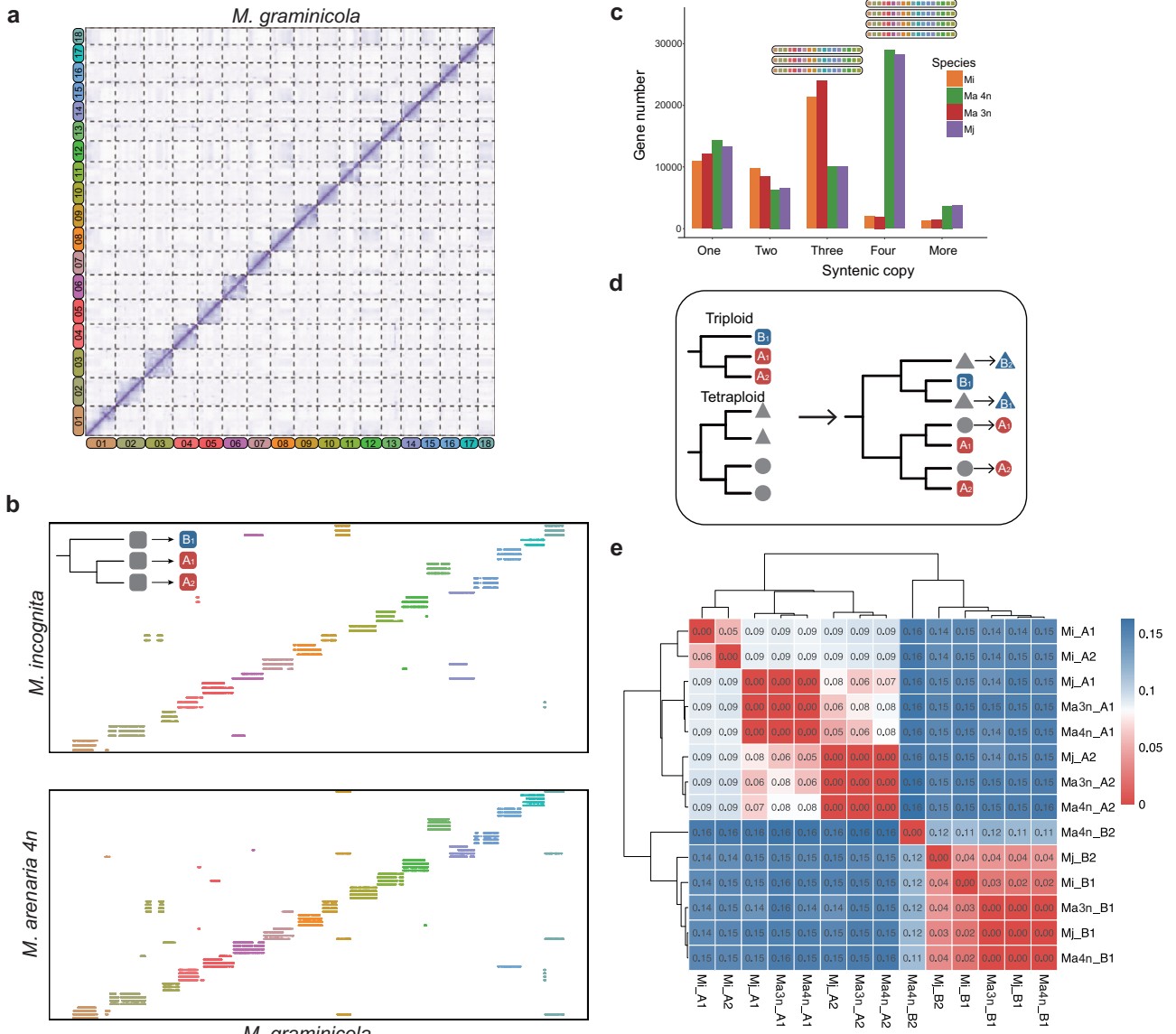

**Fig. 1 | Genome structure of *Meloidogyne*. a** Genome-wide Hi-C heatmap of 18 *M. graminicola* chromosomes. **b** Dot plot of the genomic synteny between *M. incognita* (triploid) or *M. arenaria* 4n (tetraploid) and *M. graminicola*. A schematic diagram of the subgenome distinguishing triploid species is shown in the upper left. **c** Distribution of the copy number for syntenic genes among four species. The copy number is counted from the synteny blocks between polyploid species and *M.* *graminicola*. **d** Schematic diagram of the subgenome distinguishing tetraploid species. **e** Heatmap and clustering map of synonymous substitutions (*Ks*) between 14 subgenomes (assembly version 1). The color intensity represents the median *Ks* value of all syntenic gene pairs between two subgenomes. Source data are provided as a Source Data file.

Approximately 28.1–32.9% of the genomes for the polyploid species were predicted to be repetitive sequences, whereas only 8.5% was predicted to be repetitive for the Mg genome (Supplementary Fig. 8 and Supplementary Data 7). The increase in repetitive sequences of polyploid species was consistent with previous study[22]. We identified 12,968 protein-coding genes from the Mg genome and 43,863–63,446 from the polyploid genomes (Supplementary Data 8). The BUSCO scores were 69.5–70.5% for proteomes, which was also higher than other published *Meloidogyne* genomes (Supplementary Fig. 9 and Supplementary Data 9). Synteny analysis showed that each Mg chromosome usually had a syntenic relationship with 3, 3, 4, and 4 scaffolds of Mi, Ma 3n, Ma 4n, and Mj, respectively (Fig. 1b, c; Supplementary Figs. 10 and 11), suggesting that Mi and Ma 3n were triploid, while Ma 4n and Mj were tetraploid, which was consistent with the ploidy estimation based on k-mer[32].

To explore the genome structure of polyploids, we investigated a total of 2185–4776 complete syntenic gene groups in polyploid species; each group retained a single copy of 3 or 4 homologous chromosomes in polyploid species and one copy in Mg. We conducted phylogenetic analysis of each complete syntenic gene group from each triploid nematode with the orthologue from Mg as the outgroup. The dominant topologies of these phylogenetic trees suggested an AAB genome pattern in Mi and Ma 3n (Fig. 1b and Supplementary Fig. 12a, b). The average nucleotide identity between two similar copies was approximately 95%, while it was approximately 90% between two similar copies and the divergent copy (Supplementary Fig. 12c). We thus defined the divergent copy as B. As there was no way to distinguish the two similar copies, we randomly defined them as $A_1$ and $A_2$. Further comparison between Mi and Ma 3n revealed that the genome structure of both triploids was AAB rather than one AAB and one ABB

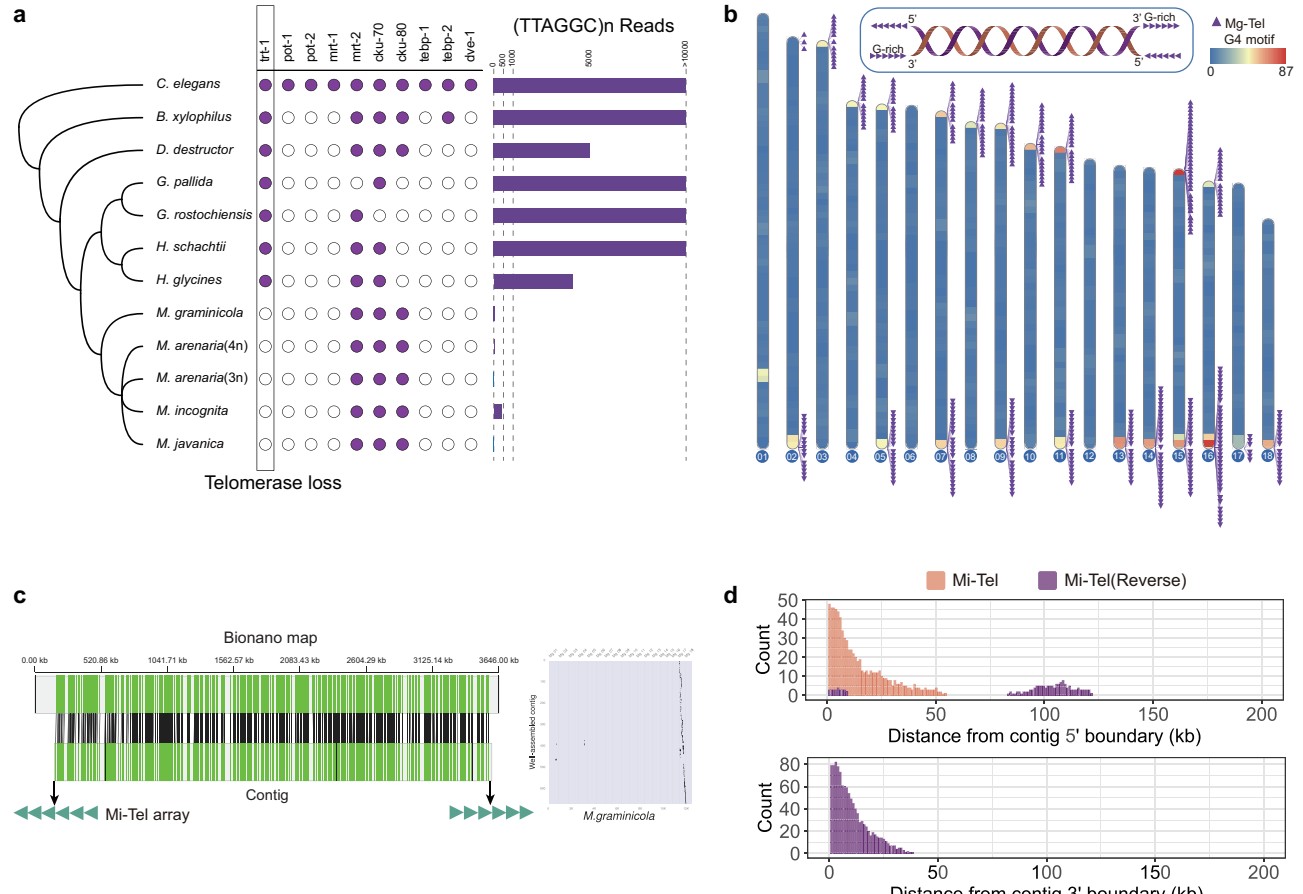

**Fig. 2 | Identification of telomeric repeats in *Meloidogyne*. a** Conservation of typical telomere-associated genes and typical telomeric repeats in Clade IV nematodes (defined by Blaxter et al.). **b** Distribution of Mg-Tel on the Mg chromosome. The position of Mg-Tel is related to G4-motif density. **c** One contig at the approximate chromosome level (tig00000250_np512512) verified by the BioNano map and synteny analysis. The green triangles at the bottom indicate Mi-Tel elements distributed at both ends of the contigs in opposite directions. **d** Distribution of Mi-Tel elements in Mi contigs. Only contigs larger than 200 kb are shown. Source data are provided as a Source Data file.

(Supplementary Fig. 13). For tetraploids, among the 15 putative phylogenetic tree topologies, those with 2:2 branches dominated, suggesting that the polyploidy pattern was AABB (Supplementary Fig. 14). We observed that one branch of tetraploids displayed a close relationship with the A subgenome of triploid species, while the other branch was similar to the B subgenome of triploid species (Fig. 1d). We finally assigned the scaffolds of tetraploids into four subgenomes based on the triploid subgenome information. In total, 230 scaffolds of polyploid species were grouped into 14 subgenomes (Supplementary Data 10). By calculating the global synonymous substitution rate ($Ks$) and synonymous third codon transversion rate (4DTv), the 14 subgenomes can be subgrouped into two obvious lineages (Fig. 1e and Supplementary Fig. 15). These data revealed that the genome structures of polyploid species were $A_1A_2B_1$ (Mi and Ma 3n) and $A_1A_2B_1B_2$ (Ma 4n and Mj).

**Identification of putative novel telomeric repeats in Meloidogyne genomes**

Clade I RKNs are characterized by a dynamic karyotype with chromosome numbers less than the expected number based on $n = 18$ as ancestral number of chromosomes, ($3n = 54$, $4n = 72$)[17], indicating the presence of chromosomal fusion events or chromosome losses. Chromosomal fusion has been reported to be related to telomere dysfunction[35]. We examined telomere-associated genes[36] (Supplementary Data 11) and typical telomeric repeats in the genomes of nematodes belonging to Clade IV[37]. We found that most nematodes,

except those belonging to the *Meloidogyne* genus, contained telomerase and typical telomeric repeats (Fig. 2a), suggesting that the ancestors of Clade IV nematodes had a telomerase-dependent telomere lengthening mechanism, which was lost during the evolution of the *Meloidogyne* genus.

In some organisms without telomerase, telomeres can be maintained by alternative lengthening of telomeres (ALT) mechanisms[38] such as those in *Drosophila*[39] and *Anopheles*[40]. We analyzed the repetitive sequences at both ends of each scaffold of all 5 RKN genomes. In the Mg genome, we identified a G-rich repetitive element (Mg-Tel) specifically located in scaffold ends (Fig. 2b and Supplementary Data 12). Mg-Tel had similar characteristics to typical telomeric repeats, such as a specific direction from 5' to 3' and putative G-quadruplex[41] enrichment, suggesting that Mg-Tel might be the alternative telomeric repeat in Mg.

In polyploid RKNs genomes, neither in contigs nor in scaffolds, we failed to detect similar elements located exclusively at the ends. The reason for this might lie in contig misanchoring in polyploid RKNs assemblies or chromosomal fusion events. Considering this, we tried to identify putative telomeric repeats from those well-assembled contigs. We screened one contig that almost reach chromosome level according to BioNano maps (Supplementary Data 13) and the collinearity with Mg, and identified a repetitive element from both ends of this contig (Mi-Tel), which was arranged in tandem with opposite direction at both ends (Fig. 2c). The repetitive sequences of Mi telomeres are highly complex. However, we have identified a conserved

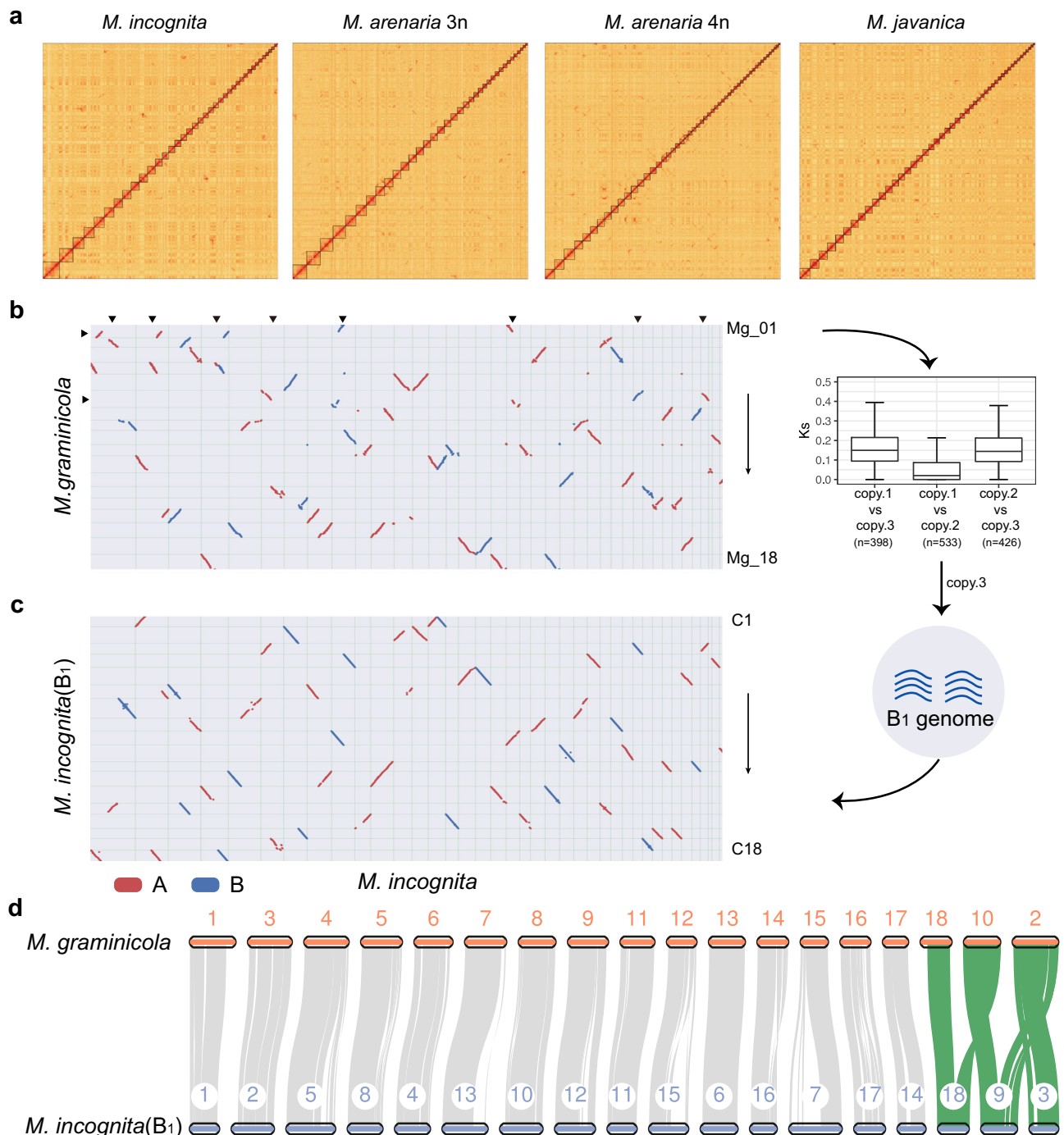

**Fig. 3 | Chromosome-level genomes of four polyploid RKNs. a** Genome-wide Hi-C heatmap for four polyploid *Meloidogyne* species. **b** Genomic synteny between *M. incognita* (v2) and *M. graminicola* (T2T). Black triangles represent some chromosomal variation between them. The B₁ subgenome was identified by the boxplot of *Ks* values for three homologous chromosomes. In box plots, the central line represents the median, the box represents the 25% and 75% percentiles, and the whiskers represent 1.5 times the interquartile range beyond the box. **c** Genomic

synteny between *M. incognita* (v2) and the B₁ subgenome of *M. incognita*. This analysis clearly shows the chromosomal fusion in *M. incognita*. **d** Chromosomal variation between the *M. graminicola* genome (T2T) and the B₁ subgenome of *M. incognita*. Gray lines indicate collinearity between two genomes, while green represent chromosomal rearrangement events. Source data are provided as a Source Data file.

66 bp motif sequence, referred to as a 'telomere element,' which we used to search for Mi-Tel (Supplementary Fig. 16). Further analysis of all the Mi contigs revealed that Mi-Tel was specifically located at the 5' terminus of the contig, with the reversed Mi-Tel elements at the 3' terminus (Fig. 2d).

To further validate whether Mi-Tel repeats have a terminal position on chromosome, we conducted fluorescence in situ hybridization (FISH) experiments. The results indicate that Mi-Tel repeats indeed

have a terminal position on *M. incognita* chromosomes (Supplementary Fig. 17), which is consistent with another study on *Meloidogyne* telomeres[27]. To investigate whether telomeres are involved in chromosomal fusion[27], we identified 4 putative chromosomal fusion events in assembly v1 (Supplementary Fig. 18a–c). To target these gaps in putative fusion points, we generated an additional library for Oxford Nanopore (ONT) long reads (Supplementary Data 14). When mapping ONT long reads to these sequences, we screened one 102-kb read

covering the ~70-kb gap between two scaffolds of Mi and found that this read contained 71 Mi-Tel elements and 93 reversed complementary Mi-Tel at the putative fusion point (Supplementary Fig. 18d–f). The average telomeric element copy number at the end of contig is 26 (Supplementary Fig. 19), indicating that fusion is not caused by telomere shortening. In addition, some short contigs also contained symmetrically distributed Mi-Tel, which might be the fragment of the fusion point (Supplementary Fig. 20). These results suggest that there might be some cases of telomere-to-telomere fusions in *M. incognita*.

We further explored whether Mi-Tel was conserved in other members of Clade I RKNs[18] and identified similar repetitive elements on contig boundaries from Ma 3n, Ma 4n, and Mj and from *M. floridensis* (Mf)[21,24], *M. luci*[42] and *M. enterolobii* (Me)[43] but not from *M. hapla* (Mh)[44] and other more distant *Meloidogyne* species (Supplementary Fig. 21). We extracted 9318 telomeric elements from four polyploid species, which exhibited differences in length (Supplementary Fig. 22a). These telomeric elements were further grouped into 823 clusters based on sequence similarity (193 clusters contain more than 10 telomeric elements, Supplementary Fig. 22b), which suggested that they vary widely. We found that some clusters were present in four species, while some were specifically present in Mi but absent from the rest (Supplementary Fig. 22c). Moreover, we observed that even for a particular contig, there was more than one cluster of telomeric elements (Supplementary Fig. 22d). We speculated that the presence of different telomeric elements on the same contig may be a result of recombination between different chromosomes, which was in line with the previous report of ALT maintenance by the recombination-associated pathway[45]. Considering that individuals without telomerase in *C. elegans* could maintain a stable karyotype after ALT establishment[46], we speculated that these alternative telomeric repeats might play an important role in karyotype stability during reproduction of the *Meloidogyne* genus.

## Construction of chromosome-level genomes for polyploid RKNs

Based on the information gained from assembly v1 concerning the duplicated genome structures and the putative telomeric repeats, we attempted to construct chromosome-level genomes for polyploid RKNs. By combining Hi-C data and BioNano maps (Supplementary Fig. 23), we constructed 4 chromosome-level genomes for polyploid RKNs, obtaining a total of 36-52 chromosomes (named assembly v2, Fig. 3a and Supplementary Figs. 24–27; Supplementary Data 15 and 16). In contrast to assembly v1, this assembly v2 was not guided by synteny relationships with the Mg genome but only Hi-C and BioNano data, and were utilized for all subsequent analyzes. Additionally, we corrected the Mg genome and formed a new version (named Mg_T2T, Supplementary Data 17), leaving each chromosome containing at least one detected telomeric array and a high collinearity with the related species *M. chitwoodi*[47] (Supplementary Fig. 28). The assembly v2 and the Mg T2T genome were utilized for all subsequent analyses.

We reannotated the genome and investigated the distribution of genomic elements among chromosomes, including coding genes, transposon elements (TEs), centromeres[48], and chromatin status[49] (Supplementary Fig. 29 and Supplementary Data 15). We found that chromosome arms contain more repetitive sequences (159 vs. 135 TEs per 100 kb for Mi, Mann–Whitney $P < 0.001$) and fewer genes (19 vs. 22 genes per 100 kb for Mi, Mann–Whitney $P < 0.001$) than chromosome centers. For chromatin status, the heterochromatin marker H3K9me3 was enriched in centromeric regions, as described in *C. elegans*[50] (Supplementary Fig. 30a, Mann–Whitney $P < 0.01$). Another marker, H3K4me3, which is the most enriched mark in Mi[49], was found to be enriched in subtelomeric region (Supplementary Fig. 30b, Mann–Whitney $P < 0.01$), contrary to those described in *C. elegans*. We focused on some genes that may contribute to fitness, including secreted protein genes[51] and horizontal gene transfer (HGT) genes[52],

which are sources of effectors and may play a role in interactions with plant hosts. We predicted 4572–6137 secreted proteins and 509–711 HGT candidates (Supplementary Data 18) and found that both types of genes are scattered among chromosomes and that there are some hotspots for HGT candidates, as reported in *A. vaga*[53] (Supplementary Fig. 31).

Our assembly v1 suggested that current chromosomes are derived from the fusion of multiple ancestral chromosomes. To reveal the genome composition of polyploid species, we first reconducted synteny analysis between the Mi and Mg genomes (Fig. 3b). We found that in addition to extensive chromosomal fusions in Mi, there were some chromosomal variations between Mi and Mg, indicating that Mg could not represent the ancestral genome of polyploid RKNs very well. The three sets of homologous chromosomes for Mi maintained a high degree of collinearity at the chromosome scale. To divide the assembly v2 into subgenomes and reconstitute the ancestral genome, we attempted to find some independent chromosomes without fusion or chromosome segments in the Mi genome as ancestral sequences. Since the $Ks$ value between A and B is significantly larger than that between the two A copies, the entire $B_1$ subgenome of Mi could be determined (Fig. 3b and Supplementary Data 19). Compared to that in the alignment of polyploid species against Mg, the continuity of the synteny block was strikingly increased in the alignment of polyploid species against the $B_1$ subgenome (Fig. 3c and Supplementary Fig. 32). We thus used the $B_1$ subgenome as the haploid reference for polyploid species. Furthermore, all the assembled chromosomes belonging to polyploid species were split into 18 chromosome sets (C1–C18) based on genomic synteny information (Supplementary Fig. 33; Supplementary Data 20–22). Given that Mi $B_1$ could represent ancestral genome, we revealed three chromosomal variations between the ancestor of Clade I RKNs and Mg (Fig. 3d).

To eventually determine the subgenome, we constructed the $Ks$ matrix for each chromosome set and calculated the average nucleotide identity between them (Supplementary Figs. 34 and 35). These datasets, in combination with our abovementioned analysis results in assembly v1, were employed to accurately distinguish different subgenomes (Supplementary Data 23 and 24). Again, we could not distinguish the sequence of $A_1$ from that of $A_2$ and thus assigned them randomly. Compared with assembly v1, assembly v2 exhibited stronger collinearity between subgenomes (Supplementary Fig. 33).

## Origin and evolution of polyploid RKNs

The divided subgenomes provide an opportunity to infer the origin and evolutionary history of polyploid RKNs. We examined 77-454 single-copy syntenic gene groups from 18 chromosome sets and constructed phylogenetic trees for each chromosome set (Supplementary Fig. 36). Overall, 17 out of the 18 phylogenetic trees obtained exhibited a topology with two lineages, the A lineage and the B lineage (Fig. 4a, b).

Several hypotheses can explain the formation of these allopolyploids. By combining these topologies, we first considered the formation of triploid species. The most likely model is that triploid species originate from hybridization between the $A_1A_2$ gamete and $B_1$ gamete (Supplementary Fig. 37a). Given that species producing unreduced gametes are common in RKNs, we raised two hypotheses, that is, $A_1A_2$ is from an unreduced gamete of diploid species with clonal reproduction or hybridization between $A_1$ and $A_2$. The A genome of Mi was separated from that of other species, indicating that the A subgenome of Mi have a different evolutionary history from that of other species. These A genomes might have originated from the same diploid ancestor (Supplementary Fig. 37b, scenario 1 and 2), but we noticed that the topology of the A lineage did not match this hypothesis (Supplementary Fig. 37b). A possible explanation was that some mechanisms could cause the loss of heterozygosity, such as gene conversion[54,55] (Supplementary Fig. 37c). Another possible explanation

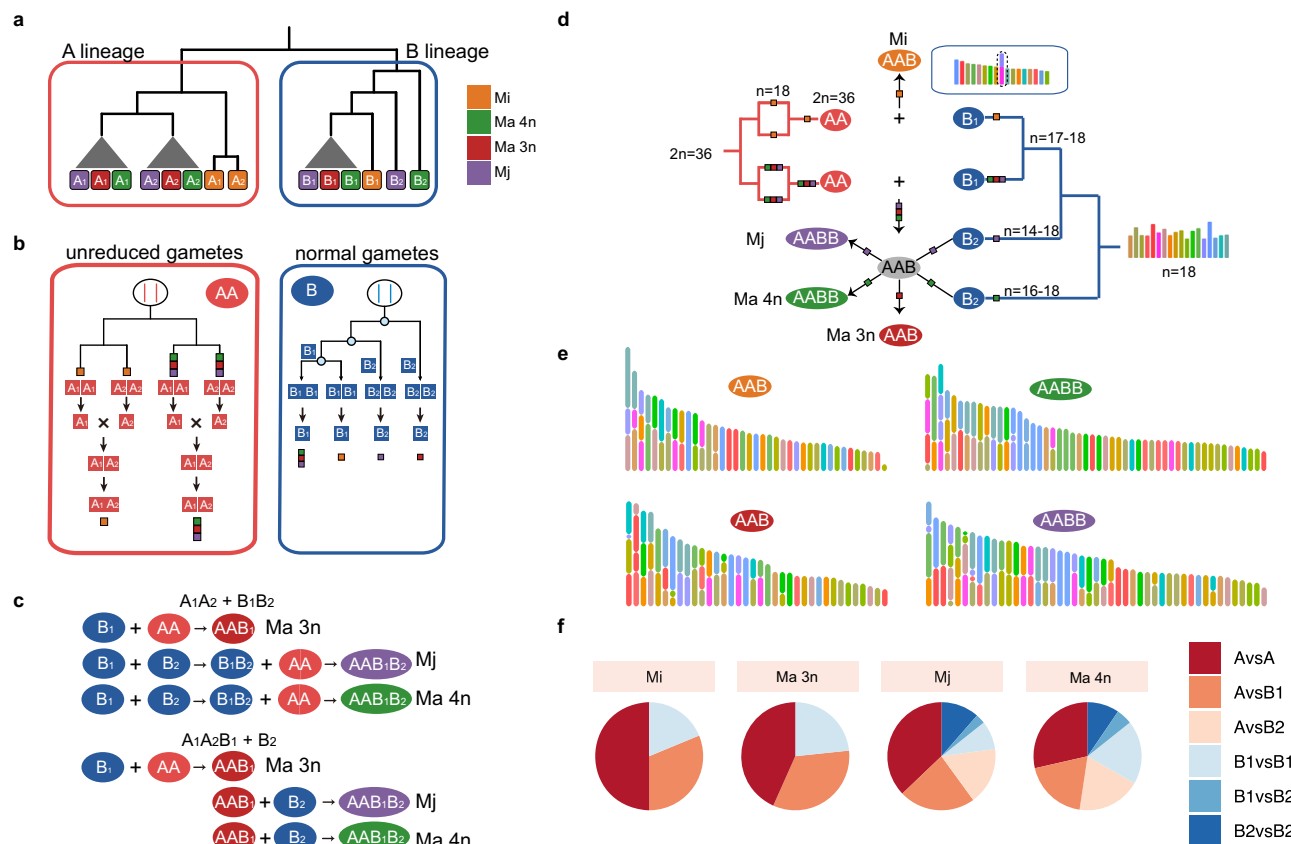

**Fig. 4 | Reticulate origin of polyploid *Meloidogyne*. a** Topological structure of the phylogenetic tree for the A and B lineages. **b** Hypotheses for the origin of the AA subgenome and B subgenome. The most likely hypotheses are that AA gametes derive from unreduced gametes of diploid hybrid species. The B subgenome derives from normal gametes of meiosis. **c** Two hypotheses of tetraploid species origin. **d** Hybridization origin model and karyotype evolution of polyploid species. **e** Reconstructed karyotype of polyploid species. Each color represents one corresponding chromosome in the ancestor ($n = 18$, C1–C18). **f** Proportion of chromosomal fusion between any two subgenomes. Source data are provided as a Source Data file.

was that the $A_1A_2$ of Mi and Ma 3n originated from two independent hybridization events (Supplementary Fig. 37b, scenario 3). Compared to extensive gene conversion, we propose that multiple hybridization events are more likely due to the higher occurrence of hybridization in RKNs. Furthermore, these hybridization events can account for the observed 5% heterozygosity in $A_1$ and $A_2$.

Second, we considered the formation of tetraploid species and found a high similarity in the $A_1A_2B_1$ subgenome (*Ks* values are approximately 0, and nucleotide identity is approximately 100%, Supplementary Figs. 34 and 35) between Ma 3n, Ma 4n and Mj. Meanwhile, the mitochondrial tree of those showed a trifurcating structure (Supplementary Fig. 2b), and no dominated bifurcating tree of those were found for nuclear genome ($A_1A_2B_1$). We hypothesized that the origin pattern of tetraploids might be $A_1A_2B_1 + B_2$ or $A_1A_2 + B_1B_2$ (Fig. 4c). The former hypothesis was more likely because it required 3 hybridization events to form Ma 3n, Ma 4n, and Mj, while the latter hypothesis required 5 hybridization events (Fig. 4d). This also suggests that Ma 3n, Ma 4n and Mj may share one common triploid ancestor ($A_1A_2B_1$). We further combined our hypothesis of species formation with mitochondrial phylogeny and noticed very low divergence (Supplementary Fig. 2, -0.29%) between polyploid species in this study with closely related species *M. luci*[42] and *M. floridensis*, suggesting that the maternal genomes of these species diverged not long ago and that hybridization occurred recently, which is exactly consistent with nuclear and mitochondrial phylogenomic analyses made on previous more fragmented versions of these genomes[22].

We attempted to elucidate the genome structure of these two closely related species by mapping corresponding reads onto the A or B subgenome. By testing our data, this strategy can roughly determine the genome composition (Supplementary Fig. 38a, b). Approximately 86% of Mf reads were aligned to the AA subgenome, indicating that Mf is composed of the A subgenome (Supplementary Fig. 38a). However, there is still a debate about whether Mf is triploid (AAA)[32] or diploid (AA)[21,56]. We checked the read depth ratio of the alternative allele in heterozygous SNPs of Mf and found a peak at 0.33 (1/3), which supported that Mf is a triploid (Supplementary Fig. 39). Similar analysis of read mapping supported that *M. luci* is a triploid of AAB (Supplementary Fig. 38c), consistent with previous study[42]; Ma samples in which the Ma 3n branch is located are triploid, while other Ma samples are tetraploid (Supplementary Fig. 40).

Given that polyploid RKNs are derived from recent hybridization, we further investigated the dynamic changes in transposable elements during evolution. The TEs landscape suggested that most TEs were old in polyploid RKN genomes while both diploid and polyploid species harbor active TE copies (Supplementary Fig. 41a). We also identified 3251 full-length long terminal repeat retrotransposons (LTR-RTs) in our five genomes (a total of 121,684 LTRs with most of them not having a complete structure), 2578 of which exhibited 100% identity within the two long terminal repeat (Supplementary Fig. 41b), indicated that some TEs (2.12% against total) are still active[57]. It has been hypothesis that the epigenetic changes after hybridization may relax the repression of TEs, leading to TEs burst[58]. However, we did not observe a burst of many TEs after recent hybridization, instead, only a few TEs were recent, and the majority of TEs were old (Supplementary Fig. 41a). These results showed that the increase of TE content in polyploid should be attributed to the TE accumulation in the common ancestor

of A and B lineage rather than the burst after hybridization. Most of the TE families could be detected in both A and B subgenomes, and no A- or B-specific TE families were found (Supplementary Fig. 41c, d), contrary to those in *X. laevis*[59]. We also investigated HGT candidates during evolution and found that at least 82% of HGTs had syntenic genes in the A or B subgenome, indicating that those HGTs were gained in a diploid ancestor (Supplementary Fig. 42).

Finally, we constructed the karyotype evolution during the hybridization process (Fig. 4d, e). In total, the current chromosomes might have undergone at least 15–35 chromosomal fusion events, of which 50% of those fusions were observed at the contig level (Supplementary Fig. 43 and Supplementary Data 25). We noticed that 24 fusions involved telomere fusion, of which 21 fusion events involved at least one telomere with more than 5 Mi-Tel elements (Supplementary Data 25). Among all the fusions, 5–14 chromosomal fusion events occurred between the A and B subgenomes (Fig. 4f), and 1 chromosomal fusion event C6-C14 ($B_1$-$B_1$) were shared in the four studied Clade I RKNs, indicating that this fusion event occurred in the ancestral stage. Given this, the chromosome number might be $n = 17$ in the $B_1$ ancestor. In addition, 4 and 2 fusion events were observed in the $B_2$ subgenomes of Ma and Mj, respectively, indicating that the chromosome number of the $B_2$ ancestor might be $n = 14$–18 in Mj and $n = 16$–18 in Ma, respectively (Fig. 4d and Supplementary Fig. 32). Obviously, fusion events between A and B or between $B_1$ and $B_2$ occurred after hybridization, suggesting that most chromosomal fusion events occurred during the post-hybrid evolution of Clade I RKNs.

## Gene expression landscape after polyploidization of Clade I RKNs

Since effector proteins play a key role in RKNs infestation of plants, such as entering plants, resisting plant immunity, establishing and maintaining feeding sites, etc. The host spectrum and infection ability of RKNs are often related to the type and quantity of the effector it secretes[60]. Therefore, we focused on 4572–6137 proteins predicted to be secreted (including putative effectors) to explore the contribution of hybridization to fitness. We found that secreted protein genes had a significantly higher level of sequence divergence between A and B subgenomes, compared with those of total genes without secreted protein genes (Mann–Whitney $P < 0.001$, Supplementary Fig. 44), indicating that hybridization might improve the diversity of secreted proteins and may contribute to expanding the host spectrum of polyploid RKNs.

To investigate the landscape of gene expression among subgenomes in different developmental stages, we focused on the complete syntenic gene groups (6764–7327 triad or tetrad genes) and strictly filtered them (Supplementary Data 26 and 27). We identified 7 and 15 expression patterns in triploids and tetraploids, respectively, using a previously reported method[61] (Fig. 5a). Approximately 80–92% of the gene groups showed a balanced pattern during the 5 life stages of four polyploid RKNs (Supplementary Data 28). Among the non-balanced patterns, the single-subgenome suppressed patterns were the most frequent (72–90%, Supplementary Data 27). The dominant pattern resulted from decrease of homoeologous gene expression rather than increased expression of the dominant gene, which was consistent with previous findings in wheat[61]. We further wondered whether the genes with a suppressed pattern were also suppressed in other species. We thus examined the normalized expression level in complete syntenic groups across species and found that the suppressed pattern was not conserved within triploids or tetraploids (Fig. 5b), indicating that the differences in homologous gene expression were most likely to occur after polyploidization.

## Discussion

In this study, we obtained 5 chromosome-level genomes, including 4 unzipped genomes of complex allopolyploid RKNs and 1 haploid genome of Mg, through multiple strategies. These genomes reveal a complex structure for those species characterized by parthenogenesis, polyploidy, and karyotypic instability. Consistent with a previous survey on chromosome number[17], we assembled 18 chromosomes for Mg and reconstructed 18 ancestral chromosomes for polyploid species; we also elucidated the chromosomal variation between them. Polyploid species have final assembled chromosome numbers ranging from 36 to 52, not multiples of 18, which supports them being derived from extensive chromosomal fusion in complete triploids or tetraploids, instead of aneuploidy[17]. Previous work also pointed out that chromosome numbers are unstable even within the same species[62], suggesting that different individuals have different chromosomal fusion patterns. Given that the polyploid RKN genomes is highly dynamic, there may be differences in reference genomes from different samples, and revealing such variation may require the construction of more reference genomes. It should be noted that the assembly of a few chromosomes is still hypothetical, and some chromosomal fusion events are only observed at the scaffold level, requiring further investigation for validation. Nonetheless, the genome assembly generated in this study is the most accurate version currently available, providing a solid foundation for subsequent studies of genetic variation and karyotype instability at the population level.

Chromosomal-level analysis identified unusual telomeres in *Meloidogyne* species. Telomeres prevent chromosomal fusion by inhibiting DNA damage response pathways[63]. We found that telomeric repeats were conserved only in closely related species, indicating their rapid evolutionary rate. Even within the same species, telomeric repeats exhibited heterogeneity in both length and sequence, the complexity observed in telomeres of polyploid RKNs is consistent with the complexity of ALT telomere sequences in human tumors. Telomeres are involved in some chromosomal fusions, but these fusions are not caused by telomere shortening. In addition to telomeric repeats, telomere-associated genes, such as shelterin components[36], are not conserved. Further work is needed to determine the specific mechanisms of chromosome protection and the genetic feature of these unusual telomeres. Moreover, ALT may influence the distribution pattern of histone modification[64]. In organisms lacking telomerase, telomeres are lengthened using ALT, a recombination-based mechanism that is initiated by DNA damage in the telomere[45,65]. ALT telomeres have a weak chromosomal protective function[66], which seems to explain the occurrence of a few telomere-to-telomere fusion events. However, both diploid and polyploid species possessed ALT, but only polyploid species exhibited extensive chromosomal fusion. Previous studies reported gene conversion[24], gene loss[67], and fragmented TE[68] in polyploid species, which was related to the DNA repair response. Recent studies have documented that the shortage of replication-associated proteins after whole-genome duplication is the reason for extensive DNA damage and karyotype instability[69]. Therefore, we speculated that polyploid RKNs harbor higher levels of DNA damage, promoting the occurrence of chromosomal fusions. We noticed that most of the full-length LTRs were recently inserted, which may be because old LTRs were affected by genome instability, resulting in the loss of their intact structure and thus making them undetectable. Additionally, mitotic asexual reproduction and holocentromeres also contributes to the successful inheritance of fused chromosomes[70].

Moreover, the genome assemblies we have generated accurately splits the A and B subgenomes, which are fully unzipped. This information helps us understand how complex RKNs arise. The published draft genomes are partially unzipped[21,22,24,26], which helped determine that there are two divergent copies, A and B; however, it is difficult to distinguish the sequence differences between $A_1$ and $A_2$ or between $B_1$ and $B_2$. In fact, varying degrees of sequence collapse led to a smaller genome size for those studies, and limited the elucidation of the reticular origin of Clade I RKNs. Using a fully unzipped sequence of

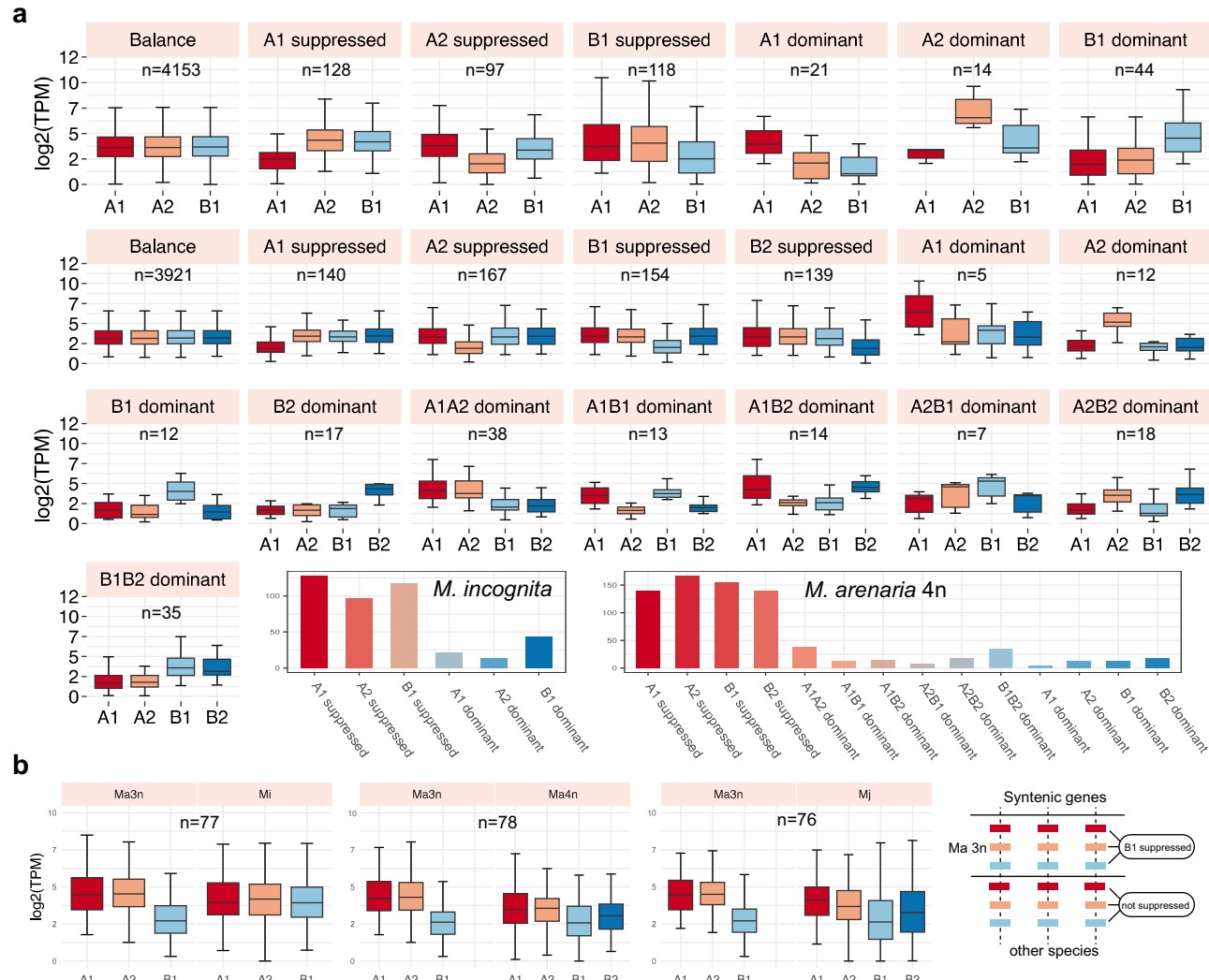

**Fig. 5 | Divergence of homologous gene expression across subgenomes.**
**a** Boxplot of TPM values across expression patterns in triploid *M. incognita* (top 7 boxplots) and tetraploid *M. arenaria* 4n (other boxplots) in the J2 stage. The histogram shows the gene group number in each expression pattern. **b** Boxplot of TPM values for syntenic genes between Ma 3n and other species. When the syntenic group in Ma 3n shows a B1 suppressed pattern, those in other species do not show a similar pattern, indicating that the expression pattern is not conserved between the species. In box plots, the central line represents the median, the box represents the 25% and 75% percentiles, and the whiskers represent 1.5 times the interquartile range beyond the box. Source data are provided as a Source Data file.

14 subgenomes, we proposed that triploid nematodes are formed by hybridization between unreduced gametes $A_1A_2$ and the haploid gamete $B_1$, while tetraploids are formed by hybridization between unreduced gametes $A_1A_2B_1$ and the haploid gamete $B_2$. The unreduced gametes $A_1A_2$ may originate from hybrid diploid species with a heterozygosity of approximately 5%, whereas all B subgenomes are derived from meiosis of a diploid species. Our hypothesis for the origin of RKN is consistent with one of the previously proposed hypotheses[19], and we speculate that one of the ancestral species for the hybrid origin of RKN is likely to be a facultative parthenogenetic species with a narrow host range that may still exist today. Although the hybridization of unreduced gametes and triploid bridges is well known in plants, they are rare in animals. We noticed that the hypothesis of hybridization between unreduced and haploid gametes is not only proposed for polyploid RKNs but also for *Otiorhynchus scaber* weevils[9], suggesting that this hypothesis may apply to the hybrid origin process of other parthenogenetic polyploid insects.

The A or B subgenomes are also useful for understanding the genome structure of another Clade I RKNs. Using read mapping, we determined that there are two current lineages in Ma, with one triploid lineage and one tetraploid lineage, indicating that the current definition of Ma is vague and that these two lineages need to be clarified and renamed. We also confirmed that *M. luci* is a triploid with an AAB genome structure[42] and that Mf is a triploid with an AAA genome structure. Several studies have generally regarded Mf as a facultative parthenogenetic diploid species[21,56], and our study suggested that Mf, at least two samples that have been sequenced, is a triploid species. Polyploid species are generally formed by the hybridization of ancestor A and ancestor B, while Mf is only composed of the A genome. Given the importance of Mf, an unzipped genome for Mf is needed to resolve this contradiction. The genome structure of AAA suggested that Mf originated from AA + A, which indicates that AA ancestors can produce both unreduced and normal gametes or that there might be multiple closely related AA species with different modes of reproduction, further broadening our understanding of polyploid nematodes. The mechanism of RKN reproduction and gametogenesis may require more diploid RKN genome sequencing and more research on cytological genetics.

The splitting of the A and B subgenomes allowed us to determine the specific genome changes that occurred in the common ancestor, in the A or B lineage, or after hybridization, which is critical for understanding the high fitness of those polyploid nematodes. Although we

observed specific distribution patterns in polyploid nematodes, such as histone modification distribution, HGT hotspots and increased TE content, it is not clear whether these changes are caused by lifestyle changes, polyploidization or parthenogenesis. Further research is needed to more closely identify diploid species, especially species that could represent the A or B ancestor. In conclusion, our work provides a genomic resource for understanding the evolution of apomictic species, polyploidy formation, and the development of pest control methods.

## Methods

### Nematode materials and species identification

Mi, Mj, and Mg were collected from farmlands in Wuhan city of Hubei province, Longyan city of Fujian province, and Changsha city of Hunan Provinces, respectively. Two Ma samples were collected from farmlands in Shenyang city of Liaoning province and Shiping city of Yunnan province. The host of Mi, Liaoning Ma 4n, Mj, Yunnan Ma 3n, and Mg collect from is tomato, tomato, okra, cabbage, and rice, respectively.

SCAR-PCR were performed to determine the species identity[71]. The nematode sample Mi, Mj, and Ma from Liaoning show a clear specific SCAR PCR band, while the nematode from Yunnan did not display a clear band at first Mi/Ma/Mj/Mh-SCAR-PCR. We sequenced and assembled the genomes of five nematodes and found that the Yunnan sample show a closet relationship with Ma, and the Liaoning sample was tetraploid and the Yunnan sample was triploid. We later identified the Ma-SCAR marker in sequencing data through virtual PCR for the Yunnan sample. It is unclear what underlies the false-negative PCR result for the absence of amplified bands in Ma-SCAR PCR of Yunnan Ma, but it could be a PCR error. Furthermore, another triploid Ma collected by us had obvious characteristic bands after Ma-SCAR PCR. For virtual PCR, we also identified the location of Ma-SCAR marker from published triploid Ma genomes (GCA_003133805.1) by using MUMmer (v4.0.0)[72]. Illumina reads were aligned into the Ma genome, and the read coverage of the Ma-SCAR region were visualized using Jbrowser to determine whether Ma-SCAR marker present or absent in samples. Thus, we named the Liaoning sample Ma 4n, and named the Yunnan sample Ma 3n.

To obtain a pure single nematode lineage for DNA sequencing, we separated a single egg mass from each species and inoculated it into Rutgers tomato (Mi, Ma and Mj) or rice (Mg) in greenhouse. After 1-4 generations of single egg mass purification, the nematodes were used for subsequent experiments.

### Nematode collection

Five stages of RKNs were used in this study. For egg collection, the nematode-infected tomato or tobacco roots were cut into 0.5 cm pieces, placed into a 500 ml beaker with 225 ml ddH$_2$O and 25 ml sodium hypochlorite added, and placed on a magnetic stirrer for 8 min at 1000 rpm. The suspension was successively passed through 18-, 60-, 100-, 200- and 500-mesh sieves and rinsed five times with water to remove sodium hypochlorite, and the eggs were retained on a 500-mesh sieve. After sucrose gradient centrifugation with a final concentration of 35%, the sucrose was passed through a 3 μm filter membrane using a negative pressure suction filter, and the eggs retained on the 3 μm filter membrane were collected for the next experiment. For J2 collection, the purified eggs were transferred to 500 mesh sieves and placed into a dish with water covering the face of the mesh. After 3−5 days of hatching, J2 could be collected in the water. For J3J4 collection, the roots treated with sodium hypochlorite to collect the egg were put into a mortar and a pestle was used to smash the roots lightly; the roots were suspended in 100 ml of water and passed through 18-, 60-, 100-, and 200-mesh sieves in sequence. J3J4 was passed through a 200-mesh sieve. After most plant debris was removed by sucrose centrifugation, J3J4 were manually picked under a microscope. For early female (female before laying eggs) collection, we collected plants

18 days after nematode infestation and picked out early females in root knots that had not produced egg masses with tweezers. For female collection, we removed the egg masses at the root knots and manually picked out the egg-laying females in the roots with tweezers.

### DNA/RNA library construction and sequencing

Approximately 100 μl eggs were purified and collected from the infected tomato to extract DNA for each nematode lineage. Genomic DNA was isolated using the CTAB method as described previously. In brief, eggs were suspended in 200 μl SDS-EB lysis buffer (50 mM Tris-HCl pH 8.0, 200 mM NaCl, 20 mM EDTA, 2% SDS, 1 mg/ml proteinase K) and transferred to liquid nitrogen. After 3 homogenizations with a pestle, the powder was collected in a prechilled 1.5 ml tube, and 300 μl of SDS-EB lysis buffer was added; then, 500 μl of 65 °C preheated CTAB buffer (100 mM Tris-HCl pH 8.0, 20 mM EDTA, 1.4 M NaCl, 2% CTAB, 1% PVP 40000) was added to the mixture and incubated at 65 °C for 30 min. Next, a phenol–chloroform-isoamyl alcohol mix (25:24:1) was used to separate the DNA and protein. After RNase A treatment, 0.75 M ammonium acetate was added, and DNA was sedimented in precooled isopropanol (0.9:1, v/v) and finally dissolved in 50–100 μl DNase-free water. For PacBio long-read sequencing, the integrity of DNA was determined with an Agilent 4200 Bioanalyzer (Agilent Technologies, Palo Alto, California), and 8 μg of genomic DNA was sheared using g-Tubes (Covaris) and concentrated with AMPure magnetic beads. The libraries (30 kb) were constructed following the protocol released from PacBio. For five nematodes, a total of one to three SMRT cells were sequenced on the PacBio Sequel I platform, generating 15−35 Gb of subreads, depending on the genome size. In addition, whole-genome shotgun resequencing was performed on the Illumina Nova-Seq platform with a 300−500 bp insert size and 150 bp paired-end read length, and 9−22 Gb reads were generated.

Five developmental stages of nematodes were selected to extract the RNA for each species. RNA extraction was performed using the TransZol Up Plus RNA Kit (TransGen Biotech, Beijinng, China), and the RNA Nano 6000 Assay Kit for the Bioanalyzer 2100 system (Agilent Technologies, CA, USA) was used to assess the RNA integrity and concentration. The RNA library preparation was performed using 2 μg of RNA per sample, and the NEBNext® Ultra™ RNA Library Prep Kit for Illumina® (#E7530L, NEB, USA) was used to generate the sequencing libraries following the manufacturer's recommendations. After library examination and cluster generation, 6 Gb of 150 bp paired-end reads were generated for each library and sequenced on an Illumina platform. Three biological replicates are generated for each sample.

### Hi-C library construction and sequencing

The Hi-C library was constructed according to a previously described method with appropriate modifications[73,74]. For each species, approximately 50 thousand newly collected eggs were pelleted (5000 rpm, 4 °C for 30 sec), resuspended in 500 μl of precooled PBS, and then cross-linked in a 1.5% final concentration (v/v) formaldehyde (Sigma) for 30 min at room temperature with rotation (15 rpm). A final concentration of 0.2 M glycine was added to quench the reaction for 5 min at room temperature with 15 rpm rotation. Eggs were washed by pelleting three times (2400 × $g$, 4 °C for 30 s) and resuspended in 1 mL of precooled PBS. After this, eggs were resuspended in 200 μl prechilled 2× nuclei purification buffer (20 mM HEPES pH 7.6, 20 mM KCl, 3 mM MgCl$_2$, 2 mM EGTA, 0.5 M sucrose, 0.05% Triton, 1 mM DTT, 0.05 M NaF, 40 mM β-glycerophosphate, 2 mM Na$_3$VO$_4$) with 1 μl protease inhibitor (Roche). The eggs were transferred to a prechilled Wheaton stainless-steel homogenizer (Wheaton catalog number 357572) and homogenized to collect nuclei. The supernatant was transferred to a new tube with 50 μl buffer. To acquire more nuclei, 200 μl of precooled 2× nuclei purification buffer was used to resuspend the remaining pellet, which was then homogenized again. A total of 5 homogenizations were performed for each sample. Finally, the

supernatant nuclei were centrifuged at $1000 \times g$ for 10 min. The pellets were resuspended in 500 μl precooled Hi-C lysis buffer (10 mM Tris-HCl pH 7.5, 10 mM NaCl, 0.2% NP-40, 1× Roche protease inhibitors) with 15 rpm rotation for 30 min at 4 °C. After centrifugation and removal of the supernatants, the pellets were washed one time with another 500 μl of precooled Hi-C lysis buffer, and 100 μl of 0.5% SDS buffer was used to resuspend the pellet, which was incubated at 62 °C for 10 min. Next, 285 μl of ddH₂O and 50 μl of 10% Triton X-100 were added to the mixture and incubated for 15 min at 37 °C.

For restriction enzyme digestion and DNA end marking, 50 μl NEBuffer2 and 15 μl 5U/μl MboI (New England Biolabs) were mixed into the suspension for digestion for 7 h at 37 °C with gentle rotation (15 rpm) followed by heat inactivation at 62 °C for 20 min. A mix of biotin-14-dATP (12.5 μl 0.4 mM biotin-14-dATP, 1.5 μl 10 mM dCTP, 1.5 μl 10 mM dGTP, 1.5 μl 10 mM dTTP, 5 μl klenow) was added to fill in the DNA overhangs at 37 °C for 90 min with 15 rpm rotation. The samples were transferred to a 2 mL tube and mixed with ligation buffer (150 μl 10× NEB T4 DNA ligase buffer, 125 μl 10% Triton X-100, 6 μl 20 mg/ml BSA, 5 μl 400 U/μl T4 DNA ligase, 662 μl ddH₂O) and rotated (15 rpm) at 16 °C overnight.

To reverse the crosslinks, the suspension was removed after centrifugation ($2400 \times g$, 4 °C for 30 s), and the pellets were resuspended in 300 μl elution buffer (Sigma) and 10 μl 20 mg/mL proteinase A and incubated at 55 °C for 30 min with shaking (600 rpm). Then, 65 μl of 5 M sodium chloride was added to the mixture and incubated at 65 °C with shaking (600 rpm) overnight.

Subsequently, the DNA was sheared by a Covaris LE220 instrument (Covaris, Woburn, MA) with 7 cycles of 30 s on and 30 s off. DNA was extracted as described above, and 30 μl nuclease-free ddH₂O was used to dissolve the DNA sediment. To remove the biotin mark from unligated fragment ends, 5 μl NEBuffer2, 2.5 μl 1 mM dNTP, 0.5 μl 10 mg/ml BSA, 3 μl T4 DNA polymerase, and 39 μl nuclease-free ddH₂O were added to the mixture and incubated at 20 °C for 4 h.

For biotinylated fragment pull-down, 100 μl 2× BB (10 mM Tris-HCl (pH 7.5); 1 mM EDTA; 2 M NaCl) suspended Dynabeads MyOne Streptavidin T1 beads (Thermo Fisher Scientific) was mixed with the sample and incubated at room temperature for 15 min with rotation (15 rpm). The beads with marked DNA were washed three times with 400 μl TWB (5 mM Tris-HCl (pH 7.5); 0.5 mM EDTA; 1 M NaCl; 0.05% Tween 20) and incubated at 55 °C for 2 min. The sample was subsequently transferred to a new tube, washed with 200 μl 10 mM Tris-HCl pH 8.0 and finally resuspended in 50 μl of 10 mM Tris-HCl buffer.

For library construction, 15 μl End Prep Mix 4 from VAHTSTM Universal DNA Library Prep Kit for Illumina® ND607 (Vazyme, Nanjing, China) was added to the mixture and incubated at 20 °C for 15 min followed by 65 °C for 15 min. Then, 25 μl Rapid Ligation buffer2, 5 μl Rapid DNA ligase, and 5 μl VAHTSTM DNA Adapters set1/set2 for Illumina® N801 (Vazyme, Nanjing, China) DNA adapter was added to the end-repaired sample and incubated at 20 °C for 15 min. The tube was placed on a magnetic stand to separate the beads and remove the suspension and then washed with 100 μl Elution buffer (Sigma). The beads were resuspended in 20 μl of elution buffer (Sigma) and heated at 98 °C for 10 min to release the ligated DNA. After brief centrifugation, the 20 μl suspension was transferred to a PCR tube, and 5 μl PCR Primer Mix 3 for Illumina and 25 μl VAHTS HiFi Amplification Mix were added to the PCR tube. The PCRs were run according to the following program: 95 °C for 3 min, 98 °C for 20 s, 60 °C for 15 s, and 72 °C for 30 s for 8–15 cycles, 72 °C for 5 min, and held at 4 °C.

After amplification, the library insert size was selected using AMPure XP beads (Beckman Coulter). A 0.6× volume of beads was mixed with the library in a 1.5 mL tube and allowed to stand for 5 min at room temperature. Then, the mixture was separated on a magnetic stand, and the suspension was transferred to a new tube. Subsequently, another 0.2× volume of beads was added; after magnetic separation, the suspension was discarded, and 700 μl 75% ethanol was used to wash the beads twice. Ethanol was removed, and 25 μl 10 mM Tris-HCl pH 8.0 was used to resuspend the air-dried beads. After separation, the supernatant with the Hi-C library was acquired. The library was quantified using Qubit (Thermo Fisher Scientific) and a 2100 Bioanalyzer instrument (Agilent). Finally, the library was pooled and sequenced on an Illumina NovaSeq platform with 150 bp paired-end reads.

### BioNano optical map construction and assembly

High molecular weight genomic DNA of Mi eggs was extracted and labeled following the protocol[75] of the BioNano PrepTM DLS DNA Labeling Kit (Bionano Genomics). Specifically, DNA was digested by the DLE-1 enzyme at 37 °C for 2 h to label it with DL-Green fluorophores. After the DNA label was applied, a Saphyr chip (Bionano Genomics) was used to load the labeled DNA, and a Saphyr instrument was used to acquire the optical map data. A total of 3,634,649 molecules with an N50 of 165 kb were obtained and assembled using BioNano Solve (v3.4.1) with nonhaplotype optional parameters and haplotype optional parameters. Subsequently, maps were subjected to hybrid scaffolding combined with polished Mi contigs. The superscaffold and sequence that did not scaffold were merged to obtain an improved assembly. The alignment between the BioNano map and genome was visualized using MapOptics[76].

### Mitochondrial phylogeny analysis and genome survey

All mitochondrial genomes, including our data and other available data, were assembled from short reads using MitoZ (v3.4)[77] with --assembler megahit --kmers_megahit 59 79 99 119 141 --clade Nematoda --requiring_taxa Nematoda parameters. For some samples that failed to be assembled using MitoZ, we performed assembly using GetOrganelle (v1.7.6.1)[78] with the -F animal_mt -R 15 -k 21,45,65,85,105,115,127 parameters. All mitochondrial genomes were annotated using MitoZ. Nucleotide sequences for each mitochondrial gene were extracted and aligned using MAFFT (v7.471)[79] with default parameters. Alignments were then concatenated into a super matrix, followed by phylogenetic tree construction using IQ-TREE (v2.0.3)[80] with -m TEST --seqtype DNA -bb 10000 parameters. The phylogenetic tree was visualized using iTol[81]. For unrooted tree, in addition to the protein coding gene, two ribosomal genes were added into the super matrix, and the tree was built using the same method. The mitochondrial divergence was calculated as mismatch site divided by sum of mismatch site and match site from multiple sequence alignment. For the genome survey, long reads of our sample were corrected using Canu (v2.2)[82]. Short reads and corrected long reads were used to count k-mer frequencies by jellyfish[83], with the option -h 100000 for polyploid nematodes and default parameters for Mg. The results were subsequently subjected to genomescope2.0 and Smudgeplot[32] to estimate genome size and ploidy with recommended parameters.

### Contig assembly

To obtain a haploid reference genome for diploid species Mg, we performed de novo genome assembly by using SMRTdenovo[84] software with default parameters based on PacBio data. Similarly, based on PacBio data, the genomes of polyploid species were assembled by using Canu (v1.9) with the additional parameters "corOutCoverage=200 batOptions = -dg 3 -db 3 -dr 1 -ca 500 -cp 50" to avoid homologous chromosome region collapse. Based on PacBio long reads and Illumina short reads, two rounds of polishing were performed by using Nextpolish (v1.1.0)[85] for the assembled contigs of the five species. The polished Mi contigs were subsequently scaffolded by optical mapping with Solve software (https://bionanogenomics.com/support/software-downloads/) to generate an improved assembly.

## Scaffold construction

Hi-C reads were mapped into the corrected contig using the Juicer (v1.5.7) pipeline[86]. The 3D-DNA (v180419) pipeline[87] was used to construct initial scaffolds with a parameter of -r 0. The scaffolds of diploid Mg were visualized and manually polished using Juicebox (v1.13.01)[88] to correct contig ordering and orientation. The final assembly of Mg had 18 chromosomes, and it was used as a haploid reference for polyploid assembly. The scaffolds of polyploid species were constructed as follows: (a) RNA-seq reads were aligned into contigs using HISAT2[89]. Then, contigs were annotated by Braker (v2.1.5)[90] based on RNA-seq data. (b) MCScanX[91] and jcvi (v1.0.3)[92] were used to identify synteny blocks between 18 chromosomes of Mg and polyploid genomes, thus resulting in each contig of 4 polyploids corresponding to each chromosome of Mg. (c) The contact position in the initial scaffolds was reordered and manually polished by using Juicebox based on the Hi-C matrix and collinearity obtained from the above step. (d) Scaffolds of 4 polyploids were named after Mg chromosomes. Eventually, RNA-seq reads and WGS short reads were mapped into the reference genome of 4 polyploids by using BWA[93] and HISAT2 to evaluate genome completeness. The BUSCO (v5.4.3) at genome model was used to evaluate the completeness of genomes in this study and other published genome, using nematoda_odb10 as a database.

## Genome collapse detection

Illumina short reads were aligned to the reference genome using BWA. Then, BEDTools (v2.29.2)[94] coverage was used to calculate the read depth in 100 kb windows. Read depths for the whole genome and each scaffold were evaluated using the median depth of those windows. The scaffolds that have significantly higher values than the average can be considered collapsed sequences.

## Hi-C heatmap construction

The Hi-C reads were aligned to the reference genome and processed with the HiC-pro (v 2.11.4) pipeline[95]. The contact heatmap was visualized using an iterative correction matrix in 150 kb windows. To facilitate the subsequent identification of homologous chromosomes, we separated the putative fused chromosome with a high contact signal but without contiguous spanning as much as possible during assembly. These high-frequency chromosomal interactions might imply chromosomal fusion in assembly version 1. For Mi, some chromosomal fusions with significantly high contact were extracted for further examination using a BioNano map. For each genome of assembly v2, iterative correction matrices in 150 kb windows were generated using the MIN_MAPQ = 10 parameters. These matrices were further used for visualization.

## Genome annotation

Repetitive elements of five *Meloidogyne* genomes were detected using the EDTA (v1.7.1) pipeline[96]. RepeatMasker (http://www.repeatmasker.org) was utilized to soft-mask repetitive regions using a repeat library obtained by EDTA. The masked genomes were used to further annotate protein-coding genes. Protein-coding genes were annotated based on transcripts and ab initio prediction evidence. Based on RNA-seq data, transcripts were de novo assembled using Trinity (v2.8.5)[97] software with default parameters. Full-length isoforms were obtained through IsoSeq3 software (https://github.com/PacificBiosciences/IsoSeq) based on long-read isoform sequencing (ISO-seq) data, and redundant sequences were removed using CD-HIT-EST (v4.8.1)[98] with default parameters. All the transcripts were integrated using PASA (v2.4.1)[99] to build a comprehensive database. RNA-seq short reads were mapped to the reference genome using HISAT2, and transcriptomes were assembled using StringTie (v2.1.1)[100] after alignment. The open reading frames (ORFs) of transcripts were predicted using TransDecoder (https://github.com/TransDecoder/). Ab initio gene prediction

was performed using AUGUSTUS (v3.3.3)[101] and GeneMark-ES (v4.64)[102]. Finally, EVidenceModeler (v1.1.1)[99] was used to integrate all evidence to obtain final gene structure information. HMMER[103] was used to scan the protein domain based on the Pfam database for gene functional annotation. Protein sequences were uploaded into the web server to perform KEGG and GO annotation with eggNOG-mapper[104]. Candidate HGT genes were identified using the Avp pipeline with uniref90 database[105]. Only genes with HGT or HGT-NT tags were considered putative HGT genes. Transmembrane domains were identified using TMHMM (2.0c)[106], and signal peptides were predicted using SignalP (v5.0)[107]. Proteins with signal peptides but without transmembrane domains were considered secreted proteins. All proteins of the five species were subjected to evaluation of genome completeness using BUSCO (v5.4.3) at proteome model with nematoda_odb10 as a database.

## Subgenome identification of assembly v1

First, we identified the synteny blocks between polyploid species and diploid Mg using MCScanX. Subsequently, we identified the complete syntenic gene groups from the synteny block between Mg and Mi or Ma 3n. Protein sequences of these gene groups were extracted, and multiple sequence alignment was performed by MAFFT with default parameters. The sequences obtained from alignment were trimmed using Gblock (0.91b)[108] with -b4=5 -b5=h, followed by phylogenetic tree construction using IQ-TREE (v1.6.12) with -m TEST. The phylogenetic tree of each gene group was rooted with the corresponding outgroup gene from Mg. Afterwards, we counted the topological structure of the phylogenetic trees using Biopython. The dominant topology was used to infer the relationship among three homologous scaffolds of Mi and Ma 3n. For example, the 63% phylogenetic trees that were constructed on three homologous scaffolds of chr1 supported topology (scaffold3, (scaffold1, scaffold2)). We defined scaffolds with only one leaf (scaffold 3) as the B subgenome. For another two minor topologies, 15% of trees supported (scaffold1, (scaffold2, scaffold3)) and 22% of trees supported (scaffold2, (scaffold1, scaffold3)). Based on the number of minor topologies, the branch with one leaf in the latter topology was defined as $A_1$ (scaffold2), and the other was defined as $A_2$ (scaffold1). For tetraploids, we first identified complete syntenic gene groups and then constructed phylogenetic trees by using the same method. We added the syntenic genes from triploid species into the identified tetraploid gene groups to obtain cross-species gene groups (Ma4n-Mi, Ma4n-Ma3n, Mj-Mi, and Mj-Ma3n). Phylogenetic trees of cross-species gene groups were constructed, as described above. The relationship presented by phylogenetic trees was used to infer the subgenome to which scaffolds of tetraploids belonged. Notably, all the initial names of A and B were temporary, and their final names were confirmed by *Ks* and 4DTv clustering.

## Calculation of Ks and 4DTv values between subgenomes

We identified syntenic gene pairs between any two subgenomes out of 14 subgenomes with MCScanX. KaKs_Calculator 2.0[109] was employed to estimate *Ka*, *Ks*, and *Ka/Ks* values between syntenic gene pairs with the YN method. A script calculate_4DTV_correction.pl was employed to calculate the 4DTv value between syntenic gene pairs. The medians of *Ks* and 4DTv values were calculated and used to cluster subgenomes to verify the inferred relationship based on phylogenetic trees. Nucleotide identity analysis was performed by using pyani[110] with the -m ANIm -g parameters. Hierarchical clustering of the 4DTv and *Ks* matrix was performed using the R package peahtmap.

## Telomere-associated gene and potential telomeric repeat analysis

To detect typical telomere structure from nematodes Clade IV, we investigated the telomere-associated genes and telomeric repeats in

11 species (*B. xylophilus* (GCA_904067135.1), *D. destructor* (GCA_022814885.1), *G. pallida* (GCA_000724045.1), *G. rostochiensis* (GCA_900079975.1), *H. schachtii* (GCA_019095935.1), *H. glycines* (GCA_004148225.1), *M. graminicola* (this study), *M. incognita* (this study), *M. arenaria* 3n (this study), *M. javanica* (this study), *M. arenaria* 4n (this study)). First, we collected 8 proteins that bind to the *C. elegans* telomeric sequence from previous research[36] (Supplementary Data 11). The DNA damage checkpoint protein MRT-2 was also included in the survey. The protein sequence of telomere genes in *C. elegans* was aligned against protein sequence in each species using BLSATP[111] with an e-value of 0.001. Telomere-associated genes in other species were identified based on reciprocal best blast hits (RBHs). Because of the large divergence between *C. elegans* and other nematodes at the protein level, we performed an iterative search based on the above detected telomere-associated genes.

To count telomere reads, we artificially constructed a telomere repeat sequence (TTAGGC)n as a reference sequence. Illumina short reads from *C. elegans* (SRR16969916), *B. xylophilus* (DRR067231), *D. destructor*[112], *G. pallida* (ERR123957), *G. rostochiensis* (ERR123958), *H. glycines* (SRR1800546), *H. schachtii* (SRR15101032) and *Meloidogyne* species in this study were mapped into that reference (TTAGGC)n with BWA, and complete match reads (150 M) were selected as telomere reads.

To identify potential telomeric elements, we first screened the repetitive elements located within 20,000 bp at the Mg chromosome border based on the RepeatMasker results. Enriched repetitive elements were further manually verified and determined as Mg-Tel. The putative G4 quadruplexes of Mg genome was identified using qgrs-cpp[113] with default parameters.

Since we did not find enriched repetitive elements in contigs or scaffolds of Mi, we tried to identify telomeric elements from well-assembled contigs. Well-assembled contigs were screened as follows: (a) the contig length was more than 1 Mb; (b) the contig maintained collinearity with one Mg chromosome; (c) the contig was completely matched to a BioNano consensus map. We selected one well-assembled contig to screen the telomeric repeats of Mi. Candidate telomeric repeats were manually validated and further determined as Mi-Tel. Through multiple sequence alignment, we found that a 66 bp region (CCCTACTGTCTACACCTATTGAATAACAACCCGTGCCTTTG GATATAACTCGGAGGTATGAAAGCC) was highly conserved, and this region was used as a seed sequence to detect telomeric repeats in other species. Alignment between BioNano map and contigs or scaffolds were performed by using Solve and visualized by using MapOptics.

In assembly v1, we tried to construct the genome in multiples of 18, thus some fused chromosomes were split into 2-3 scaffolds. We determined chromosomal fusion events as that there is an obvious Hi-C signal between the two scaffolds and a BioNano consensus map is completely aligned with these two scaffolds. To target the gaps between those fused chromosomes, we extracted DNA from eggs of the same Mi strain, WHF4-1, and sequenced it on ONT PromethION flow cells; a total of 25 Gb of data was obtained. The pore version used was R9.4.1, and the PromethION release version was 19.05.1. We screened the nanopore reads with more than 50 kb and mapped them into the two fused chromosomes using minimap2[114] with default parameters. We further seek the reads spanning the fusion point to confirm that the fusion point contained symmetrically distributed Mi-Tel. The ultralong reads spanning two chromosome boundaries were further verified using MUMmer (v4.0.0).

To investigate the relationship of telomeric repeats for polyploid species, we first extracted complete telomeric elements as sequences between two near 66 bp seed sequences and removed the elements longer than 1 kb. A total of 9318 telomeric elements were obtained. We used CD-HIT-EST to cluster those telomeric elements with the '-aS 0.8 -g 1 -sc 1 -sf 1 -c 0.9' parameters. The obtained 823

clusters were further used to investigate the distribution pattern in species and contigs.

## Chromosome-level genome construction

The ONT (Oxford Nanopore Technologies) long reads were de novo assembled using Canu (v2.0) with the same additional parameters as mentioned above, and the generated contigs were polished using Nextpolish. For Mi, hybrid assembly of ONT contigs was performed using Solve based on BioNano data. The improved scaffolds were further anchored into chromosomes by using juicer, 3D-DNA and Juicebox. For genome of Ma 3n, Ma 4n and Mj, polished PacBio contigs were anchored into chromosome by using juicer, 3D-DNA and Juicebox. The accuracy of all chromosomes was evaluated by using Hi-C heatmaps and read depth distribution. Collapsed regions are marked based on the read depth distribution.

For construction of the Mg_T2T genome, more than 10 kb PacBio reads were extracted and assembled using wtdbg2[115] with the -x sq parameter. Contigs were polished using Nextpolish and manually anchored into 18 chromosomes using Juicer, 3D-DNA and Juicebox. The published *M. chitwoodi* genome (GCA_015183035.1) was downloaded from NCBI to validate that the Mg T2T genome is at the chromosome scale.

## Subgenome identification of the chromosome-level genome

Due to the high quality and fewer rearrangements of the Mi genome, we first divided all the chromosomes of Mi into subgenomes. To this end, we performed synteny analysis of chromosomes between Mi and Mg using MCScanX. The gene located in the boundary of the synteny block was determined as the breakpoint of chromosome fusion. We thus identified three copies for each ancestral chromosome. By calculating the $Ks$ value between these subgenomes (A₁, A₂, B₁), the subgenome with great divergence in the $Ks$ value from the other two was determined to be the B₁ subgenome. Next, the fusion point of other species (Ma 3n, Mj, and Ma 4n) was identified based on their syntenic relationship with the B₁ subgenome. To obtain accurate subgenome information, all the split chromosomes of those 4 species were grouped into 18 ancestral chromosome sets. Then, a heatmap and clustering map of the $Ks$ matrix for each ancestral chromosome set were constructed, as described above. Clustering analysis showed that a copy of a genome belongs to subgenomes A or B. Between the B₁ and B₂ subgenomes of tetraploids, the subgenome with higher similarity to the B₁ subgenome of Mi was defined as the B₁ subgenome, while the other was defined as the B₂ subgenome. The two A subgenomes initially were randomly assigned into A₁ and A₂, and then re-attributed the A₁ and A₂ labels based on the $Ks$ matrix to better conform to the phylogenetic relationships.

## Phylogenetic analysis

To avoid affecting phylogenetic relationships due to misclassification of the A₁ and A₂ subgenomes, we performed phylogenetic analysis for each of 18 homologous chromosome sets. First, we performed syntenic analysis between 14 subgenomes and the B₁ subgenome and extracted complete syntenic gene groups from the synteny block. All protein sequences were extracted to perform multiple sequence alignment by MAFFT with default parameters. Then, the obtained alignment sequences were trimmed using Gblock with the -b4=5 -b5=h parameters and concatenated into a supermatrix. The phylogenetic maximum likelihood tree was constructed using IQ-TREE (v2.0.3) with the -m TEST -B 10000 parameters.

## Transposable element dynamic analysis

Repeat libraries were generated using EDTA (v2.0.0) for assembly v2 and Mg T2T genome. The landscape of all transposable elements was performed using RepeatMasker with the -s -a -inv -no_is -norna -xsmall -nolow -cutoff 225 parameters and further generated using the

calcDivergenceFromAlign.pl script. Full-length LTR retrotransposons and LTR identity were extracted from the EDTA results. For each TE family, we counted the copy number in each subgenome from the EDTA result.

### Homologous gene expression analysis

The expression level of transcripts per million (TPM) was obtained from predicted genes using Kallisto[116], with three biological replicates of each sample. The average of three replicates was considered the expression value of the gene. To ensure the accuracy of the analysis, we filtered syntenic groups as follows: (a) Only complete syntenic groups were used for analysis; (b) only syntenic groups in which each of the genes showed a < 30% coefficient of variation between biological replicates were retained; and (c) only syntenic groups in which the sum of the gene expression levels was more than 5 TPM were retained. For each filtered syntenic group, the relative expression value was normalized by the TPM value of each subgenome divided by the sum of the TPM values of all subgenomes. We calculated the Euclidean distance between the normalized gene expression value and the expected expression value. The pattern with the shortest distance was defined as the expression bias category. The number of each category was counted during five development stages. Mann–Whitney test were performed for all expression bias genes to confirm the difference expression between subgenomes.

### Distribution of genetic features in *M. incognita*

The ChIP-seq data (PRJNA725801) were downloaded from the NCBI SRA database (https://www.ncbi.nlm.nih.gov/sra). For histone modification, reads from eggs were mapped to the Mi chromosome genome using BWA, and duplications were marked using Picard tools (http://broadinstitute.github.io/picard/). BamCoverage[117] was used to calculate the RPKM value in 50 kb windows. Gene density was also calculated in 50 kb windows. For the centromere, a 19 bp box[48] (TCGGGCCTTCGGCCCTCGC) was used to identify the centromere region, allowing one base mismatch. The distribution of features was visualized using Circos[118] software. To exhibit the distribution pattern across chromosomes, we selected chromosomes without chromosomal fusion events. The RIdeogram[119] package was used to visualize the distribution pattern of genetic elements in the arms or central region and the colocalization among histone modification, centromeres, and telomeres. The 20% first and 20% last Kbs of each chromosome were considered as arms, and the rest were centers. Subtelomeric region were defined as the 250 kb region at the end of chromosomes. The 50 kb windows with the centromere motif was defined as the centromere region. Statistical tests for H3K9me3 signal in centromere region, H3K4me3 signal in subtelomeric regions, and gene density and TEs density between chromosomal arms and centers were performed using Mann–Whitney test.

### Identification of genome structure for *M. floridensis* and other *Meloidogyne* species

Whole genome sequence short reads of other root-knot nematodes (Ma, Mi, Mj, Mf, and *M. luci*) were downloaded from the NCBI SRA database (Supplementary Data 2). Reads were aligned into four polyploid RKN genomes (assembly v2) with BWA. SAMtools idxstats and BEDTools coverage were used to calculate the read count and the sequencing depth for each subgenome. We simulated the expected read count and depth when mapping samples with different ploidy levels into triploid or tetraploid genomes. These analyses suggest that Mf is composed of the A subgenome. To determine whether Mf is a triploid AAA or diploid AA, we mapped reads of Mf into a reference that can represent the A subgenome (published Mf genome GCA_003693605.1 or Mi_A$_1$ genome). Bcftools (1.8)[120] was used to identify SNPs from those alignments. We focused on the alternative allele depth of heterozygous SNPs, and there should be a peak at 0.5 when diploid and at 0.33 when triploid. We also performed similar analyses in diploid Mg. These analyses provided evidence for the genome structure of Mf.

### Fluorescent in situ hybridization of Mi-Tel repeats

Take 20 µl of purified eggs, add 400 µl of 0.005 g/L colchicine, let stand at room temperature for 3–5 h, then centrifuge and discard the supernatant. Add 1 ml of fixative solution (ethanol: acetic acid 3:1), let stand at room temperature for 5 h and then transfer to −20 °C overnight. After centrifuging to remove the fixative, resuspend the eggs with 100 µl of 45% acetic acid, take 10 µl of the suspension and drop it on a glass slide, cover it with a cover slip, tap it gently and uniformly for 30 times with the blunt tip of tweezers, and then bake it with flame for 2 s to make the cells expand when heated. Press the cover slip with your thumb for 30 s, transfer the prepared slide to −80 °C for 24 h, uncover the cover slip, and soak in absolute ethanol for 5 h or overnight. Transfer the prepared slides to a 37 °C oven for 12 h, then rinse the slides with 2× SSC for 2 min, add 100ug/ml of RNase A dropwise and treat them at 37 °C for one hour, then wash the slides with 2× SSC. After the slides were soaked in 0.01 M hydrochloric acid, 150 µl of 500 µg/ml proteinase K was added dropwise, treated at 37 °C for 20 min, and then the slides were placed in 2× SSC and shaken for 5 min. Shake slides in 4% paraformaldehyde solution for 10 min, 2× SSC shaker for 5 min, three times; shake in 70% ethanol for 3 min, 80% ethanol for 3 min, 95% ethanol for 3 min, 100% ethanol for 3 min, and dry in air for 1 h above. After the hybridization solution was prepared, it was denatured in a water bath at 85 °C for 10 min, cooled on ice for 5 min, and then 50 µl of the hybridization solution was dropped onto a glass slide, covered with a membrane, treated at 74 °C for 7 min, and incubated in a wet box at 37 °C for 16–20 h. Remove the membrane in 2×SSC solution, transfer to 1×PBS and shake for 5 min, add 100 µl 20 µg/ml DAPI and store at 4 °C overnight before microscopic examination.

### Reporting summary

Further information on research design is available in the Nature Portfolio Reporting Summary linked to this article.

## Data availability

All sequencing data have been deposited into the National Center for Biotechnology Information (NCBI) Sequence Read Archive (SRA). The PacBio, Illumina, and Hi-C data generated in this study have been deposited in SRA database under accession code PRJNA784524; The RNA-seq data generated in this study have been deposited in SRA database under accession code PRJNA786696; The BioNano data generated in this study have been deposited in BioProject under accession code PRJNA787730; The Nanopore data generated in this study have been deposited in SRA database under accession code PRJNA788579; The ISO-seq data of *M. incognita* generated in this study have been deposited in SRA database under accession code PRJNA787737; The assembly genome and annotation files generated in this study have been deposited in our laboratory website (http://bmb.hzau.edu.cn/sjxz.htm); The accession codes of data set for mitochondrial phylogenetic analysis are in Supplementary Data 2; The data set of genome and Illumina reads downloaded for telomere structure and repeat count analysis are under accession codes GCA_904067135.1, GCA_022814885.1, GCA_000724045.1, GCA_900079975.1, GCA_019095935.1, GCA_004148225.1, GCA_015183035.1, GCA_018905775.1, GCA_000172435.1, GCA_003693605.1, GCA_000751915.1, GCA_903994135.1, GCA_902706615.1, SRR16969916, DRR067231, ERR123957, ERR123958, SRR1800546, SRR15101032; The data set downloaded for histone modification analysis of *M. incognita* is under accession code PRJNA725801. The source data for Figs. 1–5 and Supplementary Figs. 1, 6–9, 12–15, 19, 21, 22, 24–27, 29, 30, 34–35, 38, 40,

41, and 44 are provided as a Source Data file. Source data are provided with this paper.

## Code availability
The codes and pipelines used in this study are openly available at Zenodo [https://doi.org/10.5281/zenodo.8383556][121].

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

## Acknowledgements

We thank Huazhong Agricultural University for technical support; Yang Wang of Yunnan Agricultural University for providing Rutgers tomato material and early guidance on nematode reproduction; Yuxi Duan and Piao Lei of Shenyang Agricultural University for providing the *M. arenaria* 4n species; Gaofeng Wang of Huazhong Agricultural University for providing *M. graminicola* species; YangYang Chen of Huazhong Agricultural University for helping in FISH experiments of *M. incognita* telomere; Xingwang Li of Huazhong Agricultural University for helping in Hi-C data analysis; Xinxin Li of Huazhong Agricultural University for the early guidance on Hi-C experiments; Wentao Wu of Yunnan Agricultural University for providing nematodes. We thank National Natural Science Foundation of China (U20A2040, M.S., and 31970076, M.S.), National Key Research and Development Program of China (2017YFD0201201, M.S.) and Hubei Hongshan Laboratory (2022hszd012, M.S.) to support this work.

## Author contributions

M.S., J.Z., and D.D. conceived the project. M.S., J.Z., and D.P. administrated the project. M.S. acquired the funding. D.D. designed the experiments. D.D. and Y.Z. performed the experiments. D.D. collected the data. C.X. and D.D. performed the data analysis. D.D., S.M., S.Z., D.B., Y.L., S.C., X.W., Z.Z., and F.L. prepared the materials. All authors contributed to manuscript preparation. C.X., D.D., J.Z., and M.S. wrote the manuscript.

## Competing interests

The authors declare no competing interests.
