## [Peer Review File · Nature Communications]

Unzipped chromosome-level genomes reveal allopolyploid nematode origin pattern as unreduced gamete hybridizationReviewers' Comments:

Reviewer #1:

Remarks to the Author:

This paper by Ming Sun et al. describes and compares the genomes of five root-knot nematode species assembled at or approaching chromosome-scale resolution. Four of the species have polyploid genomes and reproduce via mitotic parthenogenesis while a fifth species is diploid and reproduces via facultative meiotic parthenogenesis. The authors have achieved an impressive amount of work and analyses, and this has led to enormous progress in the contiguity of genome assemblies for these species. In itself, this represents an important technological achievement and a valuable resource for the whole community of nematologists and evolutionary biologists.

Analysis of these genomes led to the identification of unusual telomere repeats in these species as well as the absence of many telomere-associated proteins usually well conserve in other nematodes. Furthermore, comparative analyses of the genomes allowed confirming previous findings by different other groups on former more fragmented versions of the genomes: these species are most likely recent polyploid hybrids (Blanc-Mathieu et al. PLOS Genetics 2017, Lunt et al. PeerJ 2014, Szitenberg et al. Genome Biol Evol 2017, Jaron et al. Journal of Heredity 2021).

Overall, many analyses in this paper seems to have been inspired by the previous work by Blanc-Mathieu et al. 2017, with phylogenomics analysis of the genome copies as well as differential gene expression between the gene copies resulting from hybridization, albeit at a much higher resolution due genome assemblies based on a combination of long-read, Hi-C and optical mapping (for some). The study confirms different expression levels between the different gene copies resulting from polyploidization and suggest a new scenario for the origin of the different genome copies in the species investigated, involving gamete non-reduction and hybridization.

The authors also describe one of the root-knot nematode species they have sequenced as a new triploid species and named it *M. paraarenaria*.

Overall, the different figures and illustrations are of remarkable quality and represent in a synthetic way multiple different complex analysis and I would like to congratulate all the authors for the technical 'tour de force'.

With that said, despite the impressive amount of data generated and bioinformatics analyses performed, I regret that many of the main conclusions are not supported enough by the data and results of the analyses.

For instance:

- the whole analysis on differential expression levels seems to have been done in the absence of biological replicates. Consequently, the validity of the measured differences remains to be confirmed.
- the hypothesis that unreduced gametes have led to two of the copies in these genomes is not the sole hypothesis that could explain the observations. Multiple successive hybridization events with more or less closely related species could explain the same pattern and the level of nucleotide sequence divergence between A1 and A2 must be provided to help choosing the most likely hypothesis.
- further evidence than just a phylogeny with no bootstrap and using a single short nuclear marker sequence (ITS) is absolutely necessary before claiming and naming a new *Meloidogyne* species (*M. paraarenaria*).

Moreover, I have several major concerns regarding the methodology used to scaffold the genomes at both stages. At the first stage when *M. graminicola* is used to anchor the genomes of *Mi*, *Mj* *Ma* and *Mp* while being a distantly related *Meloidogyne* species. The authors must clarify to what extent this influences the other genome assemblies to resemble too much that of *Mg*.

At the second stage, when aiming at telomere-to-telomere genome assemblies, I also have doubts concerning the methodology which consists in forcing all the chromosomes to start and end with the newly identified telomere repeats. These efforts result at the end in a genome with less chromosomes and which is bigger than expected for *Mi*, for instance. These points should be addressed and clarified.

Furthermore, because many intermediate results are not provided (e.g., different versions of the

genome assemblies or their annotations) and some methodological points are not described with enough details, the work is mostly not reproducible, and several statements cannot be verified. I also found the discussion quite frustrating as it does not really replace or compare the new findings with previous ones found by other groups studying genome structures and evolutionary history in the same species. Curiously there are also many results that have been produced in the extended data (e.g. on HGT, on histone modification marks, on centromere) that are not mentioned in the results section or not discussed in the discussion.

Finally, syntax and grammatical English errors are present and pervasive in the whole manuscript from the abstract to the discussion and should be corrected.

For this ensemble of reasons, I would like to recommend a major revision of the manuscript and you will find below more details and suggestions to help improve the manuscript.

Major points that absolutely need to be addressed / discussed / clarified

The description of a 'new' species 'M. paraarenaria' could be very interesting in itself. However, at this stage, the evidence supporting this is a new species are not sufficient. So far, the only evidence provided is that the SCAR markers characteristic of Mi, Mj and Ma do not amplify on the so-called Mp species. However, are other SCAR markers for other Meloidogyne species been tested as well? This could completely be another already described triploid Meloidogyne species belonging to Clade I (sensu Tendingan De Ley et al.). Have Esterase profiles been determined? This is also a frequently used technique to distinguish the different Meloidogyne species. Before claiming this is a new species, several molecular tests for species identification should be done and show consistent results. Furthermore, why was the name M. paraarenaria given? Absolutely no explanation is given for this name in the main text. I understood that this was given according to the phylogenetic position (close to M. arenaria but with no support value) in Extended Data Fig. 1, based on ITS marker. However, first it is surprising to have only one ITS sequence per polyploid genome while most of the rest of the paper clearly explains that most of the genome regions exist in 3 to 4 copies and with high divergence between A and B copies. Could the author justify / explain why only one single ITS sequence per species was used and which copy? Second, using the ITS marker alone is not enough and multiple concatenated nuclear markers as well as mitochondrial markers in a phylogeny with bootstrap values should be used like in Álvarez-ortega et al. Scientific Reports 2019, for example. Would using another marker gene completely change the topology shown in this figure? So far, there is not enough convincing evidence to support this is a new species and the authors should reinforce this point.

I have an important concern about the assembly strategy for MIG species according to synteny and collinearity with Mg. This strategy will obviously allow assembling further MIG contigs and scaffolds but not based uniquely on overlap between reads or Hi-C contact data. Using such a strategy might influence MIG genome assemblies to resemble that of Mg but does not necessarily correspond to a biological reality. If major structural variations or chromosomal fusions and rearrangements occurred separately in the different branches these might be underestimated and overlooked by forcing the MIG contigs to arrange together in a colinear manner with Mg contigs. This problem is amplified by the fact that (i) MIG species are mitotic parthenogens with a lot of genome rearrangements to be expected and (ii) Mg is quite distantly related to the MIG species (there are more closely related diploid mitotic species such as M. hapla for instance). I wonder how the use of JCVI suite to anchor MIG scaffolds on Mg scaffolds would over-influence the assembly towards Mg. Could the author comment and clarify this point?

It would be informative to provide intermediate assembly results in addition to assembly version1 results. For instance, how many scaffolds were obtained by scaffolding with Hi-C data without using Mg as a guide? What were the N50 values for these intermediate versions?

An additional point that casts doubt on the validity of using Mg to anchor contigs of the other Meloidogyne species is the initial impossibility to identify the G-rich telomeric repeats in these scaffolds while an analysis at the contigs level allowed identifying them.

Another major concern is the methodology used to obtain 'telomere to telomere' genome assemblies.

I wonder to which point, by forcing all scaffolds to start and end by the identified telomeric repeats, the authors have over-assembled the genome. Indeed, the now so-called T2T assembly of *M. incognita* has a lower number of chromosomes (40) than the numbers reported in the literature. Indeed, although Triantaphyllou reported 40-46 chromosomes by cytological observations in the 80's, more recent observations (Despot-Slade et al. Mol Biol Evol 2021) count 46 chromosomes with more modern visualization and staining technology. In the same paper by Despot-Slade et al., the authors indicate that the chromosomes range in size between 0.4 and 1.5 μ m which corresponds to a size variation factor of 3.75 between the smallest and biggest chromosomes. However, in the Mi genome assembly presented here (Fig. 3) there is a size variation factor of 7.5 between the smallest and biggest chromosomes. This is also reflected by the bigger genome assembly sizes (Table 11) that now falls out of the range estimated by flow cytometry in (Blanc-Mathieu et al. PLoS Genetics, 2017). Indeed, estimated genome size for Mi was 189 (+/-15Mb) and the new assembly size is now >225 Mb. Thus, unless substantial variations in genome size exist between Mi strains across the world, these metrics cast doubt on the genome assembly validity. This possibility could be tested by performing k-mer based estimation of genome size using Jellyfish and GenomeScope 2. This is surprising that these measurements have not been done or provided in this paper. Finally, the Hi-C contact maps also cast doubts on the validity of these new T2T assemblies. Indeed, while Extended Data Fig. 3 presented rather smooth contact maps with, as expected, most of the strong signal present on the diagonal, the new contact maps (extended data figs. 18, 20-22) are more noisy with a lot of strong signal away from the diagonal suggesting extensive duplications of 100% identical scaffolds and some mis-assemblies that would deserve to be manually curated.

I also wonder how circular is the analysis of ancestral karyotype reconstruction. Again, by anchoring everything to the Mg genome, which counts 18 chromosomes, wasn't it completely expected and mandatory to infer the ancestral karyotype was composed of 18 chromosomes? How dependent to the karyotype of the meiotic genome used as an anchor is this methodology? For instance, using *M. hapla*, which counts 16 chromosomes as an outgroup species instead of *Mh* would have led to an ancestral karyotype of 16 chromosomes? Could the authors comment this point?

It should be clarified as soon as possible in the manuscript that A1 and A2 subgenomes cannot be distinguished and that only the B genome can be correctly separated based on the Ks analysis. This must be said much earlier than currently (L221) and as soon as A1 and A2 subgenomes are mentioned.

L238-240. This part concerning whole genome heterozygosity estimated from k-mer vs. 'sequence divergence' is misleading. K-mer heterozygosity gives an estimate at the whole genome level regardless of whether or not sequences are coding or non-coding. In contrast numbers extracted from table 16 only concern coding sequences and thus these values cannot be compared or put together in a same sentence because they refer to different components of the genome. This point stresses out the importance of providing an estimate of the nucleotide sequence divergence level between A1, A2 and B genomes not only in the coding regions but at the whole chromosome scale.

Reproducibility and FAIR principles:

The authors should provide all the genome assemblies (intermediate and final) as well as all the annotations to ensure reproducibility of the results they present and optimal reuse of the data they have produced for the whole community.

Other important points that need to be addressed / clarified

Abstract / Introduction

The terminology MIG for '*M. incognita* group' makes no sense from a phylogenetic or taxonomic point of view. It would be more correct to refer to 'tropical' or 'mitotic parthenogenetic' or 'Calde I' (sensu De Ley et al.) root-knot nematodes.

L40. Most animals indeed use outcrossing via sexual reproduction, but many plants and fungi use self-

fertilization or parthenogenesis.

L62-64. There is probably a confusion here. HGT events have indeed probably participated in adaptation to plant parasitism but not specifically in MIG species. This is a common feature of all plant-parasitic nematodes whether or not they have parthenogenetic reproduction. In the same way, gene duplications, genes copy losses or movements of transposable elements all participate in genome plasticity but whether these 'mutations' in a broad sense have any specific impact on the adaptability of these species remains to be demonstrated. This remains so far hypothetical.

Results

L87-88. *M. graminicola* is a facultative meiotic parthenogenetic species. I think it is important to clarify for the non-specialists that this species can do both sexual reproduction and meiotic parthenogenesis. L91-94 Important to clarify here that 44Mb for Mg corresponds to the haploid genome size where the two homologous chromosome sets have been collapsed while for the three other species (Mi, Mj and Ma) the assembly sizes are consistent with separation of the homologous chromosomes during assembly.

L101-102. In addition to CEGMA, which is an old software not maintained anymore, I would highly recommend using BUSCO as well and comparing the results in a supplementary table with all the other *Meloidogyne* species genomes available.

L107-108. Results on repetitive sequences content are interesting but would deserve a more detailed description like the relative contribution of canonical transposable elements (potentially active) and other repeats. Furthermore, in Supplementary Table 4 it would be more useful to the readers to provide (retro)transposons names rather than their EDTA codes, according to:

(https://github.com/oushujun/EDTA/blob/master/util/TE_Sequence_Ontology.txt). For example, DNA/DTA = hAT_TIR_transposon; DNA/DTA = also hAT_TIR_transposon. Then, based on this EDTA codes that refer to a same parent category, the lines should be grouped. Once this is done, DNA transposons should be separated from retrotransposons more clearly. Moreover, these results are not further discussed in the rest of the paper. It seems that the polyploid hybrid genomes have a higher proportion of repetitive elements than Mg. Interestingly similar findings were reported in (Blanc-Mathieu et al. 2017, PLoS Genetics), when the genome of *M. hapla* (diploid, meiotic) was compared to those of Mi, Mj and Ma (polyploid, mitotic).

L116-122. Results on phylogenetic relationship between the homologous gene copies are very interesting and indeed clearly suggest that two copies are more closely related to each other than the third copy. However, this does not automatically imply an AAB structure. This completely depends on the sequence divergence level between the two 'A' copies. Although the divergence is higher with B, it seems to already be quite high between 'A1' and 'A2'. Could the author provide some information about the % of nucleotide divergence between A1, A2 and B?

Extended Data Fig. 10 as most other figures of this paper is remarkably well done and interesting. It seems that the first triplet of homologous sequences in Mi does not present any synteny relationship in the other 4 *Meloidogyne* genomes. Is this genomic region in 3 copies unique to Mi?

L141-142. Please provide a reference for the classification of nematodes in Clades I – IV. Several classification systems exist, one by Hans Helder and collaborators that defines 12 Clades with Arabic numbers (1-12) and one defined by Mark Blaxter and collaborators that defines 5 Clades with Roman numbers (I- V). It seems the authors have used the classification proposed by Mark Blaxter et al. and this must be acknowledged.

L149. I suggest using the Latin species names (*Drosophila*, *Anopheles*) rather than Fly and Mosquito, as it is unsure all flies and all mosquitoes have alternative mechanisms for telomere lengthening. This is more correct to refer to the exact species.

L151 and continued. These findings on unusual telomeric repeats in the *Meloidogyne* species analyzed is particularly interesting. Could the author explain a bit more what are the typical characteristics of G-quadruplex structures for non-specialists and how they have inferred it was the case in Mg?

Concerning the unusual telomeric repeats it is unclear from the text whether Types I, II and III presented in supplementary table 9 belong respectively to Mi, Mj and Ma for instance? Or are the three types found in the three species. If yes, do they co-occur on the same regions or different types are on different contig ends? Extended Data Fig. 16 suggests all three types are present in the three

species but Type I being preferentially present in Mi. This is not clearly explained now in the text and these interesting results should be discussed. Also, it is not self-evident for non-specialists how extended Data Fig. 16 suggest recombination events between telomeric regions? Could the author further explain how this figure supports these conclusions?

Extended Data Fig. 23. This is indeed very interesting (and satisfying) to observe that the average normalized coverage seems to be uniform across the chromosomes. However, it appears that the standard deviation is much higher in Mi than for the four other Meloidogyne genomes. Do the authors have any hypothesis to explain this striking difference?

L227-229. On the origin and evolution of MIG. It is unclear what the authors mean by diploid topology in A and haploid topology in B. In Figure 4a it seems that one duplication of A is ancestral to Mj and Ma while Mi underwent a species-specific duplication of A. In contrast, each B genome from each species shows indeed a different evolutionary history with no clear species-specific duplication of group-specific duplication. Is it what the authors mean for A and B presenting respectively a diploid and haploid topology?

L233-235. Why is the previously proposed hypothesis of a hybridization between closely related species never considered or evoked for the origin of A1-A2 subgenomes. What is the level of nucleotide divergence between A1 and A2? If this divergence is high, albeit lower than B, another possible explanation would be two rounds of hybridization for Mi, one with a more closely related species (giving A1 and A2), another one with a more distantly related species giving B. Likewise, for Mj and Ma a common ancestral hybridization between two relatively close species could have given the exact same topological pattern. Hence, it would be interesting that the authors consider, discuss, and test this hypothesis too. For these reasons it is really important to provide an estimate of the level of nucleotide divergence between A1 and A2 at the whole genome level and not only for coding regions (Ks). Then, putting this divergence in relation to the very low divergence at the mitochondrial level might help clarifying which scenario is the most realistic.

L241-242. The authors seem to consider M. floridensis as a diploid A1-A2 species. However, in Jaron et al. 2021 'Genomic Features of Parthenogenetic Animals', an author already cited in ref. 19, it seems that k-mer measures using GenomeScope2 and Smudgeplot suggest M. floridensis is triploid (3n). Could the authors comment this point? The rest of the results concerning Mf being more closely related to Mi than any of the other Meloidogyne species studied is completely convincing. However, if M. floridensis is actually triploid, then the hypothesis of Mf being an ancestor of Mi (A1-A2) does not hold true anymore. And if Mf is diploid, is the divergence between its homologous chromosomes comparable to that measured between A1 and A2?

L281. I would recommend using 'putative' or 'predicted' secreted proteins. At this stage these are just bioinformatics predictions based on the presence of a signal peptide and no transmembrane region on computer-based genome annotation. Furthermore, the authors should explain / justify why putative secreted proteins would be more interesting than the rest of proteins to study an effect on fitness?

Extended Data Fig34. Please do not use 'secreted' genes, a gene is not secreted but encodes proteins that might be secreted. Furthermore, is sequence divergence measured at the amino-acid or nucleotide level? Is it measured between homologous genes or all genes? Please clarify.

L288-301. The part on differential expression of homoeologous gene copies according to the subgenome they belong to is particularly interesting. However, in this kind of analysis multi-mapped reads must be treated with particular caution as they can be fully responsible alone for the apparent variations in TPM values if not treated correctly. More importantly, it seems that no biological replicates have been produced for these transcriptomic data while it is absolutely necessary to support differential expression between the copies.

Discussion

Overall, the discussion is a bit frustrating because the many interesting results presented in the rest of the manuscript are not really discussed or put in perspective with previous analyses from the literature. For example, the hypothesis for the origin of the different genome copies is not compared to other previous hypotheses, the perspectives concerning the discovery of the new telomere repeat are not mentioned, etc...

L326-327. Neither ref. 17 nor 38 mention DNA damage repair in MIG, so please update this sentence

accordingly.

L333-337. The authors mention that genomic plasticity might be an advantage of mitotic parthenogenetic *Meloidogyne*. But are the genomes of these species actually more 'plastic' than those of meiotic facultative parthenogenetic species? Are there some data showing these genomes are particularly 'plastic'?

Methods

L557. How were the five developmental life stages distinguished. What was the morphological criteria used to differentiate J3-J4 from late J2 or from young females?

More important concern regarding the RNA-seq experiment: it does not seem that biological replicates have been conducted. Consequently, all the results for differential gene copy expression or across the life stages remain to be confirmed by using biological replicates (or qPCR).

L715-737. Genome annotation. It is important to assess the completeness of the predicted gene sets (or proteomes) using BUSCO in proteome mode and compare to previously reported protein sets for the same species in the literature. This has been done with CEGMA for the genomes but not for the predicted proteins. L734. Candidate HGTs have been detected using Alienness but with which Alien Index cutoff value and against which protein library? Moreover, nothing is said about HGTs neither in the results nor in the discussion? What was the goal of this HGT analysis and what are the main conclusions? L736. The references given for TMHMM and SignalP are incorrect, please use the references recommended by the authors of these software. Please also indicate which version of SignalP was used and with which parameters.

L743-745. Please indicate which versions of MAFFT, Gblock and IQ-Tree were used and with which options / parameters (which model of evolution for IQ-tree ? With or without bootstrap replicates?).

L777-779. Please cite all the publications for the genomes that have already been published.

L779-783. A simple BLASTP with an e-value threshold of 0.001 with *C. elegans* proteins is definitely not a sufficient criterion to decide conservation (presence /absence) of telomere-associated proteins in the other species. At least reciprocal best blast hits should be used and ideally phylogenetic analyses.

L795 and L 813. Where do the ONT long reads come from? Nothing is mentioned in the methods concerning the origin of these ONT long reads. How was the material extracted (same material)? Which sequencer was used? Which flow cell? Which basecaller with which parameters?

L805-807 and 859-861. Same remark than previously for phylogenetic analysis: which versions of MAFFT and IQTree were used, and which parameters were used for the multiple alignment and phylogeny (which model of evolution)?

L861-869. The methodology used to reconstruct the mitochondria phylogeny is not straightforward. We could expect that some contigs obtained by long reads assembly and then polished with short reads would correspond to the mitochondrial genome in each species. Was it the case? If yes, why the authors did not use an alignment of the whole mitochondrial genomes (or all their genes) to reconstruct the phylogeny? Everything was here based on SNP detected from short reads alignment on the Mi mitochondrial sequence present at the NCBI. This methodology is completely different from the one used for all the other phylogenetic analyses (multiple alignment followed by ML phylogeny). Extended Data Fig. 32 shows a very short sequences and many unknown (N) positions in these sequences, casting doubts on how reliable is this mitochondrial phylogeny on such a short sequence and many N's. Furthermore, the obtained topology is not compared to previous mitochondrial genome phylogenies such as for example in (Blanc-Mathieu et al. 2017, PLoS Genetics).

L872. The methodology to determine TPM is not clear. The authors indicate Kallisto was used but do not provide further details such as whether RNA-seq reads were first checked for QC and whether they were aligned on transcripts from ISO-seq, from StringTie or deduced from gene predictions on the genome? This combined with the absence of biological replicates for the transcriptome data cast doubts on the validity of the whole analysis on differential expression of gene copies.

L884. Which Chip-seq data have been downloaded from the NCBI? Please provide accession numbers and if these data were already published, please cite the appropriate reference. By the way, these data were just used for one Figure (extended data Fig. 24) and never further commented or even mentioned neither in the results nor in the discussion. Same remark for centromere repeats with no mention in the results or discussion. For instance, are there some histone marks specifically associated

with telomere repeats or centromere repeats?

L898. Which whole genome sequence from which source were used (please provide accession numbers)? Do these sequences correspond to previous versions of these genomes already published in the literature? If yes, please cite the appropriate references.

Correct reference to the source data

This is not sufficient to indicate that genome x has been downloaded from the NCBI or from wormbase, accession number to the bioproject or genome version should be provided and if the genome assembly has been published previously, the corresponding paper should be cited. For instance, in Extended Data Fig 15 it is impossible to tell what is the difference between *M. floridensis* 1 and 2 (two different versions? Already published?), same remark for *M. hapla* or *M. enterolobii* (already published?).

Three previous versions of *Mg* genomes have been published (Somvanchi et al. 2018, *Journal of Nematology*; Phan et al. 2020, *Ecology & Evolution*; Somvanshi et al. 2021, *Gene*), they should be at least cited and compared to this new version which is undoubtedly a substantial improvement and resolves the genome at chromosome resolution.

Syntax / grammatical errors

A lot of syntax and grammatical errors as well as sentence construction problems including in the Abstract. The whole paper needs to be extensively edited by a native English speaker.

Examples:

- L26. allopolyploidy not allopolyploid
- L26. shows a widespread not 'the' widespread
- L30. AAB and AABB, respectively for what and what?
- L41-42 the advantage of sexual reproduction not sexual species.
- L44. What does an 'inferior' position mean? Please clarify
- L46. Even surpassing not even surpasses
- L46. Their diploid sexual relatives not diploid sex.
- L92. Which was consistent with not 'in consistent with'.
- Figure 1d. Tetraploid not 'Teraploid'
- L144. Clade IV not Clade VI.
- Supplementary Table5. A gene is not secreted but its protein product yes. Please change to secreted proteins or to gene encoding putative secreted proteins. Same remark for line 302 and everywhere 'secreted gene' is incorrectly used.
- Extended Data Fig. 34. 'sequence divergence' not 'sequence divergency'
- L707. 'fused' not 'fussed'

These are just some examples to illustrate the issues and the whole manuscript needs to be extensively edited to become more correct and understandable.

Reviewer #2:

Remarks to the Author:

Review of "Phased T2T genomes reveal allopolyploid nematode origin pattern as unreduced gamete hybridization" for *Nature Communications*

The authors report four phased telomere-to-telomere de novo genome assemblies from the hybrid polyploid apomictic *Meloidogyne incognita* group (MIG): *M. incognita*, *M. arenaria* (Ma) and *M. javanica* (Mj), along with a newly discovered member: *M. paraarenaria* (Mp), and a haploid chromosome-level assembly for *M. graminicola* (Mg) – a diploid automictic outgroup. This MIG system notably has A and B subgenomes and some species are triploid (Mi, Mp) and some tetraploid (Ma, Mj). The authors resolve the relationship between these subgenomes and reconstruct a scenario for the organization of these distinct genomic combinations, involving hybrid events between reduced and unreduced gametes. Additionally, a novel alternative telomere element is reported here that appears

to have evolved in MIG following the loss of telomerase. The genomes assembled and described here appear to be very high quality and represent a huge gain to the communities interested in MIG as a pest and model for genome evolution following allopolyploidization. However, I think that the relatively shallow degree of scholarship about asexual reproduction, polyploidy, the current status of literature on the MIG system, along with minimal context for broadly interpreting the results beyond description and instances of overreaching in discussing the results prevent this text from being a compelling paper as is. Here, I outline my primary concerns, followed by some more general comments on the vagueness and missing information in the methods, and some comments about specific pieces of text.

Main comments:

I think the motivation and results need to be better contextualized. There is very little use of the prior literature except a general statement (Lines 62-64) about some genomic features possibly associated with MIG adaptability and that the previous draft genomes were fragmented (Lines 65-68). More text is warranted in the Introduction to outline what exactly is and is not known about the history of this species complex. What is the current hypothesis/hypotheses for how this species complex formed, what hybridization events have been proposed, etc. While the genome generated here are impressive, the text does not lay out their purpose in a clear manner. The Discussion is treated similarly to the Introduction; I do not get clear sense of what the status is now for our understanding of this system – what was answered and what is still to be determined. How do these results compare to other recent findings, e.g. Schoonmaker et al. 2020 G3: <https://doi.org/10.1534/g3.119.400650>? How does this text add to our understanding of genome evolution following allopolyploidy events?

The maintenance of sex is far from settled. The text indicates (Lines 41-43) that sex is advantageous because, relative to asexual reproducing lineages, sexual lineages can adapt more quickly to a changing environment. This argument is one of many hypotheses for the long-term persistence of sexual reproduction. This text needs to be updated to better reflect breadth of the field of the evolution of sex. While adaptability could be emphasized, perhaps with more examples, it should be clear that why so many organisms have sex to reproduce is not answered to date. See these recent perspectives that covered different hypotheses on the maintenance of sex and commentary on the active and ongoing debate on this question (the second also touches on geographic parthenogenesis and connections between hybridization and parthenogenesis, that should be considered more in this text):

- Neiman et al. 2018 *Evolution*: <https://doi.org/10.1111/evo.13499>
- Fujita et al. 2020 *Annu. Rev. Ecol. Evol. Syst.*: <https://doi.org/10.1146/annurev-ecolsys-011720-114900>

Similarly, the relationship between asexual reproduction and polyploidy/hybridization is not well drawn out in the text. Line 49: How polyploidy facilitates adaptive evolution in asexual taxa is not “unexplored”, perhaps “still an open question”, but there is plenty of literature on the subject. This text should include more make it clear that parthenogenesis is often associated with polyploidy, not a rarity as indicated in the text. More literature needs to be referenced on the connection between polyploidy (and hybridization) and asexual forms of reproduction, as well as more examples that the weevil study the cited.

- Otto and Whitton 2000 *Annu. Rev. Genet.*: <https://doi.org/10.1146/annurev.genet.34.1.401>
- Maciver 2016 *Trends in Parasitology*: <https://doi.org/10.1016/j.pt.2016.08.006>
- Ament-Velásquez et al. 2016 *Molecular Ecology*: <https://doi.org/10.1111/mec.13717>
- Neiman et al. 2014 *J. Evol. Biol.*

It is not clear to me that this text describes a “unique pathway of polyploidization” or how it contributes to adaptive evolution in parthenogens. The text cites a review on how unreduced gametes facilitate polyploidization (Mason and Pires 2015), which describes triploid bridges, so I am not sure how the MIG system represents a novelty. Perhaps they mean specifically in animals, as triploid bridges are well known from plants and more hypothetical in animals?

The text, starting with Abstract, claims to connect the polyploidization in MIG system to its adaptability. From the results described, I do not see such a connection. Line 76-78: I am not sure how the results here connect "genome and homologous gene expression after polyploidization" to "adaptability enhancement". Surely there is extensive work here describing evolutionary history of the subgenomes and gene expression patterns but the only discussion I can find of the data in relation to adaptation and fitness is lines 301-306, where secreted proteins are reported to have greater variation in expression compared to "other genes" (other genes is not described here, are they all other genes or a specific subset?). "Other" could perhaps be divided into other specific functional classes. Still, how exactly this result demonstrates that hybridization has contributed to an increase in fitness for these species is unclear to me. There is no comparison to a nonhybrid diploid species, nor any measure of fitness. I understand that this particular analysis focuses on variation across subgenomes and would not apply to a nonhybrid diploid but claiming that this result shows "adaptability enhancement" is a stretch. At the very least, discussion on what such variation in secreted protein expression means biologically and in terms of fitness is needed. Also, the related result, Extended Data Fig. 34 is not referenced in text. Are the results related, is that expected? Given that it is sequence divergence perhaps a connection to outgroups could be made. Furthermore, is there enrichment for types of genes/pathways fitting any of the different expression balance/dominance profiles? It seems like there is a lot of potentially interesting biology not being considered here.

The text appears to name a new species, which is potentially very important to understanding this system. As I understand it, Mp represents a previously unknown lineage of Meloidogyne that the authors have named here. I do not specifically have experience in identifying and naming new species, but I think more information about this species is generally needed. Where exactly it was collected and from what? Is it phenotypically distinct from the other species? The collection information for all 5 species in this text is vague. I think more detail is needed on the initial isolation and culturing of these nematodes. Also, the methods related to Extended Data Fig 1 are missing. The legend indicates it is an "ITS rRNA gene" tree, while the text refers to using "ITS rRNA, 28S rRNA, and COI sequences". This incongruity needs to be addressed. Also, ITS rRNA could refer to ITS1 (between 18S and 5.8S rRNA) or ITS2 (between 5.8S and 28S rRNA). Furthermore, there is no information about how the rest of the data was collected, are these BLAST hits on NCBI? I assume some universal primers were used to amplify and sequence these genes from the cultured isolates? There is also no information about how the tree was constructed: alignment, phylogenetic method, statistical support, etc. Based on that tree, Ma and Mp look like the same lineage, as do several other species pairs. Given the complex evolutionary history of these species, which includes hybridization and polyploidy, I do not see how a single gene tree is appropriate to describe a new species, especially when each species is only represented by a single sequence in this tree. Obviously, the authors generated a complete genome for this lineage demonstrating its apparent uniqueness, but I think more discussion of their motivation for doing so based on a single gene tree and negative SCAR results would improve this manuscript. For example, had researchers in the root-knot nematode community suspected another distinct lineage existed? Furthermore, I am a bit weary of declaring a new species in a system with repeated hybridization, ploidy elevations, chromosomal fusions, and karyotype change from one sample. I suspect, based on this text and other literature, that there is a great deal of intraspecific karyotypic variation among these species. I think the authors could expand on their degree of certainty that this is a novel species or part of a more complicated background of hybridization that has not been captured in previous studies.

General methodological comments:

Version numbers for programs should be included and code used for the bioinformatic analyses should be made available.

In some instance "default parameters" are mentioned as used (e.g., for Trinity [Line 721]) and other

times there is no mention of parameters (e.g. CD-HIT [Line 724] – which is think should be CD-HIT-EST for cDNA).

When outside data is brought in for additional analyses (e.g., ITS sequences, draft genomes, mitochondria) the source and method for acquiring these sequences is often not included. For example, regarding Extended Data Fig. 31: what exactly those other sequences are is not clear – are they also short reads that were mapped to the same reference mt genome mentioned in text? How were those accessions selected? There is no Mg – which was analyzed regularly throughout the rest of the text, so I am not sure how this data was collected. Additionally, for this figure, “Tree scale” is not clear (is it substitutions/site?) and there is no statistical support indicated.

Specific comments:

The Abstract does not mention that the species is a nematode; it’s in the title but should be in abstract.

Line 44: “take the inferior positions” – this sentence needs to be reworded because it makes the evolutionary process sound purposeful.

Line 73: This is the first mention of *M. graminicola* (Mg) but no description of this species. It should be clear that this is an outgroup species. It’s reproductive mode and ploidy need to be stated here.

Line 102-103: CEGMA results are reported but perhaps this could be updated to BUSCO and use a new database. For the reads mapping value – is this the short reads and/or the PacBio?

Line 107: The discrepancy in repeat content between the MIG species and Mg caught my attention. I think this result should be discussed and probably analyzed further. What is the history of activity in these species, is there a shared or unique set of expansions in the MIG species? There has been discussion previously about TE activity in these species – summarized in a review on TEs in polyploid animals (Rodriguez and Arkhipova 2018: <https://doi.org/10.1016/j.gde.2018.04.003>) that would be worth looking at. That review also describes the findings from the *Xenopus* genomes, that the *X. laevis* subgenomes were “marked” with distinct TE activity that helped distinguish the homeologous chromosomes. Given the quality of these new genome assemblies, it would be interesting to highlight TE activity.

Line 180: The introduction of *M. floridensis* (Mf) and *M. enterolobii*, and *M. hapla* seems to come out of nowhere. It is not clear what these species are or where this data comes from. Were these assemblies downloaded from NCBI? Why only these species? It appears that *M. chitwoodi*, *M. exigua*, *M. luci* also have genome assemblies available NCBI. Perhaps MIG needs to be better defined here.

Line 262-265: Context is lacking here, how is ~50 mitochondria SNVs across “different species” indicative of a very recent MIG formation? The related methods are also quite vague.

Line 292: What are the 5 life stages? It is not clear here if there were life stages that differed from one another in subgenome expression patterns.

Fig. 5: Why is this figure only “egg period”? Is it representative broadly of the findings for the other life stages? I am also not sure what the two sets of boxes in 5b are.

Reviewer #3:
None

Point-by-point response

Reviewer #1 (Remarks to the Author):

This paper by Ming Sun et al. describes and compares the genomes of five root-knot nematode species assembled at or approaching chromosome-scale resolution. Four of the species have polyploid genomes and reproduce via mitotic parthenogenesis while a fifth species is diploid and reproduces via facultative meiotic parthenogenesis. The authors have achieved an impressive amount of work and analyses, and this has led to enormous progress in the contiguity of genome assemblies for these species. In itself, this represents an important technological achievement and a valuable resource for the whole community of nematologists and evolutionary biologists.

Analysis of these genomes led to the identification of unusual telomere repeats in these species as well as the absence of many telomere-associated proteins usually well conserve in other nematodes.

Furthermore, comparative analyses of the genomes allowed confirming previous findings by different other groups on former more fragmented versions of the genomes: these species are most likely recent polyploid hybrids (Blanc-Mathieu et al. PLOS Genetics 2017, Lunt et al. PeerJ 2014, Szitenberg et al. Genome Biol Evol 2017, Jaron et al. Journal of Heredity 2021).

Overall, many analyses in this paper seems to have been inspired by the previous work by Blanc-Mathieu et al. 2017, with phylogenomics analysis of the genome copies as well as differential gene expression between the gene copies resulting from hybridization, albeit at a much higher resolution due genome assemblies based on a combination of long-read, Hi-C and optical mapping (for some).

The study confirms different expression levels between the different gene copies resulting from polyploidization and suggest a new scenario for the origin of the different genome copies in the species investigated, involving gamete non-reduction and hybridization.

The authors also describe one of the root-knot nematode species they have sequenced as a new triploid species and named it *M. paraarenaria*.

Overall, the different figures and illustrations are of remarkable quality and represent in a synthetic way multiple different complex analysis and I would like to congratulate all the authors for the technical ‘tour de force’.

With that said, despite the impressive amount of data generated and bioinformatics analyses performed, I regret that many of the main conclusions are not supported enough by the data and results of the analyses.

Response: We greatly appreciate and thank the reviewers for providing us with the precious comments on this work which have helps us greatly improved our manuscript.

In this revised version, we have heavily improved our manuscript, the main points are as follow:

1. Made the analysis and description of *M. paraarenaria* more clearly.
2. Added more information about objective evaluation of genome assembly quality.

3. Optimized and added bioinformatics analysis for telomeric element, mitochondrial phylogenetic analysis, genome structure analysis, and genome gene expression pattern.
4. Provided several hypotheses for species formation model, and choosing the best one based on our data.
5. Re-organized the discussion section.
6. Polished the language.

For instance:

- the whole analysis on differential expression levels seems to have been done in the absence of biological replicates. Consequently, the validity of the measured differences remains to be confirmed.

Response: In fact, three biological replicates are generated for each sample. We have added detailed description for methodology in our revised manuscript (**Lines 809-810**).

- the hypothesis that unreduced gametes have led to two of the copies in these genomes is not the sole hypothesis that could explain the observations. Multiple successive hybridization events with more or less closely related species could explain the same pattern and the level of nucleotide sequence divergence between A1 and A2 must be provided to help choosing the most likely hypothesis.

Response: We have added a supplementary figure (**Supplementary Fig. 36**) for several hypothesis that explain the observations, and discussed which one are most likely. We also added supplementary figures for nucleotide sequence divergence (**Supplementary Fig. 34**).

- further evidence than just a phylogeny with no bootstrap and using a single short nuclear marker sequence (ITS) is absolutely necessary before claiming and naming a new Meloidogyne species (*M. paraarenaria*).

Response: We agree with reviewer that our evidence is not sufficient for supporting “*M. paraarenaria*” is a new species. Our study is not aim to claim new species, which is not what we are good at, but found that there are two lineages of triploid and tetraploid in the existing *M. arenaria* (Ma). We suggest a more in-depth taxonomic identification of these two different ploidy Ma. We have adjusted this part, as follows:

1. We do not claim that Mp is a new species and do not make a priori judgments about its phylogeny;
2. We just considered Mp as an unknown species and sequenced it;
3. We added a description of the phylogeny of the current published Ma, which contains two lineages, and Mp represent triploid one;

4. We propose that the tetraploid Ma sample (Red lineage) are still named Ma, triploid Ma sample (Green lineage, include our Mp sample) are named “*M. paraarenaria*” (**Supplementary Fig. 1**).

Moreover, I have several major concerns regarding the methodology used to scaffold the genomes at both stages. At the first stage when *M. graminicola* is used to anchor the genomes of Mi, Mj Ma and Mp while being a distantly related Meloidogyne species. The authors must clarify to what extent this influences the other genome assemblies to resemble too much that of Mg. At the second stage, when aiming at telomere-to-telomere genome assemblies, I also have doubts concerning the methodology which consists in forcing all the chromosomes to start and end with the newly identified telomere repeats. These efforts result at the end in a genome with less chromosomes and which is bigger than expected for Mi, for instance. These points should be addressed and clarified.

Response:

1. The influence of our assembly strategy on assembly v1:
 - a) We added an objective evaluation for assembly version 1 and a description of its purpose.
 - b) The influence does not affect our purpose (estimation of Ks or nucleotide identity), but affect identification of large-scale chromosomal variations.
2. The accuracy of our assembly T2T:
 - a) We provided more clearer Hi-C heatmap (**Supplementary Fig. 21,23-25**).
 - b) Chromosomes that may be over-assembled can be confirmed in synteny analysis (**Supplementary Fig. 31**).
 - c) Size variation factor between the smallest and biggest chromosomes is because of a unique chromosomal translocation in our sample (See Response of Q5).

Furthermore, because many intermediate results are not provided (e.g., different versions of the genome assemblies or their annotations) and some methodological points are not described with enough details, the work is mostly not reproducible, and several statements cannot be verified.

Response:

1. Assemblies and annotations are uploaded in our website (<http://bmb.hzau.edu.cn/sjxz.htm>);
2. Code is available in GitHub (<https://github.com/xiecs-BMB/MIG-genome>);
3. More detailed description for methodology is included, such as parameters, program version and citation.

I also found the discussion quite frustrating as it does not really replace or compare the new findings with previous ones found by other groups studying genome structures and evolutionary history in the same species. Curiously there are also many results that have been produced in the extended data (e.g. on HGT, on histone modification marks, on centromere) that are not mentioned in the results section or not discussed in the discussion.

Response:

1. We have added some sentences for comparing with previous studies in discussion section.
 - a) palindrome structures in previous studies (**Lines 441-446**)
 - b) divergent copies of genome in previous studies (**Lines 470-475**)
2. The description of HGT genes, histone modification and centromere are now included in result section.
 - a) HGT genes distribution and syntenic analysis (**Lines 283-288 and Lines 382-385**)
 - b) Histone modification distribution patterns in relation to telomeres and centromeres (**Lines 279-283**)

Finally, syntax and grammatical English errors are present and pervasive in the whole manuscript from the abstract to the discussion and should be corrected.

Response: The manuscript has been polished in **Nature Research Editing Service**.

For this ensemble of reasons, I would like to recommend a major revision of the manuscript and you will find below more details and suggestions to help improve the manuscript.

Response: Thanks for reviewer's recommendation. These valuable comments are very helpful for us to improve the manuscript. Now we response those point in below.

Major points that absolutely need to be addressed / discussed / clarified

Q1: The description of a 'new' species 'M. paraarenaria' could be very interesting in itself. However, at this stage, the evidence supporting this is a new species are not sufficient. So far, the only evidence provided is that the SCAR markers characteristic of Mi, Mj and Ma do not amplify on the so-called Mp species. However, are other SCAR markers for other Meloidogyne species been tested as well? This could completely be another already described triploid Meloidogyne species belonging to Clade I (sensu Tendingan De Ley et al.). Have Esterase profiles been determined? This is also a frequently used technique to distinguish the different Meloidogyne species. Before claiming this is a new species, several molecular tests for species identification should be done and show consistent results. Furthermore, why was the name M.

paraarenaria given? Absolutely no explanation is given for this name in the main text. I understood that this was given according to the phylogenetic position (close to *M. arenaria* but with no support value) in Extended Data Fig. 1, based on ITS marker. However, first it is surprising to have only one ITS sequence per polyploid genome while most of the rest of the paper clearly explains that most of the genome regions exist in 3 to 4 copies and with high divergence between A and B copies. Could the author justify / explain why only one single ITS sequence per species was used and which copy? Second, using the ITS marker alone is not enough and multiple concatenated nuclear markers as well as mitochondrial markers in a phylogeny with bootstrap values should be used like in Álvarez-ortega et al. Scientific Reports 2019, for example. Would using another marker gene completely change the topology shown in this figure? So far, there is not enough convincing evidence to support this is a new species and the authors should reinforce this point.

Response: The reviewer's concern is that our evidence is not sufficient for supporting "*M. paraarenaria*" is a new species.

As said above, our study is not aim to claim a new species, which is not what we are good at, but found that there are two lineages of triploid and tetraploid in the existing *M. arenaria* (Ma). We suggest a more in-depth taxonomic identification of these two different ploidy Ma.

To make it clearly, we assembled the mitochondrial genome of our samples and constructed a mitochondrial phylogenetic tree with all published data (**Supplementary Fig. 1**). This phylogenetic tree is constructed from all mitochondrial genes, and is therefore more reliable than our previous marker gene tree.

1. Mitochondrial tree shows that Mp is located inside Ma samples and may be the same species of the published Ma sample such as MaraL32 and MaraHarA (**Supplementary Fig. 1**);
2. Mitochondrial tree also suggests that currently naming of Ma might be not a single species, but with two lineages (**Supplementary Fig. 1**);
3. To further investigate whether currently naming of Ma contains two species, we mapped reads of all published Ma data into our T2T assembly and counted the normalized reads depth from subgenomes. This analysis reveals that red lineage is proposed triploid with B2 subgenome lacking, and green is tetraploid, which confirmed that current naming Ma is not a single species but with a tetraploid species and triploid species (**Supplementary Fig. 2** and **Supplementary Fig. 39**);
4. Given the ploidy estimation and the mitochondrial phylogenetic tree, Mp may be the species of triploid Ma sample.

Based on our new information we have adjusted the description of Mp definition as follows:

1. We do not claim that Mp is a new species and also do not describe too much on the species identification of Mp;
2. We just considered Mp as an unknown species at beginning and sequenced it;
3. We added a description of the phylogeny of the current Ma, which contains two lineages, and Mp represent triploid one;
4. We propose that the tetraploid Ma sample (Red lineage in **Supplementary Fig. 1**) are still named Ma, triploid Ma sample (Green lineage in **Supplementary Fig. 1**), including our Mp sample, are named "*M. paraarenaria*".

The updated result can be found in **Lines 122-132** and methodology can be found in **Lines 911-922**.

Q2: I have an important concern about the assembly strategy for MIG species according to synteny and collinearity with Mg. This strategy will obviously allow assembling further MIG contigs and scaffolds but not based uniquely on overlap between reads or Hi-C contact data. Using such a strategy might influence MIG genome assemblies to resemble that of Mg but does not necessarily correspond to a biological reality. If major structural variations or chromosomal fusions and rearrangements occurred separately in the different branches these might be underestimated and overlooked by forcing the MIG contigs to arrange together in a colinear manner with Mg contigs. This problem is amplified by the fact that (i) MIG species are mitotic parthenogens with a lot of genome rearrangements to be expected and (ii) Mg is quite distantly related to the MIG species (there are more closely related diploid mitotic species such as *M. hapla* for instance). I wonder how the use of JCVI suite to anchor MIG scaffolds on Mg scaffolds would over-influence the assembly towards Mg. Could the author comment and clarify this point?

Response:

We agree with reviewer that our strategy (assembly v1) will influence MIG assembly. The degree of influence depends on the purpose of the assembly version.

1. The aim of assembly v1 is identifying the genome structure of MIG and to prepare for the subsequent construction of the chromosome genome;
2. Errors mainly reflect the position of the contigs in the scaffolds but not the sequence within the initial contigs. Therefore, the mis-assembly does not affect our estimation of Ks or nucleotide identity but have a greater impact on identification of large-scale chromosomal variations;
3. Assembly v1 does not absolutely depend on collinearity with Mg, it is just as an auxiliary information to build scaffolds with Hi-C data. Therefore, the accuracy of assembly v1 isn't that bad, many scaffolds are assembled completely correct and reach at chromosome level;

4. The second-stage assembly does not depend on collinearity with Mg, all of errors are corrected in our T2T genomes;
5. *M. hapla* is indeed more suitable as a haploid reference because it is more closely related to MIG, but unfortunately, our laboratory lacks this sample, so we chose Mg genome as the haploid reference. This did not affect the accuracy of our final genome assembly;

To clarify the accuracy and aim of assembly v1, we have added a sentence in revised manuscript (**Lines 145-146, Lines 148-150**).

Q3: It would be informative to provide intermediate assembly results in addition to assembly version1 results. For instance, how many scaffolds were obtained by scaffolding with Hi-C data without using Mg as a guide? What were the N50 values for these intermediate versions?

Response: Now we provided the statistic information for intermediate assembly results.

	M. incognita		M. paraarenaria		M. arenaria		M. javanica	
	N50	Scaffolds number	N50	Scaffolds number	N50	Scaffolds number	N50	Scaffolds number
Contig	2.3Mb	379	2.5Mb	331	2.4Mb	604	3.0Mb	412
Bionano improved contigs	3.6Mb	295	-	-	-	-	-	-
Contigs+Hi-C(3D-DNA)	10.6Mb	263	8.8Mb	288	9.4Mb	527	10.0Mb	418
Contigs+Hi-C(3D-DNA) + collinearity with Mg	4.4Mb	48	4.1Mb	50	4.3Mb	67	4.1Mb	64

Scaffolds constructed only with Hi-C are higher than our assembly v1. The reason for the inflated N50 is:

1. Due to MIG contigs contains homologous chromosome sequence which are of high similarity in partial region. Those homologous chromosomes sometimes show a strong Hi-C contact signal, and software may cluster those homologous chromosomes into one group.
2. Complex chromosomal fusion also generates a strong Hi-C contact signal, and it is difficult to distinguish the signal are from homologous chromosome or from fused chromosome.

Therefore, we finally discarded the assembly with Hi-C data and without Mg as a guide. Most of the genome assembly work is done in the Juicebox by manual construction. We have provided the “.assembly” file (in GitHub), which contains the information of contig position, and can restore the results of Juicebox.

Q4: An additional point that casts doubt on the validity of using Mg to anchor contigs of the other Meloidogyne species is the initial impossibility to identify the G-rich

telomeric repeats in these scaffolds while an analysis at the contigs level allowed identifying them.

Response: We agree. Actually, in assembly v1, many contigs containing telomeres are indeed mislocated when using this strategy and we fail to identify the telomeric repeats in MIG. However, this does not mean that all contigs are mislocated and there are also some well-assembled contigs that almost reach at chromosome level. Our telomeric repeat of MIG are finally identified from these **well-assembled scaffolds** instead of **all scaffolds**.

Q5: Another major concern is the methodology used to obtain ‘telomere to telomere’ genome assemblies. I wonder to which point, by forcing all scaffolds to start and end by the identified telomeric repeats, the authors have over-assembled the genome. Indeed, the now so-called T2T assembly of *M. incognita* has a lower number of chromosomes (40) than the numbers reported in the literature. Indeed, although Triantaphyllou reported 40-46 chromosomes by cytological observations in the 80’s, more recent observations (Despot-Slade et al. Mol Biol Evol 2021) count 46 chromosomes with more modern visualization and staining technology. In the same paper by Despot-Slade et al., the authors indicate that the chromosomes range in size between 0.4 and 1.5 μ m which corresponds to a size variation factor of 3.75 between the smallest and biggest chromosomes. However, in the Mi genome assembly presented here (Fig. 3) there is a size variation factor of 7.5 between the smallest and biggest chromosomes.

Response:

The reviewer raised two questions, one about the number of chromosomes and one about size variation factor between the smallest and biggest chromosomes.

The accuracy of our T2T assembly can be validated by Hi-C heatmap and collinearity between Mg genomes. Here are a few points to prove:

1. The karyotypes of MIG species exhibit a huge difference among strains (for Mi about 40-46). Although Despot-Slade reported a strain with 46 chromosomes, chromosome number in our assembly are also in the range.
2. If two real chromosomes are incorrectly assembled into one chromosome, there will be a break in the Hi-C signal (**Supplementary Figure 21, 23-25** and **Response in Q7**).
3. Our subsequent analysis revealed the fusion of each chromosome (**Figure 3d** and **Response Figure 1a**). If there are some mis-assembled chromosomes, we can further check it. For instance, in Mi, Chr1-Chr3 derive from chromosomal fusion of three original chromosomes (orange dot in **Response Figure 1a**), Chr4-Chr12 derive from chromosomal fusion of two original chromosome (purple dot), one original chromosome is spited into two chromosome (black box, Chr02 and Chr12), and one chromosomal translocation (black arrow in **Response Figure 1a**, Chr14 and Chr40, **Response Figure 1b,c**). This chromosomal translocation produces two

chromosomes, Chr14(5.8Mb) and Chr40(2Mb), which is the smallest one. In summary, 40 chromosome in our assembly could be interpreted as :

$$40 \text{ chromosomes} = 18 \times 3 (\text{expected number of triploid}) - 3 \times 2 (\text{fusion with three chromosome}) - 9 \times 1 (\text{fusion with two chromosome}) + 1 (\text{chromosome split}) = 40$$

Size variation factor between the smallest and biggest chromosomes are 7.5, which is because:

1. The smallest chromosome derives from a translocation event.
2. Given that the highly variable karyotype of Mi, the smallest chromosome generated from translocation may not be present in other samples.

Thus, if exclude this chromosomal translocation, the smallest should be Chr_39 (2.9Mb) and the size variation factor should be 4.9, which is not significantly different from the literature (3.75).

Moreover, considering the highly variable karyotype, regardless of chromosome number, the fused chromosomes will eventually be spited, and this will not affect our result. Karyotype instability in RKNs is a very interesting question, we are working on another research of *M. incognita* pan-genome, which will address the question about karyotype variation between different individuals.

Response Figure 1. Chromosomal fusion and translocation in Mi T2T genome.
 (a) Origin of 40 chromosomes in Mi. Orange dots represent current chromosomes from the fusion of three ancestral chromosomes. Purple dots represent current chromosomes

from the fusion of two ancestral chromosomes. Black box represents current chromosomes involving ancestral chromosomes splitting. Black arrows represent one chromosomal translocation event in Mi. (b) Syntenic analysis between chromosomes involved in Mi translocation event and corresponding chromosome in Mg.(c) Scenarios for chromosomal translocation event in Mi.

Q6: This is also reflected by the bigger genome assembly sizes (Table 11) that now falls out of the range estimated by flow cytometry in (Blanc-Mathieu et al. PLoS Genetics, 2017). Indeed, estimated genome size for Mi was 189 (+/-15Mb) and the new assembly size is now >225 Mb. Thus, unless substantial variations in genome size exist between Mi strains across the world, these metrics cast doubt on the genome assembly validity. This possibility could be tested by performing k-mer based estimation of genome size using Jellyfish and GenomeScope 2. This is surprising that these measurements have not been done or provided in this paper.

Response:

1. We have performed the k-mer analysis and added a supplementary table for result (**Supplementary Table 3**). We estimated the ploidy and genome size of these samples with k=21 using Illumina reads, corrected PacBio reads, and corrected Nanopore reads (Only for Mi), respectively.
2. From this analysis, haploid genome size of Mi is about 70Mb, and whole genome size for our sample is about 210Mb.
3. The estimated genome size is consistent with that of assembly v1 (213Mb), but slightly lower than that of T2T assembly(225Mb). The larger genome size for T2T assembly may be due to the use of ONT contigs during chromosome construction, where 213Mb in PacBio contigs and 225Mb in ONT contigs, which are mainly caused by the assembly software but not artificial.

Q7: Finally, the Hi-C contact maps also cast doubts on the validity of these new T2T assemblies. Indeed, while Extended Data Fig. 3 presented rather smooth contact maps with, as expected, most of the strong signal present on the diagonal, the new contact maps (extended data figs. 18, 20-22) are more noisy with a lot of strong signal away from the diagonal suggesting extensive duplications of 100% identical scaffolds and some mis-assemblies that would deserve to be manually curated.

Response: The reviewer's concern is that the Hi-C heatmaps of assembly T2T are more noisy and contain 100% duplicated region. In this revised version, we have provided new Hi-C heatmaps and explain the questions as follows:

The noisy in heatmap is due to reads of multiple alignments between homologous chromosomes. Hi-C matrix in our original manuscript is generated with allowing multiple mapping (MAPQ ≥ 1), which results in the strong signals between homologous chromosomes. These noisy could be removed by adding parameters of removing multiple mapping (MAPQ ≥ 10).

When plot Hi-C heatmap by removing multiple mapping reads, many blank areas appear on the heatmap, which is because:

(Recovery of collapsed sequence → 100% duplication region → Reads multiple mapping → blank areas in Hi-C heatmap)

1. Canu software will generate collapsed sequence when heterozygosity less than 2%. The average depth of collapsed sequences is twice as that in whole genome.
2. To achieve completely chromosome, this part of the sequence is used twice during chromosome construction.
3. Recovery of collapsed sequences will introduce some 100% duplication region, which results in many reads are multiple mapping.
4. Multiple-mapping reads will show a blank signal when using MAPQ ≥ 10 parameter, but show a normal signal when using MAPQ ≥ 1 parameter.

Notably, this only influence a few homologous chromosomes, and only in those between A1 and A2, leading an underestimation of nucleotide divergence between them, and will not influence other chromosomes. Moreover, this loss of genetic information almost exists in Mi, with few collapsed regions in other MIG members. This may suggest that the AA subgenome of Mi undergo more loss of heterozygosity (LOH), compared to the other three polyploid species. Solving this question may need more accurate sequencing data, such as PacBio HiFi reads.

We have removed those collapsed regions in further analysis, such as estimate nucleotide divergence and gene expression analysis. We also provided the Hi-C heatmaps with and without multiple mapping for four polyploid species (**Supplementary Fig. 21, 23-25**).

Q8: I also wonder how circular is the analysis of ancestral karyotype reconstruction. Again, by anchoring everything to the Mg genome, which counts 18 chromosomes, wasn't it completely expected and mandatory to infer the ancestral karyotype was composed of 18 chromosomes? How dependent to the karyotype of the meiotic genome used as an anchor is this methodology? For instance, using M. hapla, which counts 16 chromosomes as an outgroup species instead of Mh would have led to an ancestral karyotype of 16 chromosomes? Could the authors comment this point?

Response: The reviewer wants to know the logic of constructing the 18 ancestral chromosomes for MIG. In this revised version, we have added the entire process for constructing the 18 ancestral chromosomes.

To make it clearer, we have reorganized this part and added a more accurate genome of Mg (**Mg_T2T, See Response in Q50**). In this part, we aim to reconstruct the ancestral genome of MIG.

1. Based on our previous results, we know that chromosomes of Mi are derived from fusion of ancestral chromosome. To target chromosome fusion pattern, we performed the syntenic analysis between Mi and Mg. This result shows that which current chromosomes are derived from independent ancestral chromosomes and

which one from fusion of ancestral chromosomes. We also notice that there are some chromosomal variations between Mi and Mg (marked in black triangle, **Fig. 3c**), which indicate that Mg could not represent the ancestral genome of MIG very well. Moreover, those chromosomal variations are shared in three copies of Mi homologous chromosomes, indicating that there is a subset of sequences could represent Mi ancestral genome well.

2. To this end, we attempted to extract some chromosomes or chromosome segments from the Mi genome to represent Mi ancestral sequences. When using the haploid genome as a guide, those chromosomes or chromosome segments without chromosomal variation can be easily identified. For those chromosomes or chromosome segments with chromosomal variation, we need to be careful to determine this. This analysis does not strictly dependent on haploid genomes, this only changed the chromosomal variation pattern between MIG and haploid genomes, and we can always find this 18 chromosomes or chromosome segments.
3. The number of Mi ancestral genomes was 18 which was indeed to be expected as this was reported in previous studies, what we emphasize here is what those 18 chromosome sequences are.
4. In fact, many chromosomes or chromosome segments can represent ancestral chromosomes, for example, Chr_17, Chr_18 and Chr_19, they are the same to some extent. For the convenience of subsequent analysis (such as calculation of Ks or phylogeny), we selected the chromosomes belonging to B subgenome as the ancestral sequence. We eventually identified 8 chromosomes and 10 chromosome segments to represent the ancestral chromosome (Mi_B). Subsequently analysis between Mi_B and MIG genome also proved that Mi_B is very suitable as an ancestral sequence for MIG with few chromosomal variations (**Fig. 3e**).

In revised manuscript, we added some mark in **Fig. 3c** to illustrate that why Mg could not represent ancestral genome very well and added a more intuitive exhibition about chromosome variation between Mg and Mi ancestral genome (See revised manuscript **Lines 290-299** and **Fig. 3e**).

Q9: It should be clarified as soon as possible in the manuscript that A1 and A2 subgenomes cannot be distinguished and that only the B genome can be correctly separated based on the Ks analysis. This must be said much earlier than currently (L221) and as soon as A1 and A2 subgenomes are mentioned.

Response: Thanks, we now have moved this sentence to **Lines 180-181** to make it clear that A1 and A2 subgenomes cannot be distinguished and that only the B genome can be correctly separated based on the Ks analysis.

Q10: L238-240. This part concerning whole genome heterozygosity estimated from k-mer vs. 'sequence divergence' is misleading. K-mer heterozygosity gives an estimate at the whole genome level regardless of whether or not sequences are coding or non-coding. In contrast numbers extracted from table 16 only concern coding sequences and

thus these values cannot be compared or put together in a same sentence because they refer to different components of the genome. This point stresses out the importance of providing an estimate of the nucleotide sequence divergence level between A1, A2 and B genomes not only in the coding regions but at the whole chromosome scale.

Response: Thanks for raising this point. This is really a misleading sentence and we now have removed this. We originally want to provide more evidence about genome structure for *Mf*. Now this part has been rewritten (See **Response in Q28 for *M. floridensis***).

We have provided information of nucleotide identity between those subgenomes (**Supplementary Fig. 14** for assembly v1 and **Supplementary Fig. 34** for assembly T2T in revised manuscript).

Reproducibility and FAIR principles:

Q11: The authors should provide all the genome assemblies (intermediate and final) as well as all the annotations to ensure reproducibility of the results they present and optimal reuse of the data they have produced for the whole community.

Response: Thanks, we have uploaded intermediate file, script, and some command record into GitHub (<https://github.com/xiecs-BMB/MIG-genome>). Assembly and annotation are available in (<http://bmb.hzau.edu.cn/sjxz.htm>).

Other important points that need to be addressed / clarified

Abstract / Introduction

Q12: The terminology MIG for ‘*M. incognita* group’ makes no sense from a phylogenetic or taxonomic point of view. It would be more correct to refer to ‘tropical’ or ‘mitotic parthenogenetic’ or ‘Clade I’ (sensu De Ley et al.) root-knot nematodes.

Response: Thank, we have removed “MIG” terminology and used “Clade I RKNs” or “poly-ploid RKNs” terminology for those root-knot nematodes.

Q13: L40. Most animals indeed use outcrossing via sexual reproduction, but many plants and fungi use self-fertilization or parthenogenesis.

Response: Thanks, indeed many species use self-fertilization or parthenogenesis, but they are often not obligate. We here mean obligate parthenogenetic species that only use parthenogenesis without recombination, and we had adjusted this description (**Lines 41-43**).

Q14: L62-64. There is probably a confusion here. HGT events have indeed probably participated in adaptation to plant parasitism but not specifically in MIG species. This is a common feature of all plant-parasitic nematodes whether or not they have parthenogenetic reproduction. In the same way, gene duplications, genes copy losses or movements of transposable elements all participate in genome plasticity but whether these ‘mutations’ in a broad sense have any specific impact on the adaptability of these species remains to be demonstrated. This remains so far hypothetical.

Response: Thanks, we have adjusted this description. We adjusted the motivation of the Introduction to elaborate more on the hypothesis of the origin of polyploid RKNs (**Lines 78-91**).

Results

Q15: L87-88. *M. graminicola* is a facultative meiotic parthenogenetic species. I think it is important to clarify for the non-specialists that this species can do both sexual reproduction and meiotic parthenogenesis.

Response: Thanks, we have included this description (**Line 119**).

Q16: L91-94 Important to clarify here that 44Mb for Mg corresponds to the haploid genome size where the two homologous chromosome sets have been collapsed while for the three other species (Mi, Mj and Ma) the assembly sizes are consistent with separation of the homologous chromosomes during assembly.

Response: Thanks, we now have added sentence that the genome size corresponds to haploid genome size or whole genome size (**Lines 135-138**).

Q17: L101-102. In addition to CEGMA, which is an old software not maintained anymore, I would highly recommend using BUSCO as well and comparing the results in a supplementary table with all the other *Meloidogyne* species genomes available.

Response: In fact, we have performed BUSCO analysis for genome completeness assessment of RKNs, while the values obtained are not shown because they were not significantly different from previous genome versions.

We evaluated genome completeness using BUSCO (v5.4.3) and nematoda_odb10, resulting in about 61.8%-70.7% for our assembly and 40.3%- 69.9% for other available *Meloidogyne* genome.

We noticed that BUSCO value is not high for *Meloidogyne* species, possibly because nematoda_odb10 are built from proteins of *Trichinella spiralis* (Clade I), *Brugia malayi* (Clade III), *Necator americanus* (Clade V), *Loa Loa* (Clade III), and *Caenorhabditis elegans* (Clade V), while *Meloidogyne* are from Clade IV.

Some studies use BUSCO (v3.0.2) and eukaryota_odb9 to generate a higher BUSCO value, when using those versions, we get about 89.5%-97.4% for our assembly and 68.7%-96.7% for other available *Meloidogyne* genome.

We have added this part of the results to the manuscript (**Lines 154-155**) and provided the BUSCO results of the newer version (v5.4.3) (**Supplementary Fig. 5**).

Q18: L107-108. Results on repetitive sequences content are interesting but would deserve a more detailed description like the relative contribution of canonical transposable elements (potentially active) and other repeats. Furthermore, in Supplementary Table 4 it would be more useful to the readers to provide (retro)transposons names rather than their EDTA codes, according to: (https://github.com/oushujun/EDTA/blob/master/util/TE_Sequence_Ontology.txt).

For example, DNA/DTA = hAT_TIR_transposon; DNA/DTA = also hAT_TIR_transposon. Then, based on this EDTA codes that refer to a same parent category, the lines should be grouped. Once this is done, DNA transposons should be separated from retrotransposons more clearly. Moreover, these results are not further discussed in the rest of the paper. It seems that the polyploid hybrid genomes have a higher proportion of repetitive elements than Mg. Interestingly similar findings were reported in (Blanc-Mathieu et al. 2017, PLoS Genetics), when the genome of *M. hapla* (diploid, meiotic) was compared to those of *Mi*, *Mj* and *Ma* (polyploid, mitotic).

Response:

We have added more detailed description for transposable elements (TEs) in our T2T genomes (**Supplementary Fig. 40, Lines 374-382**). Analysis added are as follows:

1. Landscape of TEs.
2. Full-length LTR retrotransposons analysis.
3. TEs specific in A or B subgenome.

All of TE names are corrected using their names rather than their EDTA codes, and updated table and figure are shown in **Supplementary Fig. 7** and **Supplementary Table 6**.

We also discussed the TE content of polyploid species in our results section (**Lines 374-382**).

1. Some TEs are still ongoing.
2. Most of TEs are old, and shared in A and B subgenomes, which may suggest that the TE accumulation occur in diploid ancestor instead of burst after hybridization.

Q19: L116-122. Results on phylogenetic relationship between the homologous gene copies are very interesting and indeed clearly suggest that two copies are more closely related to each other than the third copy. However, this does not automatically imply an AAB structure. This completely depends on the sequence divergence level between the two 'A' copies. Although the divergence is higher with B, it seems to already be quite high between 'A1' and 'A2'. Could the author provide some information about the % of nucleotide divergence between A1, A2 and B?

Response:

We have added the nucleotide identity analysis between three scaffolds belonging to A₁, A₂ and B. A table have been added into **Supplementary Fig. 10c** in revised manuscript. The nucleotide divergence between A1 and A2 are about 5%, and those between A and B are about 10%. Therefore, we still define those three copies as AAB, not ABC. We also provide the nucleotide identity between all subgenome for assembly v1 (**Supplementary Fig. 14** in revised manuscript).

Q20: Extended Data Fig. 10 as most other figures of this paper is remarkably well done and interesting. It seems that the first triplet of homologous sequences in Mi does not present any synteny relationship in the other 4 Meloidogyne genomes. Is this genomic region in 3 copies unique to Mi?

Response:

These copies are not unique to Mi, which is due to the visualization method. We state as follows:

1. This analysis here are just performed between two adjacent subgenomes, which makes it appear that some chromosomes lost.
2. In assembly v1, some fused chromosomes are split, and some are not, which will result in the errors in subgenome assignment. According to syntenic analysis between Mp and Mg (**Supplementary Fig. 8**), the corresponding chromosomes in Mp_A1 is fused to another chromosome belonging to other subgenome. These collinearity relationships look not perfect, the aim of assembly v1 is not to reveal the chromosomal variation between subgenomes but to demonstrate that most genes maintain collinearity in 14 subgenomes, ensuring the accuracy of the estimation for nucleotide divergence or Ks.
3. All of errors are corrected in our subsequently T2T assembly, and similar analysis are performed in these T2T genome (**Supplementary Fig. 32**), which indicate that those subgenomes are of high similarity at chromosome scale.

Q21: L141-142. Please provide a reference for the classification of nematodes in Clades I – IV. Several classification systems exist, one by Hans Helder and collaborators that defines 12 Clades with Arabic numbers (1-12) and one defined by Mark Blaxter and collaborators that defines 5 Clades with Roman numbers (I- V). It seems the authors have used the classification proposed by Mark Blaxter et al. and this must be acknowledged.

Response: Thanks, we now have included the reference and added description in legend.

Q22: L149. I suggest using the Latin species names (*Drosophila*, *Anopheles*) rather than Fly and Mosquito, as it is unsure all flies and all mosquitoes have alternative mechanisms for telomere lengthening. This is more correct to refer to the exact species.

Response: Thanks, we have corrected.

Q23: L151 and continued. These findings on unusual telomeric repeats in the Meloidogyne species analyzed is particularly interesting. Could the author explain a bit more what are the typical characteristics of G-quadruplex structures for non-specialists and how they have inferred it was the case in Mg?

Response:

Typical characteristics of telomere is G-rich, and G-quadruples are due to the sequence being G-rich and thus have the potential to form it. We state as follows:

1. Most eukaryotes telomeric DNA are enrich in G and have potential to form G-quadruplex structures. There is evidence that G-quadruplexes can perform capping function in yeast, but not have not yet been examined in other organisms (Bryan TM. *Molecules*. 2020).
2. The core evidence determining this repetitive sequence is a telomere is the distribution and specific orientation at the end of the chromosome, which makes it different from other transposable elements, and the enrichment of G-quadruplexes is more because of telomeric sequence is rich in G. Whether G-quadruplexes function in Mg is still unknow.

We adjusted this sentence and added a reference in our manuscript (**Lines 210-213**).

Q24: Concerning the unusual telomeric repeats it is unclear from the text whether Types I, II and III presented in supplementary table 9 belong respectively to Mi, Mj and Ma for instance? Or are the three types found in the three species. If yes, do they co-occur on the same regions or different types are on different contig ends? Extended Data Fig. 16 suggests all three types are present in the three species but Type I being preferentially present in Mi. This is not clearly explained now in the text and these interesting results should be discussed. Also, it is not self-evident for non-specialists how extended Data Fig. 16 suggest recombination events between telomeric regions? Could the author further explain how this figure supports these conclusions?

Response: The reviewer wants us to clearly show the distribution pattern of telomeric repeat, and interpret the findings. In this revised version, we have added a figure to show this and added more descriptions of the findings.

We thank reviewer for careful review and concerns about those telomeric sequence. To make it clearer, we have re-organized this part as follows:

1. We optimized methodology for comparison of those telomeric repeats. The original analysis is based on phylogenetic tree. In fact, those telomeric repeats exhibit a high heterogeneity in both sequence and length, many gaps are generated when perform multiple sequence alignment for all of these, which may not be suitable for constructing a phylogenetic tree. We decided to remove the previous description (Types I, II and III and those phylogenetic tree) and employed a more suitable methodology.
2. We added new results for features of those telomeric repeats. The sequence of telomere is very complex, it is difficult for us to determine the start and end of each telomeric repeat unit. Since these sequences have a relatively conserved region of 66bp, we simply defined the complete telomeric element as two adjacent 66bp intermediate sequences. To show the difference between these telomeric repeats, we firstly extracted 9318 complete telomeric elements from four genomes and cluster them use CD-HIT-EST with -aS 0.8 -g 1 -sc 1 -sf 1 -c 0.9 -I parameter. This analysis will group the telomeric elements with more than 90% sequence similarity and more than 80% coverage into one cluster (**Supplementary Fig. 19b**). We

totally obtained 823 cluster, 193 of which contain more than 10 telomeric elements. This result indirectly reflected that telomeric repeat are of high sequence divergence. We also show the distribution of length for those telomeric elements (**Supplementary Fig. 19a**). Given that clustering is performed from all telomeric element of four species, and we next investigated the distribution pattern among different species or chromosomes. Some clusters are present in four species, while some are present in specific species, such as Cluster 0, Cluster1, Cluster2 and Cluster5 (**Supplementary Fig. 19c**). Even for a particular contig, there are more than one cluster of telomeric elements (**Supplementary Fig. 19d**).

3. We added more discussion for lengthening of those telomeric repeats. We attempt to propose a reasonable explanation for this phenomenon. To this end, we survey the research from organism that without telomerase such as some ALT (alternative lengthening of telomeres) cell line. In brief, ALT cells use a homologous recombination-based mechanism to lengthen telomere. This mechanism begins with DNA damage in the telomeric region, followed by telomere lengthening by homologous recombination using telomeric regions of other chromosomes as templates. We speculate that the presence of different telomeric elements on the same contig may be a result of recombination between different chromosomes. Our conclusions about telomeres here are just speculative and not definitive.
4. It is also worth mentioning that the telomeric region is composed of many repetitive sequences and genome assembly may have an impact on this region, thus limiting our accurate in-depth study of this issue. Addressing this issue need more techniques such as HiFi data, FISH, and more appropriate analysis methods. The aim of this paper is not to reveal the mechanism of telomere lengthening, but more to use telomeres as evidence for genome assembly. We interpreted the finding as much as possible, and a more in-depth study on telomeres may require a separate paper to complete.

In revised manuscript, we have added description for the features of telomeric element (**Lines 240-251**) and more detail of methodology (**L1086-L1091**).

Q25: Extended Data Fig. 23. This is indeed very interesting (and satisfying) to observe that the average normalized coverage seems to be uniform across the chromosomes. However, it appears that the standard deviation is much higher in Mi than for the four other Meloidogyne genomes. Do the authors have any hypothesis to explain this striking difference?

Response: We reason that in two points:

1. According to our Hi-C heatmap and similar analysis in assembly v1, Mi appears that containing more low heterozygosity regions in AA subgenomes, which cannot be phased assembled using CANU.
2. Parameter of CANU (corOutCoverage=200 "batOptions=-dg 3 -db 3 -dr 1 -ca 500 -cp 50") is recommended for PacBio data, while we used it for ONT data.

These factors may result in collapsed region or over-assembled region, which will influence normalized reads depth.

Q26: L227-229. On the origin and evolution of MIG. It is unclear what the authors mean by diploid topology in A and haploid topology in B. In Figure 4a it seems that one duplication of A is ancestral to Mj and Ma while Mi underwent a species-specific duplication of A. In contrast, each B genome from each species shows indeed a different evolutionary history with no clear species-specific duplication of group-specific duplication. Is it what the authors mean for A and B presenting respectively a diploid and haploid topology?

Response:

The diploid topology refers to the expected topology when A₁ and A₂ subgenome is assumed to be derived from diploid unreduced gametes AA.

The haploid topology refers to the expected topology when B subgenome is assumed to be derived from haploid gametes.

To make it clearer, we adjusted this part, please see Response in Q27.

Q27: L233-235. Why is the previously proposed hypothesis of a hybridization between closely related species never considered or evoked for the origin of A1-A2 subgenomes. What is the level of nucleotide divergence between A1 and A2? If this divergence is high, albeit lower than B, another possible explanation would be two rounds of hybridization for Mi, one with a more closely related species (giving A1 and A2), another one with a more distantly related species giving B. Likewise, for Mj and Ma a common ancestral hybridization between two relatively close species could have given the exact same topological pattern. Hence, it would be interesting that the authors consider, discuss, and test this hypothesis too. For these reasons it is really important to provide an estimate of the level of nucleotide divergence between A1 and A2 at the whole genome level and not only for coding regions (Ks). Then, putting this divergence in relation to the very low divergence at the mitochondrial level might help clarifying which scenario is the most realistic.

Response: The reviewer wants us to consider several hypotheses and choose the best one according to nucleotide divergence and other evidence. We have re-organized this part by discussing several possible mechanisms.

1. We added a supplementary figure to show the mechanism that led to formation of triploid species (**Supplementary Fig. 36**). In **Supplementary Fig. 36a**, we propose four possible mechanisms. Scenario **a (1)** and scenario **a (2)** describe the formation of triploids by interspecific hybridization between unreduced gametes AA and normal gametes B. Scenario **a (3)** describes the formation of triploids by interspecific hybridization between reduced gametes AA from tetraploid AAAA and normal gametes B. Scenario **a (4)** describe the other possible mechanisms but is more complicated and less likely than the first three. Diploid species with mitosis are common in root-knot nematodes, while tetraploid nematodes with meiosis have

not yet been reported. Thus, the most likely mechanism of formation of triploids is hybridization between reduced gametes AA and normal gametes B.

2. The B gametes are clearly generated from meiotic, so the question is how the AA gamete of Mi and Mp is formed. We assume that Mi and Mp are formed in the same mechanism and the situation would be too complicated if their mechanisms are different. We also added a figure to show our hypothesis (**Supplementary Fig. 36b**). Scenario **b (1)** describes that AA of Mi and Mp are from a clonal diploid species. Scenario **b (2)** describes that AA of Mi and Mp are from a hybrid species between A1A1 and A2A2. Scenario **b (3)** describes that AA of Mi and Mp are from two independent hybridization. We also exhibit the corresponding expected tree structure of those three scenarios. Considering that hybridization is a small probability event, the probability should be **b (1) > b (2) > b (3)**.
3. While only scenario **b (3)** is consistent with the observed tree structure. Divergence between homologous chromosomes (Mi A1 vs Mi A2) are smaller than between species (Mi A1 vs Mp A1) in observed tree, which is contrary to the expected topology of **b (1)** and **b (2)**. A possible explanation is that AA gametes underwent an extensive gene conversion after generation or hybridization, resulting in loss of heterozygosity between homologous chromosomes (**Supplementary Fig. 36c**). Another possible explanation is that, as mentioned in Lunt, David H et al. (PeeJ 2014) and Handoo, Z A et al. (Journal of nematology 2004), Mf (A diploid root knot nematodes) employ an automictic reproduction mechanism called first-division restitution (FDR) (**Supplementary Fig. 36d**). To our knowledge, FDR is a mechanism can generate unreduced gametes with parental heterozygosity maintained, while if crossover occurs, FDR will result in loss of heterozygosity (Brownfield L, Journal of experimental botany 2011). If AA ancestors use this mechanism for reproduction, divergence between homologous chromosomes are expected to be consistently reduced, which would also alter topological patterns. We noticed that the nucleotide divergence between A1 and A2 are about 5%, and this level of divergence may be caused by long-term parthenogenesis or from interspecific hybridization so that we could not determine that AA are from clonal diploid species or hybrid species. Considering the small probability of hybridization event and above evidence, we believe that AA are most likely from a common diploid species with clonal reproduction. In fact, the evidence is limited, and weak, other mechanisms cannot be excluded.
4. For the formation of tetraploid AABB, there are also various mechanisms with more complex situation. We found that subgenome AAB of tetraploid Ma and Mp show a high similarity (Ks about 0, and nucleotide identity about 100%) with those in Mp. This raising a possible scenario that tetraploid AABB are formed by interspecific hybridization between unreduced AAB gametes with normal gametes B. Compared with the more common scenario that AABB is formed by the fusion of AA and BB, that scenario is more likely because fewer hybridization events are needed.

5. Mitochondrial genome analysis (**Supplementary Fig. 1**) shows that these hybridization events, including species divergence with other species (*M. luci* and *M. floridensis*), are recent.

Our re-organized part can be found in revised manuscript **Lines 328-356**.

Nucleotide divergence estimation between those homologous chromosomes can be found in **Supplementary Fig 34**.

Q28: L241-242. The authors seem to consider *M. floridensis* as a diploid A1-A2 species. However, in Jaron et al. 2021 ‘Genomic Features of Parthenogenetic Animals’, an author already cited in ref. 19, it seems that k-mer measures using GenomeScope2 and Smudgeplot suggest *M. floridensis* is triploid (3n). Could the authors comment this point? The rest of the results concerning *Mf* being more closely related to *Mi* than any of the other Meloidogyne species studied is completely convincing. However, if *M. floridensis* is actually triploid, then the hypothesis of *Mf* being an ancestor of *Mi* (A1-A2) does not hold true anymore. And if *Mf* is diploid, is the divergence between its homologous chromosomes comparable to that measured between A1 and A2?

Response: The reviewer wants to know that *Mf* is triploid or diploid. In this revised version, we have provided more evidence to support *Mf* is a triploid. We thank reviewer for raising this crucial question, which prompted us to rethink the ploidy of *Mf*. This is indeed a very important and controversial issue.

1. In earlier observation (Lunt David H et al. *PeeJ* 2014 and Handoo et al. *Journal of nematology* 2004), *Mf* were described as a diploid species with meiosis.
2. However, in paper of genomescope2 (Ranallo-Benavidez et al. *Nature communications* 2020), *Mf* are predicted as a triploid based on k-mer analysis, which contradicts previous research.
3. In our analysis, we found that 86% reads could be mapped into A genome, indicating that *Mf* is composed of A genome and the genome structure should be AA (diploid) or AAA (triploid). Given previous research about cytological characterization, we simply considered *Mf* to be diploid in our original manuscript.

To confirm, we have added more analysis for genome structure of *Mf*:

1. We first investigated the ploidy of *Mf* by using Smudgeplot with two accessed data set of *Mf* (SRX2163401/SRX2163402 and ERX082408). This analysis gives different result, triploid when using first dataset and diploid when using another dataset (**Supplementary Fig. 38a**). In fact, the triploid result is reliable because the first data set is from HiSeq X Ten, which has more data size and longer read lengths, and the other set of data is from HiSeq 2000.
2. We try to provide more evidence about ploidy of *Mf*. We mapped reads into *Mf* genome (~74Mb, haploid) and called SNP from alignment. We then counted the allelic depths of heterozygous SNP (“0/1” in vcf file). If *Mf* is diploid, the alternative allele depths (AD) / total depth should be at 0.5 (for instance, in diploid

Mg, **Supplementary Fig. 38c**), while if triploid, the distribution of that value should be at 0.33. To avoid influence of the segment Mf reference genome (N50: 1.3Kb), we also performed similar analysis using Mi_A1 as reference genome. These analyses give a triploid result for Mf (**Supplementary Fig. 38b**). Therefore, given this result, we determined the Mf, at least these two samples (SRX2163401/SRX2163402 and ERX082408), are triploid species with a AAA genome structure. These finding may suggest that species described in Handoo et al. (Journal of nematology 2004) are different from sequencing sample in Lunt David H et al. (PeeJ 2014) and Szitenberg et al. (Genome biology and evolution 2017). Construction of Mf genome at chromosome level in further are crucial to resolve this question.

In revised version, we added this part in **Lines 362-369** and discussed the formation of Mf in discussion (**Lines 495-507**).

Q29: L281. I would recommend using ‘putative’ or ‘predicted’ secreted proteins. At this stage these are just bioinformatics predictions based on the presence of a signal peptide and no transmembrane region on computer-based genome annotation. Furthermore, the authors should explain / justify why putative secreted proteins would be more interesting than the rest of proteins to study an effect on fitness?

Response: As a plant pathogen, RKN secrete molecules called “effectors” to facilitate parasitism. Putative secreted proteins are major source of effectors. We therefore link sequence divergence of those putative secreted proteins to parasitic ability. We have added a sentence to explain why putative secreted protein are important and weakened the description about effector on fitness (**Lines 283-285, Lines 402-405**).

Q30: Extended Data Fig34. Please do not use ‘secreted’ genes, a gene is not secreted but encodes proteins that might be secreted. Furthermore, is sequence divergence measured at the amino-acid or nucleotide level? Is it measured between homologous genes or all genes? Please clarify.

Response: We have corrected secreted genes to secreted protein genes. The sequence divergence is measured at nucleotide level between syntenic gene pairs. Other genes refer to non-secreted protein genes. We have edited it.

Q31: L288-301. The part on differential expression of homoeologous gene copies according to the subgenome they belong to is particularly interesting. However, in this kind of analysis multi-mapped reads must be treated with particular caution as they can be fully responsible alone for the apparent variations in TPM values if not treated correctly. More importantly, it seems that no biological replicates have been produced for these transcriptomic data while it is absolutely necessary to support differential expression between the copies.

Response:

For biological replicates:

In fact, three biological replicates are generated for each sample, the description are added in **Methods Lines 809-810**.

For TPM values calculation:

As reviewer mentioned, allowing, or disallowing multi-mapped reads indeed affect the count of gene expression, especially for the more similar two A subgenomes. We refer to the strategies used by other species analysis, such as hexaploid wheat (Ramírez-González et al. science 2018) and diploid potato (Zhou et al. Nature genetics 2020). In a previous study of wheat (Borrill et al. Plant physiology 2016), they confirmed that the software Kallisto are of high accuracy for distinguish homologous expression and fast running. We optimized the methodology for gene expression analysis of homoeologous gene copies, as follows:

1. We removed the homologous gene groups with 100% identical between two A genome (from recovery of collapsed region).
2. Only homologous gene groups that each of genes showed a < 30% coefficient of variation between biological replicates were retained.
3. We removed the homologous gene groups with sum of TPM less than 5.

Updated result can be found in **Supplementary Table 21**, and the detail of methodology can be found in **Lines 1161-1174**.

Discussion

Q32: Overall, the discussion is a bit frustrating because the many interesting results presented in the rest of the manuscript are not really discussed or put in perspective with previous analyses from the literature. For example, the hypothesis for the origin of the different genome copies is not compared to other previous hypotheses, the perspectives concerning the discovery of the new telomere repeat are not mentioned, etc...

Response: As suggested by reviewer, we have re-organized the discussion section. Several results are now included:

1. Genome structure (**Lines 430-449**)
2. Alternative telomere (**Lines 451-467**).
3. Hypothesis for the origin of genome copies (**Lines 469-489**).

Q33: L326-327. Neither ref. 17 nor 38 mention DNA damage repair in MIG, so please update this sentence accordingly.

Response: We have edited this sentence and update the references.

Q34: L333-337. The authors mention that genomic plasticity might be an advantage of mitotic parthenogenetic Meloidogyne. But are the genomes of these species actually

more ‘plastic’ than those of meiotic facultative parthenogenetic species? Are there some data showing these genomes are particularly ‘plastic’?

Response: Our study shows that compared with diploid RKNs, polyploid RKNs have extensive chromosome fusion events, which is also one of the manifestations of strong genome plasticity. On the other hand, most of the polyploid RKNs have unstable karyotypes, and the variation of these karyotypes may be one of the keys to host and environmental adaptability.

Methods

Q35: L557. How were the five developmental life stages distinguished. What was the morphological criteria used to differentiate J3-J4 from late J2 or from young females?

Response: The method to distinguish the five developmental life stages was included in the Method (**Lines 756-775**). Compared with J2, late J2 is only slightly larger in size, but still significantly smaller than J3J4. We first use a sieve to filter, and then manually pick under a microscope to separate late J2 and J3J4. Similarly, the body size of young female is much larger than that of J3J4, and we directly use tweezers to pick the young female before spawning on the root, so we can completely distinguish the nematodes in these developmental stages.

Q36: More important concern regarding the RNA-seq experiment: it does not seem that biological replicates have been conducted. Consequently, all the results for differential gene copy expression or across the life stages remain to be confirmed by using biological replicates (or qPCR).

Response: All of our RNA-seq experiment content three biological replicates, we have included the description of biological replicates in Method (**Lines 809-810**).

Q37: L715-737. Genome annotation. It is important to assess the completeness of the predicted gene sets (or proteomes) using BUSCO in proteome mode and compare to previously reported protein sets for the same species in the literature. This has been done with CEGMA for the genomes but not for the predicted proteins.

Response: We evaluated genome completeness using BUSCO (v5.4.3) and *nematoda_odb10*, resulting in about 61.8%-70.7% for our assembly and 40.3%- 69.9% for other available *Meloidogyne* genome. We have added the results (**Results Lines 154-155, Supplementary Fig. 5**) and methods of BUSCO analysis (**Methods Lines 1001-1003**).

Q38: L734. Candidate HGTs have been detected using Alienness but with which Alien Index cutoff value and against which protein library? Moreover, nothing is said about HGTs neither in the results nor in the discussion? What was the goal of this HGT analysis and what are the main conclusions?

Response:

1. AI index cutoff in our first manuscript is 45, and protein library is NR. We noticed a new software (Avp) in Alienness website, and thus, we re-performed HGT detection using Avp pipeline with uniref90 database.

Updated methodology can be found in Method **Lines 997-998**

2. We now have added two analyses for HGTs:

- a) The distribution of HGTs shows that there are some hotspots for HGTs, as reported in *A. vaga* (**Supplementary Fig. 30**)
- b) Most of HGTs are shared in A and B subgenome, indicated that most of HGTs gain in A and B common ancestor (**Supplementary Fig. 41**).

Updated result can be found in Results **Lines 283-288, Lines 282-285**

Q39: L736. The references given for TMHMM and SignalP are incorrect, please use the references recommended by the authors of these software. Please also indicate which version of SignalP was used and with which parameters.

Response: We have included correct reference and added the version for SignalP and TMHMM.

Q40: L743-745. Please indicate which versions of MAFFT, Gblock and IQ-Tree were used and with which options / parameters (which model of evolution for IQ-tree? With or without bootstrap replicates?).

Response: Program versions and parameters are default parameters for MAFFT (v7.471), -b4=5 -b5=h for Gblock (0.91b), and -m TEST for IQ-TREE (v1.6.12).

Now we have included information about version and parameters. Now we have included information about version and parameters.

Q41: L777-779. Please cite all the publications for the genomes that have already been published.

Response: Now we have included the publications or accession number for those genomes.

Q42: L779-783. A simple BLASTP with an e-value threshold of 0.001 with *C. elegans* proteins is definitely not a sufficient criterion to decide conservation (presence /absence) of telomere-associated proteins in the other species. At least reciprocal best blast hits should be used and ideally phylogenetic analyses.

Response: We have used RBH analysis for telomere-associated gene, and updated **Fig. 2a** and methods (**Lines 1050-1056**).

Q43: L795 and L813. Where do the ONT long reads come from? Nothing is mentioned in the methods concerning the origin of these ONT long reads. How was the material extracted (same material)? Which sequencer was used? Which flow cell? Which basecaller with which parameters?

Response:

1. ONT data come from PromethION platform;
2. DNA for sequencing is the same Mi strain, WHF4-1;

3. The sequencer used was PromethION P48 (Oxford Nanopore Technologies, Oxford, UK);
4. The flow cell used was R9.4.1;
5. The basercaller used was Oxford Nanopore GUPPY (version 6.2.1)
6. The PromethION release version was 19.05.1.

Now we have included the description of ONT library and samples (**Methods L1076-L1079**).

Q44: L805-807 and 859-861. Same remark than previously for phylogenetic analysis: which versions of MAFFT and IQTree were used, and which parameters were used for the multiple alignment and phylogeny (which model of evolution)?

Response: The version of MAFFT is v7.471 and the version of IQTree is v2.0.3, and we used -m TEST for model test and -bb 10000 for bootstrap and we have edited it.

Q45: L861-869. The methodology used to reconstruct the mitochondria phylogeny is not straightforward. We could expect that some contigs obtained by long reads assembly and then polished with short reads would correspond to the mitochondrial genome in each species. Was it the case? If yes, why the authors did not use an alignment of the whole mitochondrial genomes (or all their genes) to reconstruct the phylogeny? Everything was here based on SNP detected from short reads alignment on the Mi mitochondrial sequence present at the NCBI. This methodology is completely different from the one used for all the other phylogenetic analyses (multiple alignment followed by ML phylogeny). Extended Data Fig. 32 shows a very short sequences and many unknown (N) positions in these sequences, casting doubts on how reliable is this mitochondrial phylogeny on such a short sequence and many N's. Furthermore, the obtained topology is not compared to previous mitochondrial genome phylogenies such as for example in (Blanc-Mathieu et al. 2017, PLoS Genetics).

Response: As suggested by reviewer, we optimized the methodology for mitochondrial phylogeny. The updated method is:

1. We assembly and annotate the mitochondrial genomes from short reads (including some published data) using MitoZ and GetOrganelle.
2. Nucleotide sequences for each mitochondrial gene are extracted and aligned using MAFFT.
3. Alignments are then concatenated into a super matrix, followed by phylogenetic tree construction using IQ-TREE.

We have added the description of methodology in revised manuscript **Methods Lines 911-922**

Q46: L872. The methodology to determine TPM is not clear. The authors indicate Kallisto was used but do not provide further details such as whether RNA-seq reads were first checked for QC and whether they were aligned on transcripts from ISO-seq, from StringTie or deduced from gene predictions on the genome? This combined with the absence of biological replicates for the transcriptome data cast doubts on the validity of the whole analysis on differential expression of gene copies.

Response: The RNA-seq reads were first checked for QC, and the TPM value are calculated from predicted genes. We generated three biological replicates for each sample. The TPM value for each gene is calculated by average of three biological replicates. We have included the detailed methodology in our revised manuscript (**Methods Lines 1161-1174**).

Q47: L884. Which Chip-seq data have been downloaded from the NCBI? Please provide accession numbers and if these data were already published, please cite the appropriate reference. By the way, these data were just used for one Figure (extended data Fig. 24) and never further commented or even mentioned neither in the results nor in the discussion. Same remark for centromere repeats with no mention in the results or discussion. For instance, are there some histone marks specifically associated with telomere repeats or centromere repeats?

Response:

1. We have added accession numbers for Chip-seq dataset (PRJNA725801)
2. We showed the distribution pattern for histone makers and found that H3K9me3 are associated with centromere and H3K4me3 are associated with subtelomeric region (**Lines 279-283, Supplementary Fig. 29**).

Q48: L898. Which whole genome sequence from which source were used (please provide accession numbers)? Do these sequences correspond to previous versions of these genomes already published in the literature? If yes, please cite the appropriate references.

Response: Accession numbers for all whole genome sequence data set can be found in **Supplementary Table 2**. These sequences are corresponded to previous versions of these genomes already published in the literature. Corresponding reference are also cited.

Correct reference to the source data

Q49: This is not sufficient to indicate that genome x has been downloaded from the NCBI or from wormbase, accession number to the bioproject or genome version should be provided and if the genome assembly has been published previously, the corresponding paper should be cited. For instance, in Extended Data Fig 15 it is impossible to tell what is the difference between *M. floridensis* 1 and 2 (two different versions? Already published?), same remark for *M. hapla* or *M. enterolobii* (already published?).

Response: We have included the accession numbers for *M. chitwoodi* (GCA_015183035.1), *M. exigua* (GCA_018905775.1), *M. hapla* (GCA_000172435.1), *M. floridensis_1* (GCA_003693605.1), *M. floridensis_2* (GCA_000751915.1), *M. enterolobii* (GCA_903994135.1) and *M. luci* (GCA_902706615.1).

Q50: Three previous versions of Mg genomes have been published (Somvanchi et al. 2018, Journal of Nematology; Phan et al. 2020, Ecology & Evolution; Somvanshi et al. 2021, Gene), they should be at least cited and compared to this new version which is undoubtedly a substantial improvement and resolves the genome at chromosome resolution.

Response:

1. We have cited previous version of Mg genome and compared the genome size and N50 (**Lines 135-138**).
2. All those published Mg genome are draft genome with N50 from 20.4kb to 294kb, while 2.5Mb for our Mg genome. To ensure that our Mg assembly is accurate, we compared the Mg genome to another closely species with good assembly quality (*M. chitwoodi*, N50=2.5Mb).
3. We noticed some error in our first Mg assembly (from smartdenovo, see below Response Figure 2). We re-assembled this genome by using flye and construct chromosome using Hi-C data. Now the new assembled Mg genome (Mg_T2T) exhibit a strongly collinearity against *M. chitwoodi* and are identified Mg-Tel at least at one end of each chromosome (**Supplementary Fig. 27**), which can prove that Mg_T2T is of high quality and at chromosome resolution.

Response Figure2 Genomic synteny of *M. graminicola* (assembly v1) with *M. chitwoodi* shows some errors in our Mg genome (blue circle).

Our new Mg genome (Mg_T2T) can be found in Result **Lines 268-271**, the methodology can be found in **Methods Lines 1115-1119**

Syntax / grammatical errors

Q51: A lot of syntax and grammatical errors as well as sentence construction problems including in the Abstract. The whole paper needs to be extensively edited by a native English speaker.

Examples:

- L26. allopolyploidy not allopolyploid
- L26. shows a widespread not 'the' widespread
- L30. AAB and AABB, respectively for what and what?
- L41-42 the advantage of sexual reproduction not sexual species.
- L44. What does an 'inferior' position mean? Please clarify
- L46. Even surpassing not even surpasses
- L46. Their diploid sexual relatives not diploid sex.
- L92. Which was consistent with not 'in consistent with'.
- Figure 1d. Tetraploid not 'Teraploid'
- L144. Clade IV not Clade VI.
- Supplementary Table5. A gene is not secreted but its protein product yes. Please change to secreted proteins or to gene encoding putative secreted proteins. Same remark for line 302 and everywhere 'secreted gene' is incorrectly used.
- Extended Data Fig. 34. 'sequence divergence' not 'sequence divergency'
- L707. 'fused' not 'fussed'

These are just some examples to illustrate the issues and the whole manuscript needs to be extensively edited to become more correct and understandable.

Response:

Thanks to the reviewers for the careful review. We have corrected above examples and polished our manuscript in Nature Research Editing Service.

Reviewer #2 (Remarks to the Author):

Review of "Phased T2T genomes reveal allopolyploid nematode origin pattern as unreduced gamete hybridization" for Nature Communications

The authors report four phased telomere-to-telomere de novo genome assemblies from the hybrid polyploid apomictic *Meloidogyne incognita* group (MIG): *M. incognita*, *M. arenaria* (Ma) and *M. javanica* (Mj), along with a newly discovered member: *M. paraarenaria* (Mp), and a haploid chromosome-level assembly for *M. graminicola* (Mg) – a diploid automictic outgroup. This MIG system notably has A and B subgenomes and some species are triploid (Mi, Mp) and some tetraploid (Ma, Mj). The authors resolve the relationship between these subgenomes and reconstruct a scenario for the organization of these distinct genomic combinations, involving hybrid events between reduced and unreduced gametes. Additionally, a novel alternative telomere element is reported here that appears to have evolved in MIG following the loss of telomerase. The genomes assembled and described here appear to be very high quality and represent a huge gain to the communities interested in MIG as a pest and model for genome evolution following allopolyploidization. However, I think that the relatively shallow degree of scholarship about asexual reproduction, polyploidy, the current status of

literature on the MIG system, along with minimal context for broadly interpreting the results beyond description and instances of overreaching in discussing the results prevent this text from being a compelling paper as is. Here, I outline my primary concerns, followed by some more general comments on the vagueness and missing information in the methods, and some comments about specific pieces of text.

Response: Thanks for positive comment of reviewer. Our paper aims to unravel the phased T2T genome of polyploid RKNs, the complex origin of hybridization, and the diverse features exhibited by polyploid genomes. It is well known that polyploid RKNs are more adaptive than diploid RKNs. Perhaps not comprehensive, but our study attempted to analyze the effect on the enhanced fitness of polyploid RKNs genome changes after polyploidization. However, Due to undersampling of diploid and polyploid RKNs, this study could not fully answer this question. We have continued to pay attention to the topics of polyploidization and adaptive enhancement in RKNs, collected a large number of RKNs populations of different ploidy across China, and carried out research on the relationship between their ploidy and fitness, and will complete the analysis and submission in the next time.

In the revised manuscript, we mainly made the following changes:

1. We have revised the description of the motivation for this paper in the Introduction, adding depth on asexual reproduction and polyploidy;
2. We listed several hypotheses about the origin of MIG reported in previous studies, and put forward several hypotheses based on the analysis results of this study, and discussed the possibility of various hypotheses;
3. We reorganized the MIG literature and revised the wording of the discussion of the results;
4. We discuss the potential relationship between telomere replacement mechanisms and chromosome fusion in RKNs and have adjusted the discussion;
5. We have increased the description of the method, such as adding some details and the version number of the software used, etc.

We provide point-to-point responses to specific questions from the reviewers as below.

Main comments:

Q52: I think the motivation and results need to be better contextualized. There is very little use of the prior literature except a general statement (Lines 62-64) about some genomic features possibly associated with MIG adaptability and that the previous draft genomes were fragmented (Lines 65-68). More text is warranted in the Introduction to outline what exactly is and is not known about the history of this species complex. What is the current hypothesis/hypotheses for how this species complex formed, what hybridization events have been proposed, etc.

Response: We thank the reviewer for the suggestions. In the revised manuscript, we cited a wider range of previous literature to explain the motivation and results of this study.

1. Early studies proposed the reticulate evolution and hybrid origins in polyploid RKNs (**Lines 77-78**), and it is the currently recognized hybrid history of RKNs complex. However, due to the failure of previous genomes to resolve clear subgenome information, researchers were unable to accurately judge the complex hybridization history of polyploid RKNs;
2. Several hypotheses about the origin of polyploid RKNs have been proposed, like polyploid RKNs origin from either intra- or interspecific hybridization following suppression of meiosis during oocyte maturation; polyploids probably originated from the hybridization between an unreduced diploid oocyte and a haploid spermatozoon; and polyploid RKNs share a recent common maternal ancestor (**Lines 78-91**). However, since no definitive ancestral species has been found and clear subgenome information is not available, a fully convincing hypothesis cannot be derived.
3. It is proposed that interspecific hybridization events between two facultative meiotic parthenogenetic diploid RKNs can occur under both laboratory and natural conditions (**Lines 93-95**).

Q53: While the genome generated here are impressive, the text does not lay out their purpose in a clear manner. The Discussion is treated similarly to the Introduction; I do not get clear sense of what the status is now for our understanding of this system – what was answered and what is still to be determined. How do these results compare to other recent findings, e.g. Schoonmaker et al. 2020 G3: <https://doi.org/10.1534/g3.119.400650?>

How does this text add to our understanding of genome evolution following allopolyploidy events?

Response: As said above, our paper aims to unravel the phased T2T genome of polyploid RKNs, the complex origin of hybridization, and the diverse features exhibited by polyploid genomes. In order to describe these points clearly, we reorganized Introduction and Discussion to lay out our motivation and results in a clear manner.

The status of MIG system is that polyploid RKNs originated from a common hybridization event, and there are multiple hypotheses of reticulation origin, but they are also controversial. For example, some research proposed that Ma and Mj are sister species but some proposed that Mi and Ma are sister species. A more consensus view is that polyploidy originated from the hybridization of diploid gametes and haploid gametes. But the specific hybridization process and the complex relationship among Ma, Mj and Mi is still unclear.

In this study, we dissected the genome structure of MIGs, distinguished their respective subgenomes; resolved the complex history of their origin evolution as hybrid between unreduced and haploid gametes based on subgenome information; proved that Ma and Mj are sister species, and there is a triploid and a tetraploid lineage in the

existing *Ma* species; revealed extensive chromosomal fusion events after RKN polyploidization; discovered alternative telomere structures in RKN, and speculated that the extensive chromosomal fusion of polyploid RKN is associated with higher levels of DNA damage in polyploid nematodes.

Q54: The maintenance of sex is far from settled. The text indicates (Lines 41-43) that sex is advantageous because, relative to asexual reproducing lineages, sexual lineages can adapt more quickly to a changing environment. This argument is one of many hypotheses for the long-term persistence of sexual reproduction. This text needs to be updated to better reflect breadth of the field of the evolution of sex. While adaptability could be emphasized, perhaps with more examples, it should be clear that why so many organisms have sex to reproduce is not answered to date. See these recent perspectives that covered different hypotheses on the maintenance of sex and commentary on the active and ongoing debate on this question (the second also touches on geographic parthenogenesis and connections between hybridization and parthenogenesis, that should be considered more in this text):

- Neiman et al. 2018 *Evolution*: <https://doi.org/10.1111/evo.13499>
- Fujita et al. 2020 *Annu. Rev. Ecol. Evol. Syst.*: <https://doi.org/10.1146/annurev-ecolsys-011720-114900>

Response: We agree with that the maintenance of sex is far from settled. Although why most organisms maintain sexual reproduction remains an unanswered question, some hypotheses, such as the faster rate of adaptive evolution and elimination of deleterious mutations than those of asexual species, partially answer this question.

The hybridization and polyploidization generate unique combinations of genetic variation in parthenogenetic species, the nematodes we study are obligate parthenogenetic species, which can be used as materials for studying the relationship among hybridization, polyploidization and parthenogenesis. We have re-summarized more information and perspectives on sexual and asexual reproduction (**Lines 41-53**).

Q55: Similarly, the relationship between asexual reproduction and polyploidy/hybridization is not well drawn out in the text. Line 49: How polyploidy facilitates adaptive evolution in asexual taxa is not “unexplored”, perhaps “still an open question”, but there is plenty of literature on the subject. This text should include more make it clear that parthenogenesis is often associated with polyploidy, not a rarity as indicated in the text. More literature needs to be referenced on the connection between polyploidy (and hybridization) and asexual forms of reproduction, as well as more examples that the weevil study the cited.

- Otto and Whitton 2000 *Annu. Rev. Genet.*: <https://doi.org/10.1146/annurev.genet.34.1.401>
- Maciver 2016 *Trends in Parasitology*: <https://doi.org/10.1016/j.pt.2016.08.006>
- Ament-Velásquez et al. 2016 *Molecular Ecology*: <https://doi.org/10.1111/mec.13717>
- Neiman et al. 2014 *J Evol Biol*:

Response: We thank the reviewers for the suggestions. It is well known that asexual reproduction often associated with polyploidy and hybridization, and our wording is not rigorous enough.

Polyploidization is an important driving force for speciation and adaptation, which has been well reported in many polyploid species. The weevil has both diploid sexual reproduction and triploid and tetraploid parthenogenesis, similar with the situation of RKNs. One of the hypotheses for the origin of parthenogenesis and polyploidy in weevils is that polyploidy results from nondisjunction of haploid chromosome sets, which is similar to our hypothesis for the hybrid origin of unreduced gametes in RKNs. It is suggested that polyploidization events involving unreduced gametes may be common in lower organisms such as insects.

We have given more consideration to the relationship between polyploidization and parthenogenesis in our Introduction based on the reviewer's suggestion (**Lines 55-66**).

Q56:It is not clear to me that this text describes a “unique pathway of polyploidization” or how it contributes to adaptive evolution in parthenogens. The text cites a review on how unreduced gametes facilitate polyploidization (Mason and Pires 2015), which describes triploid bridges, so I am not sure how the MIG system represents a novelty. Perhaps they mean specifically in animals, as triploid bridges are well known from plants and more hypothetical in animals?

Response: In this study, we found a high similarity in the $A_1A_2B_1$ subgenome (Ks values are approximately 0, and nucleotide identity is approximately 100%) between Ma, Mj and Mp. After considering various hypotheses, we proposed that the unreduced gametes hybrid with haploid gametes to form polyploids and contribute to adaptive evolution is the most likely one (**Lines 349-356**).

Our wording may not be rigorous enough for the “a unique pathway of polyploidization”, but this mode of polyploidization is rare in animals, although it has been speculated that triploid chickens may have originated from the hybridization of reduced and unreduced gametes (F Abdel-Hameed et al. 1971 Science). We have adjusted this description (**Lines 36-38**).

Q57:The text, starting with Abstract, claims to connect the polyploidization in MIG system to its adaptability. From the results described, I do not see such a connection. Line 76-78: I am not sure how the results here connect “genome and homologous gene expression after polyploidization” to “adaptability enhancement”. Surely there is extensive work here describing evolutionary history of the subgenomes and gene expression patterns but the only discussion I can find of the data in relation to adaptation and fitness is lines 301-306, where secreted proteins are reported to have greater variation in expression compared to “other genes” (other genes is not described here, are they all other genes or a specific subset?). “Other” could perhaps be divided into other specific functional classes.

Response: Our work focuses on dissecting the complex subgenome of polyploid RKNs, and the evolutionary history of the subgenome and gene expression patterns. We really tried to find evidence of polyploidization and adaptive enhancement, such as the differential expression of homologous genes after polyploidization, however, our results did not fully answer this question due to the undersampling of our diploid species.

1. Even so, the secreted proteins (including effector) are the most important weapon for RKNs to infect plants, and it plays an important role in entering plants, resisting plant immunity, establishing and maintaining feeding sites, etc.
2. Therefore, we tried to find the effector changes after polyploidization, and found that the effector genes in polyploid RKNs had a faster evolution rate than other genes (all genes except the effector genes) **Supplementary Fig. 42**.
3. Although a faster evolution rate of effectors in plant pathogens has been reported (Thierry Rouxel et al. 2011 Nature Communications), we still believe that the changes of effectors after polyploidization helped polyploid RKNs to enhance fitness and broaden the host spectrum.

As said above, we undersampled polyploid and diploid RKNs, resulting in insufficient answers to these questions in our study. We have collected more than 300 RKNs samples with different ploidy in the past few years, and conducted tests such as infectivity, aiming to answer the above questions fully in the next research.

Q58: Still, how exactly this result demonstrates that hybridization has contributed to an increase in fitness for these species is unclear to me. There is no comparison to a nonhybrid diploid species, nor any measure of fitness. I understand that this particular analysis focuses on variation across subgenomes and would not apply to a nonhybrid diploid but claiming that this result shows “adaptability enhancement” is a stretch. At the very least, discussion on what such variation in secreted protein expression means biologically and in terms of fitness is need.

Response: As the reviewer stated, demonstrating that polyploidization increases fitness requires a comparison of polyploid species to diploid species, and our study does not have an ancestor species with nonhybrid diploid of MIG, so it is more in terms of “adaptability enhancement” is an extension.

The focus of our study is to reveal the complex genome and origin history of MIG, and future work will pay more attention to the relationship between polyploidization and adaptation. As suggested by the reviewers, we have added a discussion of the biological implications of variation in secreted proteins for fitness (**Lines 402-405**).

Q59: Also, the related result, Extended Data Fig. 34 is not referenced in text. Are the results related, is that expected? Given that it is sequence divergence perhaps a connection to outgroups could be made. Furthermore, is there enrichment for types of

genes/pathways fitting any of the different expression balance/dominance profiles? It seems like there is a lot of potentially interesting biology not being considered here.

Response: The Extended Data Fig. 34 is referenced in text Lines 284 in our first manuscript. This result is a comparison of the sequence divergence between the effector gene in MIG and other non-secreted protein genes, and the result show that effector has a faster evolution rate, may be one of the driving forces of adaptive evolution for polyploid RKNs. In our revised manuscript, Extended Data Fig. 34 is renamed as **Supplementary Fig. 42** and referenced in text **Lines 409**.

As suggested by the reviewers, if more diploid outgroups could be used for comparison, especially diploids of the A or B subgenomic progenitor species corresponding to MIG, it would help us to reveal whether after polyploidization, effector expanded significantly and had a faster evolution rate than diploid, thus increasing MIG adaptability and broadening the host spectrum. We expect that more diploid RKNs of closely related species can be collected and sequenced in future studies to reveal this problem fully.

We did the enrichment analysis in our first manuscript, but found no types of genes/pathways fitting any of the different expression balance/dominance profiles. Other potential biological issues are developed in the **Discussion**.

Q60: The text appears to name a new species, which is potentially very important to understanding this system. As I understand it, Mp represents a previously unknown lineage of Meloidogyne that the authors have named here. I do not specifically have experience in identifying and naming new species, but I think more information about this species is generally needed.

Where exactly it was collected and from what? Is it phenotypically distinct from the other species? The collection information for all 5 species in this text is vague. I think more detail is needed on the initial isolation and culturing of these nematodes. Also, the methods related to Extended Data Fig 1 are missing. The legend indicates it is an “ITS rRNA gene” tree, while the text refers to using “ITS rRNA, 28S rRNA, and COI sequences”. This incongruity needs to be addressed. Also, ITS rRNA could refer to ITS1 (between 18S and 5.8S rRNA) or ITS2 (between 5.8 and 28S rRNA). Furthermore, there is no information about how the rest of the data was collected, are these BLAST hits on NCBI? I assume some universal primers were used to amplify and sequence these genes from the cultured isolates? There is also no information about how the tree was constructed: alignment, phylogenetic method, statistical support, etc. Based on that tree, Ma and Mp look like the same lineage, as do several other species pairs. Given the complex evolutionary history of these species, which includes hybridization and polyploidy, I do not see how a single gene tree is appropriate to describe a new species, especially when each species is only represented by a single sequence in this tree. Obviously, the authors generated a complete genome for this lineage demonstrating its apparent uniqueness, but I think more discussion of their motivation for doing so based on a single gene tree and negative SCAR results would improve this manuscript. For

example, had researchers in the root-knot nematode community suspected another distinct lineage existed? Furthermore, I am a bit weary of declaring a new species in a system with repeated hybridization, ploidy elevations, chromosomal fusions, and karyotype change from one sample. I suspect, based on this text and other literature, that there is a great deal of intraspecific karyotypic variation among these species. I think the authors could expand on their degree of certainty that this is a novel species or part of a more complicated background of hybridization that has not been captured in previous studies.

Response: We thank the suggestion of reviewer, the detail information of Mp as below:

1. The Mp was collected from cabbage in Shiping city of Yunnan province;
2. The phenotypically of Mp is not distinct from the other species under our tests;
3. The collection information for all 5 species were update in Method (**Lines 746-750**);
4. The initial isolation and culturing of Mp: Mp was collected from the cabbage root knots in the farmland around Shiping City. After the single egg mass was isolated in the laboratory, it was inoculated on tomato Rutgers, and the single egg mass was isolated twice in a row to ensure pure species.

The claim of the new species Mp is indeed controversial, as said above, our study is not aim to claim a new species, which is not what we are good at, but found that there are two lineages of triploid and tetraploid in the existing *M. arenaria* (Ma). We suggest a more in-depth taxonomic identification of these two different ploidy Ma.

To make it clearly, we assembled the mitochondrial genome of our samples and constructed a mitochondrial phylogenetic tree with all published data (**Supplementary Fig. 1**). This phylogenetic tree is constructed from all mitochondrial genes, and is therefore more reliable than our previous marker gene tree.

1. Mitochondrial tree shows that Mp is located inside Ma samples and may be the same species of the published Ma sample such as MaraL32 and MaraHarA (**Supplementary Fig. 1 and Supplementary Table 2 for accession number**);
2. Mitochondrial tree also suggests that currently naming of Ma might be not a single species, but with two lineages (green and red lineage in **Supplementary Fig. 1**);
3. To further investigate whether currently naming of Ma contains two species, we mapped reads of all published Ma data into our T2T assembly and counted the normalized reads depth from subgenomes. This analysis reveals that red lineage is proposed triploid with B2 subgenome lacking, and green is tetraploid, which confirmed that current naming Ma is not a single species but with a tetraploid species and triploid species (**Supplementary Fig. 2 and Supplementary Fig. 39**);
4. Given the ploidy estimation and the mitochondrial phylogenetic tree, Mp may be the species of triploid Ma sample.

Now we have adjusted the description of Mp definition as follows:

1. We do not claim that Mp is a new species and also do not describe too much on the species identification of Mp;
2. We just considered Mp as an unknown species at beginning and sequenced it;
3. We added a description of the phylogeny of the current Ma, which contains two lineages, and Mp represent the triploid one;
4. We propose that the tetraploid Ma sample (Red lineage in **Supplementary Fig. 1**) are still named Ma, triploid Ma sample (Green lineage in **Supplementary Fig. 1**), including our Mp, are named “*M. paraarenaria*”.

The updated result can be found in **Lines 122-132** and methodology can be found in **Lines 911-922**.

General methodological comments:

Q61: Version numbers for programs should be included and code used for the bioinformatic analyses should be made available.

Response: We have added version of programs and provided code for bioinformatic analyses (<https://github.com/xiecs-BMB/MIG-genome>).

Q62:In some instance “default parameters” are mentioned as used (e.g., for Trinity [Line 721]) and other times there is no mention of parameters (e.g. CD-HIT [Line 724] – which is think should be CD-HIT-EST for cDNA).

Response: Thanks, we have edited it and added parameters for other programs.

Q63:When outside data is brought in for additional analyses (e.g., ITS sequences, draft genomes, mitochondria) the source and method for acquiring these sequences is often not included. For example, regarding Extended Data Fig. 31: what exactly those other sequences are not clear – are they also short reads that were mapped to the same reference mt genome mentioned in text? How were those accessions selected? There is no Mg – which was analyzed regularly throughout the rest of the text, so I am not sure how this data was collected. Additionally, for this figure, “Tree scale” is not clear (is it substitutions/site?) and there is no statistical support indicated.

Response:

1. We have included the source, accessions, and method for all outside data in our revised manuscript.
2. We have adjusted the methodology for mt genome analysis, all available *Meloidogyne* sample are included.
3. We have added bootstrap information for all phylogenetic trees.

Specific comments:

Q64: The Abstract does not mention that the species is a nematode; it's in the title but should be in abstract.

Response: Thanks to the reviewer for the suggestion, we have stated in the Abstract that the species of this study is nematode.

Q65: Line 44: “take the inferior positions” – this sentence needs to be reworded because it makes the evolutionary process sound purposeful.

Response: Thanks to the reviewer for the suggestion, we have revised the wording to “usually exhibit weak competitiveness” (**Line 50**).

Q66: Line 73: This is the first mention of *M. graminicola* (Mg) but no description of this species. It should be clear that this is an outgroup species. It's reproductive mode and ploidy need to be stated here.

Response: Thanks to the reviewer for the suggestion, we have claimed the reproductive mode, ploidy and outgroup species for Mg (**Line 106**).

Q67: Line 102-103: CEGMA results are reported but perhaps this could be updated to BUSCO and use a new database. For the reads mapping value – is this the short reads and/or the PacBio?

Response:

1. We have added the BUSCO analysis.

We evaluated genome completeness using BUSCO (v5.4.3) and nematoda_odb10, resulting in about 61.8%-70.7% for our assembly and 40.3%- 69.9% for other available *Meloidogyne* genome. We noticed that BUSCO value is not high for *Meloidogyne* species, possibly because nematoda_odb10 are built from proteins of *Trichinella spiralis* (Clade I), *Brugia malayi* (Clade III), *Necator americanus* (Clade V), *Loa Loa* (Clade III), and *Caenorhabditis elegans* (Clade V), while *Meloidogyne* are from Clade IV. Some studies use BUSCO (v3.0.2) and eukaryota_odb9 to generate a higher BUSCO value, when using those versions, we get about 89.5%-97.4% for our assembly and 68.7%-96.7% for other available *Meloidogyne* genome. We have added this part of the results to the manuscript (L147-L148) and provided the BUSCO results of the newer version (v5.4.3) (**Supplementary Fig. 5**).

2. The reads mapping value are counted from RNA-seq short reads, and we have edited it.

Q68: Line 107: The discrepancy in repeat content between the MIG species and Mg caught my attention. I think this result should be discussed and probably analyzed further. What is the history of activity in these species, is there a shared or unique set of expansions in the MIG species? There has been discussion previously about TE activity in these species – summarized in a review on TEs in polyploid animals (Rodriguez and Arkhipova 2018: <https://doi.org/10.1016/j.gde.2018.04.003>) that

would be worth looking at. That review also describes the findings from the *Xenopus* genomes, that the *X. laevis* subgenomes were “marked” with distinct TE activity that helped distinguish the homeologous chromosomes. Given the quality of these new genome assemblies, it would be interesting to highlight TE activity.

Response: The reviewer wants us to provide more analysis of TE dynamics, we have added additionally analysis for TE activity history.

1. We firstly showed the TE landscape for four polyploid species and found that there are active TE copies in both diploid and polyploid species (**Supplementary Fig. 40a**). Most of TE copies are old (**Supplementary Fig. 40a**), suggests that TE is gradually accumulated during the ancestral state.
2. We also investigated the LTR retrotransposon in our genome. Among 3277 full-length LTR retrotransposons, 79% of them show a 100% LTR identity (**Supplementary Fig. 40b**). As mentioned in that review, many old LTR retrotransposon are fragmented in polyploid RKN genomes, which makes the majority of the full-length LTR retrotransposon detected are young. These results suggest that TE are still active in RKNs genome, and the increased TE content in polyploids is not caused by burst recently, but mainly accumulated in the ancestral state.
3. We also investigated whether some TE families are specific expansions in A or B subgenome (**Supplementary Fig. 40c,d**) as described in *Xenopus* genomes. We counted copy number of each TE family in A or B subgenome. Very few TE families show a unique distribution in A or B subgenome. These families are not evenly distributed as found in *Xenopus*, but are only clustered on a few chromosomes, that is, they are only expand in specific region in genome, but not expand in specific subgenome.

We have added this in our revised manuscript (**Lines 374-382**).

Q69: Line 180: The introduction of *M. floridensis* (Mf) and *M. enterolobii*, and *M. hapla* seems to come out of nowhere. It is not clear what these species are or where this data comes from. Were these assemblies downloaded from NCBI? Why only these species? It appears that *M. chitwoodi*, *M. exigua*, *M. luci* also have genome assemblies available NCBI. Perhaps MIG needs to be better defined here.

Response:

1. We have added accession numbers or reference for all assemblies.
2. Several assemblies are available in NCBI including *M. hapla*, *M. floridensis*, *M. luci*, *M. enterolobii*, *M. chitwoodi* and *M. exigua*. According to **Supplementary Fig. 1** (Mitochondrial polygeny), *M. floridensis*, *M. luci*, *M. enterolobii*, belong to same clade of our polyploid species (Clade I), *M. hapla* belongs to Clade II, *M. chitwoodi* and *M. exigua* belong to Clade III.

3. We have added conservative analysis of telomeric element for all available assemblies (**Supplementary Fig. 18**), and have included the accession number in legend.
4. “MIG” is not a phylogenetic terminology, and we now use “Clade I RKNs” and “polyploid RKNs” to represent these samples. The relationship of those nematodes is shown in **Supplementary Fig. 1**.

Q70: Line 262-265: Context is lacking here, how is ~50 mitochondria SNVs across “different species” indicative of a very recent MIG formation? The related methods are also quite vague.

Response: In original manuscript, we called SNP for mitochondrial genome and constructed phylogenetic tree. Now we have optimized the methodology, as follows:

1. We assembly and annotate the mitochondrial genomes from short reads (including some published data) using MitoZ and GetOrganelle.
2. Nucleotide sequences for each mitochondrial gene are extracted and aligned using MAFFT.
3. Alignments are then concatenated into a super matrix, followed by phylogenetic tree construction using IQ-TREE.
4. We could estimate the number of mutation sites between those mitochondrial genomes. The mitochondria are maternally inherited and the mitochondrial mutation rate in *C. elegans* are about 1.05×10^{-7} /site/generation. This information combined with very few mutation sites in MIG species could determine that maternal genome of these species diverged not long ago, and thus hybridization occurred recently.

Updated result can be found in **Supplementary Fig. 1**. Methodology can be found in revised manuscript **Lines 911-922**. We also edited the description for recent formation (**Lines 356-360**).

Q71: Line 292: What are the 5 life stages? It is not clear here if there were life stages that differed from one another in subgenome expression patterns.

Response: We have edited it to developmental stages. In fact, we did not compare gene expression pattern between developmental stages. We only show the expression pattern in one stage and perform similar analysis for each stage of life cycle. The summary of expression pattern can be found in **Supplementary Table 21**.

Q72: Fig. 5: Why is this figure only “egg period”? Is it representative broadly of the findings for the other life stages? I am also not sure what the two sets of boxes in 5b are.

Response:

1. We performed gene expression pattern analysis for each of five stages of life cycle (See **Supplementary Table 21**). We picked one stage to show gene expression pattern. In original manuscript, we selected egg stage, and we now selected J2 stage in revised version because of J2 stage having more filtered syntenic gene groups.
2. In fact, the results are similar in each of 5 stages, and display all of them will appear redundant, so we only selected one stage for visualization. All of data are shown in **Supplementary Table 21**.
3. The purpose of Figure 5b is to check whether the genes with repression patterns are also repressed in other species. We have edited result and legend for Fig 5b to make it clearer.

Reviewers' Comments:

Reviewer #1:

Remarks to the Author:

First of all, I would like to congratulate the authors for the remarkable efforts made to address as much as possible my previous comments and for the improvement of the manuscript. This new version is much clearer, much better written and many results are now more convincing and many of my previous points can be considered addressed.

The major concern that remains from my side is still related to the assembly strategy. In their quest to obtain T2T chromosome-scale assemblies, the authors have extensively edited manually the assembly to meet their expectations. After careful observations of the optical map data, I am not convinced this manual assembly is strongly supported.

For Mi, I would recommend only using the hybrid assembly obtained by BioNano Solve software based on the ONT contigs + optical maps without manual edition. Or at least provide this hybrid 'Solve' assembly and a dot plot alignment with the manually-edited assembly to figure out possible discrepancies.

In my opinion, all the other results provided by the authors are sufficiently convincing and interesting to justify an eventual publication in a high-standard journal like Nature Communications and these 'forced' T2T assemblies bring too much uncertainty and are not essential regarding the rest of the results.

Other main points that need to be addressed are listed below:

1-Taxonomic identity of *Meloidogyne* species and *M. para-arenaria*

I realize that the authors have deleted a very important information that was initially present in the previous version of the methods section. The information about SCAR markers is essential and absolutely needed to understand how species identities were assigned. Without this crucial information, it is impossible to understand why in the results section (L 117 – 119) it is indicated that 5 *Meloidogyne* species were collected: Mi, Mj, Ma and Mg as well as an unknown species. In the new material & methods there is no more information on how species identity had been originally determined. Because it is almost impossible to differentiate these species based on morphological characteristics, this information about SCAR markers should be reintroduced and the PCR results should be provided. Otherwise, it is impossible to understand why Mp was initially unknown and called *M. paranarenaria*.

Interestingly, and I congratulate the authors for this, the mitochondrial phylogeny I had suggested to perform (Supp. Fig 1) reveals several important things. First, it confirms that Mi is *M. incognita* and that Mj is *M. javanica*, so there is no doubt for these two species. However, it also shows that based on the mitochondrial genomes, what the author call *M. paraarenaria* falls right in the middle of some other *M. arenaria* populations (that are 3n according to the authors further analysis). Finally, the species that the authors call Ma (because it does present a Ma SCAR marker) falls in a small poorly supported phylogenetic group with two other *M. arenaria* populations (that are 4n). According to this phylogeny, it seems, indeed, that *M. arenaria* is not monophyletic and therefore maybe not a valid species.

Because this topology suggesting *M. arenaria* is paraphyletic and not a valid species is only moderately supported in the tree, maybe adding the two mitochondrial ribosomal genes in addition to the protein-coding genes in the supermatrix might help resolve better the tree. This is an important result that needs confirmation because I don't think any previous phylogeny based on mitochondrial markers had revealed *M. arenaria* was paraphyletic. My hypothesis is that, at this stage, there is not enough phylogenetic signal in the concatenated matrix to resolve the topology and I hope adding ribosomal genes will help resolve it.

Another suggestion is to perform some virtual PCR in the genomes of the 3n and 4n populations of *M.*

arenaria (if genome assemblies are available) and determine whether the SCAR marker would be present / absent in these other populations. Alternatively, and I don't know whether the authors have the possibility to perform this kind of experiments in their lab, it would be very interesting to check whether esterase profiles of Mp and Ma are the same and at the expected pattern for *M. arenaria* or are different. At least, I guess it is possible to check in the original publications of the other *M. arenaria* populations (particularly the 3n ones) which markers / information (if any) were used by the authors to state their species were *M. arenaria*.

Whatever the final conclusions will be on the validity of *M. arenaria* as a species, I would not recommend at this stage to call Mp *M. paraarenaria* but *M. arenaria* 3n and Ma should be called *M. arenaria* 4n. Indeed, substantial variations in ploidy levels have been observed among populations and isolates of *M. arenaria* and it is not surprising to observe both 3n and 4n populations for this species (Cytogenetics, cytotaxonomy and phylogeny of root-knot nematodes, Triantaphyllou, 1985). What is more surprising is that they form two separate branches in the mitochondrial phylogeny and not a single monophyletic branch, but this needs to be further clarified.

2- Anchoring Mi, Mj and Ma assemblies on Mg genome and manually editing to obtain telomeres at both ends.

I am still not convinced using the genome of Mg as an anchor to help scaffold the genomes of the polyploid genomes was an ideal strategy. I hypothesize this can include a lot of biases and over-scaffold the polyploid genomes in a multiple of 18 chromosomes.

I also don't think it is absolutely necessary to provide telomere-to-telomere assemblies. These manual editions of the genome assembly to eventually obtain telomeres at both ends of all the scaffolds might also introduce several biases.

For *M. incognita*, the authors seem to have generated a non-forced and not manually edited assembly from the contigs using optical mapping data (using Solve software L 935-936). This is a very valuable and interesting point of comparison that has not been utilized enough. According to the methods section (L 1094 – 1098), for *M. incognita*, the contigs used were those assembled from ONT data via CANU and then polished via Nextpolish.

In addition to this hybrid assembly the authors have also apparently aligned contigs the optical map data and made a manual assembly starting from contigs containing Mi-tel motifs. It is unclear whether this new assembly was only based on ordered ONT contigs or a mixture of both the ONT and PacBio contigs? For instance, in Supp. Fig. 22, in the absence of a proper legend, it is impossible to know what is represented in the upper track (data from optical mapping?) and lower track (ONT contigs, PacBio contigs?). In addition, on the first chromosome of Supp. Fig. 22, there seems to have a gap at around 8 Mb where tags of the optical map don't match tags of the contigs. Therefore, the validity of this scaffolding can be questioned. According to Supp. Table 10, it is indicated that the longest reconstructed scaffold from the molecules is 8.2 to 8.8 Mb long (haplotype vs. nonhaplotype). Therefore, chromosome I which is >14.7Mb necessarily represent the assembly of several long molecules and the support of this gap by molecules pileups is important. The authors should provide BioNano molecules pileups to allow checking for support of these regions by molecules. At this stage it is impossible to assess the validity of the assembly based on examination of Supp. Fig. 22 which also needs a legend and probably another title (I guess this is alignment of the optical map with contigs and not chromosomes). It seems that extensive manual editing was used to obtain these 40 chromosomes and without strong support for the points of junctions between the contigs (by molecule pileups for instance), the validity cannot be assessed.

Also, L262-263 it is indicated that the same strategy was used to assemble Mi, Ma3n and Ma4n. It cannot be the same strategy because neither ONT long reads nor optical map data were generated for these 3 other polyploid RKN samples. Therefore, the authors should clearly state how the strategy differed for this species as it was necessarily based solely on PacBio and Hi-C data.

Finally, Fig 2d as well as Supp. Table 15 are confusing to me and again cast doubt on the validity of the manually edited assembly. Indeed, it seems that all the biggest scaffolds (or chromosomes) result from the fusion of A and B subgenomes) while this is never observed for the other smaller chromosomes.

Actually, all the interesting results provided by the authors concerning the hypothesis about the origin

of polyploidy, the TE dynamics as well as the expression level shift between subgenomes will remain true on genomes assembled without an external species as a guide and without adding telomeres everywhere.

3-Hypothesis of unreduced gamete and other hypotheses

I appreciate and acknowledge the efforts made by the authors to explore several different hypotheses to explain the origin of A1 – A2 subgenomes (supp. Fig. 36). However, it seems that the authors make the strong assumption that the two triploid species (Mi and Mp = Ma3n) have the same unique origin for their A subgenomes (an unreduced gamete with or without previous hybridization). Then, they invoke gene conversion (36c) to explain that the observed topology of A1-A2 subgenomes does not match the expected topology of unreduced gametes (36a (1) and (2)) but rather matches their third scenario. It appears completely acceptable that some gene conversion events took place. However, it seems unrealistic that gene conversion events took place at such an extensive rate that now, most of the A1 and A2 genes in Mi and Mp (Ma3n) result from gene conversion events and that the expected topology is never observed in the 14 trees in supp. Fig 35. In all the topologies the A1-A2 subgenomes of Mi are always more closely related to one another whereas the two A1-A2 subgenomes of Mp (Ma3n) are always more closely related to those of the tetraploid species. Therefore, a more likely hypothesis is that Mi and Mp A subgenomes have different origins and that Mp A subgenomes followed an evolutionary history closer to tetraploid species. This hypothesis is consistent with the observation from supp. Fig. 1 which suggests Mp is a triploid population of *M. arenaria*.

I also acknowledge that in supp. Fig. 14b, the information about the nucleotide divergence between subgenomes is now provided. For instance, in *M. incognita*, it seems that the divergence between A1 and A2 is about 4-5% (~96% identity) while the divergence between A and B is around 9-10% (~91% identity). Given the very low divergence at the nucleotide level between the mitochondrial genomes of the Mi, Mj and Ma, I wonder whether this relatively high divergence between A1 and A2 would be compatible with divergence between former alleles? Discrepancy between this high within-species divergence at the nuclear level (A1-A2) vs the very low 'between species' divergence at the mitochondrial level rather suggest hybridization at the origin of A1-A2 in an ancestor. Indeed, mitochondrial genomes evolve faster than nuclear genomes in *C. elegans* and most animals and this is expected to be the same in *Meloidogyne*. Thus, the divergence between these RKN species is supposed to be very recent. This is not consistent with long-term parthenogenesis as hypothesized L344-345. To explain 5% divergence between A1 and A2, two alternative hypotheses can be considered. (I) High heterozygosity between A alleles was pre-existing in a non-hybrid diploid ancestor but this is in contradiction with the low heterozygosity observed in *M. graminicola* as well as the other diploid *Meloidogyne* species sequenced so far. (II) Hybridization yielded most of the 5% divergence observed between A1 and A2 in an ancestor and was inherited by Mi from an unreduced gamete. This last scenario seems to be the most likely.

Although I agree restitution of first division of meiosis will maintain high heterozygosity, I really don't see how this could explain the topology in Fig. 36d. In the opposite, this should exactly give the same result than scenario (1) and (2) in Fig. 36b. Indeed, it will maintain high divergence between A1 and A2 in the ancestor and this A1-A2 is passed to the offspring of Mi and Mp(Ma3n), then we should observe all the A1s on a branch and all the A2s on another branch, which is not the case. In that sense, I do not really understand the mention in Fig. 4b 'some mechanism that reduces the heterozygosity between homologous chromosomes'. I would say this is just the opposite and the quite high (5%) heterozygosity is maintained between A1 and A2.

Concerning the tetraploid species, Mp cannot be considered a sister species of the parent of Ma and Mj (L355-356) since the mitochondrial phylogeny (Supp. Fig. 1) clearly shows that maternally, Mp is a triploid population of Ma. Another intriguing observation is that in the nuclear phylogenetic analyses (Fig4a), Mp is systematically more closely related to Mj than Ma or any other species. This also does not support calling Mp *M. paraarenaria*.

Furthermore, it is very surprising that none of the topologies presented in Fig. 4a (and Supp. Fig. 35) corresponds to the mitochondrial topology presented in (Supp. Fig. 1). It is expected that at least one subgenome is maternally inherited and thus should follow the mitochondrial topology. In Blanc-Mathieu et al. 2017, one of the nuclear topologies (their Fig 3) exactly reproduced the mitochondrial

phylogeny (their Fig. 7 which is completely consistent with your Supp. Fig. 1). What is the hypothesis of the authors concerning the absence of a maternal topology in the subgenomes phylogenies? I am really surprised by this finding.

Inclusion of more explanation about *M. floridensis* and *M. luci* is really interesting and informative and I congratulate the authors for this addition and further clarification. It is indeed really interesting that the majority of *M. floridensis* match A genomes and this suggest a very close phylogenetic relationship with A. It is also very important that the authors have confirmed *M. floridensis* is itself most likely a triploid species and therefore cannot have passed its three subgenomes as an ancestor but only two of them. The finding that *M. luci* is triploid is consistent with and confirms previous analysis by Susic et al. *J. Nematol* 2020. Therefore this, should be mentioned and the paper (ref58) cited around L370-371.

The new Supp. Fig. 39 is also very interesting and informative and strongly suggests that there exist two different categories of *M. arenaria*: one category of triploid populations which contain Mp (and therefore should be called Ma3n) and one category of tetraploid populations which contain Ma. Interestingly this dichotomy between triploid and tetraploid populations in Ma is perfectly reflected by the mitochondrial phylogeny in Supp. Fig. 1. I guess *M. arenaria* KP202350 is not present in supp. Fig. 39 because no genome data is available for this population? In any case this finding is very important and suggests *M. arenaria* is not a valid species but a complex of species with different ploidy levels. In that sense and following a previous comment, it would be interesting to try to reinforce the support values of the mitochondrial phylogeny in Supp. Fig. 1. This might hopefully be achieved by adding the ribosomal mitochondrial genes.

TE dynamics

Similarly, the new Supp. Fig. 40 is very interesting and gives another dimension to the analysis. I congratulate the authors for this much more detailed analysis of TEs. I would just suggest more explanation for non-specialists. For instance, explain that a whole population of TEs show a peak at very high identity, suggesting they have recently expanded in these genomes and might still be active. Interestingly, similar conclusions were drawn in *M. incognita* in (Kozłowski DKL et al. 2021 *Evolutionary Applications* 'Movements of transposable elements contribute to the genomic plasticity and species diversification in an asexually reproducing nematode pest'). L379 I guess the author mean 'although' TE multiplication (or activity) is ongoingbecause hybridization is most likely to be recent.

4-Expression data done with biological replicates

It is now clearly indicated that 3 replicates were used for the 5 developmental life stages. The results are very interesting and show that although most homoeologous gene copies maintain similar expression levels between the subgenomes, some show biased expression in one or several subgenomes and this bias is not conserved across species.

To make sure I completely understand, I have the following questions:

- It is not clear to me whether expression bias in any of the five developmental life stages was considered as the minimal criterion to consider the gene had an expression bias across subgenomes or if the bias should be observed in multiple life stages. Could you please clarify?
- I don't see any statistical test for expression bias in subgenomes, only the shortest Euclidian distance is considered. Applying a statistical test to confirm the difference in the distribution of expression values is different from the rest of the subgenomes is necessary.

Other important remarks:

Data availability, reproducibility, and FAIR principles

I appreciate the authors have deposited the code they used and many associated data in a GitHub repository. However, some of their main data (genome assemblies and annotations) were deposited on a website from their laboratory. However, all these data should rather be deposited in global public repositories such as the NCBI, EMBL ENA or WormBase Parasite.

Introduction

I acknowledge and appreciate the efforts made by the authors to reinforce all the introduction and better place it in the context of the rest of the literature. However, some of the references cited seem

not to be at the appropriate position in this paragraph.

L78 ref 42 (Lunt et al. 2014 PeerJ) as well as ref 23 (Blanc-Mathieu et al. 2017 PLoS Genetics) should be cited here together with reference 20. Or even better, maybe cite first reference 20 as early hypothesis of reticulate evolution later confirmed using genome data by references 20 and 42.

L85 'combination of closely related females with more diverse 'paternal' rather than 'parental' lineages and reference 23 should be cited here as it is exactly the hypothesis made in this paper.

L91 'M. floridensis and another yet unknown species' reference 23 should be removed from the list here because this hypothesis has not been proposed in this paper.

L102- 103 I would be more cautious about the repetitive sequences identified at the ends of contigs and scaffolds. Although they were found at the extremities, exactly where telomere repeats are expected to be observed, at this stage we absolutely don't know whether they play the same role of protecting chromosome ends. Therefore, instead as 'act as the telomeres of polyploid RKN', it is safer to mention 'that might act as functional telomeres of...'. At this stage we have absolutely no evidence about their function.

Secretome

It is very interesting that, in the predicted secretome, proteins seem to feature higher sequence divergence between A and B genomes than the rest of proteins. I wonder whether higher divergence among the secretome would be a more general feature and not necessarily related to hybridization. For instance, in the rest of the Meloidogyne species that are not polyploid (e.g. M. graminicola), do you also observe more divergence in duplicated (cis, tandem or dispersed) proteins from the secretome than for the rest of the proteins?

Phased vs. unzipped terminology?

Phasing usually refers to diploid heterozygous genomes in which the two haplotypes are phased out according to SNPs between alleles. In the case of Clade I Meloidogyne, the situation is different because the different genome copies have no allelic relationship but are homoeologous subgenomes resulting from hybridization events and displaying quite high divergence. I wonder whether unzipped would be a better adjective than phased in that situation?

Repeats analysis (Supp. Fig.7 and Supp. Table 6)

The proportion of repetitive sequences (TE and other repeats) seem to be much higher in polyploid RKN than Mg. It also seems to be correlated with the genome size (the bigger the genome, the higher the percentage of repeats). Similar results were obtained in old versions of the polyploid genomes and the genome of M. hapla in (Blanc-Mathieu et al. 2017) and it was hypothesized that TE underwent multiplications in the polyploid genomes. Could the author comment these previous findings and do their new findings support or contradict this hypothesis?

Mg seems to have a proportion of low complexity repeats more than twice higher (>6%) than polyploid Meloidogyne (<3%). Do the authors have any hypothesis to explain this observation? Is it due to expansion of any particular low-complexity repeat?

Distribution of coverage along scaffolds (Supp. Fig. 26).

This Figure is interesting as it reveals variations in coverage across the scaffolds and suggests some scaffolds (partially) result from collapsed homoeologous copies. It is still intriguing that the standard deviation in Mg and Mi seems much bigger than for the two Ma (3n and 4n) genomes and for Mj. Do the authors have any hypothesis that could explain this? Furthermore, is it possible that within a same scaffold some regions are collapsed while others are not (= variations in coverage within a scaffold). Is it visible in the 100kb windows defined by the authors (L959-960)?

Prediction of protein-coding genes (methods)

The authors seem to have employed a quite sophisticated and comprehensive methodology to predict genes in the Meloidogyne genomes (including ISO-seq data in M. incognita). L1001 -1003 it is indicated that BUSCO was used to evaluate the completeness of the five predicted proteomes (not genomes I guess). Given the quality of the genomes and the gene prediction procedure, I expect protein-level BUSCO scores to be very good. However, I did not find these BUSCO results at the

protein level. A supplementary table presenting these BUSCO results and comparison with previous versions of predicted proteins in Meloidogyne species would be interesting.

AABB subgenomes in tetraploid genomes

In Supp. Fig. 12 and in the main text L183-185, it is indicated that 2:2 topologies were dominant and thus suggested an AABB subgenomes structure. This is interesting but do you also observe among the three possible 2:2 topologies one that seem dominant (A1,A2)(B1,B2) or (A1,B1)(A2,B2) or (A1,B2)(A2,B1)? I understood that at this stage of the analysis the distinction between A1 and A2 or B1 and B2 is random but at least between A and B this is possible; so is the most frequently observed topology separating the A genomes from the Bs?

Synteny between Mg and polyploid RKN and chromosomal fusions / losses (Supp. Fig. 13)

This Figure suggests that not only chromosomal fusions occur but also some chromosomal losses. For example, in Mj, there are only two copies of Mg chromosome 18 (an only in the B genomes), suggesting two copies have been lost in the A genomes (4 copies are expected in total). Conversely, some regions present in the polyploid RKN have no equivalent in Mg. For instance, the two last chromosomes in Ma B2 and Mj A1 seem to have no equivalent in Mg. Therefore, L197-199: I would say that chromosomal copy losses can also be expected in addition to fusions to explain the variations in karyotype observed in RKN.

Telomere-associated genes and telomeric repeats

It remains not completely clear how genes 'involved in telomere formation in *C. elegans*' were selected. Does the list of telomere-associated genes come from a publication? Or was just the keyword 'telomere' searched in WormBase? Could the authors clarify how they obtained this list of *C. elegans* genes involved in telomere formation (and I guess some are just involved in maintenance, or have unclear roles and are just associated with telomeres)? Maybe the authors can write a few words about the predicted or demonstrated functions of these genes and how it is related to telomere maintenance or elongation? After a rapid check, it seems all these genes have the keyword telomere in their description in Wormbase excepted *dve-1*. What kind of function *dve-1* is supposed to play regarding telomeres? Also, to remain consistent with the rest of the genes in Fig2. A, R06A4.2 as well and T12E12.3 should be renamed *tebp-1* and *tebp-2*, respectively. In the methods section L 1050-1056, I guess the author meant the *C. elegans* proteins were aligned to the proteomes of the selected species in Clade IV nematodes not the reverse? Later in the same section (1059-1060), I think there is an error on the telomere sequence, it is supposed to be (TTAGGC)*n* in *C. elegans* and not (GAATCC)*n* in the material and methods or even (TAAGCC)*n* in Figure 2a. So, please clarify exactly which repeat was searched in the genomic reads of the Clade IV nematodes. Finally, same remark as previous for BLASTp, I guess it is the telomeric repeat motifs that have been searched against the reads using BWA and not the reverse.

The finding of an unusual complex repeat in Mg located at the ends of the scaffolds is very interesting and the fact that the G-rich motif is in the 3' strand is consistent with known telomeric repeats in other species. This strongly suggests this repeated motif has replaced the classical telomeric repeat of nematodes in Mg. The authors need to explain how they have detected putative G-quadruplex forming regions in these genomes? Which software and parameters were used to identify G4 quadruplexes? This is not explained in the methods while presented as a result in Fig2 b.

The way a repetitive motif was found in Mi, is totally unclear and casts doubts on the validity of the initial scaffolding. L216-217: 'The reason for this might lie in contig misanchoring in polyploid RKNs assemblies...' This statement reinforces my doubt about the validity of using Mg as a guide to assemble Mi scaffolds as well as the other polyploid RKNs. The fact that the search for enriched motif had to be re-performed at the contig level confirms that something went completely wrong during Mg-guided scaffolding and that an assembly without a priori should be used instead of the Mg-guided one. Also, it seems really surprising that the Mi-Tel motif was initially found from one single contig considered to be well assembled by optical map. How was this specific contig chosen? It seems very unlikely by, just by chance this motif was identified from just one contig. How many contigs displayed this Mi-Tel motif at one end or at both ends? Fig. 2c is also misleading and confusing as currently represented. How many BioNano molecules have been grouped together in the upper track? What is

this extra ~100kb sequence in the upper track that does not align with the contigs (lower track). My initial guess was that this long sequence at the beginning represents telomere repeat arrays in which no BioNano tag is present. However, this is in contradiction with the lower track in which a lot of (green) tags are represented in the circled region labelled as Mi-Tel. Therefore, the authors should clarify this contradiction and indicate whether or not Mi-Tel contains the BioNano tag. If yes, then the upper track is wrong and if not, this is the lower track that is wrong.

Another point that is unclear is the telomere-to-telomere fusions of some contigs or scaffolds? Could the author explain whether they observe these fusions only at the scaffold level or also within one contig? Do the identifiers 'Mi_ChrX' represent contigs obtained by SMRT DeNovo or Mg-guided scaffolds? Or ONT contigs? In Fig. 15b Mi_Chr12 is represented 4 times in 3 different maps. It is not clear if (3) is the continuation of (2) or if it is separate molecules. In that case it would mean that Mi_Chr12 was cut into different pieces? Furthermore, confirmation with ultra-long read ONT data (>50kb) is quite weak with only one long read supporting this fusion. Rather than providing a few examples of some contigs that contain both Mi-Tel and Mi-Tel reverse in arrays, it would be more convincing to provide an information about coverage by ONT reads. What is the coverage of these candidate fusion points by ONT long reads? And is the coverage comparable to that observed just before and just after the fusions? Finally, assuming that these 4 definite end-to-end fusions are true, why these are not represented in Fig. 3? I would expect to clearly see Mi-Tel + Mi-Tel reverse arrays in the middle of some of the manually edited chromosomes in Fig3a but it is not the case. Could the author clarify whether or not these fusions actually exist and if yes indicate them in Fig3a?

Recovery of collapsed regions

It is currently not very clear how collapsed regions have been recovered. I understood they were identified based on coverage but once they have been identified, how were these regions re-separated? Were they just duplicated as 100% identical regions or unzipped using phased variants in long reads? The authors should more clearly indicate how this process was done. Furthermore, in Supp. Fig. 26 it seems that the standard deviation in coverage (depth) is much higher within Mi chromosomes (~ 0- >2) than for the rest of the RKN species (all >0 and <2). Could the author provide explanation / hypothesis about this difference in coverage distribution?

Comparison between *M. graminicola* and *M. chitwoodi* genomes (Supp. Fig. 27)

Only 20 *M. chitwoodi* contigs were included in this dot plot comparison while this version of the genome (GCA_015183035.1) contains 30 contigs. Therefore, I wonder whether the 10 other contigs simply don't match the *M. graminicola* assembly or if there is any other reason?

Distribution of features along the chromosomes.

L276 – 279. It is indicated that chromosome arms were relatively repeat rich and gene poor compared to chromosome centers (repeat poor and gene rich). However, I did not find in the methods which regions were considered arms and which ones centers? Did you select for instance the 15% first and 15% last Kbs of each chromosome compared to the rest? What were the criteria used and are these differences in numbers of TEs / Mb and genes / Mb statistically significant (no statistical test seems to have been performed). Same remark for the histone marks. Supp. Fig. 29 indeed suggest a co-localization of centromere repeats and H3K9me3 marks and an enrichment of H3K4me3 in subtelomeric regions. These two very interesting results would be more convincing if statistical tests confirmed these enrichments and co-localizations.

Discussion

The new discussion is clearly improved and much more comprehensive, and I congratulate the authors for this. However, I do not agree the strategy for genome assembly proposed by the authors should be considered a guideline for other assemblies (L447-448). In their quest to obtain T2T chromosome-level assemblies, the authors have extensively edited manually their assemblies with strong a priori concerning telomeres and synteny with a distant outgroup species. I would not recommend this strategy.

L492-494: I fully agree that according to the results provided by the authors, the definition of *M.*

arenaria as a species is vague and should be reinvestigated. At least this is what suggests Supp. Fig. 1 at the mitochondrial level. Unfortunately, the branches supporting the separation of *M. arenaria* into two clades are only poorly supported. Hopefully, addition of ribosomal mitochondrial genes to the supermatrix would allow better resolving these branches.

L495: We also 'confirmed' not 'determined' that *M. luci* is a triploid. Indeed, this was already shown in the *M. luci* genome paper by Susic et al. 2020 Journal of Nematology, and this paper should be cited there.

Typo and minor suggestions

L34: homologous gene expression "change" or "decrease" or "attenuation" rather than 'suppression'.

L106: which 'reproduces' with both sexual reproduction and meiotic parthenogenesis...

L112: I would tone down the claim that these are 'extremely highly accurate' resources. These are undoubtedly high quality and useful data but it is not necessary to provide a judgement of value in the introduction of the paper. This kind of statements and succession of superlatives might be used for the cover letter but not the paper itself.

L140-141: 'suggesting the homoeologous subgenomes have been correctly separated during genome assembly' rather than 'indicating that all...'

L154-155: the numbers provided in the text for BUSCO scores and % of reads mapping the genome do not match those indicated in Supp. Table 5. So which numbers are the correct ones?

L177: I think it is Fig. 1d here rather than 1b?

L328: 'these allopolyploids' instead of these 'allotetraploids' because two species are triploid here.

L402: 'Since effector proteins play' rather than 'Since effectors play'

L405: 'proteins predicted to be secreted' and (including putative effectors).

L421: 'decrease of homoeologous gene expression' rather than 'suppression of its homologous gene'.

The gene is not suppressed, only the expression is decreased.

L430: 'unzipped' might be more appropriate than 'phased' here?

L431-432: 'these genomes reveal a complex structure' rather than complex genome structure to avoid repeating the term 'genome'.

L737-738: (Fig5. Legend) J2 stage (not period). Fig5a: in the histogram the labels of the different categories are not in front of the bars and consequently the last bar has no label.

L794 one to three 'SMRT' Cells to avoid confusion with the cells of the organisms themselves.

L1065: I guess you want to cite Supp. Fig. 22 here?

L1087: I guess the authors mean 'seed' sequences, not 'sed' sequences?

L1165: containing 100% identical duplicated genes not '100% similarity genes'.

Reference list: reference 23 and reference 27 are the same.

Reviewer #2:

Remarks to the Author:

The additional text in the Introduction helps to better contextualize the work presented in this manuscript. Overall, I think the manuscript is much improved and I am impressed by the follow up to the extensive critical assessment from the reviews. I would like to congratulate the others on the tremendous amount of work and high-quality data generated here. If the few remaining concerns I have below are addressed I endorse this manuscript for publication.

Broader concerns:

I think referring to the Mp lineage with a new species name is still problematic. The Mp fits within a clade of other Ma triploids (according to the Mt tree), so I don't see how it could be considered a distinct species unless every branch in Figure S1 is a different species. Furthermore, because mitochondrial phylogenies may be discordant with the nuclear phylogeny and whole genome information is available here, I'm not understanding how the Mp lineage requires a distinct name from Ma based on the Mt genome, alone. I think lines 444-446 make it clear that the evolutionary history of this system is complicated and there are probably numerous lineages of distinct subgenome

combinations that might otherwise appear to be the same species, even mitochondrially.

I was glad to see some deeper analysis of the TE content in these species. I think the fact that many full-length LTR elements with 100% identity between LTR sequences were recovered in these genomes indicates ongoing activity should be more explicitly stated in the text, though. In text the contrast of the A and B sub genome similarities to what's been seen in *Xenopus* could be stated as well to help clarify what might have been expected if bursts of TEs occurred after hybridization. Essentially, I think the hypothesis of TE activity related to hybridization/polyploidy could be clarified and that these results do not support the expectation because most of the TEs are old could be more explicitly stated.

Specific comments:

Line 40: I don't want to get overly pedantic, but self-fertilization is sexual reproduction.

Lines 43-46: "partially answer" is a bit strong. I think something like "are hypotheses with empirical support" more accurately reflects the status of those ideas.

Line 370: "proved" could be changed to "supported".

Line 374: A "hypothetical evolutionary model" of what? The expectation of TE burst in hybrids and polyploids?

We thank both reviewers for their very constructive comments which help us to heavily improve the manuscript. Through careful thinking, rigorous analysis, and thorough verification, we have addressed the concerns raised by the reviewers in the following letters in a point-to-point manner.

The major improvements include (1) Figure 4 and some supplementary figures have been updated; (2) “Ma 3n” and “Ma 4n” are used according to the reviewer’s suggestion, and information of species identification are provided; (3) A more accurate description and a workflow for the T2T chromosome-scale genome assembly strategy has been added into the Method section, which shows all contigs are supported by Hi-C data to determine their location instead of ‘forcing’ them to be in relying on manual editing, such as gain telomeres at both ends; (4) We used the ONT contigs + optical maps to assemble a Mi genome without manual editing, and compared it with the T2T genome to verify the T2T assembly; (5) The hypothesis of the origin of the allopolyploid RKNs has been adjusted; (6) More analysis on the evolutionary relationship of Ma 3n, Ma 4n and Mj were performed; (7) More details in genome assembly and telomere identification have been included.

REVIEWER COMMENTS

Reviewer #1 (Remarks to the Author):

First of all, I would like to congratulate the authors for the remarkable efforts made to address as much as possible my previous comments and for the improvement of the manuscript. This new version is much clearer, much better written and many results are now more convincing and many of my previous points can be considered addressed.

The major concern that remains from my side is still related to the assembly strategy. In their quest to obtain T2T chromosome-scale assemblies, the authors have extensively edited manually the assembly to meet their expectations. After careful observations of the optical map data, I am not convinced this manual assembly is strongly supported.

Response: The major concern of the reviewers regarding our assembly strategy is caused by the lack of clarity in the description of our **Method**. Briefly, we first used BioNano map to construct scaffolds, then used Hi-C to further construct scaffolds, and then tried to find telomeres at both ends to prove that all scaffolds reached the chromosome level. We did not force assembly genome to gain telomeres at both ends, but just find the missing telomeres after scaffold construction (Response Figure 7). In other words, we scaffolded contigs into one chromosome, not to make up two telomeres, but because there is a Hi-C signal between them or a support of BioNano map. We have edited the methodology description for T2T genome construction to make it more clear. We have added proper legend in BioNano map alignment and provided Hi-C evidence for doubtful chromosome to target reviewer’s concern (please see **Response Figure 7 and 10**).

For Mi, I would recommend only using the hybrid assembly obtained by BioNano Solve software based on the ONT contigs + optical maps without manual edition. Or at least provide this hybrid ‘Solve’ assembly and a dot plot alignment with the manually-edited assembly to figure out possible discrepancies.

Response: We generated two hybrid assembly based on Bionano + PacBio contigs and Bionano +

ONT contigs, and provided the dot plot alignment between those two hybrid assembly and our T2T assembly to target reviewer's concern (please see **Response Figure 8 and 9**). The results show that our T2T assembly has high accuracy.

In my opinion, all the other results provided by the authors are sufficiently convincing and interesting to justify an eventual publication in a high-standard journal like Nature Communications and these 'forced' T2T assemblies bring too much uncertainty and are not essential regarding the rest of the results.

Response: Thanks for the positive comments on our work. In our assemblies, all contigs are supported by Hi-C data to determine their location instead of 'forcing' them to be in relying on manual editing. According to suggestions, we have provided more information and revised our manuscript to make it clearer.

Other main points that need to be addressed are listed below:

1-Taxonomic identity of Meloidogyne species and *M. para-arenaria*

I realize that the authors have deleted a very important information that was initially present in the previous version of the methods section. The information about SCAR markers is essential and absolutely needed to understand how species identities were assigned. Without this crucial information, it is impossible to understand why in the results section (L 117 – 119) it is indicated that 5 Meloidogyne species were collected: Mi, Mj, Ma and Mg as well as an unknown species. In the new material & methods there is no more information on how species identity had been originally determined. Because it is almost impossible to differentiate these species based on morphological characteristics, this information about SCAR markers should be reintroduced and the PCR results should be provided. Otherwise, it is impossible to understand why Mp was initially unknown and called *M. paranarenaria*.

Interestingly, and I congratulate the authors for this, the mitochondrial phylogeny I had suggested to perform (Supp. Fig 1) reveals several important things. First, it confirms that Mi is *M. incognita* and that Mj is *M. javanica*, so there is no doubt for these two species. However, it also shows that based on the mitochondrial genomes, what the author call *M. paraarenaria* falls right in the middle of some other *M. arenaria* populations (that are 3n according to the authors further analysis). Finally, the species that the authors call Ma (because it does present a Ma SCAR marker) falls in a small poorly supported phylogenetic group with two other *M. arenaria* populations (that are 4n). According to this phylogeny, it seems, indeed, that *M. arenaria* is not monophyletic and therefore maybe not a valid species.

Because this topology suggesting *M. arenaria* is paraphyletic and not a valid species is only moderately supported in the tree, maybe adding the two mitochondrial ribosomal genes in addition to the protein-coding genes in the supermatrix might help resolve better the tree. This is an important result that needs confirmation because I don't think any previous phylogeny based on mitochondrial markers had revealed *M. arenaria* was paraphyletic. My hypothesis is that, at this stage, there is not enough phylogenetic signal in the concatenated matrix to resolve the topology and I hope adding ribosomal genes will help resolve it.

Another suggestion is to perform some virtual PCR in the genomes of the 3n and 4n populations

of *M. arenaria* (if genome assemblies are available) and determine whether the SCAR marker would be present / absent in these other populations. Alternatively, and I don't know whether the authors have the possibility to perform this kind of experiments in their lab, it would be very interesting to check whether esterase profiles of Mp and Ma are the same and at the expected pattern for *M. arenaria* or are different. At least, I guess it is possible to check in the original publications of the other *M. arenaria* populations (particularly the 3n ones) which markers / information (if any) were used by the authors to state their species were *M. arenaria*.

Whatever the final conclusions will be on the validity of *M. arenaria* as a species, I would not recommend at this stage to call Mp *M. paraarenaria* but *M. arenaria* 3n and Ma should be called *M. arenaria* 4n. Indeed, substantial variations in ploidy levels have been observed among populations and isolates of *M. arenaria* and it is not surprising to observe both 3n and 4n populations for this species (Cytogenetics, cytotaxonomy and phylogeny of root-knot nematodes, Triantaphyllou, 1985). What is more surprising is that they form two separate branches in the mitochondrial phylogeny and not a single monophyletic branch, but this needs to be further clarified.

Summary of reviewer's question:

Q1. How RKN species identity had been originally determined.

Q2. Topology suggesting *M. arenaria* is paraphyletic and not a valid species is only moderately supported in the tree, adding the two mitochondrial ribosomal genes in the supermatrix might help resolve better the tree.

Q3. Whether the SCAR marker would be present / absent in Ma 3n and Ma 4n population.

Our response:

Q1: How RKN species identity had been originally determined.

Summary: In this study, Ma, Mi, and Mj can be clearly identified by SCAR-PCR, Mp do not have a target band in initial PCR experiment. Based on Illumina reads analysis, both Mp and Ma sample have the Ma-SCAR target sequence. From this viewpoint, the Mp sample should be Ma, but not a new species. So we have taken the very good suggestion of the reviewer, used "Ma 4n" and "Ma 3n" to replace the former Ma and Mp. In addition, our results indicate that Ma 3n, Ma 4n and Mj may have equal evolutionary positions. So we kept both "Ma 4n" and "Ma 3n" for the evolution story of Clade I root-knot nematode in the revised manuscript, and let the species problem for future work.

The reason we initially defined *M. paraarenaria* as unknown species are as follows:

When we initially perform the Mi/Ma/Mj/Mh/Me-SCAR identification, there was no target band in Mp samples. We thought that it might be a new species and thus sequenced this sample. Because the sequencing data had the highest mapping rate with published Ma genome, and its ITS rRNA gene had the closest relationship with Ma, it was originally named *M. paraarenaria*. Subsequent syntenic analysis after genome assembly found that Mp is triploid and Ma is tetraploid, which makes us think that Mp is indeed a new species different from Ma.

To finally solve the definition problem of Mp, we carried out the following work:

1. After receiving reviewers' comments, we checked the Ma-SCAR target sequence in genome sequencing data of Mp, and found that Mp indeed contains the Ma-SCAR target sequence

(**Response Figure 1**). In brief, we mapped Illumina reads against published Ma genome (GCA_003133805.1), and checked reads pileup in Ma-SCAR target region (tig00000449:174235-174642, identified by MUMmer). We found that all of Ma samples including our Mp sample completely covered the Ma-SCAR target region. Moreover, in assembled contigs of Mp, complete Ma-SCAR target sequence are found in contig_280:2099744-2100148 (identified by MUMmer). As Mp has Ma-SCAR marker, and there was no band in the first SCAR PCR experiment, it might be an PCR error in the experiment. We carefully checked the gel map of the experimental record (performed in 2018-11-01) which was initially thought to have no Ma-SCAR band of Mp, and found that Ma-SCAR had a very faint band (**Response Figure 2b left**).

2. Unfortunately, the original sequenced Mp sample was extinct in our greenhouse during the COVID-2019, we could not perform SCAR-PCR and the esterase experiment. We understand that Mp related work cannot be replicated due to sample death, but Ma of the triploid population is a common population. Our lab has another triploid Ma sample, mitochondrial tree and read depth of subgenomes indicates that this sample is the same species as Mp (**Response Figure 3**). This sample showed a strong band signal when performing Ma-SCAR PCR (**Response Figure 2b right**). Above results indicates that based on Ma-SCAR, Mp samples should be initially defined as Ma, rather than as a new species.
3. Overall, the PCR error prompted us to sequence both triploid and tetraploid Ma without our knowledge. We have added a description of the motivation for sequencing these five nematodes in **Methods (L753-767)**.

Here, we thank the reviewer for the naming suggestions of Ma 3n and Ma 4n, which resolved the contradiction between the same species identity based on Ma-SCAR and the different ploidy of species.

In revised version, we have edited Mp to Ma 3n, Ma to Ma 4n, and provided SCAR PCR results for Mi, Mj and Ma 4n, and provided virtual PCR results (reads coverage in Ma-SCAR target) for identification of Ma 3n sample (**Supplementary Fig 1**). We elaborated a clearer motivation for sequencing these five nematodes, more detailed description can be found in **Methods (L753-767)**.

Response Figure 1 Read coverage in Ma-SCAR target sequence. Reference sequences are published Ma genome (GCA_003133805.1). Ma-SCAR target region are located at `tig00000449:17235-174642`. Although PCR bands are absent in the original Ma-SCAR experiment of Mp (*M. arenaria* Yunnan) samples, reads of Mp (*M. arenaria* Yunnan) completely covered Ma-SCAR target region. This evidence supports that Mp (*M. arenaria* Yunnan) are still *M. arenaria* based on SCAR analysis. **This figure can be found in Supplementary Fig. 1.**

Response Figure 2 (a) Identification of root-knot nematode species by SCAR characteristic primers. **(b)** The left picture is the SCAR identification of the species defined as *M. paraarenaria* in the early stage of this study. At that time, only 18S was considered to have a band, but there was a very faint band in Ma-SCAR which ignored by us before. The picture on the right shows the

Ma-SCAR amplification results of another triploid Ma cultured in our laboratory. The red arrows shows the PCR bands. This figure can be found in **Supplementary Fig. 1**.

Response Figure 3 Mitochondrial tree (a) and read depth of subgenomes (b) showing that another sample in our lab is the same species as Mp.

Q2: Topology suggesting *M. arenaria* is paraphyletic and not a valid species is only moderately supported in the tree, adding the two mitochondrial ribosomal genes in the supermatrix might help resolve better the tree.

Reply:

1. As suggested by the reviewer, we have added two ribosomal genes (12S rRNA and 16S rRNA) for phylogenetic tree building. We also modified the way bootstrap values are presented in the mitochondrial tree. After the addition of two ribosomal genes, there was no significant improvement in the bootstrap values (**Response Figure 4**).
2. To avoid the impact of outgroups (Me) on the topology, we provided unrooted tree (**Response Figure 5**). Unrooted tree shows that Mj, Ma_3n, and Ma_4n originated from one ancestral node and quickly diverged into three independent lineages, and these three samples do not exhibit a clear binary phylogenetic relationship but display a trifurcating structure.
3. So, we predicted that the reason for moderately supported may be: (1) Ma 4n, Ma 3n and Mj are too similar to each other in mitochondrial genome; (2) Ma 4n, Ma 3n and Mj form a trifurcating tree rather than a bifurcating tree.

In revised version, we have added this unrooted tree into **Supplementary Fig 2**.

Response Figure 4 Mitochondrial tree for Clade I RKNs. Left tree is built from protein coding gene, right tree is built from protein coding gene and rRNA gene.

Response Figure 5 Unrooted tree for Clade I RKNs, except for *M. enterolobii*. The maximum-likelihood tree is constructed using mitochondrial protein coding gene and ribosomal genes. This figure can be found in **Supplementary Fig. 2**.

Q3: Whether the SCAR marker would be present / absent in Ma 3n and Ma 4n population.

Reply:

Both Ma 3n and Ma 4n contain the expected Ma-SCAR target sequence, making it difficult to distinguish between these two populations using SCAR PCR (**Response Figure 1 and 2**). The genome sequencing of these two distinct Ma samples in this study was made possible due to an initial SCAR PCR error. Without this error, we would have only sequenced one Ma sample instead of two.

2- Anchoring Mi, Mj and Ma assemblies on Mg genome and manually editing to obtain telomeres at both ends.

I am still not convinced using the genome of Mg as an anchor to help scaffold the genomes of the polyploid genomes was an ideal strategy. I hypothesize this can include a lot of biases and over-scaffold the polyploid genomes in a multiple of 18 chromosomes.

I also don't think it is absolutely necessary to provide telomere-to-telomere assemblies. These manual editions of the genome assembly to eventually obtain telomeres at both ends of all the scaffolds might also introduce several biases.

For *M. incognita*, the authors seem to have generated a non-forced and not manually edited assembly from the contigs using optical mapping data (using Solve software L 935-936).

This is a very valuable and interesting point of comparison that has not been utilized enough. According to the methods section (L 1094 – 1098), for *M. incognita*, the contigs used were those assembled from ONT data via CANU and then polished via Nextpolish.

In addition to this hybrid assembly the authors have also apparently aligned contigs the optical map data and made a manual assembly starting from contigs containing Mi-tel motifs. It is unclear whether this new assembly was only based on ordered ONT contigs or a mixture of both the ONT and PacBio contigs? For instance, in Supp. Fig. 22, in the absence of a proper legend, it is impossible to know what is represented in the upper track (data from optical mapping?) and lower track (ONT contigs, PacBio contigs ?). In addition, on the first chromosome of Supp. Fig. 22, there seems to have a gap at around 8 Mb where tags of the optical map don't match tags of the contigs. Therefore, the validity of this scaffolding can be questioned. According to Supp. Table 10, it is indicated that the longest reconstructed scaffold from the molecules is 8.2 to 8.8 Mb long (haplotype vs. nonhaplotype). Therefore, chromosome I which is >14.7Mb necessarily represent the assembly of several long molecules and the support of this gap by molecules pileups is important. The authors should provide BioNano molecules pileups to allow checking for support of these regions by molecules. At this stage it is impossible to assess the validity of the assembly based on examination of Supp. Fig. 22 which also needs a legend and probably another title (I guess this is alignment of the optical map with contigs and not chromosomes). It seems that extensive manual editing was used to obtain these 40 chromosomes and without strong support for the points of junctions between the contigs (by molecule pileups for instance), the validity cannot be assessed.

Also, L262-263 it is indicated that the same strategy was used to assemble Mi, Ma3n and Ma4n. It cannot be the same strategy because neither ONT long reads not optical map data were generated for these 3 other polyploid RKN samples. Therefore, the authors should clearly state how the strategy differed for this species as it was necessarily based solely on PacBio and Hi-C data.

Finally, Fig 2d as well as Supp. Table 15 are confusing to me and again cast doubt on the validity

of the manually edited assembly. Indeed, it seems that all the biggest scaffolds (or chromosomes) result from the fusion of A and B subgenomes) while this is never observed for the other smaller chromosomes.

Actually, all the interesting results provided by the authors concerning the hypothesis about the origin of polyploidy, the TE dynamics as well as the expression level shift between subgenomes will remain true on genomes assembled without an external species as a guide and without adding telomeres everywhere.

Summary of reviewer's question:

Q1: Strategy that using Mg as an anchor to help scaffold the genomes of the polyploid genomes may be not ideal.

Q2: T2T assemblies obtained manually might introduce several biases and be over-assembled.

Q3: The hybrid assembly obtained by BioNano Solve software based on the ONT contigs + optical maps without manual edition should be compared with Mi T2T genome.

Q4: Clarify the new assembly is based ONT contigs or PacBio contigs.

Q5: Supp. Fig 22 need a proper legend.

Q6: The authors should provide evidence, such as BioNano molecules pileups, to support junctions between contigs in chromosome I.

Q7: Assembly strategies for the other 3 polyploid RKN samples need to clearly state.

Q8: Fig 2d as well as Supp. Table 15 need to clarify

Our response:

Q1: Strategy that using Mg as an anchor to help scaffold the genomes of the polyploid genomes may be not ideal.

Reply:

We generated two versions of assembly, in which assembly v1 use Mg as an anchor, while **assembly T2T do not.**

For assembly V1 construction:

Contigs ordering and orientation were initial determined by Hi-C in Juicebox. Most work about adjustment of contigs position and orientation has been done in this stage. For a few left contigs with uncertain position and orientation from Hi-C signal, we referred to their collinearity with Mg to confirm it. In addition, many chromosomal fusion events were uncertain at this stage, so we also referred to their collinearity with Mg. We only used Mg as an anchor at this stage.

Although assembly V1 does not reach the chromosome level, it does not mean that assembly V1 contain big mistakes. We provided a dot plot of assembly V1 vs assembly T2T (**Response Figure 6**), which indicated that many scaffolds have already reached chromosome level.

The errors in assembly V1 are mainly reflected in: (1) some real chromosomes are segmented into 2-3 scaffolds in assembly V1; (2) the orientation of a few contigs is reversed, especially the contigs at the borders.

Assembly V1 clearly reveals the genome structure of polyploid genome are AAB or AABB, and from a few scaffolds of assembly V1, we identified the telomeres. This information is critical to our subsequent construction of T2T genome. In fact, our current experience is sufficient to directly construct the T2T genome, but if removed assembly V1, many prior information will have no source.

Response Figure 6 The dot plot of syntenic region between Assembly V1 and Assembly T2T. The syntenic region are identified using MUMmer (--mum), and only 1-to-1 alignment region are shown.

Q2: T2T assemblies obtained manually might introduce several biases and be over-assembled.

Reply:

1. Briefly, we first used BioNano map to construct scaffolds, then used Hi-C to further construct scaffolds, and then tried to find telomeres at both ends to prove that all scaffolds reached the chromosome level. We did not force assembly genome to gain telomeres at both ends, but just find the missing telomeres after scaffold construction (**Response Figure 7**). In other words, we scaffolded contigs into one chromosome, not to make up two telomeres, but because there is a Hi-C signal between them or a support of Bionano map.
2. We re-stated the methods for T2T genome construction as follows:

For Mi:

- a. PacBio and ONT raw data are assembled into contigs through Canu. We aligned the contigs of PacBio and ONT to the BioNano map. Overall, we used ONT contigs for construction because they are longer than PacBio contigs. For some genome regions, PacBio contigs may be longer, so we replaced ONT contigs with corresponding PacBio contigs.
- b. Based on BioNano map, we generated scaffolds and searched Mi-tel from scaffolds. Among those scaffolds, some reached at T2T level, some have only one end of telomere, and some have no telomere. This step is equivalent to Solve hybrid assembly, but in process of manual construction, we used MUMmer to fill in some gaps based on contigs overlap.

- c. All of scaffolds and other remaining ONT contigs were used to construct chromosomes based on Hi-C data by using Juicer + 3D-DNA + Juicebox. Thus, many chromosomes, such as the longest chromosome, are finally constructed based on Hi-C data but not based on BioNano consensus map. In this step (Juicebox), Mi-Tels could help determine the position and orientation of the boundary contig in Juicebox.
- d. We checked the collapsed region by calculating the sequencing depth in 100kb windows. The breakpoints of the collapse regions were determined using MUMmer. The collapsed region was duplicated as 100% identical sequence and added into the corresponding position according to Hi-C heatmap.
- e. For chromosomes still missing telomeres, we try to find the missing telomeres as follows:
 - i. Extending chromosomal boundary with long reads or other contigs using minimap2.
 - ii. Checking whether the chromosomal boundary is collapsed with other chromosomal boundary, and recovery it.
- f. Accuracy of all chromosomes were evaluated by using Hi-C heatmap, Bionano map alignment and read depth distribution.

For Ma(3n), Ma(4n) and Mj:

- a. PacBio raw data was assembled into contigs by Canu, and then scaffolds were constructed based on Hi-C data using Juicer and 3D-DNA. Results of 3D-DNA were loaded into Juicebox to construct chromosomes, and this round of assembly are only used Hi-C signal as evidence but not used Mg as an anchor. Similarly, telomere motif could help determine the position and orientation of the boundary contig when using Juicebox.
 - b. Recovery collapsed region and finding the missing telomeres as described above.
 - c. Accuracy of all chromosomes were evaluated by using Hi-C heatmap and read depth distribution.
3. Manually edition is mainly reflected in: (1) After confirmed the contigs ordering and orientation, we added 100*N between those contigs. (2) If gap of contigs can be filled with contigs overlap, we concatenate the contigs using the corresponding sequences. (3) Collapsed region are duplicated into corresponding position.

The methodology description in our original manuscript is not very clear. In revised version of manuscript, we have edited the description for methodology of T2T genome construction to make it more clear (Method Line 1138-1165; Workflow: Supplementary Fig. 21).

Response Figure 7 Workflow for constructing T2T assembly for Mi (a) and other 3 polyploid RKNs (b).

Q3: The hybrid assembly obtained by BioNano Solve software based on the ONT contigs + optical maps without manual edition should be compared with Mi T2T genome.

Reply:

As reviewer suggested, we generated two hybrid assembly based on Bionano + PacBio contigs and Bionano + ONT contigs, and provided the dot plot alignment between those two hybrid assembly and our T2T assembly (**Response Figure 8 and 9**). Those two hybrid assembly without manual edition show a high collinearity with our T2T genome, suggesting our T2T assembly is overall accurate.

Response Figure 8 The dot plot of syntenic region between hybrid assembly based on ONT + Bionano and Assembly T2T. The syntenic region are identified using MUMmer (--mum), and only 1-to-1 alignment region are shown.

Response Figure 9 The dot plot of syntenic region between hybrid assembly based on PacBio + Bionano and Assembly T2T. The syntenic region are identified using MUMmer (--mum), and only 1-to-1 alignment region are shown.

Q4: Clarify the new assembly is based ONT contigs or PacBio contigs.

Reply:

Overall, we used ONT contigs for construction because ONT contig length are larger than PacBio contigs. For some genome regions, PacBio contigs may be longer, so we replaced ONT contigs with corresponding PacBio contigs.

We have now included this in **Methods (Line 1140-1143)**.

Q5: Supp. Fig 22 need a proper legend.

Reply:

In supplementary Figure 22, the top track is chromosome sequence, the bottom track is Bionano consensus map. For some chromosomes, Bionano may lack evidence to prove junctions between contigs, but Hi-C signal could further support it.

We have now included this in figure legend (**Supplementary Fig. 23**).

Q6: The authors should provide evidence, such as BioNano molecules pileups, to support junctions between contigs in chromosome I.

Reply:

1. We understand that the reviewer considers that the chromosome I is constructed by Bionano, and there is a Gap in Bionano, so the accuracy of the chromosome is doubtful, and more evidence is needed to prove junctions between contigs.
2. Notably, in assembly T2T, some chromosomes can be directly constructed by Bionano consensus map to the telomere-to-telomere level, and the remaining chromosomes, such as chromosome I, are constructed by Hi-C data. The figure below shows the Hi-C signal between 12 contigs inside chromosome I (**Response Figure 10**). There is a strong Hi-C signal between the contigs inside this chromosome, which can prove that this is a chromosome rather than an over-assembly of multiple chromosomes.

3. The gap shown in Bionano map at around 8 Mb may be due to the error in position and orientation of those short contigs (**Response Figure 10**). Of course, the reason for the gap may also be an assembly error in the Bionano consensus map. Except for the questionable short contigs in the middle, at least the overall assembly of the chromosome I is correct. As suggested by reviewer, we try to provide the molecules pileups, but we failed to visualize the alignment between the raw molecules and the genome. This analysis seems to require a software IrysView, but the software is currently unavailable for download from Bionano website. In fact, the molecule pileups may not support the position and orientation of these short contigs either.
4. In addition, for non-model species, requiring 100% accuracy for whole chromosome seems too stringent, especially for these allopolyploid nematodes with complex genome. In our genome, most chromosomes can be jointly verified by Hi-C and Bionano consensus map, and only a small number of chromosomes may have a few errors in the position and orientation of contigs inside it. Nonetheless, this level of genome assembly is already of high quality compared with the genomes of other plant-parasitic nematodes and is sufficient to address scientific questions involved in this species.

Response Figure 10 Hi-C heatmap between contigs inside chromosome I. The arrow shows some short contigs that are difficult to determine the orientation and position, corresponding to the position of the gap in Bionano map. The bottom table shows the position of each contig inside chromosome I.

Q7: Assembly strategies for the other 3 polyploid RKN samples need to clearly state.

Reply:

For Ma(3n), Ma(4n) and Mj:

- a. PacBio raw data was assembled into contigs by Canu, and then scaffolds were constructed based on Hi-C data using Juicer and 3D-DNA. Results of 3D-DNA are loaded into Juicebox to construct chromosomes, and this round of assembly are only using Hi-C signal as evidence but not using Mg as an anchor. Similarly, telomere motif could help determine the position and orientation of the boundary contig when using Juicebox.
- b. Recovery collapsed region and finding the missing telomeres as described above.
- c. Accuracy of all chromosomes were evaluated by using Hi-C heatmap and read depth distribution.

Now we have included this in **Methods (Line 1162-1165)**.

Q8: Fig 2d(3d) and Supp. Table 15 need to clarify.

Reply:

All of T2T genomes are constructed without Mg as an anchor.

For Supp. Table 15:

Supp. Table 15 is the breakpoint position of the fused chromosome for Mi T2T genome.

Hi-C, Bionano and Mi-Tel at both ends have indicated that the genome assembly is almost correct and has reached the chromosome level. Through synteny analysis, it can be known that smaller chromosomes are unfused chromosomes directly inherited from diploid ancestors, while long chromosomes are fused from ancestral chromosomes. Subsequent analysis needs to split the fused chromosomes into 2-3 ancestral chromosomes, while the unfused chromosomes do not need to be split. What we did was to use the Mg genome as the diploid ancestral genome to determine the roughly breakpoint position of the fused chromosome. Supplementary Table 15 records information on breakpoints of those fused chromosome.

For Fig 3d:

Fig. 3d is used to illustrate that the Mi B genome is more suitable as the ancestral reference of the polyploid genome than the Mg genome

It should be noted that there are three chromosomal rearrangements between the Mg genome and the diploid ancestor of the polyploid genome. Therefore, it is necessary to pay attention to the genome region involving chromosomal rearrangements when identifying the roughly breakpoint position.

After we split the chromosome into 3 copies, we got the whole Mi B genome. When performing synteny analysis between Mi B genome and Mi T2T genome, we found that there were no chromosomal rearrangements as in Mg, except for chromosomal fusion and fission, which suggested that Mi B were more suitable as ancestral chromosomes. For the other three chromosomes, we replaced Mg with the Mi B genome to determine the roughly breakpoint position of fused chromosomes.

3-Hypothesis of unreduced gamete and other hypotheses

I appreciate and acknowledge the efforts made by the authors to explore several different hypotheses to explain the origin of A1 – A2 subgenomes (supp. Fig. 36). However, it seems that the authors make the strong assumption that the two triploid species (Mi and Mp = Ma3n) have the same unique origin for their A subgenomes (an unreduced gamete with or without previous hybridization). Then, they invoke gene conversion (36c) to explain that the observed topology of A1-A2 subgenomes does not match the expected topology of unreduced gametes (36a (1) and (2)) but rather matches their third scenario. It appears completely acceptable that some gene conversion events took place. However, it seems unrealistic that gene conversion events took place at such an extensive rate that now, most of the A1 and A2 genes in Mi and Mp (Ma3n) result from gene conversion events and that the expected topology is never observed in the 14 trees in supp. Fig 35. In all the topologies the A1-A2 subgenomes of Mi are always more closely related to one another whereas the two A1-A2 subgenomes of Mp (Ma3n) are always more closely related to those of the tetraploid species. Therefore, a more likely hypothesis is that Mi and Mp A subgenomes have different origins and that Mp A subgenomes followed an evolutionary history closer to tetraploid species. This hypothesis is consistent with the observation from supp. Fig. 1 which suggests Mp is a triploid population of *M. arenaria*.

I also acknowledge that in supp. Fig. 14b, the information about the nucleotide divergence between subgenomes is now provided. For instance, in *M. incognita*, it seems that the divergence between A1 and A2 is about 4-5% (~96% identity) while the divergence between A and B is around 9-10% (~91% identity). Given the very low divergence at the nucleotide level between the mitochondrial genomes of the Mi, Mj and Ma, I wonder whether this relatively high divergence between A1 and A2 would be compatible with divergence between former alleles? Discrepancy between this high within-species divergence at the nuclear level (A1-A2) vs the very low ‘between species’ divergence at the mitochondrial level rather suggest hybridization at the origin of A1-A2 in an ancestor. Indeed, mitochondrial genomes evolve faster than nuclear genomes in *C. elegans* and most animals and this is expected to be the same in *Meloidogyne*. Thus, the divergence between these RKN species is supposed to be very recent. This is not consistent with long-term parthenogenesis as hypothesized L344-345. To explain 5% divergence between A1 and A2, two alternative hypotheses can be considered. (I) High heterozygosity between A alleles was pre-existing in a non-hybrid diploid ancestor but this is in contradiction with the low heterozygosity observed in *M. graminicola* as well as the other diploid *Meloidogyne* species sequenced so far. (II) Hybridization yielded most of the 5% divergence observed between A1 and A2 in an ancestor and was inherited by Mi from an unreduced gamete. This last scenario seems to be the most likely.

Although I agree restitution of first division of meiosis will maintain high heterozygosity, I really don't see how this could explain the topology in Fig. 36d. In the opposite, this should exactly give the same result than scenario (1) and (2) in Fig. 36b. Indeed, it will maintain high divergence between A1 and A2 in the ancestor and this A1-A2 is passed to the offspring of Mi and Mp(Ma3n), then we should observe all the A1s on a branch and all the A2s on another branch, which is not the case. In that sense, I do not really understand the mention in Fig. 4b ‘some mechanism that reduces the heterozygosity between homologous chromosomes’. I would say this is just the opposite and the quite high (5%) heterozygosity is maintained between A1 and A2.

Summary:

Reviewer offers their views on our hypothesis.

1. The A subgenome of Mi and Mp have different origins:

- (a) Gene conversion won't be so extensive, and trees in supp. Fig 35 does not support this.
- (b) First-division restitution will give the same result as scenario (1) and scenario (2), but not scenario (3).

2. How does 5% divergence between A1 and A2 came from:

- (a) The RKN species diverged recently, and the divergence between A1 and A2 were not accumulated through long-term parthenogenesis, but more like from hybridization.
- (b) The hybridization origin of A subgenome also explain this.

Response:

We would like to thank the reviewer for their in-depth reading and consideration of our manuscript, as well as for their professional suggestions. After learning about the reviewer's description of the influence of gene conversion and first-division restitution on phylogenetic topology, we excluded these two possible hypotheses and accepted that Scenario (3) is the most likely model for the origin of A subgenome. At the same time, Scenario (3) also explained that %5 heterozygosity between A1A2 comes from hybridization.

In the revised version, we edited as follows:

- (a) We removed the explanation about the First-division restitution (FDR) part and described that gene conversion could not be so extensive (**Line 351-354**).
- (b) We modified the figures involving the pattern of origin, such as Fig. 4a, Fig. 4b, and Fig. 4d (**Fig. 4 and Supplementary Fig. 37**).
- (c) We removed the unclear description in Fig. 4b 'some mechanism that reduces the heterozygosity between homologous chromosomes'.

Concerning the tetraploid species, Mp cannot be considered a sister species of the parent of Ma and Mj (L355-356) since the mitochondrial phylogeny (Supp. Fig. 1) clearly shows that maternally, Mp is a triploid population of Ma. Another intriguing observation is that in the nuclear phylogenetic analyses (Fig4a), Mp is systematically more closely related to Mj than Ma or any other species. This also does not support calling Mp *M. paraarenaria*.

Furthermore, it is very surprising that none of the topologies presented in Fig. 4a (and Supp. Fig. 35) corresponds to the mitochondrial topology presented in (Supp. Fig. 1). It is expected that at least one subgenome is maternally inherited and thus should follow the mitochondrial topology. In Blanc-Mathieu et al. 2017, one of the nuclear topologies (their Fig 3) exactly reproduced the mitochondrial phylogeny (their Fig. 7 which is completely consistent with your Supp. Fig. 1). What is the hypothesis of the authors concerning the absence of a maternal topology in the subgenomes phylogenies? I am really surprised by this finding.

Summary of reviewer's question:

What is the phylogenetic relationship between Ma(3n), Ma(4n) and Mj, and which subgenome is maternally inherited and follow the mitochondrial topology.

Response:

1. It should be noted that the mitochondrial phylogenetic tree indeed suggest that Ma is not

monophyletic, but could not determine which of those species is the ancestral node.

2. For mitochondrial phylogeny of Mi, Ma_3n, Ma_4n and Mj, the closest outgroup we can find is *M. enterolobii* (Me). When rooted using Me, we can get the topology similar to Supp. Fig 1 (**Response Fig 11a,b**), and when rooted using the midpoint, the topology becomes different (**Response Fig 11c**). Me is quite divergent from these four species (**Response Fig 11a**) and thus may have an impact on mitochondrial topology.
3. To avoid the influence of outgroups, we provide an unrooted tree (**Response Fig 11d**), which shows that Mi and Mj/Ma_3n/Ma_4n can be clearly distinguished. The topology of Mj, Ma_3n and Ma_4n show a trifurcating structure, and it is difficult to determine which species at the ancestor node.
4. Similarly, for the nuclear genome (A1A2B1), Mi and Mj/Ma_3n/Ma_4n can be clearly distinguished, but it is difficult to determine the relationship between Mj/Ma_3n/Ma_4n. For three possible topologies, (Mj, (Ma_3n, Ma_4n)), ((Mj, Ma_3n), Ma_4n) and ((Mj, Ma_4n), Ma_3n), according to Supp. Fig. 36, the number of trees supporting each of those three topologies is about the same.

In conclusion:

Regardless of mitochondrial or nuclear genomes(A1A2B1), it is more suitable to use a trifurcating tree to represent the phylogenetic relationship of Mj, Ma_3n and Ma_4n. No obvious binary phylogenetic topology is formed among them, so it is difficult to determine which subgenome represents maternal inheritance. For origin of tetraploid species, we believed that there was a triploid intermediate species (A1A2B1) that rapidly diverges into three lineages, one of which is Ma 3n, one lineage subsequently hybridizes with Ma4n_B2 to form Ma 4n, and the other lineage hybridizes with Mj_B2 to form Mj.

To make it clearer, we have provided the unrooted mitochondrial tree (**Supplementary Fig. 2b**), and collapsed the topology of Mj, Ma_3n and Ma_4n in Fig. 4a. We have also edited the description for hypothesis of tetraploid species formation (**Line 358-360, 363-364**).

Response Figure 11

- (a) Phylogenetic tree of Clade I RKNs, rooted with *M. enterolobii*.
- (b) Phylogenetic tree of Clade I RKNs, rooted with *M. enterolobii* and ignoring branch length.
- (c) Phylogenetic tree of Clade I RKNs, removed *M. enterolobii* and rooted with midpoint.
- (d) Unrooted tree for Clade I RKNs, removed *M. enterolobii*.

Inclusion of more explanation about *M. floridensis* and *M. luci* is really interesting and informative and I congratulate the authors for this addition and further clarification. It is indeed really interesting that the majority of *M. floridensis* match A genomes and this suggest a very close phylogenetic relationship with A. It is also very important that the authors have confirmed *M. floridensis* is itself most likely a triploid species and therefore cannot have passed its three subgenomes as an ancestor but only two of them. The finding that *M. luci* is triploid is consistent with and confirms previous analysis by Susic et al. J. Nematol 2020. Therefore this, should be mentioned and the paper (ref58) cited around L370-371.

The new Supp. Fig. 39 is also very interesting and informative and strongly suggests that there exist two different categories of *M. arenaria*: one category of triploid populations which contain Mp (and therefore should be called Ma3n) and one category of tetraploid populations which contain Ma. Interestingly this dichotomy between triploid and tetraploid populations in Ma is perfectly reflected by the mitochondrial phyloegeny in Supp. Fig. 1. I guess *M. arenaria* KP202350 is not present in supp. Fig. 39 because no genome data is available for this population? In any case this finding is very important and suggests *M. arenaria* is not a valid species but a complex of species with different ploidy levels. In that sense and following a previous comment, it would be interesting to try to reinforce the support values of the mitochondrial phylogeny in Supp.

Fig. 1. This might hopefully be achieved by adding the ribosomal mitochondrial genes.

Summary:

1. Previous paper about ploidy of *M.luci* needs to be cited.
2. *M. arenaria* KP202350 is not present in supp. Fig. 39.

Response:

We thank the reviewers for affirming our analysis of the genome structure of *Mf*, *M.luci*, *Ma 3n* and *Ma 4n*.

1. We have mentioned and cited this reference now (**Line 380**).
2. KP202350 is accession number for *M. arenaria* mitochondrial genome, the genome sequencing data for this sample are not available.
3. The new mitochondrial tree that added ribosomal mitochondrial genes can be found in **Supplementary Fig 2b**.

TE dynamics

Similarly, the new Supp. Fig. 40 is very interesting and gives another dimension to the analysis. I congratulate the authors for this much more detailed analysis of TEs. I would just suggest more explanation for non-specialists. For instance, explain that a whole population of TEs show a peak at very high identity, suggesting they have recently expanded in these genomes and might still be active. Interestingly, similar conclusions were drawn in *M. incognita* in (Kozłowski DKL et al. 2021 Evolutionary Applications ‘Movements of transposable elements contribute to the genomic plasticity and species diversification in an asexually reproducing nematode pest’). L379 I guess the author mean ‘although’ TE multiplication (or activity) is ongoingbecause hybridization is most likely to be recent.

Response:

1. We have added the explanation for 100% LTR and mentioned Kozłowski DKL et al. 2021 (**Line 386-388**).

2. For L379:

We know that TE may become reactive or burst after recent hybridization. But we found that active TEs (young TEs) are only a minority and most TEs are old. In this case, the increased TE content in polyploid should be attributed to the accumulation of old TEs but not burst of young TE. We have edited the description for this in revised manuscript (**Line 388-396**).

4-Expression data done with biological replicates

It is now clearly indicated that 3 replicates were used for the 5 developmental life stages. The results are very interesting and show that although most homoeologous gene copies maintain similar expression levels between the subgenomes, some show biased expression in one or several subgenomes and this bias is not conserved across species.

To make sure I completely understand, I have the following questions:

- It is not clear to me whether expression bias in any of the five developmental life stages was considered as the minimal criterion to consider the gene had an expression bias across subgenomes or if the bias should be observed in multiple life stages. Could you please clarify?
- I don't see any statistical test for expression bias in subgenomes, only the shortest Euclidian distance is considered. Applying a statistical test to confirm the difference in the distribution of expression values is different from the rest of the subgenomes is necessary.

Summary:

1. What is the criterion for expression bias.
2. Statistical test is needed for expression bias genes.

Response:

1. Expression bias or balance is only considered in one stage. For instance, gene group X show an expression bias pattern in egg stage but show a balanced pattern in J2 stage. The analysis of gene expression bias was separately performed in 20 groups (4 species x 5 developmental stages).
2. In revised version, we have added a supplementary table for all statistical test of gene expression different analysis among subgenomes (**Supplementary Table 25**).

Other important remarks:

Data availability, reproducibility, and FAIR principles

I appreciate the authors have deposited the code they used and many associated data in a GitHub repository. However, some of their main data (genome assemblies and annotations) were deposited on a website from their laboratory. However, all these data should rather be deposited in global public repositories such as the NCBI, EMBL ENA or WormBase Parasite.

Response:

The genome assemblies were uploaded to NCBI in BioProject PRJNA784524 with BioSample number SAMN33562970 to SAMN33562974 and SAMN33562119 to SAMN33562123.

Introduction

I acknowledge and appreciate the efforts made by the authors to reinforce all the introduction and better place it in the context of the rest of the literature. However, some of the references cited seem not to be at the appropriate position in this paragraph.

L78 ref 42 (Lunt et al. 2014 PeerJ) as well as ref 23 (Blanc-Mathieu et al. 2017 PLoS Genetics) should be cited here together with reference 20. Or even better, maybe cite first reference 20 as early hypothesis of reticulate evolution later confirmed using genome data by references 20 and 42.

Response:

We have modified the references here (**Line 78-79**).

L85 'combination of closely related females with more diverse 'paternal' rather than 'parental' lineages and reference 23 should be cited here as it is exactly the hypothesis made in this paper.

Response:

Thanks, we replaced 'parental' with 'paternal' and added the citations (**Line 86**).

L91 ‘M. floridensis and another yet unknown species’ reference 23 should be removed from the list here because this hypothesis has not been proposed in this paper.

Response:

We have removed this sentence.

L102- 103 I would be more cautious about the repetitive sequences identified at the ends of contigs and scaffolds. Although they were found at the extremities, exactly where telomere repeats are expected to be observed, at this stage we absolutely don’t know whether they play the same role of protecting chromosome ends. Therefore, instead as ‘act as the telomeres of polyploid RKN’, it is safer to mention ‘that might act as functional telomeres of...’. At this stage we have absolutely no evidence about their function.

Response:

Thanks, we have modified this sentence (**Line 102**).

Secretome

It is very interesting that, in the predicted secretome, proteins seem to feature higher sequence divergence between A and B genomes than the rest of proteins. I wonder whether higher divergence among the secretome would be a more general feature and not necessarily related to hybridization. For instance, in the rest of the Meloidogyne species that are not polyploid (e.g. M. graminicola), do you also observe more divergence in duplicated (cis, tandem or dispersed) proteins from the secretome than for the rest of the proteins?

Response:

Since the genome of diploid Mg did not reach the unzipped genome level, we were unable to calculate whether secreted proteins from the two sets of genomes in diploid nematode had lower sequence divergence than in polyploids. We speculated that the higher divergence of secreted protein sequences in polyploid nematodes may be related to hybridization and revised the wording (**Line 423-425**).

Phased vs. unzipped terminology?

Phasing usually refers to diploid heterozygous genomes in which the two haplotypes are phased out according to SNPs between alleles. In the case of Clade I Meloidogyne, the situation is different because the different genome copies have no allelic relationship but are homoeologous subgenomes resulting from hybridization events and displaying quite high divergence. I wonder whether unzipped would be a better adjective than phased in that situation?

Response: Thanks, we agree with the reviewer and changed the terminology “phased” to “unzipped” in the revised manuscript.

Repeats analysis (Supp. Fig.7 and Supp. Table 6)

The proportion of repetitive sequences (TE and other repeats) seem to be much higher in polyploid RKN than Mg. It also seems to be correlated with the genome size (the bigger the genome, the higher the percentage of repeats). Similar results were obtained in old versions of the polyploid genomes and the genome of M. hapla in (Blanc-Mathieu et al. 2017) and it was hypothesized that TE underwent multiplications in the polyploid genomes. Could the author comment these previous findings and do their new findings support or contradict this hypothesis?

Mg seems to have a proportion of low complexity repeats more than twice higher (>6%) than polyploid *Meloidogyne* (<3%). Do the authors have any hypothesis to explain this observation? Is it due to expansion of any particular low-complexity repeat?

Summary:

Q1: Comment on previous findings on TE multiplications, and whether the new findings support or contradict the previous hypothesis?

Q2: Hypotheses about the increase of low-complexity repeat content in Mg.

Response:

Q1: Comment on previous findings on TE multiplications, and whether the new findings support or contradict the previous hypothesis?

Blanc-Mathieu et al. 2017 observed TEs multiplications in polyploids, but it is not clear whether the multiplications are due to reproductive mode or hybridization.

In the revised version, we have stated that TE content is increased in polyploid species and mentioned Blanc-Mathieu et al. 2017 (**Line 165-166**). In the latter part of the paper, we further analyzed TE history and raised our point (**Line 388-396**). We show that although a fraction of TEs is active, most of TEs are old, and we believed that TEs multiplications in polyploids is due to accumulation in diploid ancestor but not hybridization. If TEs multiplications are related to recent hybridization, then most TEs should be young. Whether TEs multiplications is related to the mode of reproduction is still unclear.

Q2: Hypotheses about the increase of low-complexity repeat content in Mg.

We checked the copy number of low-complexity repeats in five genomes, and the following table shows the repeats that more than 500 copies of repeat in Mg.

Mg genome are a haploid genome, while other genomes contain 3-4 copies of genome. In this case, polyploid genomes would be expected to have 3-4 times the copy number of low-complexity repeats compared to Mg if no expansion. However, we noticed that the copy number of many low-complexity repeats of Mg is roughly equivalent to that of the polyploid genome, indicated that the low-complexity are weak expanded in Mg. Moreover, Mg genome is relatively small (45Mb) and weak expansion of repeat elements would result in an increase in proportion.

Repeat	Mg	Mi	Ma_3n	Mj	Ma_4n
(TTC)n	505	884	952	1182	1242
(ATTTA)n	513	678	670	896	897
(TA)n	514	449	438	566	606
(TTA)n	515	696	892	864	1171
(ATAA)n	522	873	838	1187	1146
(AT)n	613	563	610	793	754
(TAA)n	646	818	919	1136	1222
(TAATT)n	700	711	774	947	978
(ATTA)n	729	805	837	1069	1086
(AATTT)n	744	1100	1355	1603	1570

(AAT)n	770	1016	1128	1341	1412
(TTAT)n	796	1085	1196	1396	1549
(ATT)n	826	1026	1112	2371	1505
(AATA)n	852	1310	1423	1940	1992
(TAAT)n	996	1000	1025	1304	1333
(TTTAA)n	1020	1256	1310	1746	1790
(TATT)n	1045	1432	1585	1924	1954
(A)n	1067	1008	1106	1579	1438
(ATTT)n	1339	2176	2254	2840	2932
(TTTA)n	1363	1799	1897	2391	2507
GA-rich	1488	2852	2921	3992	3932
(TTAA)n	2209	1981	2068	2708	2714
(AATT)n	2739	2375	2505	3276	3236
(T)n	2864	3683	3918	4981	5152
A-rich	16237	34801	37310	48375	47957

Distribution of coverage along scaffolds (Supp. Fig. 26).

This Figure is interesting as it reveals variations in coverage across the scaffolds and suggests some scaffolds (partially) result from collapsed homoeologous copies. It is still intriguing that the standard deviation in Mg and Mi seems much bigger than for the two Ma (3n and 4n) genomes and for Mj. Do the authors have any hypothesis that could explain this? Furthermore, is it possible that within a same scaffold some regions are collapsed while others are not (= variations in coverage within a scaffold). Is it visible in the 100kb windows defined by the authors (L959-960)?

Summary:

Q1: In Supp. Fig. 26, the standard deviation in Mi seems much bigger than for the two Ma (3n and 4n) genomes and for Mj.

Q2: Is it possible that within a same scaffold some regions are collapsed while others are not.

Response:

Q1:

We guess that the Illumina library of Mi are not very good, which is also found in kmer-based genome survey. The model fit in Mi Illumina library is 87.22%, while Illumina library of other samples are 94.64% - 96.18% (**Supplementary Table 3**).

Q2:

Yes, it is possible. To show the depth of those 100kb windows, we now replaced the boxplot of Supp. Fig. 26 as a figure of read depth distribution among chromosomes (**Supplementary Fig. 27**).

Prediction of protein-coding genes (methods)

The authors seem to have employed a quite sophisticated and comprehensive methodology to predict genes in the Meloidogyne genomes (including ISO-seq data in *M. incognita*). L1001 -1003 it is indicated that BUSCO was used to evaluate the completeness of the five predicted proteomes (not genomes I guess). Given the quality of the genomes and the gene prediction procedure, I

expect protein-level BUSCO scores to be very good. However, I did not find these BUSCO results at the protein level. A supplementary table presenting these BUSCO results and comparison with previous versions of predicted proteins in *Meloidogyne* species would be interesting.

Response:

Now we have provided BUSCO score for genomes and proteomes in revised version.

BUSCO score for genomes: **Supplementary Fig. 6** and **Supplementary Table 6** and **Line 156-157**.

BUSCO score for proteomes: **Supplementary Fig. 9** and **Supplementary Table 9** and **Line 168-170**.

AABB subgenomes in tetraploid genomes

In Supp. Fig. 12 and in the main text L183-185, it is indicated that 2:2 topologies were dominant and thus suggested an AABB subgenomes structure. This is interesting but do you also observe among the three possible 2:2 topologies one that seem dominant (A1,A2)(B1,B2) or (A1,B1)(A2,B2) or (A1,B2)(A2,B1)? I understood that at this stage of the analysis the distinction between A1 and A2 or B1 and B2 is random but at least between A and B this is possible; so is the most frequently observed topology separating the A genomes from the Bs?

Response:

In fact, the AABB here is just a symbol and does not mean A or B subgenome. It only proves that the tetraploid is composed of two pairs of similar genome copies.

To target reviewer's concern, we rechecked the 2:2 topologies by combing subgenome information. We confirmed that the most frequently observed topology is (A, A) (B, B).

Synteny between Mg and polyploid RKN and chromosomal fusions / losses (Supp. Fig. 13)

This Figure suggests that not only chromosomal fusions occur but also some chromosomal losses. For example, in Mj, there are only two copies of Mg chromosome 18 (an only in the B genomes), suggesting two copies have been lost in the A genomes (4 copies are expected in total). Conversely, some regions present in the polyploid RKN have no equivalent in Mg. For instance, the two last chromosomes in Ma B2 and Mj A1 seem to have no equivalent in Mg. Therefore, L197-199: I would say that chromosomal copy losses can also be expected in addition to fusions to explain the variations in karyotype observed in RKN.

Response:

Conclusion: Chromosomal copy are not losses in polyploid RKNs, according to synteny analysis between Mg and polyploid RKNs (**Supplementary Fig. 10 and 11**). Similar analysis based on our T2T genome (**Supplementary Fig. 33**) also confirm that all of chromosomal copy are present in those polyploid RKNs genome. Chromosomal copy loss is due to visualization, not real.

Some chromosomes appear to be lost is because:

1. The synteny analysis here use assembly V1, which does not reach the chromosome level, and some chromosomal copy may be mis-assigned to other subgenomes.
2. The synteny analysis here only shows the collinearity between two adjacent subgenome.

We know that this type of visualization in **Supp. Fig. 13** is indeed commonly used to display

patterns of chromosomal rearrangements. Here, we just want to show that most genes remain collinearity across 14 subgenomes to ensure that there are enough syntenic genes to estimate nucleotide divergence or Ks estimation.

To avoid misunderstanding to readers, we decided to remove this figure.

Telomere-associated genes and telomeric repeats

It remains not completely clear how genes ‘involved in telomere formation in *C. elegans*’ were selected. Does the list of telomere-associated genes come from a publication? Or was just the keyword ‘telomere’ searched in WormBase? Could the authors clarify how they obtained this list of *C. elegans* genes involved in telomere formation (and I guess some are just involved in maintenance, or have unclear roles and are just associated with telomeres)? Maybe the authors can write a few words about the predicted or demonstrated functions of these genes and how it is related to telomere maintenance or elongation? After a rapid check, it seems all these genes have the keyword telomere in their description in Wormbase excepted *dve-1*. What kind of function *dve-1* is supposed to play regarding telomeres? Also, to remain consistent with the rest of the genes in Fig2. A, R06A4.2 as well and T12E12.3 should be renamed *tebp-1* and *tebp-2*, respectively. In the methods section L 1050-1056, I guess the author meant the *C. elegans* proteins were aligned to the proteomes of the selected species in Clade IV nematodes not the reverse? Later in the same section (1059-1060), I think there is an error on the telomere sequence, it is supposed to be (TTAGGC)*n* in *C. elegans* and not (GAATCC)*n* in the material and methods or even (TAAGCC)*n* in Figure 2a. So, please clarify exactly which repeat was searched in the genomic reads of the Clade IV nematodes. Finally, same remark as previous for BLASTp, I guess it is the telomeric repeat motifs that have been searched against the reads using BWA and not the reverse.

Summary:

- Q1. How telomere-associated genes are selected?
- Q2. Methodology used in telomere-associated genes search (L1050-1056).
- Q3. Correct telomere sequence.
- Q4. Methodology used in telomeric repeat search.

Response:

Q1: How telomere-associated genes are selected?

The telomere-associated genes are collected from Dietz, Sabrina et al. “The double-stranded DNA-binding proteins *TEBP-1* and *TEBP-2* form a telomeric complex with *POT-1*.” *Nature communications*. In this paper, Dietz, Sabrina et al. used mass spectrometry and pulldown to identify 8 proteins that bind to the *C. elegans* telomeric sequence (*POT-1*, *POT-2*, *MRT-1*, *CKU-70/CKU-80*, *TEBP-1*, *TEBP-2*, and *DVE-1*). In addition, we also included *MRT-2* in our survey. It is unclear what function *DVE-1* plays in telomeres, but it has been detected in telomere-binding proteins, so we also included this gene.

In revised version, we have added the description for collection of telomere-associated gene and added a table for functions of those genes (**Line 1079-1081, Supplementary Table 11**).

In figure 2a, R06A4.2 and T12E12.3 are now renamed *tebp-1* and *tebp-2*, respectively.

Q2: Methodology used in telomere-associated genes search (L1050-1056).

We have edited the method description of BLASTp (**Line 1081-1083**).

Q3: Correct telomere sequence.

We thank the reviewer for their careful examination of our manuscript.

Sequence (TTAGGC)_n and (TAAGCC)_n are both telomere repeat, but the direction or starting position are different (See below, sequence of *C. elegans* chromosome I). Sequence (GAATCC)_n is a typo, it should be (GCCTAA)_n in our originally manuscript. Now we have edited all telomere repeat as (TTAGGC)_n.

```
>I
GCCTAAGCCTAAGCCTAAGCCTAAGCCTAAGCCTAAGCCTAAGCCTAAGCCTAAGCCTAA GCCTAAGCCTAAGCCTAA
GCCTAAGCCTAAGCCTAAGCCTAAGCCTAAGCCTAAGCCTAAGCCTAAGCCTAAGCCTAAGCCTAAGCCTAAGCCTAA
GCCTAAGCCTAAGCCTAAGCCTAAGCCTAAGCCTAAGCCTAAGCCTAAGCCTAAGCCTAAGCCTAAGCCTAAGCCTAAGCCTAA
GCCTAAGCCTAAGCCTAAGCCTAAGCCTAAGCCTAAGCCTAAGCCTAAGCCTAAGCCTAAGCCTAAGCCTAAGCCTAA
. . .
AGTTCTTTATATCATTGCGCCGATCCATAGATATTGCTGATGATTGCGTCGAGGCGCTTGAGTCTGAAGCTTTGGAA
GTTTCGGTTGTTGCATCTACTTTTGGATACTAATGATGTTGCGGTATTGGTCTTAGGCTTAGGCTTAGGCTTAGG
CTTAGGCTTAGGCTTAGGCTTAGGCTTAGGCTTAGGCTTAGGCTTAGGCTTAGGCTTAGGCTTAGGCTTAGGCTTAGG
CTTAGGCTTAGGCTTAGGCTTAGGCTTAGGCTTAGGCTTAGGCTTAGGCTTAGGCTTAGGCTTAGGCTTAGGCTTAGG
```

Q4: Methodology used in telomeric repeat search.

Methods for searching for telomere repeats:

We artificially generated a telomeric repeat sequence (TTAGGC TTAGGC...TTAGGC)_n as reference, then, Illumina reads of each sample are mapped into that reference (TTAGGC TTAGGC.....TTAGGC)_n, and complete match reads (150M) were selected as telomere reads. In revised version, we have edited the description to make it clearer (Line 1088-1094).

The finding of an unusual complex repeat in Mg located at the ends of the scaffolds is very interesting and the fact that the G-rich motif is in the 3' strand is consistent with known telomeric repeats in other species. This strongly suggests this repeated motif has replaced the classical telomeric repeat of nematodes in Mg. The authors need to explain how they have detected putative G-quadruplex forming regions in these genomes? Which software and parameters were used to identify G4 quadruplexes? This is not explained in the methods while presented as a result in Fig2 b.

Response:

The G4 quadruplexes were identified using qgrs-cpp with default parameters. We have included the description in method (Line 1098-1099).

The way a repetitive motif was found in Mi, is totally unclear and casts doubts on the validity of the initial scaffolding. L216-217: 'The reason for this might lie in contig misanchoring in polyploid RKNs assemblies...' This statement reinforces my doubt about the validity of using Mg as a guide to assemble Mi scaffolds as well as the other polyploid RKNs. The fact that the search for enriched motif had to be re-performed at the contig level confirms that something went completely wrong during Mg-guided scaffolding and that an assembly without a priori should be used instead of the Mg-guided one. Also, it seems really surprising that the Mi-Tel motif was initially found from one single contig considered to be well assembled by optical map. How was this specific contig chosen? It seems very unlikely by, just by chance this motif was identified from just one contig. How many contigs displayed this Mi-Tel motif at one end or at both ends? Fig. 2c is also misleading and confusing as currently represented. How many BioNano molecules have been grouped together in the upper track? What is this extra ~100kb sequence in the upper

track that does not align with the contigs (lower track). My initial guess was that this long sequence at the beginning represents telomere repeat arrays in which no BioNano tag is present. However, this is in contradiction with the lower track in which a lot of (green) tags are represented in the circled region labelled as Mi-Tel. Therefore, the authors should clarify this contradiction and indicate whether or not Mi-Tel contains the BioNano tag. If yes, then the upper track is wrong and if not, this is the lower track that is wrong.

Summary:

Q1. When searching for telomeric repeat sequences initially, it should be performed at the contig level, not just assembly V1.

Q2. How was the well-assembled contig chosen?

Q3. How many contigs displayed this Mi-Tel motif at one end or at both ends?

Q4. How many BioNano molecules have been grouped together in the upper track of Fig. 2c?

Q5. What is this extra ~100kb sequence in the upper track that does not align with the contigs (lower track) and does the Mi-Tel contains BioNano tags.

Response:

Q1: When searching for telomeric repeat sequences initially, it should be performed at the contig level, not just assembly V1.

Reply:

We tried to screen for repetitive sequences that are present in most of scaffolds ends (about 20kb near ends). For instance, in Mg, there are 36 (18 x 2) border region, we check whether the repeat family are present in most border region, such as 30 of 36. For contigs, there are so many border regions (contigs number x 2), thus it may be difficult to find repeats that exist in most of those regions.

We re-searched enriched motif at the contigs level, and added a description about searching at the contigs level (**Line 223**).

Q2: How was the well-assembled contig chosen?

Reply:

1. Well assembled contigs means that the contigs reach at the chromosome level when construct contigs.
2. We just aligned the PacBio contigs to the Bionano map and observed a complete alignment of this contigs (tig00000250_np512512) with the Bionano map, we guessed that this contigs reached the chromosome level. We then investigated the sequences at both ends of this contig, thus identifying Mi-Tel. Since then, we have not paid any further attention to other contigs.
3. In fact, we further confirmed that this contigs have reached the chromosome level by checking the collinearity between this contig and Mg chromosome.

In the revised version, we added the collinearity analysis between this contig and Mg as evidence for selecting well assembled contig (**Fig. 2c**). We have also added more description about selecting well-assembled contigs in **Methods (Line 1101-1105)**:

- (a) The contig length is more than 1Mb.

(b) Contig maintains collinearity with a chromosome of Mg.

(c) Contig presents a complete alignment with a Bionano map.

Q3: How many contigs displayed this Mi-Tel motif at one end or at both ends?

Reply:

We re-examined the distribution of Mi-Tel in contigs, and among the 65 contigs larger than 1Mb, 29 had only one end, 7 had both ends, and 29 had neither end.

Q4: How many BioNano molecules have been grouped together in the upper track of Fig. 2c?

Reply:

3,634,649 molecules were assembled into 143 consensus maps. The upper track is just one of those consensus maps, not grouped from several consensus maps.

Q5: What is this extra ~100kb sequence in the upper track (Fig. 2c) that does not align with the contigs (lower track) and does the Mi-Tel contains BioNano tags.

Reply:

The region not containing the tag in the upper track is the telomeric repeat array. We checked the contig shown in Fig. 2c (tig00000250_np512512) for the position of telomeric repeats and the position of the restriction site (Bionano tag). The telomeric region is 1-15142 and 3428506-3436655, while the first tag of tig00000250_np512512 at 21894, last tag at 3424693, indicating that the telomeric region does not contain BioNano tags.

It seems that the tag exists in telomeric region because our circle marker range is too large, we just want to show that Mi-Tel is at the end of the contigs.

In revised version, we changed the circle mark to an arrow to avoid misunderstanding for readers (Fig. 2c).

Another point that is unclear is the telomere-to-telomere fusions of some contigs or scaffolds? Could the author explain whether they observe these fusions only at the scaffold level or also within one contig? Do the identifiers 'Mi_ChrX' represent contigs obtained by SMRT DeNovo or Mg-guided scaffolds? Or ONT contigs? In Fig. 15b Mi_Chr12 is represented 4 times in 3 different maps. It is not clear if (3) is the continuation of (2) or if it is separate molecules. In that case it would mean that Mi_Chr12 was cut into different pieces? Furthermore, confirmation with ultra-long read ONT data (>50kb) is quite weak with only one long read supporting this fusion. Rather than providing a few examples of some contigs that contain both Mi-Tel and Mi-Tel reverse in arrays, it would be more convincing to provide an information about coverage by ONT reads. What is the coverage of these candidate fusion points by ONT long reads? And is the coverage comparable to that observed just before and just after the fusions? Finally, assuming that these 4 definite end-to-end fusions are true, why these are not represented in Fig. 3? I would expect to clearly see Mi-Tel + Mi-Tel reverse arrays in the middle of some of the manually edited chromosomes in Fig3a but it is not the case. Could the author clarify whether or not these fusions actually exist and if yes indicate them in Fig3a?

Summary:

- Q1. Fusions are found in scaffold level or also within one contig?
- Q2. What is Identifier “Mi_ChrX” of 15b?
- Q3. Coverage of these candidate fusion points by ONT long reads?
- Q4. End-to-end fusion should be represented in Fig. 3a.

Response:

Q1: Fusions are found in scaffold level or also within one contig?

Conclusion: Fusions are found between scaffolds of assembly V1.

1. In assembly V1, we tried to construct the genome in multiples of 18, so some fused chromosomes were split into 2-3 scaffolds. Each scaffold maintains collinearity with the one Mg chromosome, suggested that they were once an independent chromosome.
2. Thus, if there is an obvious Hi-C signal between the two scaffolds of assembly V1 and a complete alignment between single Bionano consensus map with those two scaffolds, it means that those two scaffolds were once two separate chromosomes, but subsequently fused in the Mi genome.
3. In Supp Fig. 15, we just confirmed four fusion events based on the obvious Hi-C signal between the two scaffolds, and the alignment between single Bionano map with two scaffolds.

In revised version, we added more description for identification of fusions in assembly V1 (**Supplementary Fig. 16 and Line 235-237 and Methods Line 1115-1118**).

Q2: What is Identifier “Mi_ChrX” of 15b?

Identifiers of sequence is the scaffolds ID of assembly V1.

In revised version, we have edited the Supp. Fig. 15b to complete the scaffolds ID (**Supplementary Fig. 16**).

Q3: Coverage of these candidate fusion points by ONT long reads?

We thank reviewer for their suggestion.

1. We understand the strategy for aligning the contigs (maybe generated from short reads) into chromosomes-level genome, and determining the junction or the sequence of gap between two contigs through the coverage of ONT reads. In our situation, both assembly V1 and assembly T2T are assembled from long reads. The gap in assembly v1 is still a gap in assembly T2T (See below).
2. The exact sequence of these fusion points is unknown, they may have 70Kb-200Kb of N, according to the BioNano map. Our ONT data is a standard library, the average read length is 15kb, and there are only 1565 reads over 100kb. In this case, due to too many N, there are very few reads that can cross the gap, we only find one ONT reads cover it. Other reads could not completely map into these fusion points. If we have an ultra-long ONT library containing many reads over 100kb, providing coverage of these fusion point will be achievable.

Q4: End-to-end fusion should be represented in Fig. 3a.

Four fusion events in Supp Fig. 15b are:

Scaffold ID in assembly V1	Chromosome ID in assembly T2T
Mi_Chr4_11 + Mi_Chr9_24 =	Mi_Chr_08
Mi_Chr11_29 + Mi_Chr12_31 =	Mi_Chr_09
Mi_Chr12_32 + Mi_Chr6_17 =	Mi_Chr_07
Mi_Chr7_10 + Mi_Chr12_33 =	Mi_Chr_05

In revised version, we have marked all the Mi-Tel in Fig. 3a as much as possible. Those four fusion events in Supp Fig. 15b are now represented in **Fig. 3a**.

Of course, not all fusion sites contain Mi-Tel + Mi-Tel reverse arrays, they may be lost during evolution after chromosome fusion, or remain gaps (NNNN)n in the genome.

Recovery of collapsed regions

It is currently not very clear how collapsed regions have been recovered. I understood they were identified based on coverage but once they have been identified, how were these regions re-separated? Were they just duplicated as 100% identical regions or unzipped using phased variants in long reads? The authors should more clearly indicate how this process was done. Furthermore, in Supp. Fig. 26 it seems that the standard deviation in coverage (depth) is much higher within Mi chromosomes (~ 0- >2) than for the rest of the RKN species (all >0 and <2). Could the author provide explanation / hypothesis about this difference in coverage distribution?

Summary:

Q1. How collapsed regions have been recovered?

Q2. Why standard deviation in coverage (depth) is much higher within Mi chromosomes (~ 0- >2) than for the rest of the RKN species?

Response:

Q1: How collapsed regions have been recovered?

The collapsed region could be identified from read depth and Hi-C heatmap (See below).

In revised version, we have added the detail description of recovery of collapsed regions (**Line 1152-1156**).

(a) We identified the collapsed region by calculating the sequencing depth in 100kb windows.

(b) MUMmer was used to determine the breakpoints of the collapse regions.

(c) The collapsed region was duplicated as 100% identical sequence and added into the corresponding position according to Hi-C heatmap.

We initially want to make the genome look more complete by recovering the collapsed regions, because without recovering, some genome regions appear to be lost. Now it seems that recovering the collapsed region is not necessary because the collapsed region will still be removed in the subsequent analysis, it may be better to mark the collapsed region but not recover it.

Q2: Why standard deviation in coverage (depth) is much higher within Mi chromosomes (~ 0- >2) than for the rest of the RKN species?

We guess that the Illumina library of Mi is not very good, which is also found in kmer-based genome survey. The model fit in Mi Illumina library is 87.22%, while Illumina library of other samples are 94.64% - 96.18% (**Supplementary Table 3**).

Comparison between *M. graminicola* and *M. chitwoodi* genomes (Supp. Fig. 27)

Only 20 *M. chitwoodi* contigs were included in this dot plot comparison while this version of the genome (GCA_015183035.1) contains 30 contigs. Therefore, I wonder whether the 10 other contigs simply don't match the *M. graminicola* assembly or if there is any other reason?

Response:

The remaining 10 contigs are not shown because they are too short to be easily displayed. They account for only 2.3% of the *M. chitwoodi* genome (47.5Mb), here are the lengths of the remaining 10 contigs.

Contig ID	Length(bp)	matched Mg ID
JACZZP010000021.1	217383	Mg_Chrr_14
JACZZP010000022.1	172047	Mg_Chrr_2
JACZZP010000023.1	153282	-
JACZZP010000024.1	153258	-
JACZZP010000025.1	130989	-
JACZZP010000026.1	99878	Mg_Chrr_14
JACZZP010000027.1	62198	Mg_Chrr_1
JACZZP010000028.1	46796	-

JACZZP010000029.1	46375	-
JACZZP010000030.1	46139	-

Distribution of features along the chromosomes.

L276 – 279. It is indicated that chromosome arms were relatively repeat rich and gene poor compared to chromosome centers (repeat poor and gene rich). However, I did not find in the methods which regions were considered arms and which ones centers? Did you select for instance the 15% first and 15% last Kbs of each chromosome compared to the rest? What were the criteria used and are these differences in numbers of TEs / Mb and genes / Mb statistically significant (no statistical test seems to have been performed). Same remark for the histone marks. Supp. Fig. 29 indeed suggest a co-localization of centromere repeats and H3K9me3 marks and an enrichment of H3K4me3 in subtelomeric regions. These two very interesting results would be more convincing if statistical tests confirmed these enrichments and co-localizations.

Summary:

Q1. How to define the arms and centers?

Q2. Statistical test for TEs/Mb and Genes/Mb.

Q3. Statistical test for histone mark enrichment.

Response:

Q1: How to define the arms and centers?

The 20% first and 20% last Kbs of each chromosome are considered as arms, and rest are centers.

Q2: Statistical test for TEs/Mb and Genes/Mb.

Density of genes and TEs are now counted in 100kb windows, and statistically test is performed using Mann-Whitney two-side test.

Q3: Statistical test for histone mark enrichment.

Chip-seq signal of H3K9me3 in centromere region and H3K4me3 in subtelomeric regions are significantly higher than those in whole genome. Mann-Whitney two-side test are performed.

In revised version, we have added the definition for chromosome arms, centers, centromere region and subtelomeric region in **Method (Line 1243-1249)** and added statistical test for those finding in result.

Discussion

The new discussion is clearly improved and much more comprehensive, and I congratulate the authors for this. However, I do not agree the strategy for genome assembly proposed by the authors should be considered a guideline for other assemblies (L447-448). In their quest to obtain T2T chromosome-level assemblies, the authors have extensively edited manually their assemblies with strong a priori concerning telomeres and synteny with a distant outgroup species. I would not recommend this strategy.

Response: As we described above, rather than relying on extensive manual editing or constructing a complex polyploid unzipped genome against the diploid Mg genome, our assembly strategy

relies on multiple sequencing technologies, and joint analysis approaches to resolve the complexities of allopolyploid genome information. Perhaps our approach is not broadly generalizable, we agree with the reviewer and have revised the wording (**Line 462-464**).

L492-494: I fully agree that according to the results provided by the authors, the definition of *M. arenaria* as a species is vague and should be reinvestigated. At least this is what suggests Supp. Fig. 1 at the mitochondrial level. Unfortunately, the branches supporting the separation of *M. arenaria* into two clades are only poorly supported. Hopefully, addition of ribosomal mitochondrial genes to the supermatrix would allow better resolving these branches.

Response: We have added the ribosomal mitochondrial genes to construct the new tree (**Supplementary Fig. 2b**).

L495: We also ‘confirmed’ not ‘determined’ that *M. luci* is a triploid. Indeed, this was already shown in the *M. luci* genome paper by Susic et al. 2020 Journal of Nematology, and this paper should be cited there.

Response: We revised the wording and cited the literature (**Line 510**).

Typos and minor suggestions

L34: homologous gene expression “change” or “decrease” or “attenuation” rather than ‘suppression’.

Response: Thanks, we replaced “suppression” with “decrease”.

L106: which ‘reproduces’ with both sexual reproduction and meiotic parthenogenesis...

Response: We have modified this sentence to make it clearer (**Line 103-107**).

L112: I would tone down the claim that these are ‘extremely highly accurate’ resources. These are undoubtedly high quality and useful data but it is not necessary to provide a judgement of value in the introduction of the paper. This kind of statements and succession of superlatives might be used for the cover letter but not the paper itself.

Response: We have deleted the “extremely highly”.

L140-141: ‘suggesting the homoeologous subgenomes have been correctly separated during genome assembly’ rather than ‘indicating that all....’

Response: Thanks, we adjusted the description (**Line 141-142**).

L154-155: the numbers provided in the text for BUSCO scores and % of reads mapping the genome do not match those indicated in Supp. Table 5. So which numbers are the correct ones?

Response: Thanks for the careful review, we have checked the BUSCO value, Supp. Table 5 is correct. We have corrected the reads mapping ratio and BUSCO score in result (**Line 154-158**).

L177: I think it is Fig. 1d here rather than 1b?

Response: Fig. 1b shows the 3:1 or 4:1 syntenic relationship between polyploid species and *Mg*. Fig. 1d shows that how to distinguish subgenome. In **Line 172**, we just want to show the ploidy of those polyploid species, thus, here should be Fig. 1b.

L328: 'these allopolyploids' instead of these 'allotetraploids' because two species are triploid here.

Response: We have changed this word in **Line 342**.

L402: 'Since effector proteins play' rather than 'Since effectors play'

Response: We have modified this sentence (**Line 415**).

L405: 'proteins predicted to be secreted' and (including putative effectors).

Response: We have modified this sentence (**Line 418-419**).

L421: 'decrease of homoeologous gene expression' rather than 'suppression of its homologous gene'. The gene is not suppressed, only the expression is decreased.

Response: We agree with reviewer and replaced "suppression of its homologous gene" with "decrease of homoeologous gene expression" (**Line 435**).

L430: 'unzipped' might be more appropriate than 'phased' here?

Response: We have replaced all the "phased" to "unzipped" in our revised manuscript.

L431-432: 'these genomes reveal a complex structure' rather than complex genome structure to avoid repeating the term 'genome'.

Response: We have deleted this "genome".

L737-738: (Fig5. Legend) J2 stage (not period). Fig5a: in the histogram the labels of the different categories are not in front of the bars and consequently the last bar has no label.

Response: Thanks to the reviewer for the careful review, we have updated **Fig.5a**.

L794 one to three 'SMRT' Cells to avoid confusion with the cells of the organisms themselves.

Response: We have added the 'SMRT' to avoid confusion with the cells of the organisms themselves (**Line 815**).

L1065: I guess you want to cite Supp. Fig. 22 here?

Response: This figure was used to verify accuracy of genome, we have cited this figure in the **Result** part (**Line 275**).

L1087: I guess the authors mean 'seed' sequences, not 'sed' sequences?

Response: Thanks, we have corrected this word.

L1165: containing 100% identical duplicated genes not '100% similarity genes'.

Response: Thanks, we have replaced 'similarity genes' with 'identical duplicated genes' (**Line 1218-1219**).

Reference list: reference 23 and reference 27 are the same.

Response: We have removed reference 27.

Reviewer #2 (Remarks to the Author):

The additional text in the Introduction helps to better contextualize the work presented in this manuscript. Overall, I think the manuscript is much improved and I am impressed by the follow up to the extensive critical assessment from the reviews. I would like to congratulate the others on the tremendous amount of work and high-quality data generated here. If the few remaining concerns I have below are addressed I endorse this manuscript for publication.

Response: We thank the reviewer for the positive comments, and thank the reviewer for the constructive comments that helped us improve the quality of this paper as a whole.

Broader concerns:

I think referring to the Mp lineage with a new species name is still problematic. The Mp fits within a clade of other Ma triploids (according to the Mt tree), so I don't see how it could be considered a distinct species unless every branch in Figure S1 is a different species. Furthermore, because mitochondrial phylogenies may be discordant with the nuclear phylogeny and whole genome information is available here, I'm not understanding how the Mp lineage requires a distinct name from Ma based on the Mt genome, alone. I think lines 444-446 make it clear that the evolutionary history of this system is complicated and there are probably numerous lineages of distinct subgenome combinations that might otherwise appear to be the same species, even mitochondrially.

Response:

We agree with the reviewers that Mp could not be claimed as a new species.

1. We provided the Ma-SCAR result and unrooted Mt tree of those polyploid RKNs (**Supplementary Fig. 1 and 2b**). We found that Mp and Ma are indistinguishable when using marker genes (Ma-SCAR), and a trifurcating tree were formed between Mp, Ma and Mj. This evidence is indeed not strong enough to support Mp as a new species. Based on the existing nematode species identification method, Mp should be identified as *M. arenaria* but not a new species.
2. The main reason we initially defined Mp as a new species was that Mp is triploid, and Ma is tetraploid.
3. In our revised manuscript, we now do not claim that Mp is a new species, but defined Mp as Ma 3n, while tetraploid Ma is defined as Ma 4n.
4. The motivation for sequencing Ma 3n and Ma 4n was explained in **Method (Line 753-765)**.

I was glad to see some deeper analysis of the TE content in these species. I think the fact that many full-length LTR elements with 100% identity between LTR sequences were recovered in these genomes indicates ongoing activity should be more explicitly stated in the text, though. In text the contrast of the A and B sub genome similarities to what's been seen in *Xenopus* could be stated as well to help clarify what might have been expected if bursts of TEs occurred after

hybridization. Essentially, I think the hypothesis of TE activity related to hybridization/polyploidy could be clarified and that these results do not support the expectation because most of the TEs are old could be more explicitly stated.

Response:

As suggested by reviewer, we revised as follows:

1. We explicitly stated that TE are still active according to 100% identity between LTR, and mentioned a previous study (**Line 386-388**).
2. We explicitly stated that no A- or B- specific TE family are found, contrary to those in *X. laevis* (**Line 392-394**)
3. We explicitly stated that the increase of TE content in polyploid species are attributed to accumulation in the common ancestor of A and B lineage rather than the burst after hybridization. (**Line 394-396**)

Specific comments:

Line 40: I don't want to get overly pedantic, but self-fertilization is sexual reproduction.

Response: Thanks, we have deleted "or self-fertilization".

Lines 43-46: "partially answer" is a bit strong. I think something like "are hypotheses with empirical support" more accurately reflects the status of those ideas.

Response: Thanks, we have modified this description (**Line 45-46**).

Line 370: "proved" could be changed to "supported".

Response: We have replaced "proved" with "supported".

Line 374: A "hypothetical evolutionary model" of what? The expectation of TE burst in hybrids and polyploids?

Response: "hypothetical evolutionary model" is "those RKNs are formed from several round hybridization". We have edited it this sentence to make it clearer (**Line 383**).

Reviewers' Comments:

Reviewer #1:

Remarks to the Author:

I would like to congratulate once again the authors for the efforts deployed to improve further their manuscript and for their comprehensive point-by-point response.

Most of the points raised have been correctly addressed and many points that remained unclear in the previous version are now clarified.

However, a preprint recently posted in bioRxiv (<https://doi.org/10.1101/2023.03.29.534350>) confirmed my concerns and doubts about the validity of the T2T assemblies.

Indeed, in this analysis the authors have used FISH experiments and confirmed a repeat similar to the Mi-tel identified here was present at telomeric locations in *M. incognita*, but only at one end of the chromosomes.

I am afraid in the light of these new biological results contradicting the T2T assemblies, several parts of the manuscript should be revised accordingly.

Below are comments about the main points, most of which having been correctly addressed.

1- Taxonomic identity of the samples and Ma-3n vs. Ma-4n

I would like to thank the authors for the detailed explanations that clarify the situation regarding the previous Mp isolate which is now called Ma-3n to highlight the difference with Ma-4n.

I am convinced by the explanations provided by the authors concerning species identities. I also personally downloaded the genome assemblies and confirm the relevant SCAR markers regions are present in the corresponding species. Having identified two different branches of Ma, one being 3n and the other one being 4n is very interesting and strongly suggests Ma is not a single valid species. This also highlights using just one marker for species identification might be insufficient in the future for these species. According to the mitochondrial phylogenetic analysis of the authors, Ma-3n and Ma-4n are as different as Mjav is different from Minc.

Interestingly the two other Ma-3n populations (HarA, and L32) originate from one single same paper (Szitenberg et al. 2017, Genome Biol Evol). This paper already cited as ref24. Should be cited also as a source for the mitochondrial analysis. Unfortunately, in this ref24 paper, there is no precise information about how species were identified (which markers were used). The authors in Szitenberg et al. just mentioned species identification was done in the laboratories which provided the isolate and confirmed by sequencing but with no further explanation.

Therefore, I fully agree with the authors that this point about Ma-3n vs. Ma-4n deserves further study but is beyond the scope of the current paper. I think this discovery opens interesting perspective that must be studied in more detail in the future and published in a nematology journal. Such a future analysis should ideally include SCAR, esterase as well as morphological and host compatibility tests. The final conclusions might be that *M. arenaria* is not a valid species and that at least two different cytogenetic types with different mitochondrial genomes exist.

This is a pity the initial Yunnan Ma-3n sample was lost during COVID-19 crisis and, indeed, the absence of Ma SCAR band in the initial analysis might be due to initial PCR failure. The presence of the *M. arenaria* SCAR sequence in the genome assembly tends to support this possibility.

Supplementary Figure 1: b: please indicate what is present in each track of the gel, a legend is missing there. Is it the Ma SCAR primers everywhere or all the SCAR primers for all the tested species in each track? This information must be provided. Moreover, in the legend, the authors cannot write: "at that time, only 18S was considered to have a band, but there was a very faint band in Ma-SCAR which ignored by us before." This is completely acceptable in the point-by-point response to the reviewers but not in the paper itself or the supplementary information. I would suggest not to add this new gel with the other Ma-3n population and only state that although only a weak band was observed with the Ma SCAR marker initially, analysis of the genome confirms the presence of the Ma SCAR marker in this population.

2- 'T2T' genomes and manual edition of the assemblies

A preprint recently posted in bioRxiv (<https://doi.org/10.1101/2023.03.29.534350>) by Zotta-Mota et al. brings several clarifications and important results about the telomeres of these root-knot nematodes. In this paper, the authors have sequenced the genomes of three of the species studied here (Mi, Mj and Ma-4n) using ONT + Illumina. Using a different approach, the authors have identified similar repeats mostly at one extremity of multiple contigs in these species. The authors then wanted to confirm whether these repeats had telomeric positions in the chromosomes or their presence at one extremity of multiple contigs was due to an assembly artifact. Using FISH experiments in *M. incognita*, the authors confirmed the repeat has a telomeric position. However, this telomeric repeat is present at only one end of the chromosomes and never observed at both ends. This surprising finding was confirmed by clear observations both at Metaphase and Prophase stages and was consistent with this repeat being present at only one extremity in the contigs as well.

This preprint in bioRxiv confirms with FISH that the Mi-Tel repeat identified by Dadong Dai et al. has indeed a telomeric position and can now be considered as a telomeric repeat. This important biological confirmation should be cited in the paper.

However, in the study by Zotta-Mota et al., Mi-tel is clearly shown to be only at one end of the chromosomes. These findings confirm my doubts about the validity of the manually edited 'chromosomes' in the 'T2T' assembly presented here. I completely understand that the authors guided their assemblies with this idea in mind that all the chromosomes should start and end with telomeric repeats. However, unlike in most other eukaryotes, it is now biologically confirmed that *M. incognita* chromosomes do not have a same telomeric repeat at both ends. Therefore 'telomere to telomere' assembly or 'T2T' with Mi-Tel does not make sense as it does not apply to this species. This most likely also do not apply to Mj and Ma as the telomeric repeat was also found mostly at one single end of the contigs in this study.

These findings, which contradicts the T2T assemblies, also explain several surprising metrics, which were therefore probably due to artifacts caused by the assembly strategy.

- The size of the 'T2T' genome assembly(225Mb), is bigger and out of the range estimated by flow cytometry (189 +/- 15 Mb in Blanc-Mathieu et al. 2017 PLoS Genetics) and also bigger than k-mer estimates in the present study (ca. 212 Mb both with ONT and PacBio, Table S3).
- The number of Mi chromosomes (40) in the T2T assembly is smaller than counts recently made with modern technique in *M. incognita* on two different populations 46 in (Despot-Slade et al. Mol Biol Evol 2021) and also 45-47 in the bioRxiv preprint by ZottaMota et al.).
- The size of the biggest chromosomes seems inconsistent with measured sizes in (Despot-Slade et al. Mol Biol Evol 2021). Indeed, the *C. elegans* chromosomes range between 14 - 22 Mb in size and measure 5µm in Metaphase. In comparison, *M. incognita* chromosomes are much smaller (0.4 - 1.5µm long) and it is difficult to imagine they reach 14 Mb.

In contrast, the metrics obtained in the assembly V1 in which telomeres have not been manually added at both ends is more consistent with biological observations.

- The assembly size is 213Mb which is closer to the k-mer and flow cytometry estimates.
- The number of scaffolds is 48, which is very close to the counted number of chromosomes (45-47).

Furthermore, at the contig level assembly, most contigs >1Mb have Mi-tel at only one end. Finally, in the 'T2T' assembly, 11 of the 12 biggest chromosomes present internal Mi-tel repeats and probably represent assembly artefacts as no Mi-tel fluorescence was observed in the middle of a chromosome in *M. incognita* in FISH experiments.

For this ensemble of reasons and in the light of these new results posted in bioRxiv that confirmed my doubts: these T2T manual assemblies should be removed from the revised version of the paper as they do not correspond to a biological reality (as confirmed by FISH). The title and abstract should be

revised accordingly (no T2T for the polyploid species), as well as the rest of the manuscript.

In my evaluations of the previous versions of the paper, I already suggested not to include the extensively manually edited genome assemblies. The rest of the results presented by the authors remain interesting enough and of high-quality and these T2T assemblies finally provide no added value to the findings.

Some of the new analyses provided by the authors in the point-by-point response also do not further support the T2T assembly. In the dotplot analysis of either BioNano+PacBio or BioNano+ONT vs. the T2T, chromosome 1 is always composed of multiple different and relatively short BioNano scaffolds and contains several internal Mi-Tel repeats (Response Figures 8 & 9). Similarly, the end of chromosome 2 (which contains the Mi-Tel repeats) is supported by a different BioNano contig than the rest of chromosome 2 in PacBio + BioNano dotplot against T2T.

I appreciate and acknowledge that the authors provide additional Hi-C data to try to support their T2T assembly. However, and as the authors are very aware of, because Mi is AAB there are necessarily multiple repeated and multi-positioned contacts from Hi-C data between A1 and A2, this can lead to parasite noise in the Hi-C map. This seems to be the case in Response Figure 10.

3- Hypothesis of unreduced gametes / hybridization and other hypotheses

The authors have taken into consideration the suggestion of recent independent hybridizations for Mi and Ma-3n to explain the 4-5% divergence between the A1 and A2 subgenomes, closer relation of A1-A2 within species than between species in their tree topologies and very low divergence at the mitochondrial level.

The authors have profoundly changed this part of the manuscript and now propose this scenario is the most likely.

- Mitochondrial topology

Concerning the mitochondrial topology, I agree with the authors that changing the root of the tree (rooted with Me as an outgroup vs. rooted at the midpoint) changes the sense of reading although all the species remain monophyletic. Although Me is indeed quite distant compared to the rest of the Meloidogyne here, I do not see any problem using it as an outgroup species to root the tree. It seems that the mid-point topology looks different mainly because the Mi from this study produces a very long branch compared to the rest of the Mi. I think adding *M. enterolobii* as an outgroup is the most reasonable choice so far because, it is the most closely related species available even if relatively distant from the rest and using it as an outgroup allows giving a direction for reading the tree. In supplementary Fig. 12 the authors have used Mg as an outgroup which is even more distantly related than Me. Therefore, I do not understand the problem raised by using Me, which is much more closely related, as an outgroup for the mitochondrial phylogeny.

- 18 Nuclear Topologies (supplementary Fig. 36)

I still find really surprising that the 18 nuclear tree topologies are so different one from another and that there does not seem to have a dominant majority topology. The only strong and consistent signal is the separation of the A from the B subgenomes. However, within A and within B the topology seems to be changing in each tree with no main constant signal. And again, none of this topology corresponds to the mitochondrial topology.

I wonder to what extent the manually edited 'T2T' genomes might be responsible for these inconsistent topologies. Indeed, if on some occasions A1 contigs have been fused to A2 contigs to artificially obtain telomeres at both ends, this might completely blur the phylogenetic signal.

In the light of major point #2, it is now clear that there is no chromosome with Mi-Tel at both ends in *M. incognita* and this is probably also the case in *M. javanica* and *M. arenaria*.

I strongly recommend performing the same phylogenetic analyses at the nuclear level between the subgenomes without prior manual editing of the assemblies for either multiples of 18 chromosomes or Mi-tel at both ends. Even if this will probably result in more trees and with less concatenated genes, this will exclude the risk of A1-A2 fusions or even A-B fusions. By eliminating this risk, it will be very

interesting and informative to assess whether some main majority topologies seem to emerge now, because so far this is really surprising that the topologies of each of the 18 groups diverge so much from one group to the other.

- Other important points to be considered.

I found a bit worrying that the explanation for the high standard deviation in coverage (depth) in Mi would be due to low quality Illumina data for Mi. These same reads have been used to polish the genome so if they are of low quality, this raises some concerns. The authors should proceed with a QC of the Illumina read and appropriate cleaning with FastP or Trimmomatic according to the possible problems identified in QC before using them for k-mer analyses or polishing the genome.

Finally, a point that needs further improvement concerns the way collapsed regions have been resolved. I agree on the method to detect them (based on double read coverage). However, I disagree that duplicating them at 100% identity is always correct. Indeed, many regions at double coverage probably represent collapse between A1 and A2 with both copies having low divergence albeit not necessarily 0%. Therefore, duplicating the region at 100% identity does not represent the reality of this divergence. It would be more appropriate to use linkage information between SNPs using long reads to try to phase and unzip these collapsed regions and represent their true divergence.

Reviewer #2:

Remarks to the Author:

Again, I would like to congratulate the authors on their thorough response to reviews. The methods section seems to be improved regarding clarity in the assembly strategy and quality control. I think designating Ma 3n and 4a is a much better approach to describing and discussing these data and reflect some really interesting findings. I think the additional supplemental figures are helpful and I can see how the SCAR for Ma 3n was interpreted as not having a band. I recommend accepting this manuscript for publication. I have just a few minor comments.

I agree that a term distinct from "phased" is appropriate here and "unzipped" seems useful. I think the authors could clarify what is meant by "unzipped" in the Introduction, as it is (at least not yet) not regularly applied to this scenario. Basically, line 104 could include an "i.e., separated subgenomes" or something along those lines. Later (line 366) "divided subgenomes" is used, so that may be how "unzipped" can be defined.

Line 169: for proteome[s]

Line 289: HGT is not defined in text as "horizontally transferred genes".

Lines 351-354: I agree with the assessment from Reviewer #1 that gene conversion probably couldn't have been so extensive to explain the observations, but I would refrain from saying gene conversion "could not extensively occur" because it sounds too certain, especially without a reference to back up that claim. I think just saying that you weigh that as unlikely compared to multiple hybridizations – where hybridization is regular – is fine.

Lines 760-761: I suggest changing the language to indicate that it is unclear what underlies the false-negative PRC result.

REVIEWER COMMENTS

Reviewer #1 (Remarks to the Author):

I would like to congratulate once again the authors for the efforts deployed to improve further their manuscript and for their comprehensive point-by-point response.

Most of the points raised have been correctly addressed and many points that remained unclear in the previous version are now clarified.

Response: Thanks to the reviewers for their careful review, which greatly helped us improve the quality of this manuscript.

However, a preprint recently posted in bioRxiv (<https://doi.org/10.1101/2023.03.29.534350>) confirmed my concerns and doubts about the validity of the T2T assemblies.

Indeed, in this analysis the authors have used FISH experiments and confirmed a repeat similar to the Mi-tel identified here was present at telomeric locations in *M. incognita*, but only at one end of the chromosomes.

I am afraid in the light of these new biological results contradicting the T2T assemblies, several parts of the manuscript should be revised accordingly.

Response:

The work of this bioRxiv has set an example in the study of telomeres of *M. incognita* chromosomes, laying the foundation for us to carry out more detailed studies of Mi chromosomes. However, regarding the conclusion in this study that only one end of Mi's chromosomes has telomeres, we think further research is needed. A reasonable guess is that the telomeres at both ends of the chromosome may have different lengths, and the number of telomere repeats at one end is relatively less, resulting in weak or even no FISH signal.

Considering this possibility, we used Mi's egg in this study to perform FISH experiments to verify the position of telomeres. The results showed that many chromosomes appeared to have a telomere signal at only one end, but there were also multiple chromosomes with signals at both ends, and the signal at one end was significantly stronger than the signal at the other end (**Response Fig. 1**). Moreover, we also found that there are indeed some chromosomes with telomere signals inside. These results suggest that our guess is credible.

To address the concerns of the reviewers regarding manual editing of the genome, we have generated an assembly called **assembly v2**, which has not undergone any manual editing.

Assembly V2 was constructed using a more standardized workflow:

Mi: ONT contigs (Canu) + Illumina (Nextpolish) + BioNano (Solve) + HiC (Juicer+3D-DNA+Juicebox)

Ma3n/Ma4n/Mj: PacBio contigs (Canu) +Illumina (Nextpolish) + HiC (Juicer+3D-DNA+Juicebox)

Assembly V2 does not have manually added telomeres, manually recovered collapsed regions, or any manual editing. Furthermore, to enhance reproducibility, we have provided the assembly file that can be imported into Juicebox to reproduce our assembly results. All subsequent analyses were re-performed using the updated assembly v2. All related figures, tables, and data have been updated accordingly (**Fig. 3-5, Supplementary Fig. 21-41, Supplementary Table 6,8,15-25**).

The assembly file used in Juicebox are uploaded in GitHub (https://github.com/xiecs-BMB/MIG-genome/tree/master/Assembly_V2/juicebox_file). The genome file of assembly v2 were uploaded to NCBI in BioProject PRJNA784524 and our Lab website (<http://bmb.hzau.edu.cn/info/1031/1381.htm>)

Response Fig. 1 Fluorescent in situ hybridization of Mi-Tel indicated its location on the *M. incognita* chromosome. (a) The Mi-Tel FISH results of some Mi chromosomes are displayed. Fluorescent signals at both ends of multiple chromosomes can be clearly observed, as well as fluorescent signals inside the chromosomes. We can clearly observe that the signal at one end of some chromosomes is stronger than that at the other end. (b-d) The Mi-Tel FISH results of all the Mi chromosomes are displayed, the large green patch at the top of **Response Fig. 1d** is the autofluorescence of the eggshell. Fluorescent signals at both ends of multiple chromosomes also can be observed. Mi-Tel repeats with FITC fluorophore were used as probes for FISH, chromosomes are counterstained with DAPI, scale bar = 5 μm .

Below are comments about the main points, most of which having been correctly addressed.

1- Taxonomic identity of the samples and Ma-3n vs. Ma-4n

I would like to thank the authors for the detailed explanations that clarify the situation regarding the previous Mp isolate which is now called Ma-3n to highlight the difference with Ma-4n.

I am convinced by the explanations provided by the authors concerning species identities. I also personally downloaded the genome assemblies and confirm the relevant SCAR markers regions are present in the corresponding species. Having identified two different branches of Ma, one being 3n and the other one being 4n is very interesting and strongly suggests Ma is not a single valid species. This also highlights using just one marker for species identification might be insufficient in the future for these species. According to the mitochondrial phylogenetic analysis of the authors, Ma-3n and Ma-4n are as different as Mjav is different from Minc.

Response: We thank the reviewer for the positive comments.

Interestingly the two other Ma-3n populations (HarA, and L32) originate from one single same paper (Szitenberg et al. 2017, Genome Biol Evol). This paper already cited as ref24. Should be cited also as a source for the mitochondrial analysis. Unfortunately, in this ref24 paper, there is no precise information about how species were identified (which markers were used). The authors in Szitenberg et al. just mentioned species identification was done in the laboratories which provided the isolate and confirmed by sequencing but with no further explanation.

Therefore, I fully agree with the authors that this point about Ma-3n vs. Ma-4n deserves further study but is beyond the scope of the current paper. I think this discovery opens interesting perspective that must be studied in more detail in the future and published in a nematology journal. Such a future analysis should ideally include SCAR, esterase as well as morphological and host compatibility tests. The final conclusions might be that *M. arenaria* is not a valid species and that at least two different cytogenetic types with different mitochondrial genomes exist.

Response: We thank the reviewer for the affirmation. Similarly, we also believe that the taxonomy of Ma deserves further in-depth study in the future, especially the need for further research by nematode taxonomists to define it.

This is a pity the initial Yunnan Ma-3n sample was lost during COVID-19 crisis and, indeed, the absence of Ma SCAR band in the initial analysis might be due to initial PCR failure. The presence of the *M. arenaria* SCAR sequence in the genome assembly tends to support this possibility.

Response: Although Yunnan Ma-3n died in COVID-19, Ma-3n is a common population, and our subsequent PCR of other Ma-3n samples also confirmed the presence of Ma-specific SCAR bands.

Supplementary Figure 1: b: please indicate what is present in each track of the gel, a legend is missing there. Is it the Ma SCAR primers everywhere or all the SCAR primers for all the tested species in each track? This information must be provided. Moreover, in the legend, the authors cannot write: “at that time, only 18S was considered to have a band, but there was a very faint band in Ma-SCAR which ignored by us before.” This is completely acceptable in the point-by-point response to the reviewers but not in the paper itself or the supplementary information. I would suggest not to add this new gel with the other Ma-3n population and only state that although only a weak band was observed with the Ma SCAR marker initially, analysis of the

genome confirms the presence of the Ma SCAR marker in this population.

Response: Thanks to the reviewer for pointing out the issue, we have revised the description in this legend (**Line 1412-1415**) and removed the new gel with the other Ma-3n population. The primers used for each track are marked in the Supplementary Figure 1a and 1b, and the templates of all tracks are the same template, so as to test which specific primers can amplify the bands of the sample, and to determine the species.

2- 'T2T' genomes and manual edition of the assemblies

A preprint recently posted in bioRxiv (<https://doi.org/10.1101/2023.03.29.534350>) by Zotta-Mota et al. brings several clarifications and important results about the telomeres of these root-knot nematodes. In this paper, the authors have sequenced the genomes of three of the species studied here (Mi, Mj and Ma-4n) using ONT + Illumina. Using a different approach, the authors have identified similar repeats mostly at one extremity of multiple contigs in these species. The authors then wanted to confirm whether these repeats had telomeric positions in the chromosomes or their presence at one extremity of multiple contigs was due to an assembly artifact. Using FISH experiments in *M. incognita*, the authors confirmed the repeat has a telomeric position. However, this telomeric repeat is present at only one end of the chromosomes and never observed at both ends. This surprising finding was confirmed by clear observations both at Metaphase and Prophase stages and was consistent with this repeat being present at only one extremity in the contigs as well.

This preprint in bioRxiv confirms with FISH that the Mi-Tel repeat identified by Dadong Dai et al. has indeed a telomeric position and can now be considered as a telomeric repeat. This important biological confirmation should be cited in the paper.

Response:

The preprint in bioRxiv have been cited in revised manuscript (**Line 248-250**). Thanks for suggestion. However, we performed FISH experiments on telomeres of *M. incognita* samples in this study and found that there were telomere signals at both ends of multiple chromosomes. Due to the large intra-species variation of the root-knot nematode population, there are large differences in the number of chromosomes, etc., we do not rule out that other populations do have telomeres with only one end of all chromosomes that can be observed with fluorescent signals.

However, in the study by Zotta-Mota et al., Mi-tel is clearly shown to be only at one end of the chromosomes. These findings confirm my doubts about the validity of the manually edited 'chromosomes' in the 'T2T' assembly presented here. I completely understand that the authors guided their assemblies with this idea in mind that all the chromosomes should start and end with telomeric repeats. However, unlike in most other eukaryotes, it is now biologically confirmed that *M. incognita* chromosomes do not have a same telomeric repeat at both ends. Therefore 'telomere to telomere' assembly or 'T2T' with Mi-Tel does not make sense as it does not apply to this species. This most likely also do not apply to Mj and Ma as the telomeric repeat was also found mostly at one single end of the contigs in this study.

Response:

As suggested by the reviewers, we have removed the T2T genome of the polyploid RKNs

and provided a new version of the assembly that has not undergone any manual editing. All subsequent analyses have been re-performed using the newly provided genome (assembly v2).

These findings, which contradicts the T2T assemblies, also explain several surprising metrics, which were therefore probably due to artifacts caused by the assembly strategy.

- The size of the 'T2T' genome assembly(225Mb), is bigger and out of the range estimated by flow cytometry (189 +/- 15 Mb in Blanc-Mathieu et al. 2017 PLoS Genetics) and also bigger than k-mer estimates in the present study (ca. 212 Mb both wit ONT and PacBio, Table S3).
- The number of Mi chromosomes (40) in the T2T assembly is smaller than counts recently made with modern technique in *M. incognita* on two different populations 46 in (Despot-Slade et al. Mol Biol Evol 2021) and also 45-47 in the bioRxiv preprint by ZottaMota et al.).
- The size of the biggest chromosomes seems inconsistent with measured sizes in (Despot-Slade et al. Mol Biol Evol 2021). Indeed, the *C. elegans* chromosomes range between 14 - 22 Mb in size and measure 5µm in Metaphase. In comparison, *M. incognita* chromosomes are much smaller (0.4 – 1.5µm long) and it is difficult to imagine they reach 14 Mb.

In contrast, the metrics obtained in the assembly V1 in which telomeres have not been manually added at both ends is more consistent with biological observations.

- The assembly size is 213Mb which is closer to the k-mer and flow cytometry estimates.
- The number of scaffolds is 48, which is very close to the counted number of chromosomes (45-47).

Response:

The original version of the T2T genome has been removed and replaced with the unedited version (assembly v2). We still want to emphasize that the genome size, number of chromosomes, and the biggest chromosome size are not results of the assembly strategy.

For genome size:

The genome assembly at the contig level has reached 225Mb, and there are still some sequences that remain partially collapsed. At least the contigs size are not affected by manual editing, and the subsequent scaffolding process does not introduce new sequences, except for a few N's.

In our newly unedited assembly (assembly v2), the total assembly size is also 225Mb, consisting of 39 scaffolds. If we only consider the size of those 39 scaffolds, it amounts to 211Mb. This means that 6% of the total assembly size are unscaffolded, which may contribute to the larger genome size to some extent.

Moreover, in *M. incognita* populations, the number of chromosomes varies greatly between populations, and the genome size of different populations may be different. Therefore, the size estimated by flow cytometry (189 +/- 15 Mb in Blanc-Mathieu et al. 2017 PLoS Genetics) may not fully represent the genome size of all *Mi* populations.

For chromosome number:

Assembly v2 consists of 39 scaffolds, but based on the distribution of read depths, it was observed that two whole chromosomes have significantly higher sequencing depths than the average, indicating that these two chromosomes are likely to be collapsed. Therefore, the real chromosome number should be 39-41, depending on whether the collapsed chromosomes have

fused with other chromosomes.

We understand that the chromosome number of 39-41 is smaller than other Mi populations as reported. In fact, this is not caused by our assembly strategy but rather reflects the true nature of this organism. Literature suggests that different Mi populations have varying chromosome numbers. The Mi strain used in this study may differ from other Mi samples, resulting in differences in the chromosome number. Based on this, we counted the number of chromosomes of Mi in this study, and multiple karyotypes photos showed that the number of chromosomes was between 39-41 (**Response Fig. 1b-d and Response Fig. 2**). Besides, we measured the lengths of chromosomes using imageJ and found that they were between 0.5-2.5um. Furthermore, differences in chromosome condensation state may result from differences in colchicine treatment, among others, would have contributed to the observed significant chromosome length differences. At the same time, comparing the karyotypes of *M. incognita* in this study, it can be found that there are chromosomes with large differences in length, while the length of chromosomes in bioRxiv (<https://doi.org/10.1101/2023.03.29.534350>) by Zotta-Mota et al. is more uniform, indicating that these are two obviously different *M. incognita* populations.

On the other hand, we are conducting a genome-wide root-knot nematode population genome in China, which contains multiple resequencing data of root-knot nematodes downloaded from the Internet. We found that root-knot nematodes are mainly divided into three clades. The Mi samples in this study are in the smaller branch. When we applied the same assembly strategy to another Mi sample collected in our lab, we obtained 44-45 chromosomes (**Response Fig. 3**). These results indicate that the chromosome size is not determined by the assembly strategy but the number of chromosomes differed between different populations of root-knot nematodes.

Response Fig. 2 The chromosome karyotype of Mi in this study. It can be counted to 39-41 chromosomes, and the chromosome size measurement on the left shows that the size range is between 0.5-2.5 um. Chromosomes are counterstained with DAPI.

Response Fig. 3 phylogenetic tree of 300 Mi sample (unpublished data). Black dot represents the published Mi sample in NCBI. Most of published Mi sample are located in the blue branch, while our sample are located in the red branch.

For the biggest chromosome size:

Different Mi strains have varying numbers of chromosomes, and the reason for this difference could be attributed to variations in the patterns of chromosome fusion. The longest chromosome in our newly provided assembly v2 is 15Mb. According to synteny analysis, this longest chromosome is the result of fusion of three ancestral chromosomes. In assembly v2, we do not assume the presence of telomeres at both ends, so we do not need to forcefully merge two unrelated chromosomes into one. In this case, the reason for anchoring contigs to chr1 is because the Hi-C signal inside Chr1 is continuous and strong. If it's two chromosomes, it's hard to know where to break from. Therefore, we believe that the longest chromosome with 15Mb size is real, at least supported by Hi-C evidence. Furthermore, our cytological evidence also indicated the presence of significantly longer chromosomes than the others in the Mi samples of this study (**Response Fig. 1 and 2**).

Of course, we understand the concern of the reviewers regarding the potential noise in Hi-C data between homologous chromosomes. However, in the case of chr1, there are no homologous chromosomes present. According to syntenic analysis (**Response Fig 4, also Fig. 3c in main text**), chr1 is a result of the fusion of C7(A1), C14(A1), and C18(B1). The internal Hi-C signals are unlikely to be caused by multiple repeated contacts between homologous chromosomes. We have also presented a case demonstrating the noise in Hi-C signals between homologous chromosomes (**See Response Fig 6 in below**). In the case of Chr 1, the Hi-C signals should be real rather than noise from homologous chromosomes.

Response Fig 4 Hi-C map and syntenic analysis of Chr1.

Furthermore, at the contig level assembly, most contigs >1Mb have Mi-tel at only one end. Finally, in the ‘T2T’ assembly, 11 of the 12 biggest chromosomes present internal Mi-tel repeats and probably represent assembly artefacts as no Mi-tel fluorescence was observed in the middle of a chromosome in *M. incognita* in FISH experiments.

Response: We performed Mi-Tel analysis with the newly assembled chromosome-level v2 genome and the contig-level assembly, and found that there are Mi-Tel at both ends of some of the scaffolds and contigs (**Response Fig. 5**). Besides, our Mi-Tel FISH experiment found the fluorescent signal in the middle of some chromosome (**Response Fig. 1**), suggests that the chromosome fusion event did occur in the Mi sample in this study, while the lack of signal in other chromosomes may be due to the lack of fusion or the weak signal caused by the less number of Mi-Tel repeats at the fusion site.

Response Fig. 5 The Mi-Tel distribution on the scaffold and contig level of Mi, Ma-3n, Ma-4n, and Mj.

For this ensemble of reasons and in the light of these new results posted in bioRxiv that confirmed my doubts: these T2T manual assemblies should be removed from the revised version of the paper

as they do not correspond to a biological reality (as confirmed by FISH). The title and abstract should be revised accordingly (no T2T for the polyploid species), as well as the rest of the manuscript.

Response:

As suggested by the reviewers, we have removed the T2T genome of the polyploid RKNs and provided a new version of the assembly that has not undergone any manual editing. All subsequent analyses have been re-performed using the newly provided genome (assembly v2). But we think that the conclusion of this bioRxiv may only represent the population used in their experiments, and our biological experiments also show that there are telomere repeat sequences at both ends of chromosomes (There may be more chromosomes with fewer repeat sequences at one or both ends, so that only one end has FISH signal or even no signal at both ends).

In my evaluations of the previous versions of the paper, I already suggested not to include the extensively manually edited genome assemblies. The rest of the results presented by the authors remain interesting enough and of high-quality and these T2T assemblies finally provide no added value to the findings.

Some of the new analyses provided by the authors in the point-by-point response also do not further support the T2T assembly. In the dotplot analysis of either BioNano+PacBio or BioNano+ONT vs. the T2T, chromosome 1 is always composed of multiple different and relatively short BioNano scaffolds and contains several internal Mi-Tel repeats (Response Figures 8 & 9). Similarly, the end of chromosome 2 (which contains the Mi-Tel repeats) is supported by a different BioNano contig than the rest of chromosome 2 in PacBio + BioNano dotplot against T2T. I appreciate and acknowledge that the authors provide additional Hi-C data to try to support their T2T assembly. However, and as the authors are very aware of, because Mi is AAB there are necessarily multiple repeated and multi-positioned contacts from Hi-C data between A1 and A2, this can lead to parasite noise in the Hi-C map. This seems to be the case in Response Figure 10.

Response:

T2T genome of the polyploid RKNs has been removed in revised manuscript, and we added a new version of the assembly that has not undergone any manual editing.

Mi is a triploid species, which means that Mi has three copies ($A_1A_2B_1$) of each ancestral chromosome. There may be duplicated segments between the three copies, which is expected to result in some Hi-C noise. Notably, Hi-C noise only exists between homologous chromosomes, such as C1_A1 vs C1_A2, and not between non-homologous chromosomes, such as C1_A1 vs C9_A2. In the case of chr1, as shown in **Response Fig 4**, chr1 is formed by the fusion of C7(B_1), C14(A_1), and C18(A_1), which are not homologous chromosomes. Therefore, there would not be significant noise between them that would result in incorrect assembly. Furthermore, even between homologous chromosomes, the Hi-C noise has a very weak signal that does not affect the correct assembly (**Response Fig 6**).

Response Fig 6 Hi-C noise between homologous chromosomes. Here, we present the four homologous chromosomes of Ma4n, and the arrows indicate regions where Hi-C noise from duplicated segments. These noise signals are very weak compared to the real Hi-C signals and therefore do not impact the assembly.

3- Hypothesis of unreduced gametes / hybridization and other hypotheses

The authors have taken into consideration the suggestion of recent independent hybridizations for Mi and Ma-3n to explain the 4-5% divergence between the A1 and A2 subgenomes, closer relation of A1-A2 within species than between species in their tree topologies and very low divergence at the mitochondrial level.

The authors have profoundly changed this part of the manuscript and now propose this scenario is the most likely.

Response: Thanks again to the reviewers for their suggestions.

- Mitochondrial topology

Concerning the mitochondrial topology, I agree with the authors that changing the root of the tree (rooted with Me as an outgroup vs. rooted at the midpoint) changes the sense of reading although all the species remain monophyletic. Although Me is indeed quite distant compared to the rest of the Meloidogyne here, I do not see any problem using it as an outgroup species to root the tree. It seems that the mid-point topology looks different mainly because the Mi from this study produces a very long branch compared to the rest of the Mi. I think adding *M. enterolobii* as an outgroup is the most reasonable choice so far because, it is the most closely related species available even if relatively distant from the rest and using it as an outgroup allows giving a direction for reading the tree. In supplementary Fig. 12 the authors have used Mg as an outgroup which is even more distantly related than Me. Therefore, I do not understand the problem raised by using Me, which is much more closely related, as an outgroup for the mitochondrial phylogeny.

Response:

The mitochondrial topology structure obtained using Me as the root and the topology structure obtained using the midpoint rooting are different. Even if we remove the Mi samples with longer branch lengths, the topology structure of the midpoint rooting does not change significantly (**Response Fig 7c and 7d**). In the absence of more suitable outgroup species (Me is

distant relatively from others), we tend to use the midpoint rooting. Moreover, the topology structure obtained using the midpoint rooting can be simplified as shown in **Response Fig 7e**, which is more similar to the topology structure of the nuclear genome (**Response Fig 7**). In supplementary Fig. 12, using Mg as the root or using the midpoint as the root does not affect the topology structure.

To avoid causing confusion, we thus provide an unrooted tree for the mitochondrial genome.

Response Fig 7 Phylogenetic tree among multiple species of root-knot nematode.

(a-b) Phylogenetic tree of Clade I RKNs, rooted with *M. enterolobii*.

(c-e) Phylogenetic tree of Clade I RKNs, removed *M. enterolobii* and rooted with midpoint.

- 18 Nuclear Topologies (supplementary Fig. 36)

I still find really surprising that the 18 nuclear tree topologies are so different one from another and that there does not seem to have a dominant majority topology. The only strong and consistent signal is the separation of the A from the B subgenomes. However, within A and within B the topology seems to be changing in each tree with no main constant signal. And again, none of this topology corresponds to the mitochondrial topology.

I wonder to what extent the manually edited 'T2T' genomes might be responsible for these inconsistent topologies. Indeed, if on some occasions A1 contigs have been fused to A2 contigs to artificially obtain telomeres at both ends, this might completely blur the phylogenetic signal.

In the light of major point #2, it is now clear that there is no chromosome with Mi-Tel at both ends in *M. incognita* and this is probably also the case in *M. javanica* and *M. arenaria*.

I strongly recommend performing the same phylogenetic analyses at the nuclear level between the subgenomes without prior manual editing of the assemblies for either multiples of 18 chromosomes or Mi-tel at both ends. Even if this will probably result in more trees and with less concatenated genes, this will exclude the risk of A1-A2 fusions or even A-B fusions. By eliminating this risk, it will be very interesting and informative to assess whether some main majority topologies seem to emerge now, because so far this is really surprising that the topologies of each of the 18 groups diverge so much from one group to the other.

Response:

We have performed phylogenetic analysis using newly generated assembly v2. In assembly v2, we did not obtain telomeres at both ends through manual editing. As mentioned in **Response Fig 6**, the Hi-C signal between A1 and A2 copies is very weak, which would not cause A1-A2 fusions during genome assembly.

According to **supplementary Fig 34**, all 18 phylogenetic trees support the topology structure shown below (**Response Fig 8a**), except for the topology structure within the gray box or black triangular (**Response Fig 8a**). If the black triangular part ($A_1A_2B_1$ of Ma3n/Ma4n/Mj) is collapsed, these 18 trees show a consistent dominant topology structure. Furthermore, the A1 tree, A2 tree and B1 tree can be simplified as shown in **Response Fig 8b**, which is consistent with the topology structure of the mitochondrial genome (midpoint rooting).

In the revised version, we have added the **Response Fig 8a** to the **Supplementary Fig 34** to assist readers in understanding the dominant topology structure of these trees.

Response Fig 8

(a) Statistics of the topological structure of 18 phylogenetic trees.

(b) Simplified A1, A2, and B1 trees, which are consistent with the structure of the Mit tree (simplified and midpoint rooting).

- Other important points to be considered.

I found a bit worrying that the explanation for the high standard deviation in coverage (depth) in Mi would be due to low quality Illumina data for Mi. These same reads have been used to polish the genome so if they are of low quality, this raises some concerns. The authors should proceed with a QC of the Illumina read and appropriate cleaning with FastP or Trimomatic according to the possible problems identified in QC before using them for k-mer analyses or polishing the genome.

Response:

Thanks for suggestion. All sequencing data underwent quality control and cleaning before use. We re-checked the QC report of Mi Illumina data (see below) and found that the quality is high. Thus, this data will not influence the polish step of contigs. The high standard deviation in depth may be attributed to the library preparation process. In the revised version, we have provided additional Illumina data from the same Mi sample to perform sequencing depth analysis. The newly generated depth distribution plot appears to be better (see **Supplementary Fig 22**). The additional Illumina data are also upload in NCBI SRR24770443.

Finally, a point that needs further improvement concerns the way collapsed regions have been resolved. I agree on the method to detect them (based on double read coverage). However, I disagree that duplicating them at 100% identity is always correct. Indeed, many regions at double coverage probably represent collapse between A1 and A2 with both copies having low divergence albeit not necessarily 0%. Therefore, duplicating the region at 100% identity does not represent the reality of this divergence. It would be more appropriate to use linkage information between SNPs using long reads to try to phase and unzip these collapsed regions and represent their true divergence.

Response:

Thanks for suggestion. In our newly generated v2 version, to avoid excessive editing of the genome, we did not split the collapsed regions. The larger folding regions are marked in **Supplementary Fig 22-25**. These collapsed regions were also filtered out in our subsequent analyses (such as gene expression level analysis or phylogenetic analysis).

Reviewer #2 (Remarks to the Author):

Again, I would like to congratulate the authors on their thorough response to reviews. The methods section seems to be improved regarding clarity in the assembly strategy and quality control. I think designating Ma 3n and 4a is a much better approach to describing and discussing these data and reflect some really interesting findings. I think the additional supplemental figures are helpful and I can see how the SCAR for Ma 3n was interpreted as not having a band. I

recommend accepting this manuscript for publication. I have just a few minor comments.

Response: We thank the reviewer for the positive comments, your suggestions have greatly improved our manuscript.

I agree that a term distinct from “phased” is appropriate here and “unzipped” seems useful. I think the authors could clarify what is meant by “unzipped” in the Introduction, as it is (at least not yet) not regularly applied to this scenario. Basically, line 104 could include an “i.e., separated subgenomes” or something along those lines. Later (line 366) “divided subgenomes” is used, so that may be how “unzipped” can be defined.

Response: Thanks to the reviewers for their suggestions, we have made adjustments and revisions to the manuscript.

Line 169: for proteome[s]

Response: Edited, thanks.

Line 289: HGT is not defined in text as “horizontally transferred genes”.

Response: Edited, thanks.

Lines 351-354: I agree with the assessment from Reviewer #1 that gene conversion probably couldn't have been so extensive to explain the observations, but I would refrain from saying gene conversion “could not extensively occur” because it sounds too certain, especially without a reference to back up that claim. I think just saying that you weigh that as unlikely compared to multiple hybridizations – where hybridization is regular – is fine.

Response: We have edited the sentence as suggested (**Lines 350-355**), thanks.

Lines 760-761: I suggest changing the language to indicate that it is unclear what underlies the false-negative PRC result.

Response: We have edited this language in **Lines 762-764** as suggested.

Reviewer #1 (Remarks to the Author):

Following my previous evaluation of their paper, the authors have considerably modified this new revised version and I would like to congratulate them for that and all the additional work that appears in the point-by-point response.

The authors now propose a non-manually edited version of their *Meloidogyne* genomes V2. These new versions are not edited to add telomeres at both ends of the scaffolds and the collapsed regions have not been artificially un-collapsed by producing two duplicated 100% identical subgenomes.

I really think at this stage of our knowledge of the karyotype and chromosomes of these *Meloidogyne* species, these are much more reasonable representations of the polyploid RKN genomes.

Indeed, the exact number of chromosomes, as well as how many have Mi-tel repeats at one end, both ends, in the middle or not at all in the polyploid RKN remains uncertain. The preliminary FISH analyses provided by the authors do not allow to precisely determine the number of chromosome and it is difficult to tell whether some (and how many) chromosomes have Mi-tel at both ends because many chromosomes overlap in the FISH images and are not well separated. Therefore, there are too many unknown characteristics at this stage to guide genome assembly and scaffolding with strong a priori assumptions and these new unguided genomes V2 are more representative of the state of our knowledge.

I completely agree with the authors that the number of chromosomes in these species is likely to vary from one population to another, and possibly involving some chromosomal fusions / fissions. These karyotypic variations might indeed explain part of the differences in genome assemblies and FISH observations between this paper and the one posted in bioRxiv by Zotta-Mota et al on other populations of the same species.

Overall, I now consider this major point has been addressed by the authors.

Response:

Thanks for the reviewer's comments, your comments have made us a lot better. Our FISH results are mainly used to confirm that Mi-Tel is located at the terminal position of the chromosome. We will not describe whether Mi-Tel exists at both ends of the chromosome in this work but will analyze it in detail in the next Mi population genetics work. We have therefore incorporated the FISH results into the manuscript results and added a figure (**Supplementary Fig. 16**).

I have only one other major point concerning the evidence supporting 'extensive' chromosomal fusions that, in my opinion, needs to be addressed.

The rest are more minor yet important suggestions that, I hope, will help clarify further the paper, which is now of high quality and comprehensive.

- MAJOR POINT: evidence for extensive chromosomal fusions

The whole sections on chromosomal fusions are very interesting as it might explain several idiosyncrasies in the assembly. However, currently some explanations are missing to help the reader understand exactly the data supporting (or not) these putative fusions.

For assemblies V1, it is very important to explain clearly that fusions are predicted between McScanX linkage groups syntenic to Mgram and considered ancestral

chromosome parts. Indeed, it took me a long time to understand the nomenclature of scaffolds Mi_ChrXX_XX in Figures 2f as well as Figures S16 and S17. Only TableS10 allowed me to fully understand that this nomenclature was according to synteny with the 18 Mgram chromosomes. Providing a supplementary figure similar to FigS31 but with assemblies V1 and Mg chromosomes like in previous versions of the manuscript might help the readers fully understand.

Response:

Thanks for suggestion. We have mentioned how fusions are predicted in legend and added a figure into supplementary Fig. 17 to show the nomenclature of assembly v1.

Initially, the analysis of chromosome fusions in assembly v1 was conducted with the aim of providing more comprehensive evidence for Mi-Tel as telomeres. Now, Mi-Tel as telomeres has been thoroughly validated through FISH. The analysis of chromosomal fusion in assembly v1 just indicated the presence of some telomere-to-telomere fusion cases and lacked strong evidence. Therefore, we have tone down this part in the main text by moving some results into the supplementary files (**Supplementary Fig. 17**).

Response Fig 1

Looking into more details at Figure S16, I realized that two of the four putative fusions of ancestral chromosomes involved V1 scaffolds present in two copies rather than the three expected, namely in (1) Mi_Chr9_24 and in (3) Mi_Chr6_17. For Mi_Chr6_17 this might represent a case of collapse between A1 and A2. Have these contigs higher coverage in Fig S7 for instance? If that is the case, then there is an alternative hypothesis. Indeed, this might represent collapse between one subgenome having Mi-tel repeats and the other one not having Mi-tel. In that case, the collapsed consensus will have Mi-tel but it is still possible that the true contact information is with the other subgenome not having Mi-tel. The alternative version without Mi-tel being absent in the assembly because of the collapse these contact information cannot be represented. Therefore, for these two putative fusion events, it cannot be excluded that this results from assembly collapse and artefact of Hi-C and BioNano. Indeed, if A1 and A2 are very similar, BioNano Optical map will produce exactly the same barcode for the two molecules and also fuse them as a single molecule with no possibility to make any difference between the two.

Response:

For Mi_Chr6_17, based on syntenic analysis, the other copy has been assigned to other scaffolds due to fusion and fission (**Response Fig 2**). The sequence depth of Mi_Chr6_17 is also consistent with the whole genome and does not exhibit a higher coverage. We have updated the Figure S16 to show all of homoeologous copies as reviewer suggested (**Supplementary Fig. 17**).

Response Fig 2

Ideally, to disentangle these cases of putative fusions, it would be interesting to provide a similar figure than S16a but including Hi-C data for all the homoeologous copies. For instance, include MiChr4_10 and 4_12 as well as 9_25 in addition to 4_11 and 9_24 for (1) and include 12_31, 12_33 as well as 6_16 for (3). It is very important to show whether contact data support alternative linkage information as strongly or not than the one proposed by the authors to support their telomere-to-telomere fusions.

Response:

We have updated the Figure S16 to show all of homoeologous copies as reviewer suggested (**Supplementary Fig. 17**).

Response Fig 3

Furthermore, in FigureS16 b(4), Mi_Chr7_10 does not exist in the assembly or Table S10. I guess it is rather Mi_Chr7_18? This would be consistent with Figure 2f and Figure S17. Please clarify this point and also explain why Mi_Chr12_33 is represented two times in Figure S16b (4)?

Response:

Thanks for carefully check. We have updated the FigureS16 to correct this typo. Two times of Mi_Chr12_33 is because originally wanted to show a gap in Mi_Chr12_33 and now are updated (**Supplementary Fig. 17**).

Besides, I maintain that FigureS17 with only one single read supporting the telomere-to-telomere fusion does not really make sense. More coverage of this junction with ultra-long reads would be necessary to unequivocally support that fusion.

Response:

We agree with the reviewer's point that supporting telomere-to-telomere fusions with only one single read evidence is weak. We have made efforts to find more supporting reads for telomere-to-telomere fusions. However, due to the gap size of approximately 70Kb, it requires long reads (>100kb) to span the gap, and such reads constitute only a small portion of our data. As a result, we did not find more supporting reads.

The initial purpose of conducting this part of the work was to provide more evidence for Mi-Tel being telomeres. Now, it has been confirmed through FISH that Mi-Tel serves as telomeres (both in our study and another research). This part of the content only indicates the presence of telomere-to-telomere fusion cases, which are not the main point of telomere section of our paper. Therefore, we have decided to tone down this aspect in main text (**Line 245-255**).

Overall, with only 4 examples of telomere-to-telomere fusions, two of which potentially involving collapsed scaffolds, the evidence is too weak at this stage to unequivocally support these putative fusions.

Therefore, I would suggest toning down this claim like 'we identified some putative chromosomal fusions in Minc' (as nothing is said about putative fusions in the other Meloidogyne species as well in this part of the paper).

Response:

We agree with the reviewers and have edited the sentence as suggested (**Line 245-247**).

Concerning assemblies V2, if I understand well, the authors have realized synteny was much better conserved between Minc B1 subgenome and the other subgenomes than with Mgram. Therefore, they considered B1 as a better representation of the ancestral karyotype and based their reconstruction of karyotype evolution on this new reference.

Response:

Yes. We considered B1 as a better representation of the ancestral karyotype.

L403-404: The authors estimate that at least 15-35 chromosomal fusions occurred, including between the A and B subgenomes. This is a strong assumption that requires strong evidence. How many of these putative fusions are telomere-to-telomere fusions?

Response:

We agree with reviewer that those fusions are assumption and need more evidence. We have added a supplementary figure (**Supplementary Fig. 41**) and table (**Supplementary table 25**) to show how many of these putative fusions are present at the contig level. There are 100 fusions (Mi:15, Ma3n:28, Ma4n:22, Mj:35), among them, 50 fusion are present at the contig level. We also investigated the amount of Mi-Tel(+) and Mi-Tel(-) near the fusion site. Only 9 chromosomal fusion events were found with Mi-Tel(+) >5 and Mi-Tel(-) >5. This indicates that not all fusions are telomere-to-telomere fusions.

I understand these fusions are according to Hi-C scaffolding of the genomes which are blurry and noisy due to the complex polyploid nature of the genomes. As further discussed in the rest of my comments, all the biggest scaffolds are not strongly supported by Hi-C contact data. Therefore, the number of deduced fusion events is highly dependent on these unsure yet as good as possible assemblies.

Assessing whether some of these putative fusions are supported by genomic reads is very important and so far this has not been the case for V1 putative telomere-to-telomere fusions (only one supported by one single read). To determine whether most putative fusions are a consequence of the scaffolding it is essential to describe how many of these putative fusions were already present at the contig level.

Response:

We agree with reviewer that those fusions are assumption and need more evidence. In order to check whether chromosomal fusion present at the contig level, we marked the gaps between contigs for each fused chromosome. For example (**Response Fig 4**), Ma3n_Chromosome_03 consists of 4 contigs, involving two chromosome fusion events. Among them, the fusion between C12-C8 is present at the contig level, while C8-C17 are not. The remaining information can be found in the **Supplementary Table 25, Supplementary Fig 41 and Line 414-417**.

Response Fig 4

In their point-by-point response (Response Figure 5), the authors clearly show that the scaffolding of the contigs is responsible for many telomere-to-telomere fusions in assembly V1. Indeed, many contigs that had a high number of Mi-tel repeats at only one side have disappeared during the scaffolding and been fused in scaffolds that have internal Mi-tel repeats. Therefore, I wonder, if at the contig level there is evidence of fusions between the subgenomes, including between A and B or if most of these are generated during the scaffolding. It is important to verify that point because a major claim of the paper is that there are extensive chromosomal fusions.

Response:

Yes, we agree that chromosome fusion is one of the main innovative aspects of this paper and indeed requires more evidence. Information about chromosome fusion, and whether it can be supported in contig level, can be found in the **Supplementary table 25**. Some A-B fusions are present at contig level.

If it turns out that most putative fusions appear after the scaffolding, I would recommend toning down this claim of 'extensive chromosomal fusions'. Instead, I would rather indicate that scaffolding of the genomes suggests multiple chromosomal fusions that will need to be further validated in the future.

Response:

Thanks for suggestion. We agree with reviewers that some fusion (about 50%) are only speculative and need to be further validated. We have added a sentence in discussion (**Line 472-474**).

- Other important yet not major point: Inconsistency between genome assembly sizes and k-mer / flow cytometry estimates

I completely agree with the authors that genome size might vary from one population to another, and this could explain differences between flow cytometry estimates made by Blanc-Matheu et al. on completely different populations of Minc, Mjav and Mare.

However, the authors should discuss the inconsistencies between their own genome assembly sizes and k-mer estimates (made by themselves on the same populations). Indeed, for Minc, k-mer estimates suggest a genome assembly size of $\sim 3 \times 71 \text{ Mb} = 213 \text{ Mb}$ (Table S3, excluding Illumina data which provided inconsistent k-mer estimates). The convergent Nanopore and PacBio k-mer estimates of 213Mb, is exactly consistent with the 213.3 Mb of their Minc assembly V1. However, the 225Mb of their assembly V2 is substantially bigger and no explanation is given for that. Concerning Mjav and Mare3n or Mare4n, same remark: the k-mer estimates are substantially lower than genome assembly sizes and in this case both for V1 and V2 assemblies. Could the authors provide an explanation / hypothesis for this difference?

Response:

The initial k-mer estimation was performed using KMC+GenomeScope2, and the estimated value was $71 \times 3 = 213$. After using Jellyfish (-h = 1,000,000) as recommended by the reviewer, along with GenomeScope2, we obtained an estimation of 73-74Mb ($\times 3 = 219$ -222Mb). Now, the updated genome size estimation is closer to 225Mb. Additionally, assembly v1 used PacBio contigs (213Mb), while assembly v2 used ONT contigs (225Mb). The difference of 10Mb is attributed to using different versions of Canu and different datasets.

The description of the methodology is included in **Method** and **Supplementary Fig. 21**.

From my own experience, Jellyfish + GenomeScope2 can provide underestimation of the genome size if Jellyfish is left with default parameter on genomes that contain many repeats. I suggest the authors to change the option '-h' to 1,000,000 in the jellyfish count command and check whether this changes the k-mer estimates to values more consistent with their genome assemblies.

Response:

Thanks for suggestion. We have added -h 1,000,000 parameter (default is 100,000, which is used for Mg) in kmer analysis of polyploid species, and updated the data in **Supplementary Table 3**.

Sample	Genome Haploid Length(bp)	Model Fit(%)
M. incognita		
Illumina	54,551,869	87.22%
PacBio(canu corrected)	73,055,525	96.79%
Nanopore(canu corrected)	74,207,583	95.06%
M. arenaria 3n		
Illumina	75,056,360	94.64%
PacBio(canu corrected)	74,223,416	95.15%
M. arenaria 4n		
Illumina	74,995,859	96.07%

PacBio(canu corrected)	73,632,344	96.81%
M. javanica		
Illumina	68,282,235	95.12%
PacBio(canu corrected)	73,480,081	96.56%
M. graminicola		
Illumina	41,291,280	96.18%
PacBio(canu corrected)	36,326,612	97.49%

Whatever the results of these new Jellyfish genome size estimates, another observation is very surprising: genome assembly sizes did not substantially change in assembly V2 compared to V1 for Mjav and the two Mare but changed substantially for Minc. What could explain this substantial difference? Is there any additional step that took place in Minc that was not applied to Mjav and the two Mare? The only difference I see is that BioNano data was used for Minc and not for the others.

In their point-by-point response, the authors claim that the Minc genome size at the contig level reached 225Mb with some sequences still partially collapsed. However, In Table S5, the genome size at the contig level is 213.3 Mb. I also downloaded Minc genome V1 and I confirm it is 213.3Mb big. Given that there are only 37.2 kb of N's in Minc genome V2 what could explain the >10Mb difference between assemblies V1 and V2?

Response:

We have two Mi contigs (from ONT data and from PacBio data). PacBio contigs is 213Mb (canu1.9, **Supplementary Table 5**), and ONT contigs is 225Mb (canu2.0, **Supplementary Table 14**). The difference of 10Mb is attributed to using different versions of Canu and different datasets. ONT contigs are used to construct assembly v2 (motioned in **Supplementary Fig 21 and Method Line 1139-1141**). Those two versions of contigs can be found in our website (Mi_Pacbio.fasta.zip and Mi_ONT.fasta.zip).

In Table S5 I would suggest replacing Unscaffolds by Unscaffolded contigs (if my guess this represents contigs that were not scaffolded is correct).

Response:

Thanks for suggestion. We have edited it (**Supplementary Table 5 and 15**).

- OTHER REMARKS / SUGGESTIONS

ABSTRACT

l28-29: I suggest writing 'identified a putative novel telomeric repeat'. Indeed, at this stage, although it is very likely that this repeat is telomeric, whether it actually participates in telomeric functions remains to be determined in the future. And this phrasing would be more consistent with what is written in the introduction.

Response:

Edited as suggested, thanks (**Line 29**).

INTRODUCTION

L60: 'weevil' is not a species but an umbrella name that groups many species. It would be more correct to give the exact name 'Otiorynchus scaber weevils'.

Response:

Edited as suggested, thanks (Line 61).

L94: although crosses between *M. fallax* and *M. chitwoodii* gave F1 hybrids (ref. 25), the egg masses had several critical defects and no functional F2 was obtained, so these hybrids were considered sterile. I think this is important to indicate that although hybridization was carried out successfully, the offspring was viable but sterile...

Response:

Yes, the interspecific hybridization of these two root-knot nematodes did produce sterile offspring, and we cited this paper to show the possibility of interspecific hybridization and polyploidization of root-knot nematodes in nature.

L101-102 "we identified two repetitive sequences that replace the typical telomere sequence and that might act as functional telomeres of polyploid and diploid RKNs which defines the boundary of the chromosome". In fact, this statement poses a problem for the polyploid RKN species (but not for *Mgra*). What seems to be sure and supported by this analysis as well as ref.42 in bioRxiv is that the canonical TTAGGC_n nematode telomeric repeats are absent from the RKN. However, whether the newly identified repeats replace TTAGGC_n is unsure. Although ref.42 in bioRxiv confirms a terminal position of Mi-Tel on most *M. incognita* chromosomes with FISH, the repeat seems to be at only one end (which is consistent with the genome assemblies of ref42 not only in *Minc* but also *Mjav*, *Mare* and *Mluci* at the contig level). Therefore, whether the Mi-tel repeat is present at both ends or mostly at one end remains unsure. Although the authors now provide some preliminary FISH results on *M. incognita* eggs, the analysis is not conclusive enough regarding whether the repeat is mostly at one or two ends (in many cases this can be due to several chromosomes overlapping in the same slide as they are not well separated). Furthermore, in their *M. incognita* assembly V2, only 6 of the 39 chromosomes have the Mi-tel repeat at both ends (and nowhere else) and correctly oriented in the + strand and - strand. A total of 12 chromosomes do not show Mi-tel repeated at any end at all. Consequently, it cannot be assumed that Mi-tel completely replaces the canonical nematode telomeric repeat and defines chromosome boundaries for the polyploid RKN.

As also written by the authors in the discussion, further work would be needed both at the cytogenetic and genome assembly levels to definitely conclude on whether the repeat is mostly at one chromosome end possibly at both in some others.

However, the RKN telomeres are not the main scope of this paper, and many other interesting results are presented and supported by comprehensive analyses. Therefore, to avoid unnecessary further delay in publication, I suggest to simply rephrase this part in a more neutral way which would avoid any over-interpretation. Something like: 'canonical nematode telomeric repeats were not found in these RKN genomes, instead we identified two new types of repeats, respectively in the diploid and polyploid species enriched at the extremities of contigs / scaffolds. Recent cytogenetics analyses have confirmed a terminal position of the repeat on *Minc* chromosomes (ref.42) suggesting it is a telomeric repeat'

Response:

We agree with the reviewer's suggestion that further work is needed to validate these findings. Therefore, in the paper, we only describe that the telomeres are located in the terminal position without emphasizing whether they are located at both ends. We have used a more neutral phrasing to describe the conclusion of both our study and bioRxiv (**Line 102-107**).

L107-108 "Unzipping the genome into several subgenomes is crucial " I suggest "...for a comprehensive and realistic representation of polyploid RKNs."

Response:

Edited as suggested, thanks(**Line 111-112**).

RESULTS

- GC%

It would be interesting to provide GC% for the different assemblies V1 as well as V2 and briefly discuss whether they differ from previous genomes for the same species or are similar.

Response:

We have added GC% information in **Supplementary Table 5 and 15**.

In this study, the GC content ranged from 30.1% to 30.3%, which is similar to the previous genomes in Blanc-Mathieu et al. 2017 (29.8%-30.0%).

- Exploration of the genome structure of Clade I RKNs

L132-133: the results even suggest that Ma3n and Ma4n are two different species that cannot be distinguished with SCAR markers but are clearly different based on their mitochondrial genomes.

Response:

Edited as suggested, thanks (**Line 136-138**).

L133 134: 'we propose naming the triploid Ma sample as Ma3n and the tetraploid sample as Ma4n in the rest of our study.

Response:

Edited as suggested, thanks (**Line 139-140**).

L137-138 the genome size of Mgram estimated by k-mers is 36.3 Mb with PacBio data and 41.3 Mb with Illumina data. (Table S3). Therefore, the assembly size is consistent with the higher estimates.

Response:

Edited as suggested, thanks (**Line 145**).

L141: I understand here that you confirm the genome size at the contig level for Minc is 213 Mb and not 225?

Response:

The k-mer analysis were updated (**Supplementary Table 3**).

L146: I suggest 'all the subgenomes' rather than all the haplotypes.

Response:

Edited as suggested, thanks (**Line 153**).

L151: I suggest 'This draft preliminary assembly was used to elucidate...'

Response:

Edited as suggested, thanks (**Line 158**).

L161: '... suggesting that assemblies version 1 were mostly complete with a low degree of collapsed subgenomes.'

Response:

Edited as suggested, thanks (**Line 168-169**).

- Identification of telomeric repeats for Meloidogyne

I suggest changing the title of this section by 'Identification of putative novel telomeric repeats in Meloidogyne genomes'.

Response:

Edited as suggested, thanks (**Line 209**).

L204 -206: to facilitate comprehension, I would indicate here 'based on n=18 as ancestral number of chromosomes.' And I would also indicate that chromosome fusions is not the sole scenario and whole chromosome losses are also possible.

Response:

Edited as suggested, thanks (**Line 211-213**).

Lines 248-250 "248 Fig. 18). Moreover, we noted that another study on Meloidogyne telomeres, utilizing fluorescence in situ hybridization (FISH), reveals that Mi-Tel is present at the ends of the chromosomes (ref. 42)"

This is not completely exact, ref. 42 concludes that Mi-Tel is present at one end of most of the chromosomes and not 'at the ends'. To avoid unnecessary controversy or debates, I would suggest using a more neutral phrasing like 'reveals that Mi-Tel repeats have a terminal position on M. incognita chromosomes'

Response:

Thanks for suggestion, we have used a more neutral phrasing to describe the conclusion of both our study and that ref (**Line 241-245**).

L255 and in M. luci as well (+ cite the reference for the genome sequence here).

Response:

Reference is now included, thanks (**Line 259**).

L261-262 some were specifically present in Minc but absent from the rest.

Response:

Edited as suggested, thanks (**Line 266**).

- Construction of chromosome-level genomes for polyploid RKNs

L274: I suggest 'based on the information gained from assemblies V1 concerning the duplicated genome structures and the putative telomeric repeats, we attempted to construct.... Also indicate somewhere that for Minc, Nanopore data was used to assemble the contigs in contrast to PacBio for all the other ones.

Response:

Edited as suggested, thanks (Line 278-279).

L277: '36-52 chromosomes' does not match with '36-47 scaffolds' in Figure S21 but matches numbers given in Table S15. Please clarify which numbers are correct.

Response:

Thanks for carefully check. 36-52 is correct, and we have edited it.

L283: I suggest 'in contrast to assemblies V1, this assemblies V2 were not guided by synteny relationships with the Mg T2T2 genome but only Hi-C and / or optical map data and were utilized for all subsequent analyzes'.

Response:

Edited as suggested, thanks (Line 283-285).

Figure S22: I completely understand the authors have done their best to scaffold the Minc genome based on Hi-C and Optical map data and this is not an easy task at all owing to the complexity of the genome structure and parasite contact noise caused by the copies as well as problematic collapsed regions. However, it should still be noted that the seven biggest Minc scaffolds show blurry contact data. For example, some contigs in scaffold 1 show strong contact data with other contigs in scaffold 2. Overall, all the biggest scaffolds are showing blurry / noisy contact signals. Unfortunately, I do not think anything can be done at that stage to improve the situation. However, this still casts some doubt on the reality of the biggest chromosomes.

In their point-by-point response, the authors indicate they have measured chromosomes up to 2.5µm in length in Minc. Considering chromosomes in *C. elegans* measure ca. 5 µm for 15 – 20 Mb, the biggest Minc chromosomes should be 8 – 10 Mb big. The biggest Minc scaffolds are bigger, and none are unequivocally supported by Hi-C data. However, to avoid long debate, I rather suggest the author to indicate (maybe in the discussion?) that this current representation of the genome remains hypothetical and might change in the future with technological progress. Interestingly, the Hi-C signals seem to be more uniform and less blurry in the other Meloidogyne assemblies V2 (Ma3n, Ma4n and Mj).

Response:

We thank the reviewer for their understanding our efforts in genome assembly. We agree with the reviewer's comments that the assembly of some chromosomes is still hypothetical and has not been one-to-one corresponded with the chromosome number.

We have added a sentence to the discussion section to mention this point (Line 472-474).

Figure S31. Please indicate that C1 – C18 are given according to synteny with Minc B1 genome (considered as representative of the haploid ancestral state if I understand correctly).

Response:

Added it into figure legend, thanks (**Supplementary Fig. 31**).

- Origin and evolution of polyploid RKN

Figure S34: I congratulate the authors for having re-done the whole phylogenomics analysis according to the non-manually edited assembly. One suggestion to make the figure easier to understand for the readers: I would rename Trees 1_18 to Trees C1_C18. This way, it will be more straightforward to understand the 18 trees come from concatenated single-copy genes from the 18 groups of synteny and 14 subgenomes of Figure S31 (If I understand correctly).

Response:

Yes. We have updated this figure as suggested, thanks (**Supplementary Fig. 34**).

Line 339: I rather propose '17/18 trees exhibited....' Indeed, tree#17 is the only one not showing a clear separation between the A and B subgenomes.

Response:

Edited as suggested, thanks (**Line 343**).

The other major separation that can be observed is within the A subgenomes, in which A1-A2 from Minc hold an outgroup position relative to the rest of the A genomes from the other species in 15/18 trees (3 trees C1, C2 and C5 show Minc A1-A2 in the middle). This result suggests the A subgenomes of Minc have a different evolutionary history from the A subgenomes of the other species.

Then, within the A lineage, the majority signal is a separation between the A1 and A2 subgenomes from Ma3n, Ma4n and Mj. If A1 and A2 have actually been attributed randomly to the subgenomes this A1-A2 separation has very low probability to be observed. I rather understand that the authors have re-attributed the A1 and A2 labels to the subgenomes according to the tree topologies? Is that the case? If yes it should be indicated in the results and / or methods.

Response:

Yes, A1 and A2 are initially divided randomly, for easier understanding and observation, we re-attributed the A1 and A2 labels.

This information has been added in methods (**Line 1171-1173**).

Furthermore, the fact that the A1 subgenomes are clearly separated from the A2 subgenomes in most trees for Ma3n, Ma4n and Mj suggests these three species have a common origin for their A subgenomes and this topology is completely consistent with the hypothesis of the authors (FigS35 b 1-2 and c 1-2). Therefore, given these new results, I do not find that the 'topology of the A lineage did not match (L347) their hypothesis. In contrast, this perfectly matches.

The only thing that should be explained (in my opinion) is that the parent that provided A subgenomes in Minc is different from the parent that provided the A subgenomes in the other species. Otherwise, the rest of the scenario is completely valid (clonal diploid with high heterozygosity (1) or hybrid diploid (2) as donors of the A genomes.

Response:

Yes, we agree. We added a sentence to indicate the A subgenomes of Mi have a different evolutionary history from the A subgenomes of the other species, and also edited the sentence of previous L347 (**Line 352-356**).

L369-370 "suggesting that the maternal genomes of these species diverged not long ago and that

hybridization occurred recently." I would add "which is exactly consistent with nuclear and mitochondrial phylogenomics analyses made on previous more fragmented versions of these genomes (Blanc-Mathieu et al. 2017).

Response:

Edited as suggested, thanks (**Line 379-381**).

- Gene expression landscape after polyploidization of Clade I RKNs

These results are very interesting and convincing both regarding the higher divergence between subgenomes for the secreted genes vs. the rest of the genes and also for the identification of gene expression decrease for one subgenome. These two elements indeed suggest that allo-polyploidization has favored functional divergence between the gene copies as also previously shown in Blanc-Mathieu et al. 2017 at the gene expression pattern level.

Response:

Thanks for comment.

- DISCUSSION

L459: It is important to note here that the palindromes initially identified in the genome of *Adineta vaga* in Flot et al. Nature 2013 have not been later confirmed by their new assembly of the genome based on long reads (Simion P. et al. Science Advances 2021). After discussion with the authors of these papers, they confirmed that the palindromes initially identified were most probably due to assembly errors in the first version of the genome. Concerning the *Meloidogyne*, it was not indicated whether the palindromes identified were the result of chromosome fusions between the subgenomes or independent segmental duplications. Therefore, I would not necessarily make such a statement.

Response:

Thanks for the reviewer's feedback. We indeed did not investigate the specific situation of palindromes in root-knot nematodes, so we decided to remove this sentence.

L477: "...but only polyploid species exhibited extensive chromosome fusion." If we consider these chromosome fusions as true (but see my previous comments on the lack of strong

evidence), the explanation might lie in the reproductive mode. Indeed, while the diploid species do meiosis and must maintain a number of chromosomes that can be divided in two, this is not the case for the polyploid mitotic species. Therefore, big chromosomal rearrangements might not be counter-selected in the mitotic species while they are highly deleterious in the meiotic diploid species and thus counter-selected.

Response:

Yes, we agree. We believe that the mode of reproduction and the holocentromere both facilitate the occurrence of chromosomal fusion. We have added this into discussion (**Line 497-498**).

L485: I suggest 'Moreover, the genome assemblies we have generated...'

Response:

Edited as suggested, thanks (**Line 500**).

L504: please indicate 'Otiorynchus scaber weevils'

Response:

Edited as suggested, thanks (**Line 519**).

L512-516. I found very interesting the hypothesis that *M. floridensis* has an AAA triploid genome. However, I do not exactly understand how this conclusion was drawn and based on which results. In Figure S7a, on the left, *M. floridensis* is proposed to be triploid by SmudgePlot but AAB. How did the authors reach the conclusion that it is in fact AAA? Could it be in fact AA with AA similar to the A genomes of *Minc* and a completely different B genome not closely related to any B genomes of the other species studied here?

Response:

Firstly, 86% of the reads can be aligned to the A genome, indicating that it could be AAA or AA. Then, by analyzing the read depth of alleles (if AA then peak at 1/2, if AAA then peak at 1/3), we can infer that it is most likely AAA (**Line 386-390**). At least, we can confirm that *Mf* should be triploid. The reviewer might propose another possibility, that *Mf* has a genome structure like AAC. The size of the *Mf* genome assembly is approximately 70Mb, which is equivalent to a haploid genome size in CladeI root-knot nematodes. This suggests that during the assembly process, due to the high similarity between the three copies, they collapsed into one copy. If the third copy differs significantly from A and B, even based on illumina data, it is unlikely to collapse into one copy. Based on the current data and overall considerations, the possibility of AAA is the highest. We are also interested in *Mf*, but due to the lack of materials, we are unable to conduct in-depth research. We look forward to other groups providing long-read data to thoroughly address this issue.

- METHODS

Lines 950-953: Short reads and corrected long reads were subjected to genomescope2.0 and Smudgeplot31 to estimate genome size and ploidy with recommended parameters.

Which software was used to enumerate k-mers? Was it JellyFish or KMC ? And which parameters were used? This information is important given the inconsistency between genome size estimations with kmers and genome assembly size in Minc.

Response:

The detailed information about kmer analysis (JellyFish+ genomescope2.0) is now included (Line 953-954).

Line 1155 "Subgenome identification of the telomere-to-telomere genome". To be consistent with the rest of the manuscript and since these genomes are not called "T2T" anymore, this title should be changed to 'chromosome level' rather than T2T.

Response:

Edited it, thanks!

- DATA AVAILABILITY:

The authors have now released their annotated genome assemblies V2 in the NCBI, which will greatly facilitate re-use and further analysis!

I think Assemblies T2T should be removed from the laboratory website download page to avoid any confusion with assemblies V2.

On the laboratory website data download page, what is Mh_assembly_V1 ? Is it *Meloidogyne* hapla?

Response:

We have removed T2T version in our website.

Mh_assembly_V1 is a preliminary assembly version of *Meloidogyne hapla*, we will construct the chromosome level genome in next project. Thanks for your interest in our work.

- ACKNOWLEDGEMENTS

lines 1387-1389: since the FISH results have not been included neither in the supplementary material nor in the results (because these are too preliminary and not conclusive enough to be included), I am afraid they cannot be referred to in the acknowledgement section.

Response:

Our FISH results are only used to illustrate that Mi-Tel has a terminal position on *M. incognita* chromosome, and our results can clearly show that most of these Mi-Tel signals are located at the end of the chromosome, so we put this result in the Supplementary figures and update the Methods (Line 241-244, Line 1249-1269 and Supplementary Fig. 16).

Reviewer #3 (Remarks to the Author):

This manuscript provides solid convincing data that the diploid Mg genome lacks telomerase reverse transcriptase and maintains its telomeres via the ALT pathway. Moreover, the diploid Mg genome represents a reflection of an ancestral state to related polyploid *Meloidogyne* species, as both diploid and polyloid M species use related ALT telomere maintenance elements. It is unusual to find head to head telomere fusions in

nature, so it is intriguing that these might be characteristic of polyploidization, given that the Mg diploid genome lacks telomere fusions and maintains its telomeres via ALT. The Mg genome appears to be high quality with almost all telomeres accounted for (TtoT). Moreover, genomes of several related polyploid nematodes are created, and telomeres for many chromosomes of polyploid chromosomes are defined. The authors indicate that telomeres of the polyploid species can be sorted into 'telomere clusters' and that some telomere clusters are unique to specific species whereas other telomere clusters are shared between polyploid species that were derived from a common diploid ancestor that lacked telomerase. These are interesting observations regarding the heterogeneity, inheritance and evolution of telomere sequences of ALT telomeres, although the authors describe telomere clusters in vague terms without clearly defining how they differ in sequence or length.

A number of HGT events are documented for polyploids (hundreds), but analysis of these sequences is not presented in detail. Remarkably, the authors observe consistent clusters of HGT events at specific segments of some chromosomes of polyploid species. This might provide insight into the mechanism of HGT, but little information is provided about genes inserted at HGT clusters, which might be related to sites of transposon insertion or to specific epigenomic states that some chromosome domains may adopt. It is possible that the authors' Hi-C data might reveal features of HGT hotspots.

The synteny analysis is impressive and helps to clarify how polyploid genomes evolved and are structured, reflecting 18 ancestral chromosomes. Clade V species contain 6 chromosomes, and it has been speculated that 18 to 22 ancestral chromosomes may have been present in very distantly related species, reflecting conservation of gene order for more than 500 million years of evolution. It would be interesting if the authors could show that the 18 chromosomes of Mg correspond to the highly ancestral state proposed by Simakov 2022. They may not, because the Mg genome lacks canonical telomeres and maintains telomeres via ALT, and ALT is almost always activated when canonical telomeres become very short and a number of telomere fusions occur.

The authors show that transposon levels are increased in polyploid strains but speculate that this increase occurred in diploid genomes prior to polyploidization. However, I suggest that the authors might be able to create a more general model, where a transposition burst might occur after polyploidization, which might lead to chromosome instability that drives chromosome fusion in a manner that helps to shape polyploid genomes. HGT events might be related to or integrated with transposon activation. The ALT telomeres might be related to some fusion events, but the significance of ALT telomere fusions in the context of polyploidization will only become apparent when genomes of polyploid nematodes telomerase-positive are characterized. Overall, this is an interesting and generally well written manuscript that is appropriate for Nature Communications.

Response:

We appreciate the reviewer's positive feedback and the professional suggestions regarding ALT telomeres, transposons, and horizontal gene transfer (HGT). In particular, the insights

on ALT telomeres are profound. It is important to note that our intention in including these aspects in the paper was to provide a more comprehensive perspective rather than making significant breakthroughs in these specific fields. Our main focus has been on the genomic structure, chromosome fusion, and origins of polyploid root-knot nematodes. Therefore, the analysis of these aspects was relatively simple. Many of the mentioned questions are intriguing, and we believe they are worth pursuing through independent research.

Comments:

1. It would be helpful to show the Mg-Tel repeat sequence in Fig. 2. It has little similarity to the TTAGGC canonical repeat sequence. ALT typically involves amplification of canonical telomere repeat sequences, and the authors could point out that there is no example of the TTAGGC sequence in Mg-Tel. However, there are many examples of TTA and some examples of GGC: these could be highlighted and queried to determine if they are present at a frequency that is more likely than would be expected by chance. If not, then the Mg-Tel sequence bears essentially no similarity to the ancestral TTAGGC telomere repeat, despite the G-rich nature of Mg-Tel on the strand that runs 5' to 3' towards the end of the chromosome.

Response:

Thanks for suggestion. We have marked TTA and GGC in Mg-Tel sequence (**Supplementary Table 12**).

2. 'As expected the read contained 71 Mi-Tel elements and 93 reversed complementary Mi-Tel elements.' This represents much telomeric DNA at the site of the telomere fusion. How does the element number compare with the number of Mi-Tel or Mg-Tel elements at natural telomeres? Is there evidence for telomere shortening prior to fusion? If not, what might be the cause of the telomere fusion event? Evidence for this might be apparent based on the sequence at the site of telomere fusion: it would be of interest if the authors could show this predicted sequence.

Response:

We counted the copy number of Mi-Tel in chromosome ends (mean = 40, median = 21). It is hard to say that telomere-telomere fusions are caused by a reduction in telomere copy numbers. We thank the reviewers for raising many questions and hypotheses. However, our current understanding of telomeres in root-knot nematodes is quite limited. We just identified a sequence that specifically locates at the chromosome end, which may function as an alternative telomere. Many questions still need to be confirmed, such as the start and end of the telomere repeat unit and so on. Thus, many hypotheses are hard to address in this paper in a few sentences, and these works deserve to be addressed in the future using more accurate sequencing techniques, such as HiFi data.

3. 'We extracted 9318 telomeric elements from 4 polyploid species which exhibited differences in length". The length in Fig S20 varies from 300 bp to 1000 bp. Does this mean that the length of the repeat unit varies significantly at telomeres, even in the same species? Is the sequence similar in all species? To understand the similarity, it might be helpful to show the sequence of one telomere repeat unit per species.

Response:

We thank reviewers for their interest in this section. In fact, these repetitive sequences of telomeres are highly complex and may contain multiple variants. Through multiple sequence alignments, we have preliminarily identified a 66bp motif that is relatively conserved. In order to describe the features of complex telomere sequences, we define the sequence between two adjacent 66bp motifs as a telomeric element. Indeed, even within the same species, these telomeric repeats exhibit significant variations in both length and sequence. The only thing that can be confirmed is that this 66bp motif is indeed located at the chromosome ends (based on FISH). To be frank, this part of the analysis is still relatively rudimentary. This result can only indicate that these sequences are highly complex, but the variation patterns between them are not clear. Below is the result of multiple sequence alignment after removing redundant sequences using CD-HIT. We still believe that based on the current genomic assembly data and the understanding of root-know nematode telomeres, the question about the evolution of repeat sequences is not sufficiently resolved.

In revised version, we added a sentence in discussion to describe that these complex telomere sequences deserve further study (**Line 483-485**).

4. Note that even for human ALT cancers, the sequences at telomeres of specific cell lines may not be the same and are not well understood. A recent PLOS Genetics paper from the Baird group demonstrates that telomeres of cells that become ALT can display related sequences at their telomeres, but that the telomere repeat sequence is quite heterogeneous, perhaps reflecting the sequence heterogeneity of telomeres near the telomere-subtelomere boundary of one short chromosome end that is used to create a template that is used for ALT: <https://www.ncbi.nlm.nih.gov/pmc/articles/PMC9678338/>

Response:

We thank reviewers for their comments and the recommended literature. Indeed, the length and sequence heterogeneity of ALT telomeres are significant, and even in cancer cells, they are not well understood. Furthermore, investigating the patterns of ALT telomere sequences still requires a amount of work, which is beyond the scope of our research. We hope for your understanding regarding the limitations of our study in terms of telomere sequence features.

5. 'Some telomere clusters were found in each species whereas some were specific to specific species'. The term cluster is not well defined. What distinguishes the different telomere clusters? Are these different types of telomere sequence repeats? What are the features of telomere clusters that are only found in specific species? Are they characterized by a specific sequence or a specific length? What are the characteristics of the shared telomere clusters? It would be helpful to see one sequence example of each of the 19 telomere clusters.

Response:

Thank you for raising the questions. To characterize the sequence variations in telomere sequences, we used CD-HIT (-aS 0.8 -g 1 -sc 1 -sf 1 -c 0.9) to cluster 9,318 telomere sequences into clusters. If the sequence similarity is greater than 90% and the coverage is greater than 80%, they will be clustered together. There are many clusters, indicating that there are significant differences among these sequences. The purpose of this analysis was solely to demonstrate the significant heterogeneity in telomere sequences. As for a more detailed analysis, this is probably beyond the scope of this paper and may require follow-up independent research. Another reason that limited our in-depth investigation is that telomere sequences were assembled using ONT or PacBio reads, which may lack the accuracy of those complex and repetitive telomere sequences. In the future, new genome assemblies based on HiFi sequencing data may be more suitable for further analysis.

For more clarity, we mention that these clusters are defined based on sequence similarity in figure legend (**Line 1576-1577**).

6. It appears that only species that lacked telomerase and used ALT for telomere maintenance created these polyploids. One possibility is that the ALT telomere sequence repeat evolved in diploid nematodes that gave rise to AAB or AABB or AAA polyploid species, and that the telomere clusters (telomere sequence repeats?) that are shared between polyploid species reflects their ancestral diploid telomeres. If this is the case, then the polyploid chromosomes derived from A1 or A2 or B diploid genomes may have share

common telomere sequences (clusters) that reflect their ancestral diploid ALT telomere sequence repeats. In this case, the B chromosomes would possess common telomere sequences (telomere clusters?) that less commonly occur on A1 or A2 chromosomes, at least for chromosomes that have not fused.

Response:

Polyploid root-knot nematodes are formed by hybridization of A or B genomes. We agree that ALT telomeres likely underwent evolution in the A or B ancestral stages. In theory, there could exist A- or B-specific telomere repeats. In fact, we had previously hypothesized this and investigated the proportion of telomeres from the A or B subgenome in each cluster. However, we did not find any specific telomere repeats associated with A or B subgenome. We speculate that it is possible that telomeres from both A and B genomes have been mixed through homologous recombination (HR). Again, these studies on telomere origins are better suited for exploration in the form of independent papers (also based on HiFi data).

7. What fraction of the extensive chromosome fusions in Mi are due to telomere fusions?

Response:

Among the 100 putative chromosome fusion events, only 9 fusion events were detected with Mi-Tel(+) ≥ 5 and Mi-Tel(-) ≥ 5 . This indicates that most fusion sites lack telomere repeats in their vicinity. As the reviewer mentioned, most fusion events are likely not telomere fusions but rather caused by double-strand breaks in non-telomeric regions of two chromosomes.

We have now added a supplementary table to show that information (**Supplementary table 25**).

8. What is the nature of the horizontally transferred genes at the hotspots for HGT? This might provide insight into a mechanism of horizontal gene transfer. Are these derivatives of the same gene family that have duplicated and diverged? How do the genes present in hot spots compare with those that are stochastically distributed? How many HGT genes come from plants or soil bacteria or other nematodes? It is possible that a fraction of HGT events reflect transposons that moved horizontally between species, but it is not clear if transposons are in the HGT category or if they are binned in a separate genomic parasite category.

Response:

We appreciate the reviewers' interest in horizontal gene transfer (HGT) and their feedback. We have now provided functional annotations for these HGT genes. It is important to note that these HGT events are only hypothetical, and further investigation would require additional work, such as constructing phylogenetic trees and examining the presence of transposons around HGT genes. However, the aim of this paper was to address the structure and origin of those polyploid root-knot nematode genome. We simply aim to examine the distribution of HGT on the chromosomes, and delving deeper into the study of HGT lies beyond the scope of our research objectives. A comprehensive analysis would require a significant amount of work and space. Therefore, these findings may be better

suiting for a separate paper, focusing on the origin, functionality, and correlation with transposons of HGT.

In revised version, we have added a supplementary table to describe the gene name and predicted function of those HGT candidates (**Supplementary Table 19**)

9. It is terrific that the authors were able to define the chromosome number and features of synteny between related chromosomes in polyploid species, and that phylogenetic models were created for each chromosome.

Response:

We thank reviewers for the comments.

10. What does it mean that 2578 of 3251 LTR retrotransposons had 100% LTR identity? Does this mean that all 2578 had the same LTR sequence? If so, this defines a single highly active LTR retrotransposon. How many types of full length LTR transposons are defined by the remaining ~850 transposons, if LTR sequence identity defines transposon type?

Response:

Long Terminal Repeat Retrotransposons (LTR-RTs) are transposable elements characterized by having two long terminal repeats (LTRs, black arrows in the diagram) flanking an internal coding region. The newly inserted LTR-RT has completely identical LTR sequences on both sides, and over time, they diverge from each other. Here, we are referring to each of the 2,578 LTR-RTs, where the sequences on both sides are identical, indicating that they have been recently inserted. In other words, the LTR sequences on both sides of LTR-RT_1 is identical, but they may differ from the LTR sequences of LTR-RT_2. We have edited this sentence to avoid ambiguity (**Line 398-400**).

11. 'We did not observe a burst of many TEs after hybridization (Fig. S39a)'. I respectfully disagree. Compared to the diploid Mg where 0.5% of the genome is TEs, the polyploid species have 2.4-2.8% of their genomes as TEs. This could suggest a consistent dramatic burst of transposition upon polyploidization or that there is a TE burst prior to polyploidization. It is possible that endogenous transposons rather than those introduced by HGT are responsible for a TE burst.

Response:

We agree that polyploid species must have undergone transposon expansions, and these expansions may have occurred after hybridization or in the diploid ancestors. A key indicator is the high similarity of mitochondrial genomes in these species, indicating that hybridization is a very recent event (**Line 374-381**). If the TE burst occurred after the recent hybridization, the peak of transposons should be close to position 0; however, since

the peak is located in the middle (10-20), it indicates that the majority of transposons are not young and likely underwent a TE burst in the diploid ancestors (**Supplementary Fig 39a**).

12. If a TE burst occurs upon each round of polyploidization, why would there be A or B-specific transposons? The transposons that are active and moved upon polyploidization would move again upon AA and then again when AAB or AAA ploidy was created. Inactive TEs that cannot move non-autonomously or solo LTRs might be specific to A or B chromosomes.

Response:

Yes, exactly. We just wanted to investigate whether there are A- or B-specific TE families, as described in *X. laevis*. This conclusion is indeed not sufficient to infer a transposon burst, and we have rearranged the sentence order accordingly (**Line 406-408**).

13. Fig. S39a could be better explained. Why are the Kimura substitution levels for transposons so much higher in polyploids than in Mg?

Response:

The TE content in polyploids is higher than in Mg. In fact, we have mentioned this earlier in the text, and previous studies have also pointed this out, so we did not repeat it here. What we are investigating here is whether the transposon expansion in polyploids occurred before or after hybridization. Given that hybridization is a very recent event, and the majority of transposon peaks are not close to 0, we believe that the transposon burst likely occurred in the diploid ancestor stage.

14. If most chromosome fusion events occurred post-hybridization, and if a TE burst occurred post-hybridization, then the TE burst might generate chromosome instability (DSBs from DNA transposon excision) that might help to explain some chromosome fusions.

Response:

We believe that the majority of transposons underwent a burst in the diploid ancestor stage, and therefore, we did not discuss the contribution of transposon bursts to genome stability. However, we do not rule out the possibility that some transposons may indeed affect genome instability. If some transposons are still highly active in the Mi genome, the transposon insertion polymorphism in the Mi population might be significant. We will follow up on this hypothesis using genomic data from the Mi population, which may be our next research project.

15. 'ALT telomeres have a weak chromosome protective function, which seems to explain why chromosome fusion occurs.' Some telomere instability may be a feature of ALT tumors (rapid telomere shortening events), but the point to ALT is to maintain stable telomeres that are essential for tumor genome survival. The authors may be referring to work on ALT cancer cells. However, during activation of ALT in cancer, telomeres shorten due to lack of telomerase, and telomeres become very short, and chromosome fusions always occur prior to activation of the ALT mechanism. All ALT positive tumors possess telomere fusions, and

it is formally possible that the process of telomere fusion may be mechanistically linked to activation of ALT. The authors data may instead suggest that an aspect of polyploidization drives chromosome fusion. This might be apparent if genomes of telomerase-positive polyploid nematodes have been characterized.

Response:

We appreciate the reviewer for providing us with more professional and insightful perspectives on the ALT telomeres. We agree that even in ALT species, chromosomes are still stable, such as Mg in this paper. We indeed lean towards polyploidization being the main cause of chromosome fusion. Currently, it can be confirmed that all root-knot nematodes are telomerase-negative, while the closest relative, the cyst nematode, is telomerase-positive. The comparison between these two species may help reveal some characteristics of ALT.

We have further tone down this statement (weak protective function) in the discussion (**Line 488-490**).

16. There is a caveat to the logic that the Mg diploid genome lacks telomere fusions but that polyploid *Meloidogyne* genomes possess chromosome fusions, some of which may be telomere fusions. The caveat is that it is virtually certain that the diploid Mg chromosomes must contain telomere fusions. This is because a diploid ancestor of Mg possessed telomerase and canonical (TTAGGC)_n telomeres, but then telomerase was lost, followed by critical telomere shortening, creation of telomere fusions, and activation of the ALT telomere maintenance mechanism: there are many examples of this in cancer genome evolution. So, there might be short (TTAGGC)_n telomere tracts (100 to 300 bp long) present at the sites of telomere fusion in the Mg genome, but these telomere fusion events might be very old and no longer visible. In contrast, the telomere fusions evident in polyploid *Meloidogyne* species occurred after activation of ALT telomere maintenance, and may reflect relatively recently created telomere fusions. If the polyploid *Meloidogyne* chromosome fusions commonly involve ALT telomeres, my guess is that direct telomere-telomere fusions would be rare, and that if a telomere is involved in a chromosome fusion event, then there would be a single telomere present at the site of chromosome fusion, and that the other site of fusion would be a DNA double-strand break created in a distinct chromosome, which might be some distance from a telomere. This is because the function of telomeres is to promote chromosome end protection, and even ALT cancer cells display stable chromosome karyotypes because ALT telomeres are good at telomere end protection. However, it is also possible that polyploidization promotes fusion of long telomeres of chromosomes in species where ALT is used for telomere maintenance. In this case, a few or many fused chromosomes might contain two head to head telomeres at their fusion breakpoints.

Response:

We thank the reviewers for their insights into ALT telomere and chromosomal fusion.

We agree with the reviewers that those root-knot nematodes underwent ancient chromosomal fusion at the ancestral stage. Our additional results indicate that telomere-to-telomere fusions are indeed rare, as most fusion sites show absent of Mi-Tel (**Supplementary Table 25**). It appears that most fusions result from double-strand

breaks in non-telomeric regions of chromosomes, further indicating the role of ALT telomeres in the protection of chromosome ends.

17. 'Although we observed specific distribution patterns in polyploid nematodes, such as histone modification distribution, HGT hotspots and increased TE content, it is not clear whether these changes are caused by lifestyle changes, polyploidization, parthenogenesis or ALT telomeres.' Why would ALT telomeres cause HGT hotspots or increased TE content, especially if ALT were present in the ancestral diploids?

Response:

We speculate that ALT might be related to the distribution of histone modifications, while TE content might be associated with polyploidy and parthenogenesis. We have revised this sentence (**Line 485-486**). Thanks for suggestion.

18. The authors may be the first to discover HGT hotspots in nematode chromosomes. Is there evidence for such hotspots outside of nematodes? This is quite interesting and could be relevant to understanding the basis of HGT. It would appear that elements of chromosome instability, like chromosome fusion, increased transposon content, HGT hotspots might be coupled to polyploidization events, and in this context some or all of these chromosome instability elements might be coupled.

Response:

We appreciate the reviewer's interest in HGT. In fact, this part of the work refers to this paper (Simion, Paul et al. Science advances). In rotifer *Adineta vaga*, Simion et al. also reported those HGT hotspots. We believe that the study of HGT functions, origins, and evolution in these root-knot nematodes warrants a separate independent paper to be thoroughly investigated.

Reviewers' Comments:

Reviewer #1:

Remarks to the Author:

Following my previous evaluation of their paper, the authors have considerably modified this new revised version and I would like to congratulate them for that and all the additional work that appears in the point-by-point response.

The authors now propose a non-manually edited version of their *Meloidogyne* genomes V2. These new versions are not edited to add telomeres at both ends of the scaffolds and the collapsed regions have not been artificially un-collapsed by producing two duplicated 100% identical subgenomes.

I really think at this stage of our knowledge of the karyotype and chromosomes of these *Meloidogyne* species, these are much more reasonable representations of the polyploid RKN genomes.

Indeed, the exact number of chromosomes, as well as how many have Mi-tel repeats at one end, both ends, in the middle or not at all in the polyploid RKN remains uncertain. The preliminary FISH analyses provided by the authors do not allow to precisely determine the number of chromosome and it is difficult to tell whether some (and how many) chromosomes have Mi-tel at both ends because many chromosomes overlap in the FISH images and are not well separated. Therefore, there are too many unknown characteristics at this stage to guide genome assembly and scaffolding with strong a priori assumptions and these new unguided genomes V2 are more representative of the state of our knowledge.

I completely agree with the authors that the number of chromosomes in these species is likely to vary from one population to another, and possibly involving some chromosomal fusions / fissions. These karyotypic variations might indeed explain part of the differences in genome assemblies and FISH observations between this paper and the one posted in bioRxiv by Zotta-Mota et al on other populations of the same species.

Overall, I now consider this major point has been addressed by the authors.

I have only one other major point concerning the evidence supporting 'extensive' chromosomal fusions that, in my opinion, needs to be addressed.

The rest are more minor yet important suggestions that, I hope, will help clarify further the paper, which is now of high quality and comprehensive.

- MAJOR POINT: evidence for extensive chromosomal fusions

The whole sections on chromosomal fusions are very interesting as it might explain several idiosyncrasies in the assembly. However, currently some explanations are missing to help the reader understand exactly the data supporting (or not) these putative fusions.

For assemblies V1, it is very important to explain clearly that fusions are predicted between McScanX linkage groups syntenic to Mgram and considered ancestral chromosome parts. Indeed, it took me a long time to understand the nomenclature of scaffolds Mi_ChrXX_XX in Figures 2f as well as Figures S16 and S17. Only TableS10 allowed me to fully understand that this nomenclature was according to synteny with the 18 Mgram chromosomes. Providing a supplementary figure similar to FigS31 but with assemblies V1 and Mg chromosomes like in previous versions of the manuscript might help the readers fully understand.

Looking into more details at Figure S16, I realized that two of the four putative fusions of ancestral chromosomes involved V1 scaffolds present in two copies rather than the three expected, namely in (1) Mi_Chr9_24 and in (3) Mi_Chr6_17. For Mi_Chr6_17 this might represent a case of collapse between A1 and A2. Have these contigs higher coverage in Fig S7 for instance? If that is the case, then there is an alternative hypothesis. Indeed, this might represent collapse between one subgenome having Mi-tel repeats and the other one not having Mi-tel. In that case, the collapsed consensus will have Mi-tel but it is still possible that the true contact information is with the other subgenome not having Mi-tel. The alternative version without Mi-tel being absent in the assembly because of the

collapse these contact information cannot be represented. Therefore, for these two putative fusions events, it cannot be excluded that this results from assembly collapse and artefact of Hi-C and BioNano. Indeed, if A1 and A2 are very similar, BioNano Optical map will produce exactly the same barcode for the two molecules and also fuse them as a single molecule with no possibility to make any difference between the two.

Ideally, to disentangle these cases of putative fusions, it would be interesting to provide a similar figure than S16a but including Hi-C data for all the homoeologous copies. For instance, include MiChr4_10 and 4_12 as well as 9_25 in addition to 4_11 and 9_24 for (1) and include 12_31, 12_33 as well as 6_16 for (3). It is very important to show whether contact data support alternative linkage information as strongly or not than the one proposed by the authors to support their telomere-to-telomere fusions.

Furthermore, in FigureS16 b(4), Mi_Chr7_10 does not exist in the assembly or Table S10. I guess it is rather Mi_Chr7_18? This would be consistent with Figure 2f and Figure S17. Please clarify this point and also explain why Mi_Chr12_33 is represented two times in Figure S16b (4)?

Besides, I maintain that FigureS17 with only one single read supporting the telomere-to-telomere fusion does not really make sense. More coverage of this junction with ultra-long reads would be necessary to unequivocally support that fusion.

Overall, with only 4 examples of telomere-to-telomere fusions, two of which potentially involving collapsed scaffolds, the evidence is too weak at this stage to unequivocally support these putative fusions.

Therefore, I would suggest toning down this claim like 'we identified some putative chromosomal fusions in Minc' (as nothing is said about putative fusions in the other Meloidogyne species as well in this part of the paper).

Concerning assemblies V2, if I understand well, the authors have realized synteny was much better conserved between Minc B1 subgenome and the other subgenomes than with Mgram. Therefore, they considered B1 as a better representation of the ancestral karyotype and based their reconstruction of karyotype evolution on this new reference.

L403-404: The authors estimate that at least 15-35 chromosomal fusions occurred, including between the A and B subgenomes. This is a strong assumption that requires strong evidence. How many of these putative fusions are telomere-to-telomere fusions?

I understand these fusions are according to Hi-C scaffolding of the genomes which are blurry and noisy due to the complex polyploid nature of the genomes. As further discussed in the rest of my comments, all the biggest scaffolds are not strongly supported by Hi-C contact data. Therefore, the number of deduced fusion events is highly dependent on these unsure yet as good as possible assemblies.

Assessing whether some of these putative fusions are supported by genomic reads is very important and so far this has not been the case for V1 putative telomere-to-telomere fusions (only one supported by one single read). To determine whether most putative fusions are a consequence of the scaffolding it is essential to describe how many of these putative fusions were already present at the contig level.

In their point-by-point response (Response Figure 5), the authors clearly show that the scaffolding of the contigs is responsible for many telomere-to-telomere fusions in assembly V1. Indeed, many contigs that had a high number of Mi-tel repeats at only one side have disappeared during the scaffolding and been fused in scaffolds that have internal Mi-tel repeats. Therefore, I wonder, if at the contig level there is evidence of fusions between the subgenomes, including between A and B or if most of these are generated during the scaffolding. It is important to verify that point because a major claim of the paper is that there are extensive chromosomal fusions.

If it turns out that most putative fusions appear after the scaffolding, I would recommend toning down this claim of 'extensive chromosomal fusions'. Instead, I would rather indicate that scaffolding of the

genomes suggests multiple chromosomal fusions that will need to be further validated in the future.

- Other important yet not major point: Inconsistency between genome assembly sizes and k-mer / flow cytometry estimates

I completely agree with the authors that genome size might vary from one population to another, and this could explain differences between flow cytometry estimates made by Blanc-Matheu et al. on completely different populations of Minc, Mjav and Mare.

However, the authors should discuss the inconsistencies between their own genome assembly sizes and k-mer estimates (made by themselves on the same populations). Indeed, for Minc, k-mer estimates suggest a genome assembly size of $\sim 3 \times 71 \text{ Mb} = 213 \text{ Mb}$ (Table S3, excluding Illumina data which provided inconsistent k-mer estimates). The convergent Nanopore and PacBio k-mer estimates of 213Mb, is exactly consistent with the 213.3 Mb of their Minc assembly V1. However, the 225Mb of their assembly V2 is substantially bigger and no explanation is given for that. Concerning Mjav and Mare3n or Mare4n, same remark: the k-mer estimates are substantially lower than genome assembly sizes and in this case both for V1 and V2 assemblies. Could the authors provide an explanation / hypothesis for this difference?

From my own experience, Jellyfish + GenomeScope2 can provide underestimation of the genome size if Jellyfish is left with default parameter on genomes that contain many repeats. I suggest the authors to change the option '-h' to 1,000,000 in the jellyfish count command and check whether this changes the k-mer estimates to values more consistent with their genome assemblies.

Whatever the results of these new Jellyfish genome size estimates, another observation is very surprising: genome assembly sizes did not substantially change in assembly V2 compared to V1 for Mjav and the two Mare but changed substantially for Minc. What could explain this substantial difference? Is there any additional step that took place in Minc that was not applied to Mjav and the two Mare? The only difference I see is that BioNano data was used for Minc and not for the others.

In their point-by-point response, the authors claim that the Minc genome size at the contig level reached 225Mb with some sequences still partially collapsed. However, In Table S5, the genome size at the contig level is 213.3 Mb. I also downloaded Minc genome V1 and I confirm it is 213.3Mb big. Given that there are only 37.2 kb of N's in Minc genome V2 what could explain the >10Mb difference between assemblies V1 and V2?

In Table S5 I would suggest replacing Unscaffolds by Unscaffolded contigs (if my guess this represents contigs that were not scaffolded is correct).

- OTHER REMARKS / SUGGESTIONS

ABSTRACT

l28-29: I suggest writing 'identified a putative novel telomeric repeat'. Indeed, at this stage, although it is very likely that this repeat is telomeric, whether it actually participates in telomeric functions remains to be determined in the future. And this phrasing would be more consistent with what is written in the introduction.

INTRODUCTION

L60: 'weevil' is not a species but an umbrella name that groups many species. It would be more correct to give the exact name 'Otiorynchus scaber weevils'.

L94: although crosses between *M. fallax* and *M. chitwoodii* gave F1 hybrids (ref. 25), the egg masses had several critical defects and no functional F2 was obtained, so these hybrids were considered sterile. I think this is important to indicate that although hybridization was carried out successfully, the offspring was viable but sterile...

L101-102 "we identified two repetitive sequences that replace the typical telomere sequence and that might act as functional telomeres of polyploid and diploid RKNs which defines the boundary of the

chromosome". In fact, this statement poses a problem for the polyploid RKN species (but not for Mgra). What seems to be sure and supported by this analysis as well as ref.42 in bioRxiv is that the canonical TTAGGCn nematode telomeric repeats are absent from the RKN. However, whether the newly identified repeats replace TTAGGCn is unsure. Although ref.42 in bioRxiv confirms a terminal position of Mi-Tel on most *M. incognita* chromosomes with FISH, the repeat seems to be at only one end (which is consistent with the genome assemblies of ref42 not only in Minc but also Mjav, Mare and Mluci at the contig level). Therefore, whether the Mi-tel repeat is present at both ends or mostly at one end remains unsure. Although the authors now provide some preliminary FISH results on *M. incognita* eggs, the analysis is not conclusive enough regarding whether the repeat is mostly at one or two ends (in many cases this can be due to several chromosomes overlapping in the same slide as they are not well separated). Furthermore, in their *M. incognita* assembly V2, only 6 of the 39 chromosomes have the Mi-tel repeat at both ends (and nowhere else) and correctly oriented in the + strand and - strand. A total of 12 chromosomes do not show Mi-tel repeated at any end at all. Consequently, it cannot be assumed that Mi-tel completely replaces the canonical nematode telomeric repeat and defines chromosome boundaries for the polyploid RKN.

As also written by the authors in the discussion, further work would be needed both at the cytogenetic and genome assembly levels to definitely conclude on whether the repeat is mostly at one chromosome end possibly at both in some others.

However, the RKN telomeres are not the main scope of this paper, and many other interesting results are presented and supported by comprehensive analyses. Therefore, to avoid unnecessary further delay in publication, I suggest to simply rephrase this part in a more neutral way which would avoid any over-interpretation. Something like: 'canonical nematode telomeric repeats were not found in these RKN genomes, instead we identified two new types of repeats, respectively in the diploid and polyploid species enriched at the extremities of contigs / scaffolds. Recent cytogenetics analyses have confirmed a terminal position of the repeat on Minc chromosomes (ref.42) suggesting it is a telomeric repeat'

L107-108 "Unzipping the genome into several subgenomes is crucial " I suggest "...for a comprehensive and realistic representation of polyploid RKNs."

RESULTS

- GC%

It would be interesting to provide GC% for the different assemblies V1 as well as V2 and briefly discuss whether they differ from previous genomes for the same species or are similar.

- Exploration of the genome structure of Clade I RKNs

L132-133: the results even suggest that Ma3n and Ma4n are two different species that cannot be distinguished with SCAR markers but are clearly different based on their mitochondrial genomes.

L133 134: 'we propose naming the triploid Ma sample as Ma3n and the tetraploid sample as Ma4n in the rest of our study.

L137-138 the genome size of Mgram estimated by k-mers is 36.3 Mb with PacBio data and 41.3 Mb with Illumina data. (Table S3). Therefore, the assembly size is consistent with the higher estimates.

L141: I understand here that you confirm the genome size at the contig level for Minc is 213 Mb and not 225?

L146: I suggest 'all the subgenomes' rather than all the haplotypes.

L151: I suggest 'This drat preliminary assembly was used to elucidate...'

L161: '... suggesting that assemblies version 1 were mostly complete with a low degree of collapsed subgenomes.'

- Identification of telomeric repeats for Meloidogyne

I suggest changing the title of this section by 'Identification of putative novel telomeric repeats in Meloidogyne genomes'.

L204 -206: to facilitate comprehension, I would indicate here 'based on n=18 as ancestral number of chromosomes.' And I would also indicate that chromosome fusions is not the sole scenario and whole chromosome losses are also possible.

Lines 248-250 "248 Fig. 18). Moreover, we noted that another study on Meloidogyne telomeres, utilizing fluorescence in situ hybridization (FISH), reveals that Mi-Tel is present at the ends of the chromosomes (ref. 42)"

This is not completely exact, ref. 42 concludes that Mi-Tel is present at one end of most of the chromosomes and not 'at the ends'. To avoid unnecessary controversy or debates, I would suggest using a more neutral phrasing like 'reveals that Mi-Tel repeats have a terminal position on M. incognita chromosomes'

L255 and in M. luci as well (+ cite the reference for the genome sequence here).

L261-262 some were specifically present in Minc but absent from the rest.

- Construction of chromosome-level genomes for polyploid RKNs

L274: I suggest 'based on the information gained from assemblies V1 concerning the duplicated genome structures and the putative telomeric repeats, we attempted to construct.... Also indicate somewhere that for Minc, Nanopore data was used to assemble the contigs in contrast to PacBio for all the other ones.

L277: '36-52 chromosomes' does not match with '36-47 scaffolds'in Figure S21 but matches numbers given in Table S15. Please clarify which numbers are correct.

L283: I suggest 'in contrast to assemblies V1, this assemblies V2 were not guided by synteny relationships with the Mg T2T2 genome but only Hi-C and / or optical map data and were utilized for all subsequent analyzes'.

Figure S22: I completely understand the authors have done their best to scaffold the Minc genome based on Hi-C and Optical map data and this is not an easy task at all owing to the complexity of the genome structure and parasite contact noise caused by the copies as well as problematic collapsed regions. However, it should still be noted that the seven biggest Minc scaffolds show blurry contact data. For example, some contigs in scaffold 1 show strong contact data with other contigs in scaffold 2. Overall, all the biggest scaffolds are showing blurry / noisy contact signals. Unfortunately, I do not think anything can be done at that stage to improve the situation. However, this still casts some doubt on the reality of the biggest chromosomes.

In their point-by-point response, the authors indicate they have measured chromosomes up to 2.5µm in length in Minc. Considering chromosomes in C. elegans measure ca. 5 µm for 15 – 20 Mb, the biggest Minc chromosomes should be 8 – 10 Mb big. The biggest Minc scaffolds are bigger, and none are unequivocally supported by Hi-C data. However, to avoid long debate, I rather suggest the author to indicate (maybe in the discussion?) that this current representation of the genome remains hypothetical and might change in the future with technological progress. Interestingly, the Hi-C signals seem to be more uniform and less blurry in the other Meloidogyne assemblies V2 (Ma3n, Ma4n and Mj).

Figure S31. Please indicate that C1 – C18 are given according to synteny with Minc B1 genome (considered as representative of the haploid ancestral state if I understand correctly).

- Origin and evolution of polyploid RKN

Figure S34: I congratulate the authors for having re-done the whole phylogenomics analysis according to the non-manually edited assembly. One suggestion to make the figure easier to understand for the readers: I would rename Trees 1_18 to Trees C1_C18. This way, it will be more straightforward to understand the 18 trees come from concatenated single-copy genes from the 18 groups of synteny and 14 subgenomes of Figure S31 (If I understand correctly).

Line 339: I rather propose '17/18 trees exhibited....' Indeed, tree#17 is the only one not showing a clear separation between the A and B subgenomes.

The other major separation that can be observed is within the A subgenomes, in which A1-A2 from Minc hold an outgroup position relative to the rest of the A genomes from the other species in 15/18 trees (3 trees C1, C2 and C5 show Minc A1-A2 in the middle). This result suggests the A subgenomes of Minc have a different evolutionary history from the A subgenomes of the other species.

Then, within the A lineage, the majority signal is a separation between the A1 and A2 subgenomes from Ma3n, Ma4n and Mj. If A1 and A2 have actually been attributed randomly to the subgenomes this A1-A2 separation has very low probability to be observed. I rather understand that the authors have re-attributed the A1 and A2 labels to the subgenomes according to the tree topologies? Is that the case? If yes it should be indicated in the results and / or methods.

Furthermore, the fact that the A1 subgenomes are clearly separated from the A2 subgenomes in most trees for Ma3n, Ma4n and Mj suggests these three species have a common origin for their A subgenomes and this topology is completely consistent with the hypothesis of the authors (FigS35 b 1-2 and c 1-2). Therefore, given these new results, I do not find that the 'topology of the A lineage did not match (L347) their hypothesis. In contrast, this perfectly matches.

The only thing that should be explained (in my opinion) is that the parent that provided A subgenomes in Minc is different from the parent that provided the A subgenomes in the other species. Otherwise, the rest of the scenario is completely valid (clonal diploid with high heterozygosity (1) or hybrid diploid (2) as donors of the A genomes.

L369-370 "suggesting that the maternal genomes of these species diverged not long ago and that hybridization occurred recently." I would add 'which is exactly consistent with nuclear and mitochondrial phylogenomics analyses made on previous more fragmented versions of these genomes (Blanc-Mathieu et al. 2017).

- Gene expression landscape after polyploidization of Clade I RKNs

These results are very interesting and convincing both regarding the higher divergence between subgenomes for the secreted genes vs. the rest of the genes and also for the identification of gene expression decrease for one subgenome. These two elements indeed suggest that allo-polyploidization has favored functional divergence between the gene copies as also previously shown in Blanc-Mathieu et al. 2017 at the gene expression pattern level.

- DISCUSSION

L459: It is important to note here that the palindromes initially identified in the genome of *Adineta vaga* in Flot et al. Nature 2013 have not been later confirmed by their new assembly of the genome based on long reads (Simion P. et al. Science Advances 2021). After discussion with the authors of these papers, they confirmed that the palindromes initially identified were most probably due to assembly errors in the first version of the genome. Concerning the *Meloidogyne*, it was not indicated whether the palindromes identified were the result of chromosome fusions between the subgenomes or independent segmental duplications. Therefore, I would not necessarily make such a statement.

L477: "...but only polyploid species exhibited extensive chromosome fusion." If we consider these

chromosome fusions as true (but see my previous comments on the lack of strong evidence), the explanation might lie in the reproductive mode. Indeed, while the diploid species do meiosis and must maintain a number of chromosomes that can be divided in two, this is not the case for the polyploid mitotic species. Therefore, big chromosomal rearrangements might not be counter-selected in the mitotic species while they are highly deleterious in the meiotic diploid species and thus counter-selected.

L485: I suggest 'Moreover, the genome assemblies we have generated...'

L504: please indicate 'Otiorynchus scaber weevils'

L512-516. I found very interesting the hypothesis that *M. floridensis* has an AAA triploid genome. However, I do not exactly understand how this conclusion was drawn and based on which results. In Figure S7a, on the left, *M. floridensis* is proposed to be triploid by SmudgePlot but AAB. How did the authors reach the conclusion that it is in fact AAA? Could it be in fact AA with AA similar to the A genomes of *Minc* and a completely different B genome not closely related to any B genomes of the other species studied here?

- METHODS

Lines 950-953: Short reads and corrected long reads were subjected to genomescope2.0 and Smudgeplot31 to estimate genome size and ploidy with recommended parameters. Which software was used to enumerate k-mers? Was it JellyFish or KMC ? And which parameters were used? This information is important given the inconsistency between genome size estimations with kmers and genome assembly size in *Minc*.

Line 1155 "Subgenome identification of the telomere-to-telomere genome". To be consistent with the rest of the manuscript and since these genomes are not called "T2T" anymore, this title should be changed to 'chromosome level' rather than T2T.

- DATA AVAILABILITY:

The authors have now released their annotated genome assemblies V2 in the NCBI, which will greatly facilitate re-use and further analysis!

I think Assemblies T2T should be removed from the laboratory website download page to avoid any confusion with assemblies V2.

On the laboratory website data download page, what is *Mh_assembly_V1* ? Is it *Meloidogyne* hapla?

- ACKNOWLEDGEMENTS

lines 1387-1389: since the FISH results have not been included neither in the supplementary material nor in the results (because these are too preliminary and not conclusive enough to be included), I am afraid they cannot be referred to in the acknowledgement section.

Reviewer #3:

Remarks to the Author:

This manuscript provides solid convincing data that the diploid *Mg* genome lacks telomerase reverse transcriptase and maintains its telomeres via the ALT pathway. Moreover, the diploid *Mg* genome represents a reflection of an ancestral state to related polyploid *Meloidogyne* species, as both diploid and polyloid *M* species use related ALT telomere maintenance elements. It is unusual to find head to head telomere fusions in nature, so it is intriguing that these might be characteristic of polyploidization, given that the *Mg* diploid genome lacks telomere fusions and maintains its telomeres via ALT. The *Mg* genome appears to be high quality with almost all telomeres accounted for (TtoT).

Moreover, genomes of several related polyploid nematodes are created, and telomeres for many chromosomes of polyploid chromosomes are defined. The authors indicate that telomeres of the polyploid species can be sorted into 'telomere clusters' and that some telomere clusters are unique to specific species whereas other telomere clusters are shared between polyploid species that were derived from a common diploid ancestor that lacked telomerase. These are interesting observations regarding the heterogeneity, inheritance and evolution of telomere sequences of ALT telomeres, although the authors describe telomere clusters in vague terms without clearly defining how they differ in sequence or length.

A number of HGT events are documented for polyploids (hundreds), but analysis of these sequences is not presented in detail. Remarkably, the authors observe consistent clusters of HGT events at specific segments of some chromosomes of polyploid species. This might provide insight into the mechanism of HGT, but little information is provided about genes inserted at HGT clusters, which might be related to sites of transposon insertion or to specific epigenomic states that some chromosome domains may adopt. It is possible that the authors' Hi-C data might reveal features of HGT hotspots.

The synteny analysis is impressive and helps to clarify how polyploid genomes evolved and are structured, reflecting 18 ancestral chromosomes. Clade V species contain 6 chromosomes, and it has been speculated that 18 to 22 ancestral chromosomes may have been present in very distantly related species, reflecting conservation of gene order for more than 500 million years of evolution. It would be interesting if the authors could show that the 18 chromosomes of Mg correspond to the highly ancestral state proposed by Simakov 2022. They may not, because the Mg genome lacks canonical telomeres and maintains telomeres via ALT, and ALT is almost always activated when canonical telomeres become very short and a number of telomere fusions occur.

The authors show that transposon levels are increased in polyploid strains but speculate that this increase occurred in diploid genomes prior to polyploidization. However, I suggest that the authors might be able to create a more general model, where a transposition burst might occur after polyploidization, which might lead to chromosome instability that drives chromosome fusion in a manner that helps to shape polyploid genomes. HGT events might be related to or integrated with transposon activation. The ALT telomeres might be related to some fusion events, but the significance of ALT telomere fusions in the context of polyploidization will only become apparent when genomes of polyploid nematodes telomerase-positive are characterized. Overall, this is an interesting and generally well written manuscript that is appropriate for Nature Communications.

Comments:

1. It would be helpful to show the Mg-Tel repeat sequence in Fig. 2. It has little similarity to the TTAGGC canonical repeat sequence. ALT typically involves amplification of canonical telomere repeat sequences, and the authors could point out that there is no example of the TTAGGC sequence in Mg-Tel. However, there are many examples of TTA and some examples of GGC: these could be highlighted and queried to determine if they are present at a frequency that is more likely than would be expected by chance. If not, then the Mg-Tel sequence bears essentially no similarity to the ancestral TTAGGC telomere repeat, despite the G-rich nature of Mg-Tel on the strand that runs 5' to 3' towards the end of the chromosome.

2. 'As expected the read contained 71 Mi-Tel elements and 93 reversed complementary Mi-Tel elements.' This represents much telomeric DNA at the site of the telomere fusion. How does the element number compare with the number of Mi-Tel or Mg-Tel elements at natural telomeres? Is there evidence for telomere shortening prior to fusion? If not, what might be the cause of the telomere fusion event? Evidence for this might be apparent based on the sequence at the site of telomere fusion: it would be of interest if the authors could show this predicted sequence.

3. 'We extracted 9318 telomeric elements from 4 polyploid species which exhibited differences in length'. The length in Fig S20 varies from 300 bp to 1000 bp. Does this mean that the length of the repeat unit varies significantly at telomeres, even in the same species? Is the sequence similar in all species? To understand the similarity, it might be helpful to show the sequence of one telomere repeat unit per species.

4. Note that even for human ALT cancers, the sequences at telomeres of specific cell lines may not be the same and are not well understood. A recent PLOS Genetics paper from the Baird group demonstrates that telomeres of cells that become ALT can display related sequences at their telomeres, but that the telomere repeat sequence is quite heterogeneous, perhaps reflecting the sequence heterogeneity of telomeres near the telomere-subtelomere boundary of one short chromosome end that is used to create a template that is used for ALT:
<https://www.ncbi.nlm.nih.gov/pmc/articles/PMC9678338/>

5. 'Some telomere clusters were found in each species whereas some were specific to specific species'. The term cluster is not well defined. What distinguishes the different telomere clusters? Are these different types of telomere sequence repeats? What are the features of telomere clusters that are only found in specific species? Are they characterized by a specific sequence or a specific length? What are the characteristics of the shared telomere clusters? It would be helpful to see one sequence example of each of the 19 telomere clusters.

6. It appears that only species that lacked telomerase and used ALT for telomere maintenance created these polyploids. One possibility is that the ALT telomere sequence repeat evolved in diploid nematodes that gave rise to AAB or AABB or AAA polyploid species, and that the telomere clusters (telomere sequence repeats?) that are shared between polyploid species reflects their ancestral diploid telomeres. If this is the case, then the polyploid chromosomes derived from A1 or A2 or B diploid genomes may have share common telomere sequences (clusters) that reflect their ancestral diploid ALT telomere sequence repeats. In this case, the B chromosomes would possess common telomere sequences (telomere clusters?) that less commonly occur on A1 or A2 chromosomes, at least for chromosomes that have not fused.

7. What fraction of the extensive chromosome fusions in Mi are due to telomere fusions?

8. What is the nature of the horizontally transferred genes at the hotspots for HGT? This might provide insight into a mechanism of horizontal gene transfer. Are these derivatives of the same gene family that have duplicated and diverged? How do the genes present in hot spots compare with those that are stochastically distributed? How many HGT genes come from plants or soil bacteria or other nematodes? It is possible that a fraction of HGT events reflect transposons that moved horizontally between species, but it is not clear if transposons are in the HGT category or if they are binned in a separate genomic parasite category.

9. It is terrific that the authors were able to define the chromosome number and features of synteny between related chromosomes in polyploid species, and that phylogenetic models were created for each chromosome.

10. What does it mean that 2578 of 3251 LTR retrotransposons had 100% LTR identity? Does this mean that all 2578 had the same LTR sequence? If so, this defines a single highly active LTR retrotransposon. How many types of full length LTR transposons are defined by the remaining ~850 transposons, if LTR sequence identity defines transposon type?

11. 'We did not observe a burst of many TEs after hybridization (Fig. S39a)'. I respectfully disagree. Compared to the diploid Mg where 0.5% of the genome is TEs, the polyploid species have 2.4-2.8% of their genomes as TEs. This could suggest a consistent dramatic burst of transposition upon polyploidization or that there is a TE burst prior to polyploidization. It is possible that endogenous

transposons rather than those introduced by HGT are responsible for a TE burst.

12. If a TE burst occurs upon each round of polyploidization, why would there be A or B-specific transposons? The transposons that are active and moved upon polyploidization would move again upon AA and then again when AAB or AAA ploidy was created. Inactive TEs that cannot move non-autonomously or solo LTRs might be specific to A or B chromosomes.

13. Fig. S39a could be better explained. Why are the Kimura substitution levels for transposons so much higher in polyploids than in Mg?

14. If most chromosome fusion events occurred post-hybridization, and if a TE burst occurred post-hybridization, then the TE burst might generate chromosome instability (DSBs from DNA transposon excision) that might help to explain some chromosome fusions.

15. 'ALT telomeres have a weak chromosome protective function, which seems to explain why chromosome fusion occurs.' Some telomere instability may be a feature of ALT tumors (rapid telomere shortening events), but the point to ALT is to maintain stable telomeres that are essential for tumor genome survival. The authors may be referring to work on ALT cancer cells. However, during activation of ALT in cancer, telomeres shorten due to lack of telomerase, and telomeres become very short, and chromosome fusions always occur prior to activation of the ALT mechanism. All ALT positive tumors possess telomere fusions, and it is formally possible that the process of telomere fusion may be mechanistically linked to activation of ALT. The authors data may instead suggest that an aspect of polyploidization drives chromosome fusion. This might be apparent if genomes of telomerase-positive polyploid nematodes have been characterized.

16. There is a caveat to the logic that the Mg diploid genome lacks telomere fusions but that polyploid Meloidogyne genomes possess chromosome fusions, some of which may be telomere fusions. The caveat is that it is virtually certain that the diploid Mg chromosomes must contain telomere fusions. This is because a diploid ancestor of Mg possessed telomerase and canonical (TTAGGC)_n telomeres, but then telomerase was lost, followed by critical telomere shortening, creation of telomere fusions, and activation of the ALT telomere maintenance mechanism: there are many examples of this in cancer genome evolution. So, there might be short (TTAGGC)_n telomere tracts (100 to 300 bp long) present at the sites of telomere fusion in the Mg genome, but these telomere fusion events might be very old and no longer visible. In contrast, the telomere fusions evident in polyploid Meloidogyne species occurred after activation of ALT telomere maintenance, and may reflect relatively recently created telomere fusions. If the polyploid Meloidogyne chromosome fusions commonly involve ALT telomeres, my guess is that direct telomere-telomere fusions would be rare, and that if a telomere is involved in a chromosome fusion event, then there would be a single telomere present at the site of chromosome fusion, and that the other site of fusion would be a DNA double-strand break created in a distinct chromosome, which might be some distance from a telomere. This is because the function of telomeres is to promote chromosome end protection, and even ALT cancer cells display stable chromosome karyotypes because ALT telomeres are good at telomere end protection. However, it is also possible that polyploidization promotes fusion of long telomeres of chromosomes in species where ALT is used for telomere maintenance. In this case, a few or many fused chromosomes might contain two head to head telomeres at their fusion breakpoints.

17. 'Although we observed specific distribution patterns in polyploid nematodes, such as histone modification distribution, HGT hotspots and increased TE content, it is not clear whether these changes are caused by lifestyle changes, polyploidization, parthenogenesis or ALT telomeres.' Why would ALT telomeres cause HGT hotspots or increased TE content, especially if ALT were present in the ancestral diploids?

18. The authors may be the first to discover HGT hotspots in nematode chromosomes. Is there evidence for such hotspots outside of nematodes? This is quite interesting and could be relevant to

understanding the basis of HGT. It would appear that elements of chromosome instability, like chromosome fusion, increased transposon content, HGT hotspots might be coupled to polyploidization events, and in this context some or all of these chromosome instability elements might be coupled.

Reviewers' Comments:

Reviewer #1:

Remarks to the Author:

Dear authors,

Congratulations for the remarkable amount of work done for this paper and all the additional work accomplished during the different rounds of revisions of the paper.

This paper presents high-quality data and many sophisticated analyses with remarkably well-designed figures and graphs.

The paper, the genomes and all the analyses represent a valuable resource not only for the community of plant nematologist but also more broadly for evolutionary biologists interested in the fate of genes and genomes after hybridization and polyploidization.

All my previous major concerns have been correctly addressed.

All the other points have also been clarified.

I now recommend publication of the paper and I would like to congratulate again all the authors for their tenacious and meticulous work.

I only have a few minor suggestions that the authors can be believed to address with no need to be further reviewed.

L94-95 'it is important to indicate that although hybridization occurred, it yielded a sterile offspring in this study'

L112 'polyploid RKN genomes' not 'polyploid RNKs'

L225 'located at scaffold ends' not 'in chromosome ends'

L1576 Supp. Fig. 20 'identity greater' not 'similarity greater'

Supp. Fig. 26b I guess the arrowhead represents Mg-Tel not Mi-Tel

Reviewer #3:

Remarks to the Author:

Please see uploaded PDF file that is easier to interpret.

The authors have responded to all reviewer comments. In some cases, the responses are appropriate. I have a few remaining concerns that might improve how the manuscript reads.

Comment:

2. 'As expected the read contained 71 Mi-Tel elements and 93 reversed complementary Mi-Tel elements.' This represents much telomeric DNA at the site of the telomere fusion. How does the element number compare with the number of Mi-Tel or Mg-Tel elements at natural telomeres? Is there evidence for telomere shortening prior to fusion? If not, what might be the cause of the telomere fusion event? Evidence for this might be apparent based on the sequence at the site of telomere fusion: it would be of interest if the authors could show this predicted sequence

Response:

We counted the copy number of Mi-Tel in chromosome ends (mean = 40, median = 21). It is hard to say that telomere-telomere fusions are caused by a reduction in telomere copy numbers. We thank

the reviewers for raising many questions and hypotheses. However, our current understanding of telomeres in root-knot nematodes is quite limited. We just identified a sequence that specifically locates at the chromosome end, which may function as an alternative telomere. Many questions still need to be confirmed, such as the start and end of the telomere repeat unit and so on. Thus, many hypotheses are hard to address in this paper in a few sentences, and these works deserve to be addressed in the future using more accurate sequencing techniques, such as HiFi data.

I thank the authors for counting the copy number of MiTel at telomeres: this graph is informative. Based on the data (telomere range from 10 to 300 MiTel copies, with most telomeres shorter than 40 copies). The authors could show this data and state that the apparent telomere fusion is not due to telomere erosion but instead due to fusion of long telomeres. This is important because ALT telomeres are known to have varied lengths, ranging from very long to very short, and one possibility is that short ALT telomeres are prone to fusion. The authors have the opportunity to clarify what they have learned: the telomere fusion they observe was created by fusion of long telomeres. This conclusion is mechanistically important because chromosome fusion is a central theme of this manuscript.

Comment:

3. 'We extracted 9318 telomeric elements from 4 polyploid species which exhibited differences in length'. The length in Fig S20 varies from 300 bp to 1000 bp. Does this mean that the length of the repeat unit varies significantly at telomeres, even in the same species? Is the sequence similar in all species? To understand the similarity, it might be helpful to show the sequence of one telomere repeat unit per species.

Response: We thank reviewers for their interest in this section. In fact, these repetitive sequences of telomeres are highly complex and may contain multiple variants. Through multiple sequence alignments, we have preliminarily identified a 66bp motif that is relatively conserved. In order to describe the features of complex telomere sequences, we define the sequence between two adjacent 66bp motifs as a telomeric element. Indeed, even within the same species, these telomeric repeats exhibit significant variations in both length and sequence. The only thing that can be confirmed is that this 66bp motif is indeed located at the chromosome ends (based on FISH). To be frank, this part of the analysis is still relatively rudimentary. This result can only indicate that these sequences are highly complex, but the variation patterns between them are not clear. Below is the result of multiple sequence alignment after removing redundant sequences using CD-HIT. We still believe that based on the current genomic assembly data and the understanding of root- know nematode telomeres, the question about the evolution of repeat sequences is not sufficiently resolved. In revised version, we added a sentence in discussion to describe that these complex telomere sequences deserve further study (Line 483-485).

Thank you for this explanation. It would be informative to readers if the authors explained in the manuscript text that the repetitive sequences of Mi telomeres are highly complex, but that they identified a 66 bp motif that is a conserved of their 'telomere elements'. Showing the 66 bp conserved sequence in a main figure in the paper seems reasonable and easy to do. This concrete if rudimentary insight regarding what Mi telomeres actually look like will be satisfying, even if a future paper on the complicated structures of Mi telomeres were published in a future study.

Comment:

4. Note that even for human ALT cancers, the sequences at telomeres of specific cell lines may not be the same and are not well understood. A recent PLOS Genetics paper from the Baird group demonstrates that telomeres of cells that become ALT can display related sequences at their telomeres, but that the telomere repeat sequence is quite heterogeneous, perhaps reflecting the sequence heterogeneity of telomeres near the telomere-subtelomere boundary of one short

chromosome end that is used to create a template that is used for ALT:

<https://www.ncbi.nlm.nih.gov/pmc/articles/PMC9678338/>

Response:

We thank reviewers for their comments and the recommended literature. Indeed, the length and sequence heterogeneity of ALT telomeres are significant, and even in cancer cells, they are not well understood. Furthermore, investigating the patterns of ALT telomere sequences still requires a amount of work, which is beyond the scope of our research. We hope for your understanding regarding the limitations of our study in terms of telomere sequence features.

I was asking the authors to comment on the heterogeneity that they have already observed, not to investigate patterns of ALT telomere sequences in a manner that is beyond the scope of their research. Perhaps simply add a sentence or two to the current manuscript to indicate that the complexity apparent in telomeres of wild Mi nematodes is consistent with the complexity of ALT telomere sequences in human tumors. This places the current paper in context, even if the understanding of Mi telomeres based on long read sequence analysis is rudimentary.

Comment:

7. What fraction of the extensive chromosome fusions in Mi are due to telomere fusions?

Response:

Among the 100 putative chromosome fusion events, only 9 fusion events were detected with Mi-Tel(+) ≥ 5 and Mi-Tel(-) ≥ 5 . This indicates that most fusion sites lack telomere repeats in their vicinity. As the reviewer mentioned, most fusion events are likely not telomere fusions but rather caused by double-strand breaks in non-telomeric regions of two chromosomes. We have now added a supplementary table to show that information (Supplementary table 25). Thank you for clarifying this matter.

Comment:

The authors state that there are only 9 telomere-to-telomere fusion events. However, the authors also observe at least 8 fusion events that involve one telomere with more than 5 long telomere elements. The authors could point out that this means that long ALT telomeres were often involved in fusion events, either with other telomeres or with other chromosome segments. The authors may have evidence that short ALT telomeres with 1-5 copies of a telomere element fuse, but it is not clear if these 1-5 copies are at telomeres or at internal segments of the genome.

Comment:

10. What does it mean that 2578 of 3251 LTR retrotransposons had 100% LTR identity? Does this mean that all 2578 had the same LTR sequence? If so, this defines a single highly active LTR retrotransposon. How many types of full length LTR transposons are defined by the remaining ~850 transposons, if LTR sequence identity defines transposon type?

Response: Long Terminal Repeat Retrotransposons (LTR-RTs) are transposable elements characterized by having two long terminal repeats (LTRs, black arrows in the diagram) flanking an internal coding region. The newly inserted LTR-RT has completely identical LTR sequences on both sides, and over time, they diverge from each other. Here, we are referring to each of the 2,578 LTR-RTs, where the sequences on both sides are identical, indicating that they have been recently inserted. In other words, the LTR sequences on both sides of LTR-RT_1 is identical, but they may differ from the LTR sequences of LTR-RT_2. We have edited this sentence to avoid ambiguity (Line 398-400). So, the authors are suggesting that 2578 of 3251 transposon insertions are recent, such that their LTR match perfectly. This seems consistent with a recent stimulation of transposon activity.

Comment:

11. 'We did not observe a burst of many TEs after hybridization (Fig. S39a)'. I respectfully disagree. Compared to the diploid Mg where 0.5% of the genome is TEs, the polyploid species have 2.4-2.8% of their genomes as TEs. This could suggest a consistent dramatic burst of transposition upon polyploidization or that there is a TE burst prior to polyploidization. It is possible that endogenous transposons rather than those introduced by HGT are responsible for a TE burst.

Response:

We agree that polyploid species must have undergone transposon expansions, and these expansions may have occurred after hybridization or in the diploid ancestors. A key indicator is the high similarity of mitochondrial genomes in these species, indicating that hybridization is a very recent event (Line 374-381). If the TE burst occurred after the recent hybridization, the peak of transposons should be close to position 0; however, since the peak is located in the middle (10-20), it indicates that the majority of transposons are not young and likely underwent a TE burst in the diploid ancestors (Supplementary Fig 39a).

I am unclear about this logic. 'It indicates that the majority of transposons are not young' is at odds with the authors' data that 'We identified 3251 full length LTR retrotransposons in our 5 genomes, 2578 of which (79%) of which exhibited 100% identity between the LTR repeats.' This means that the vast majority of LTR retrotransposons (80%) are very young. Yet the authors conclude the above sentence in the text 'of which exhibited 100% identity between the LTR repeats, indicated that some TEs are still active'. This indicates that most TEs are still active or were recently very active.

Then, in the text, the authors conclude that 'we did not observe a burst of TEs after recent hybridization, instead only a few TEs were recent and most TEs were old'. How is it possible that only a few TEs were old if 80% of LTR transposons had identical 5' and 3' LTRs that indicate very recent activity??

The authors go on to state 'Most of the TE families could be detected in A and B subgenomes': This phrase should probably be rewritten to state 'Most of the TE families could be detected in both A and B subgenomes, and no A- or B- specific TE families were found'. The logic here needs to be better explained. Within a single species, there is introduction of new transposons as the species evolves, such that individual strains within a species will differ by hundreds to thousands of new transposons. Therefore distinct A- and B- species would possess thousands of different transposons on their chromosomes. The fact that no A- or B-subgenome specific transposons were detected therefore means that most transposon families moved from the A subgenome to the B subgenome, or vice versa. Therefore, it is likely that the large increase in transposon copy number observed in polyploid strains occurred during of after polyploidization.

Despite the above logic, the authors seem convinced based on their data in Fig S39 that most transposons are old and moved only in the diploid ancestors. I do not see how this is possible. The authors need to very clearly explain the premise of the data in Fig. S39 and why this means that most transposons are old and do not move. If the authors have clear independent lines of evidence for the hypothesis that most transposons are old and do not move, please convey this information. Please also consider the alternative hypothesis that TE bursts might have occurred during or after polyploidization. And then perhaps speculate on how these distinct possibilities will be resolved in future work.

Comment:

13. Fig. S39a could be better explained. Why are the Kimura substitution levels for transposons so much higher in polyploids than in Mg?

Response:

The TE content in polyploids is higher than in Mg. In fact, we have mentioned this earlier in the text, and previous studies have also pointed this out, so we did not repeat it here. What we are investigating here is whether the transposon expansion in polyploids occurred before or after hybridization. Given that hybridization is a very recent event, and the majority of transposon peaks are not close to 0, we believe that the transposon burst likely occurred in the diploid ancestor stage.

Fig. S39a is a supplemental figure that has space to explain what the Kimura substitution levels mean. Was this mentioned with references in the text? Perhaps the authors could add a sentence or two to Fig. S39a about why Kimura substitution levels are likely to mean a diploid burst, if it is at all possible to differentiate a diploid from polyploid burst. If a diploid burst, then this would mean that a factor triggers dramatic transposon expansion prior to polyploidy.

REVIEWERS' COMMENTS

Reviewer #1 (Remarks to the Author):

Dear authors,

Congratulations for the remarkable amount of work done for this paper and all the additional work accomplished during the different rounds of revisions of the paper.

This paper presents high-quality data and many sophisticated analyses with remarkably well-designed figures and graphs.

The paper, the genomes and all the analyses represent a valuable resource not only for the community of plant nematologist but also more broadly for evolutionary biologists interested in the fate of genes and genomes after hybridization and polyploidization.

All my previous major concerns have been correctly addressed.

All the other points have also been clarified.

I now recommend publication of the paper and I would like to congratulate again all the authors for their tenacious and meticulous work.

I only have a few minor suggestions that the authors can be believed to address with no need to be further reviewed.

Response: We thank the reviewers for their contributions to improving the quality of the paper. All of minor suggestions have been edited in revised version.

L94-95 'it is important to indicate that although hybridization occurred, it yielded a sterile offspring in this study'

Response: We have included this information.

L112 'polyploid RKN genomes' not 'polyploid RNKs'

Response: We have edited this sentence.

L225 'located at scaffold ends' not 'in chromosome ends'

Response: We have changed this word.

L1576 Supp. Fig. 20 'identity greater' not 'similarity greater'

Response: We have changed this word.

Supp. Fig. 26b I guess the arrowhead represents Mg-Tel not Mi-Tel

Response: Thanks, we have updated this figure.

Reviewer #3 (Remarks to the Author):

Please see uploaded PDF file that is easier to interpret.

The authors have responded to all reviewer comments. In some cases, the responses are appropriate. I have a few remaining concerns that might improve how the manuscript reads.

Summary: Reviewer #3 provided 7 more suggestions in our previous round of responses. We now address them one by one in below. The **green text** represents the comments and responses from the previous round. The **black text** represents the reviewer's comments this time, and our responses are marked in **blue**.

The problem mainly involves telomeres and transposable elements (TE)

For telomere:

As suggested by the reviewer, we added two supplementary figures to show the copy number of Mi-Tel and the 66 bp motif of Mi-Tel. In the discussion, the heterogeneity of telomeric elements and its relationship with chromosomal fusion are also described.

For TE activity:

The reviewer's confusion is because what we describe in the article is Full-length LTR (only 2.6% of all LTR), not all LTR copies. We have edited the description in **Result** and added a sentence to the **Discussion** describing that old LTRs may become so fragmented due to genomic instability that they cannot be detected. Additionally, a description of kimura distance is added as suggested.

Comment:

2. 'As expected the read contained 71 Mi-Tel elements and 93 reversed complementary Mi-Tel elements.' This represents much telomeric DNA at the site of the telomere fusion. How does the element number compare with the number of Mi-Tel or Mg-Tel elements at natural telomeres? Is there evidence for telomere shortening prior to fusion? If not, what might be the cause of the telomere fusion event? Evidence for this might be apparent based on the sequence at the site of telomere fusion: it would be of interest if the authors could show this predicted sequence

Response:

We counted the copy number of Mi-Tel in chromosome ends (mean = 40, median = 21). It is hard to say that telomere-telomere fusions are caused by a reduction in telomere copy numbers. We thank the reviewers for raising many questions and hypotheses. However, our current understanding of telomeres in root-knot nematodes is quite limited. We just identified a sequence that specifically locates at the chromosome end, which may function as an alternative telomere. Many questions still need to be confirmed, such as the start and end of the telomere repeat unit and so on. Thus, many hypotheses are hard to address in this paper in a few sentences, and these works deserve to be addressed in the future using more accurate sequencing techniques, such as HiFi data.

I thank the authors for counting the copy number of MiTel at telomeres: this graph is informative. Based on the data (telomere range from 10 to 300 MiTel copies, with most telomeres shorter than 40 copies). The authors could show this data and state that the apparent telomere fusion is not due to telomere erosion but instead due to fusion of long telomeres. This is important because ALT telomeres are known to have varied lengths, ranging from very long to very short, and one possibility is that short ALT telomeres are prone to fusion. The authors have the opportunity to clarify what they have learned: the telomere fusion they observe was created by fusion of long telomeres. This conclusion is mechanistically important because chromosome fusion is a central theme of this manuscript.

Response:

We thank the reviewers for their interpretation of our data. We have now added a supplementary figure (Supplementary Fig 19) to describe the copy number of Mi-Tel at telomere and state that telomere fusion is not due to telomere shortening.

Comment:

3. 'We extracted 9318 telomeric elements from 4 polyploid species which exhibited differences in

length”. The length in Fig S20 varies from 300 bp to 1000 bp. Does this mean that the length of the repeat unit varies significantly at telomeres, even in the same species? Is the sequence similar in all species? To understand the similarity, it might be helpful to show the sequence of one telomere repeat unit per species.

Response: We thank reviewers for their interest in this section. In fact, these repetitive sequences of telomeres are highly complex and may contain multiple variants. Through multiple sequence alignments, we have preliminarily identified a 66bp motif that is relatively conserved. In order to describe the features of complex telomere sequences, we define the sequence between two adjacent 66bp motifs as a telomeric element. Indeed, even within the same species, these telomeric repeats exhibit significant variations in both length and sequence. The only thing that can be confirmed is that this 66bp motif is indeed located at the chromosome ends (based on FISH). To be frank, this part of the analysis is still relatively rudimentary. This result can only indicate that these sequences are highly complex, but the variation patterns between them are not clear. Below is the result of multiple sequence alignment after removing redundant sequences using CD-HIT. We still believe that based on the current genomic assembly data and the understanding of root- know nematode telomeres, the question about the evolution of repeat sequences is not sufficiently resolved. In revised version, we added a sentence in discussion to describe that these complex telomere sequences deserve further study (Line 483-485).

Thank you for this explanation. It would be informative to readers if the authors explained in the manuscript text that the repetitive sequences of Mi telomeres are highly complex, but that they identified a 66 bp motif that is a conserved of their ‘telomere elements’. Showing the 66 bp conserved sequence in a main figure in the paper seems reasonable and easy to do. This concrete if rudimentary insight regarding what Mi telomeres actually look like will be satisfying, even if a future paper on the complicated structures of Mi telomeres were published in a future study.

Response:

Thanks to the reviewers for their suggestions. We show a schematic diagram of the multiple sequence alignment and label the 66 bp sequence (**Lines 235-238**, Supplementary Fig 16). We also stated the complexity of the sequence in the discussion (**Lines 492-496**).

Comment:

4. Note that even for human ALT cancers, the sequences at telomeres of specific cell lines may not be the same and are not well understood. A recent PLOS Genetics paper from the Baird group demonstrates that telomeres of cells that become ALT can display related sequences at their telomeres, but that the telomere repeat sequence is quite heterogeneous, perhaps reflecting the sequence heterogeneity of telomeres near the telomere-subtelomere boundary of one short chromosome end that is used to create a template that is used for ALT: <https://www.ncbi.nlm.nih.gov/pmc/articles/PMC9678338/>

Response:

We thank reviewers for their comments and the recommended literature. Indeed, the length and sequence heterogeneity of ALT telomeres are significant, and even in cancer cells, they are not well understood. Furthermore, investigating the patterns of ALT telomere sequences still requires a amount of work, which is beyond the scope of our research. We hope for your understanding regarding the

limitations of our study in terms of telomere sequence features.

I was asking the authors to comment on the heterogeneity that they have already observed, not to investigate patterns of ALT telomere sequences in a manner that is beyond the scope of their research. Perhaps simply add a sentence or two to the current manuscript to indicate that the complexity apparent in telomeres of wild Mi nematodes is consistent with the complexity of ALT telomere sequences in human tumors. This places the current paper in context, even if the understanding of Mi telomeres based on long read sequence analysis is rudimentary.

Response:

Thanks for suggestions. We have described the complexity of telomeres in the discussion (**Lines 492-496**).

Comment:

7. What fraction of the extensive chromosome fusions in Mi are due to telomere fusions?

Response:

Among the 100 putative chromosome fusion events, only 9 fusion events were detected with Mi-Tel(+) ≥ 5 and Mi-Tel(-) ≥ 5 . This indicates that most fusion sites lack telomere repeats in their vicinity. As the reviewer mentioned, most fusion events are likely not telomere fusions but rather caused by double-strand breaks in non-telomeric regions of two chromosomes. We have now added a supplementary table to show that information (Supplementary table 25).

Thank you for clarifying this matter.

Comment:

The authors state that there are only 9 telomere-to-telomere fusion events. However, the authors also observe at least 8 fusion events that involve one telomere with more than 5 long telomere elements. The authors could point out that this means that long ALT telomeres were often involved in fusion events, either with other telomeres or with other chromosome segments. The authors may have evidence that short ALT telomeres with 1-5 copies of a telomere element fuse, but it is not clear if these 1-5 copies are at telomeres or at internal segments of the genome.

Response:

Thanks for suggestion. We have modified this sentence to describe the telomere fusion event more objectively (**Lines 425-427**). "We noticed that 24 fusions involved telomere fusion, of which 21 fusion events involved at least one telomere with more than 5 Mi-Tel elements." This sentence also further suggested that chromosomal fusions are not due to telomere shortening.

Comment:

10. What does it mean that 2578 of 3251 LTR retrotransposons had 100% LTR identity? Does this mean that all 2578 had the same LTR sequence? If so, this defines a single highly active LTR retrotransposon. How many types of full length LTR transposons are defined by the remaining ~850 transposons, if LTR sequence identity defines transposon type?

Response: Long Terminal Repeat Retrotransposons (LTR-RTs) are transposable elements characterized by having two long terminal repeats (LTRs, black arrows in the diagram) flanking an internal coding region. The newly inserted LTR-RT has completely identical LTR sequences on both sides, and over time, they diverge from each other. Here, we are referring to each of the 2,578 LTR-RTs, where the

sequences on both sides are identical, indicating that they have been recently inserted. In other words, the LTR sequences on both sides of LTR-RT_1 is identical, but they may differ from the LTR sequences of LTR- RT_2. We have edited this sentence to avoid ambiguity (Line 398-400).

So, the authors are suggesting that 2578 of 3251 transposon insertions are recent, such that their LTR match perfectly. This seems consistent with a recent stimulation of transposon activity.

Response:

The 3251 LTRs described here refers specifically to the **full-length LTR**, which is just a small portion of **all LTRs** (121,684 LTR copy, from EDTA results). In the full-length LTR (about 2.6% of all LTRs), it is true that 80% are recently inserted, but that doesn't mean all LTRs are recent insertions. We speculate that some old LTRs may have lost their complete structure, making them undetectable by software and only detectable through homology-based alignment methods. In the revised version, we describe this in the discussion (**Lines 511-514**).

Comment:

11. 'We did not observe a burst of many TEs after hybridization (Fig. S39a)'. I respectfully disagree. Compared to the diploid Mg where 0.5% of the genome is TEs, the polyploid species have 2.4-2.8% of their genomes as TEs. This could suggest a consistent dramatic burst of transposition upon polyploidization or that there is a TE burst prior to polyploidization. It is possible that endogenous transposons rather than those introduced by HGT are responsible for a TE burst.

Response:

We agree that polyploid species must have undergone transposon expansions, and these expansions may have occurred after hybridization or in the diploid ancestors. A key indicator is the high similarity of mitochondrial genomes in these species, indicating that hybridization is a very recent event (Line 374-381). If the TE burst occurred after the recent hybridization, the peak of transposons should be close to position 0; however, since the peak is located in the middle (10-20), it indicates that the majority of transposons are not young and likely underwent a TE burst in the diploid ancestors (Supplementary Fig 39a).

I am unclear about this logic. 'It indicates that the majority of transposons are not young' is at odds with the authors' data that 'We identified 3251 full length LTR retrotransposons in our 5 genomes, 2578 of which (79%) of which exhibited 100% identity between the LTR repeats.' This means that the vast majority of LTR retrotransposons (80%) are very young. Yet the authors conclude the above sentence in the text 'of which exhibited 100% identity between the LTR repeats, indicated that some TEs are still active'. This indicates that most TEs are still active or were recently very active

Then, in the text, the authors conclude that 'we did not observe a burst of TEs after recent hybridization, instead only a few TEs were recent and most TEs were old'. How is it possible that only a few TEs were old if 80% of LTR transposons had identical 5' and 3' LTRs that indicate very recent activity??

Response: We identified 3251 **full-length LTRs** in five genomes. It is worth noting that this refers only to **full-length LTRs** but not **all LTRs**. According to the results from EDTA, a total of 121,684 LTRs were identified based on homology-based strategies, with most of them being incomplete or fragmented.

We speculate that this may be because old LTRs have lost their full-length structure due to

genomic instability, while newly inserted LTRs have not yet been affected by genomic instability. Full-length LTRs are identified based on structural detection, and old LTRs, due to their loss of intact structure, cannot be detected. Full-length LTRs account for only 2.6% of all LTRs, and most of LTRs are fragmented. To estimate the age of most TEs, we perform kimura distance-based copy divergence analyses. The younger the TE, the closer its kimura value is to 0. If most of TE are newly inserted, then there will be peak at Kimura= 0. However, we observed that only a small number of TEs exhibit Kimura values of 0, while the majority of TEs show peaks around 10-20.

We realize that our previous descriptions of results were not clear enough and could easily be misunderstood by readers. In the revised version, we have modified these descriptions to make the meaning clearer (**Lines 403-409**).

The authors go on to state ‘Most of the TE families could be detected in A and B subgenomes’: This phrase should probably be rewritten to state ‘Most of the TE families could be detected in both A and B subgenomes, and no A- or B- specific TE families were found’. The logic here needs to be better explained. Within a single species, there is introduction of new transposons as the species evolves, such that individual strains within a species will differ by hundreds to thousands of new transposons. Therefore distinct A- and B- species would possess thousands of different transposons on their chromosomes. The fact that no A- or B-subgenome specific transposons were detected therefore means that most transposon families moved from the A subgenome to the B subgenome, or vice versa. Therefore, it is likely that the large increase in transposon copy number observed in polyploid strains occurred during of after polyploidization.

Response: Thanks for suggestion, the sentence has been rewritten (**Lines 415-417**).

To explore the activity history of TE more clearly, we drew a schematic diagram. The kimura distance can be used to roughly estimate the “age” of TE. The younger TE is, the closer it is to 0.

According to mitochondrial evidence, hybridization is a very recent event, and if TE burst after hybridization, the kimura distribution should be as shown in case 3. Similarly, if most TEs moved from the A subgenome to the B subgenome, then they must have been inserted after hybridization between AA and B, and the kimura distance should be as shown in case 3. The kimura distribution in our study is more in line with the scenario depicted in case 1 or case 2. The situation in case 2 was described in *Xenopus laevis* (doi: 10.1038/nature19840). In order to detect whether there is a situation similar to case 2 in our study, we detected whether there are A- or B- specific TE family. The results shows that no A- or B- specific TE families were found. Therefore, scenario 2 is ruled out and we considered that scenario 1 is the most likely scenario.

Despite the above logic, the authors seem convinced based on their data in Fig S39 that most transposons are old and moved only in the diploid ancestors. I do not see how this is possible. The authors need to very clearly explain the premise of the data in Fig. S39 and why this means that most transposons are old and do not move. If the authors have clear independent lines of evidence for the hypothesis that most transposons are old and do not move, please convey this information. Please also consider the alternative hypothesis that TE bursts might have occurred during or after polyploidization. And then perhaps speculate on how these distinct possibilities will be resolved in future work.

Response: Thanks for suggestion, we have added a description of kimura substitution levels to estimate TE age in the S39 legend.

The Kimura distance used for estimating the age of TE is a commonly employed method by other researchers (doi: 10.1111/pbi.13018; doi: 10.1186/s13100-020-00218-8).

The distribution of the kimura substitution level in this study suggests that the age of most TEs is old, while polyploidization is a recent event, so we considered that most TEs are accumulated in the ancestor instead of during or after polyploidization.

Comment:

13. Fig. S39a could be better explained. Why are the Kimura substitution levels for transposons so much higher in polyploids than in Mg?

Response:

The TE content in polyploids is higher than in Mg. In fact, we have mentioned this earlier in the text, and previous studies have also pointed this out, so we did not repeat it here. What we are investigating here is whether the transposon expansion in polyploids occurred before or after hybridization. Given that hybridization is a very recent event, and the majority of transposon peaks are not close to 0, we believe that the transposon burst likely occurred in the diploid ancestor stage.

Fig. S39a is a supplemental figure that has space to explain what the mean. Was this mentioned with references in the text? Perhaps the authors could add a sentence or two to Fig. S39a about why Kimura substitution levels are likely to mean a diploid burst, if it is at all possible to differentiate a diploid from polyploid burst. If a diploid burst, then this would mean that a factor triggers dramatic transposon expansion prior to polyploidy.

Response:

Thanks for suggestion, we have added a description of kimura distance to estimate TE age in the S39 legend, and we have cited this figure in main text. Furthermore, our description in the previous round was not very accurate, and TE is gradually accumulated in diploids, rather than bursting out. How these TEs accumulated in the diploid ancestor of RKNs, such as whether there are any triggering factors that lead to this result, requires more work in the future.